# Learning Utilities from Demonstrations in Markov Decision Processes

## Abstract

Our goal is to extract useful knowledge from demonstrations of behavior in sequential decision-making problems. Although it is well-known that humans commonly engage in *risk-sensitive* behaviors in the presence of stochasticity, most Inverse Reinforcement Learning (IRL) models assume a *risk-neutral* agent. Beyond introducing model misspecification, these models do not directly capture the risk attitude of the observed agent, which can be crucial in many applications. In this paper, we propose a novel model of behavior in Markov Decision Processes (MDPs) that explicitly represents the agent's risk attitude through a *utility* function. We then define the Utility Learning (UL) problem as the task of inferring the observed agent's risk attitude, encoded via a utility function, from demonstrations in MDPs, and we analyze the partial identifiability of the agent's utility. Furthermore, we devise two provably efficient algorithms for UL in a finite-data regime, and we analyze their sample complexity. We conclude with proof-of-concept experiments that empirically validate both our model and our algorithms.

## 1 Introduction

The ultimate goal of Artificial Intelligence (AI) is to construct artificial rational autonomous agents (Russell & Norvig, 2010). Such agents will interact with each other and with human beings to achieve the tasks that *we* assign to them. In this vision, a crucial feature is being able to correctly model the *observed* behavior of other agents. This allows a variety of applications: $(i)$ *descriptive*, to understand the intent of the observed agent (Russell, 1998), $(ii)$ *predictive*, to anticipate the behavior of the observed agent (potentially in new scenarios) (Arora & Doshi, 2021), $(iii)$ *normative*, to imitate the observed agent because they are behaving in the "right way" (Osa et al., 2018).

Nowadays, Inverse Reinforcement Learning (IRL) provides the most popular and powerful models, i.e., simplified representations, of the behavior of the observed agent, named "expert". Under the so-called "reward hypothesis" (Sutton & Barto, 2018), that has been recently re-interpreted in terms of properties of preferences over trajectories (Shakerinava & Ravanbakhsh, 2022; Bowling et al., 2023), IRL algorithms construct *reward functions* representing the objectives and the desires of the expert. Depending on the application, different models can be adopted. For instance, Ng & Russell (2000) considers the expert as an exact expected return maximizer, while Ramachandran & Amir (2007) and Ziebart et al. (2008) assume that the probability with which actions and trajectories, respectively, are played is proportional to their fraction of optimality (i.e., of expected return).

All these models assume that the expert is a *risk-neutral* agent, i.e., an agent interested in the maximization of the *expected* return. However, there are many scenarios in which rational agents (Föllmer & Schied, 2016), as well as humans (Kahneman & Tversky, 1979; Kreps, 1988), adopt *risk-sensitive* strategies in the presence of stochasticity. In the most general case, agents are not only interested in the *expected* return, but in the full *distribution* of the return (Bellemare et al., 2023). Popular examples in this context include agents who aim to maximize the *expected* return while trying to minimize the variance (Mannor & Tsitsiklis, 2011; Tamar et al., 2012), agents interested in the optimization of the Conditional Value-at-Risk (CVaR) (Rockafellar & Uryasev, 2000), or in rewards volatility (Bisi et al., 2020). IRL models, thus, incur in mis-specification, which can crucially affect the descriptive, predictive, and normative power of the inferred reward function (Skalse & Abate, 2023; 2024; Chan et al., 2021).

**Related works.** To overcome this limitation, some authors have analyzed the *risk-sensitive IRL* problem (Ratliff & Mazumdar, 2020; Majumdar et al., 2017; Cao et al., 2024), in which either the learner is provided with the reward function of the expert and it must infer some parameters representing the risk attitude, or the learner must infer both the reward function and the risk attitude from the given demonstrations (Ratliff & Mazumdar, 2020; Majumdar et al., 2017; Chen et al., 2019; Cheng et al., 2023). Nevertheless, these works suffer from major limitations that prevent the adoption of the proposed algorithms in real-world applications. For instance, they either make demanding assumptions (e.g., Boltzmann policies like in Ratliff & Mazumdar (2020) and Cao et al. (2024), which hypothesize the expert to play each action exactly proportionally to its $Q$-value), or consider rather limited settings (e.g., the "prepare-react model" of Majumdar et al. (2017), that imposes too much structure in the expert's behavior and in the environment's dynamics).

An analogous line of research focuses on the problem of learning the risk attitude of an agent from demonstrations in certain decision-making settings other than Markov Decision Processes (MDPs) (Chajewska et al., 2001; Shukla et al., 2017; Lei, 2020). Even though the powerful model of von Neumann-Morgenstern (vNM) *utility functions* (von Neumann & Morgenstern, 1947; Kreps, 1988) is adopted for representing the risk attitude of the expert, these works only focus on "coarse" sequential decision-making settings like decision trees (Chajewska et al., 2001), that do not provide the rich expressivity of MDPs (there is no notion of reward function). A more detailed analysis, along with additional related works, is provided in Appendix A.

**Our proposal.** In this paper, we formalize, characterize, and analyze the problem of inferring the *risk attitude* of an agent, encoded with a utility function, from demonstrations of behavior in MDPs. The main contributions of this paper are listed below. The proofs of all results are in Appendix C-E.

- Motivated by a real-world example, we propose a simple yet powerful *model* of behavior in MDPs, that separates the objective (reward) from the risk attitude (utility) of an agent (Section 3).
- We introduce *Utility Learning* (UL) as the problem of *inferring the risk attitude* of an agent in MDPs, and we characterise the partial identifiability of the expert's utility (Section 4).
- We present and theoretically analyze two novel algorithms, **CATY-UL** and **TRACTOR-UL**, for efficiently solving the Utility Learning problem with finite data (Section 5).
- We conclude the paper with proof-of-concept experiments that serve as an empirical validation of both the proposed model and the presented algorithms. (Section 6).

## 2 PRELIMINARIES

The main paper's notation is below. Additional notation for the supplemental is in Appendix B.

**Notation.** For any $N \in \mathbb{N}$, we write $[\![N]\!] := \{1, \ldots, N\}$. Given set $\mathcal{X}$, we denote by $\Delta^{\mathcal{X}}$ the probability simplex on $\mathcal{X}$. Given $\mathcal{X} \subseteq \mathbb{R}^d, y \in \mathbb{R}^d$, we define $\Pi_{\mathcal{X}}(y) := \arg\min_{x \in \mathcal{X}} \|y - x\|_2$. A real-valued function $f : \mathbb{R} \to \mathbb{R}$ is *L-Lipschitz* if, for all $x, y \in \mathbb{R}$, we have $|f(x) - f(y)| \leqslant L|x - y|$. $f$ is *increasing* if, for all $x < y \in \mathbb{R}$, it holds $f(x) \leqslant f(y)$, and it is *strictly-increasing* if $f(x) < f(y)$. The probability distribution that puts all its mass on $z \in \mathbb{R}$ is denoted by $\delta_z$ and is called the *Dirac delta*. We represent probability measures on finite support as finite mixtures of Dirac deltas.

**Markov Decision Processes (MDPs).** A tabular episodic Markov Decision Process (MDP) (Puterman, 1994) is a tuple $\mathcal{M} = (\mathcal{S}, \mathcal{A}, H, s_0, p, r)$, where $\mathcal{S}$ and $\mathcal{A}$ are the finite state ($S := |\mathcal{S}|$) and action ($A := |\mathcal{A}|$) spaces, $H$ is the time horizon, $s_0 \in \mathcal{S}$ is the initial state, $p : \mathcal{S} \times \mathcal{A} \times [\![H]\!] \to \Delta^{\mathcal{S}}$ is the transition model, and $r : \mathcal{S} \times \mathcal{A} \times [\![H]\!] \to [0, 1]$ is the *deterministic* reward function. The interaction of an agent with $\mathcal{M}$ generates trajectories. Let $\Omega_h := (\mathcal{S} \times \mathcal{A})^{h-1} \times \mathcal{S}$ be the set of state-action trajectories of length $h$ for all $h \in [\![H + 1]\!]$, and $\Omega := \Omega_{H+1}$. A deterministic *non-Markovian* policy $\pi = \{\pi_h\}_{h \in [\![H]\!]}$ is a sequence of functions $\pi_h : \Omega_h \to \mathcal{A}$ that, given the history up to stage $h$, i.e., $\omega = (s_1, a_1 \ldots, s_{h-1}, a_{h-1}, s_h) \in \Omega_h$, prescribes an action. A *Markovian* policy $\pi = \{\pi_h\}_{h \in [\![H]\!]}$ is a sequence of functions $\pi_h : \mathcal{S} \to \mathcal{A}$ that depend on the current state only. We use $g : \bigcup_{h \in \{2, \ldots, H+1\}} \Omega_h \to [0, H]$ to denote the return of a (partial) trajectory $\omega \in \Omega_h$, i.e., $g(\omega) := \sum_{h' \in [\![h-1]\!]} r_{h'}(s_{h'}, a_{h'})$. With abuse of notation, we denote by $\mathbb{P}_{p,r,\pi}$ the probability distribution over trajectories of any length induced by $\pi$ in $\mathcal{M}$ (we omit $s_0$ for simplicity), and by $\mathbb{E}_{p,r,\pi}$ the expectation w.r.t. $\mathbb{P}_{p,r,\pi}$. We define the *return distribution* $\eta^{p,r,\pi} \in \Delta^{[0,H]}$ of policy $\pi$ as $\eta^{p,r,\pi}(y) := \sum_{\omega \in \Omega: g(\omega)=y} \mathbb{P}_{p,r,\pi}(\omega)$ for all $y \in [0, H]$. The set of possible returns at $h \in [\![H + 1]\!]$ is $\mathcal{G}_h^{p,r} := \{y \in [0, h-1] \mid \exists \omega \in \Omega_h, \exists \pi : g(\omega) = y \wedge \mathbb{P}_{p,r,\pi}(\omega) > 0\}$, and $\mathcal{G}^{p,r} := \mathcal{G}_{H+1}^{p,r}$.

We remark that $\mathcal{G}_h^{p,r}$ has finite cardinality for all $h$. The performance of policy $\pi$ is given by $J^\pi(p,r) := \mathbb{E}_{p,r,\pi}[\sum_{h=1}^H r_h(s_h, a_h)]$, and note that $J^\pi(p,r) = \mathbb{E}_{G\sim\eta^{p,r,\pi}}[G]$. We define the optimal performance as $J^*(p,r) := \max_\pi J^\pi(p,r)$, and the optimal policy as $\pi^* \in \arg\max_\pi J^\pi(p,r)$.

**Risk-Sensitive Markov Decision Processes (RS-MDPs).** A Risk-Sensitive Markov Decision Process (RS-MDP) (Wu & Xu, 2023) is a pair $\mathcal{M}_U := (\mathcal{M}, U)$, where $\mathcal{M} = (\mathcal{S}, \mathcal{A}, H, s_0, p, r)$ is an MDP, and $U \in \mathfrak{U}$ is a utility function in set $\mathfrak{U} := \{U' : [0,H] \to [0,H] \mid U'(0) = 0, U'(H) = H \wedge U'$ is strictly-increasing and continuous$\}$. Differently from Wu & Xu (2023), w.l.o.g., our utilities satisfy $U(H) = H$ to settle the scale. The interaction with $\mathcal{M}_U$ is the same as with $\mathcal{M}$, and the notation described earlier still applies, except for the performance of policies. The performance of policy $\pi$ is $J^\pi(U; p, r) := \mathbb{E}_{p,r,\pi}[U(\sum_{h=1}^H r_h(s_h, a_h))]$, and note that $J^\pi(U; p, r) = \mathbb{E}_{G\sim\eta^{p,r,\pi}}[U(G)]$. We define the optimal performance as $J^*(U; p, r) := \max_\pi J^\pi(U; p, r)$, the optimal policy as $\pi^* \in \arg\max_\pi J^\pi(U; p, r)$, and the set of optimal policies for $\mathcal{M}_U$ as $\Pi^*_{p,r}(U)$.

**Enlarged state space approach.** In MDPs, there always exists a *Markovian* optimal policy (Puterman, 1994), but in RS-MDPs this does not hold. The *enlarged state space approach* (Wu & Xu, 2023) is a method, proposed by Bäuerle & Rieder (2014), to compute an optimal policy in a RS-MDP. Given RS-MDP $\mathcal{M}_U = (\mathcal{S}, \mathcal{A}, H, s_0, p, r, U)$, we construct the *enlarged* state space MDP $\mathfrak{E}[\mathcal{M}_U] = (\{\mathcal{S} \times \mathcal{G}_h^{p,r}\}_{h\in\llbracket H\rrbracket}, \mathcal{A}, H, (s_0, 0), \mathfrak{p}, \mathfrak{r})$, with a different state space $\mathcal{S} \times \mathcal{G}_h^{p,r}$ at each $h$.[1] For every $h \in \llbracket H \rrbracket$ and $(s, y, a) \in \mathcal{S} \times \mathcal{G}_h^{p,r} \times \mathcal{A}$, the reward function $\mathfrak{r}$ is $\mathfrak{r}_h(s, y, a) = U(y + r_h(s, a))\mathbb{1}\{h = H\}$, while the dynamics $\mathfrak{p}$ assigns to the next state $(s', y') \in \mathcal{S} \times \mathcal{G}_{h+1}^{p,r}$ the probability: $\mathfrak{p}_h(s', y'|s, y, a) := p_h(s'|s, a)\mathbb{1}\{y' = y + r_h(s, a)\}$. In words, the state space is enlarged with a component that keeps track of the cumulative reward in the original RS-MDP, and the reward $\mathfrak{r}$, bounded in $[0, H]$, provides the utility of the accumulated reward at the end of the episode. A Markovian policy $\psi = \{\psi_h\}_{h\in\llbracket H\rrbracket}$ for $\mathfrak{E}[\mathcal{M}_U]$ is a sequence of mappings $\psi_h : \mathcal{S} \times \mathcal{G}_h^{p,r} \to \mathcal{A}$. Being an MDP, we adopt for $\mathfrak{E}[\mathcal{M}_U]$ the same notation presented earlier for MDPs, by replacing $p, r, \pi$ with $\mathfrak{p}, \mathfrak{r}, \psi$. Let $\psi^*$ be the optimal *Markovian* policy for $\mathfrak{E}[\mathcal{M}_U]$. Then, Theorem 3.1 of Bäuerle & Rieder (2014) shows that the (non-Markovian) policy $\pi^*$, defined for all $h \in \{2, \ldots, H\}$ and $\omega \in \Omega_h$ as $\pi_h^*(\omega) := \psi_h^*(s_h, \sum_{h'\in\llbracket h-1\rrbracket} r_{h'}(s_{h'}, a_{h'}))$, and $\pi_1^*(s_0) = \psi_1^*(s_0, 0)$, is optimal for $\mathcal{M}_U$.

**Inverse Reinforcement Learning (IRL).** IRL aims to recover the reward function of an expert agent from demonstrations of behavior (Russell, 1998). In the literature (e.g., Ng & Russell (2000); Ziebart et al. (2008); Ramachandran & Amir (2007)), various assumptions are made on how the expert's policy $\pi^E$ is generated from the expert's MDP $\mathcal{M} = (\mathcal{S}, \mathcal{A}, H, s_0, p, r^E)$. Given the expert's MDP without reward $(\mathcal{S}, \mathcal{A}, H, s_0, p)$, the expert's policy $\pi^E$, and the specific assumption considered, the IRL objective is to recover the reward $r^E$.

**Miscellaneous.** For $L > 0$, we write $\mathfrak{U}_L := \{U \in \mathfrak{U} \mid U$ is $L$-Lipschitz$\}$. For any finite set $\mathcal{X} \subseteq [0, H]$ we define $\overline{\mathfrak{U}}^{\mathcal{X}} := \{\overline{U} \in [0, H]^{|\mathcal{X}|} \mid \exists U \in \mathfrak{U}, \forall x \in \mathcal{X} : \overline{U}(x) = U(x)\}$, and $\overline{\mathfrak{U}}_L^{\mathcal{X}} := \{\overline{U} \in \overline{\mathfrak{U}}^{\mathcal{X}} \mid \exists U \in \mathfrak{U}_L, \forall x \in \mathcal{X} : \overline{U}(x) = U(x)\}$. We will denote by $\mathcal{M}_{\overline{U}}$ some RS-MDPs with $\overline{U} \in \overline{\mathfrak{U}}^{\mathcal{X}}$.

# 3 MOTIVATION AND PROBLEM SETTING

In this section, we begin by motivating the need for a more expressive model of behavior in MDPs. Next, we propose a risk-aware model and we justify it. We conclude with some observations.

**Existing models for representing behavior.** Our goal is to develop an algorithm, that permits to learn a "*good*" model of behavior of an agent from demonstrations in an MDP. In this context, the most common models present in the literature enforce a structure made of **two** components: $(i)$ a *reward function*, that represents the objective of the agent, and $(ii)$ a *planning method*, that describes *how the behavior of the agent is generated given its objective*. Crucially, the planning method is assumed to be *known*,[2] thus, all the information about the behavior must hold inside the reward (the objective) that can be *learned*. Popular examples include IRL (Ng & Russell, 2000), entropy-regularized IRL, (Ziebart et al., 2008), and Bayesian IRL (Ramachandran & Amir, 2007).

---

[1] Actually, Bäuerle & Rieder (2014) use state space $\mathcal{S} \times \mathbb{R}_{\geqslant 0}$, while Wu & Xu (2023) use $\mathcal{S} \times [h-1]$ for all $h \in \llbracket H \rrbracket$. Instead, we consider sets $\mathcal{S} \times \{\mathcal{G}_h^{p,r}\}_h$ to capture the minimal size required.

[2] Indeed, Armstrong & Mindermann (2018) have demonstrated that "it is impossible to uniquely decompose a policy into a planning algorithm and reward function", but we need to impose some structure to the problem.

**Limitations.** Our insight is that these models are *not expressive enough* to model human behavior in the presence of stochasticity in many common situations, as shown in the following example.

**Example 3.1.** *Consider the MDP on the side, where you can reach state $s$ having already earned either $0€$ or $100€$ (in this example reward is money). From $s$, you can take either the "risky" action*

*$a_{risky}$, that provides you with $200€$ with probability (w.p.) $1/2$ or $0€$ otherwise, or the "safe" action $a_{safe}$, that provides you always with $50€$.* What action would you play in state $s$? *Risk-averse (Kahneman & Tversky, 1979) people might go with $a_{safe}$ when landed on $s$ with $0€$, and with $a_{risky}$ otherwise, while risk-seeking people might*

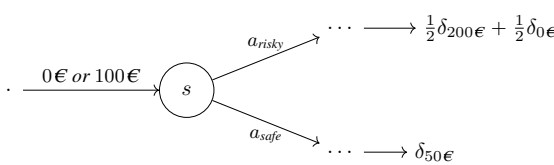

*always go with $a_{risky}$. Simply put, the current state $s$ is not sufficient for predicting behavior, because people decide to take risks depending on how much money (i.e., reward) they earned so far.*

In other words, people might exhibit a *non-Markovian* behavior dependent on both the state *and* the cumulated rewards, which is not contemplated by the aforementioned IRL models.[3]

**Our proposal.** We propose to *explicitly* represent the risk attitude by constructing a model with **three** components: $(i)$ a *reward function*, i.e., the objective; $(ii)$ a *utility function*, i.e., the risk attitude, $(iii)$ a (known) *planning method*, i.e., how the behavior of the agent is generated given its objective *and its risk-attitude*. Formally, we model the expert as an optimal agent in a RS-MDP:

$$\pi^E \in \underset{\pi}{\arg\max}\, \mathbb{E}_{p,r,\pi}\Big[ U\Big( \sum_{h=1}^{H} r_h(s_h,a_h) \Big) \Big], \tag{1}$$

where $(i)$ $r$ is the reward, $(ii)$ $U \in \mathfrak{U}$ is the utility, and $(iii)$ the principle of maximization of the expected utility is the planning method. There are many arguments that support this model:

1. it generalizes the IRL model of Ng & Russell (2000) by modelling the risk attitude through $U$;
2. it is justified by the famous expected utility theory (von Neumann & Morgenstern, 1947);[4]
3. it explains the existence of non-Markovian optimal policies (see Bäuerle & Rieder (2014));
4. the corresponding planning problem enjoys practical tractability (Wu & Xu, 2023).

**Some considerations.** If the utility $U$ is linear, the RS-MDP $\mathcal{M}_U$ admits a *Markovian* optimal policy. Otherwise, the more $U$ deviates from linearity, the more non-Markovian policies *may* out-perform Markovian policies, which may incur in a finite loss of performance, as shown below.

**Proposition 3.1.** *There exists a RS-MDP with horizon $H = 4$ in which the difference between the optimal performance and the performance of the best Markovian policy is $0.5$.*

Next, we observe that also any deterministic RS-MDP admits an optimal *Markovian* policy. Intuitively, in absence of risk (i.e., stochasticity) the utility function plays no role.

**Proposition 3.2.** *Given any RS-MDP with deterministic transition model $p$ and reward function $r$, if the utility $U$ is increasing, then, there exists a Markovian optimal policy.*

Finally, if we restrict to Markovian policies, we note that non-stationarity (i.e., the dependence of the policy on the stage $h$) and stochasticity (i.e., if the policy prescribes a lottery over actions instead of a single action) can improve the performance even in stationary environments. Intuitively, they permit to consider larger ranges of return distributions w.r.t. stationary deterministic policies.

**Proposition 3.3.** *There exists a RS-MDP with stationary transition model and reward in which the best Markovian policy is non-stationary, and the best stationary Markovian policy is stochastic.*

## 4  UTILITY LEARNING

In this section, we formalise the Utility Learning problem, we characterise the partial identifiability of the true utility from demonstrations, and we analyze the inferred utilities for applications.

---

[3]Re-modelling the MDP including the reward into the state would make the optimal policy Markovian. Yet, this mathematical device would incur in various issues, as explained in Appendix C.2.

[4]The set of prizes is $\mathcal{G}^{p,r}$, and each policy $\pi$ is a choice that induces a lottery $\eta^{p,r,\pi}$ over prizes.

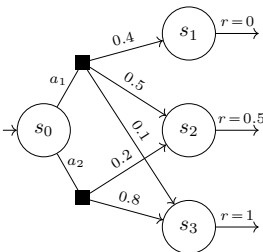 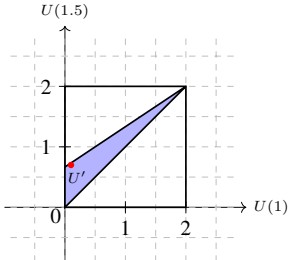 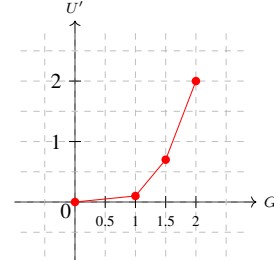

Figure 1: (Left) the MDP of Example 4.1. (Middle) its set of feasible utilities with a sample utility $U'$. (Right) plot of $U'$ with linear interpolation; being convex, it represents a risk-lover agent (Bäuerle & Rieder, 2014).

**Learning from demonstrations under the new model.** In Section 3, we described a model that parametrizes the behavior of an agent through two components: a reward $r$, and a utility $U$. Given demonstrations of behavior in an MDP, Eq. (1) defines three different learning problems:

1. **Utility Learning (UL)**: given $r$, learn $U$.
2. **Inverse Reinforcement Learning (IRL)**: given $U$, learn $r$.
3. **IRL + UL**: learn both $r$ and $U$.

Problem 3 is the most interesting (and challenging), because it makes the least assumptions, while Problem 2 has been extensively studied in literature when $U$ is linear (Ng & Russell, 2000) (but not in detail for other choices of $U$). In this paper, we focus on Problem 1, which we name *Utility Learning (UL)*, for two reasons. First, there exist relevant applications of UL per se (see the last part of this section). Second, understanding UL represents a significant step toward solving Problem 3.

**Partial identifiability in utility learning.** In the *exact* UL setting (where $s_0, p, \pi^E$ are known), by definition, we are given a policy $\pi^E$ and an MDP $\mathcal{M} = (\mathcal{S}, \mathcal{A}, H, s_0, p, r)$, and the goal is to find the expert's utility function $U^E \in \mathfrak{U}$ that satisfies $J^*(U^E; p, r) = J^{\pi^E}(U^E; p, r)$, i.e., that makes $\pi^E$ optimal in RS-MDP $\mathcal{M}_{U^E}$. Does the knowledge of $\pi^E$ and $\mathcal{M}$ suffice to *uniquely* identify $U^E$? Analogously to IRL (Cao et al., 2021; Kim et al., 2021; Skalse et al., 2023), the answer is *negative*, as shown in the following example.

**Example 4.1.** *Let* $\mathcal{M} = (\mathcal{S}, \mathcal{A}, H, s_0, p, r)$ *be the MDP in Fig. 1 (left), where* $H = 2, r_1(s_0, a_1) = 1, r_1(s_0, a_2) = 0.5$, *and all other values of* $p, r$ *are drawn in the figure. Note that* $\mathcal{G}^{p,r} = \{0, 1, 1.5, 2\}$. *Let* $\pi^E$ *be the expert policy, that prescribes* $a_1$ *in* $s_0$. *Then, a utility* $U \in \mathfrak{U}$ *makes* $\pi^E$ *optimal for* $\mathcal{M}_U$ *if playing* $a_1$ *is better than playing* $a_2$: $J^{\pi^E}(U; p, r) = 0.1U(2) + 0.5U(1.5) + 0.4U(1) \geqslant 0.8U(1.5) + 0.2U(1)$. *Thus, all the utilities* $U \in \mathfrak{U}$, *that assign to* $G = 1, G = 1.5$ *any of the values coloured in blue in Fig. 1 (middle), are equally-plausible candidates to be* $U^E$.

Example 4.1 shows that $U^E$ is just *partially identifiable* from demonstrations. In particular, we cannot uniquely identify the value of $U^E$ at points in the set $\mathcal{G}^{p,r}$, and we do not have information on $U^E$ at the other points $[0, H] \setminus \mathcal{G}^{p,r}$. Similarly to Metelli et al. (2023), we formalize the set of utilities "compatible" with $\pi^E$ in $\mathcal{M}$ by introducing the notion of *feasible utility set*:[5]

**Definition 4.1** (Feasible Utility Set). *Let* $\mathcal{M} = (\mathcal{S}, \mathcal{A}, H, s_0, p, r)$ *be an MDP, and let* $\pi^E$ *be the (potentially non-Markovian) expert policy. The* feasible utility set $\mathcal{U}_{p,r,\pi^E}$ *contains all the utilities that make* $\pi^E$ *optimal for RS-MDP* $\mathcal{M}_U$. *Formally:* $\mathcal{U}_{p,r,\pi^E} := \{U \in \mathfrak{U} \mid J^{\pi^E}(U; p, r) = J^*(U; p, r)\}$.

**Usage and transferability of utilities.** UL is a problem setting for inferring the risk attitude of an agent. Once learned, we might "use" the computed utility $\widehat{U}$ for $(i)$ *predicting* the behavior of the expert in a new environment, $(ii)$ *imitating* the expert, or $(iii)$ *assessing* how valuable a certain policy is from the viewpoint of the expert. However, due to partial identifiability, $\widehat{U}$ cannot be close to $U^E$ more than the worst utility in the feasible set $\mathcal{U}_{p,r,\pi^E}$. Is this "ambiguity" tolerated by the applications above? The following propositions answer *negatively* for all $(i), (ii)$, and $(iii)$ of them, but, fortunately, Proposition 4.6 shows that adding more data can solve the issue.

Let us begin with $(i)$. In our model, a utility $\widehat{U}$ permits to predict the behavior of an agent with utility $U^E$ in a new MDP $\mathcal{M}'$ if $\widehat{U}$ and $U^E$ induce in $\mathcal{M}'$ the same optimal policies. Nevertheless, *not* all the utilities in the feasible set satisfy this property for all the possible MDPs, as shown below:

---

[5]In Appendix D we provide a more explicit expression of the feasible utility set.

**Proposition 4.1** (Transfer to a new transition model). *There exist two MDPs $\mathcal{M} = (\mathcal{S}, \mathcal{A}, H, s_0, p, r)$, $\mathcal{M}' = (\mathcal{S}, \mathcal{A}, H, s_0, p', r)$, with $p \neq p'$, for which there exists a policy $\pi^E$ and a pair of utilities $U_1, U_2 \in \mathfrak{U}$ such that: $U_1, U_2 \in \mathcal{U}_{p,r,\pi^E}$ and $\Pi^*_{p',r}(U_1) \cap \Pi^*_{p',r}(U_2) = \{\}$.*

**Proposition 4.2** (Transfer to a new reward). *There exist two MDPs $\mathcal{M} = (\mathcal{S}, \mathcal{A}, H, s_0, p, r)$, $\mathcal{M}' = (\mathcal{S}, \mathcal{A}, H, s_0, p, r')$, with $r \neq r'$, for which there exists a policy $\pi^E$ and a pair of utilities $U_1, U_2 \in \mathfrak{U}$ such that: $U_1, U_2 \in \mathcal{U}_{p,r,\pi^E}$ and $\Pi^*_{p,r'}(U_1) \cap \Pi^*_{p,r'}(U_2) = \{\}$.*

Intuitively, we are saying that transferring the learned utility $\widehat{U}$ to an MDP with a different reward or transition model might cause it to induce optimal policies other than those induced by $U^E$ there.

Consider now $(ii)$. To perform a meaningful imitation, due to the practical difficulty of computing optimal policies, we require that any policy with an almost optimal performance for $\widehat{U}$ has also a "good" performance for $U^E$, but this does not always hold:

**Proposition 4.3.** *There exists an MDP $\mathcal{M} = (\mathcal{S}, \mathcal{A}, H, s_0, p, r)$ and a policy $\pi^E$ for which there exists a pair of utilities $U_1, U_2 \in \mathcal{U}_{p,r,\pi^E}$ such that, for any $\epsilon \geqslant 0$ smaller than some constant, there exists a policy $\pi_\epsilon$ such that $J^*(U_1; p, r) - J^{\pi_\epsilon}(U_1; p, r) = \epsilon$ and $J^*(U_2; p, r) - J^{\pi_\epsilon}(U_2; p, r) \geqslant 1$.*

Finally, concerning $(iii)$, the fact that $\widehat{U}$ and $U^E$ provide close values of performance for all the policies seems a desirable requirement, i.e., asking for small $d^{\mathrm{all}}_{p,r}(U^E, \widehat{U}) := \max_\pi \left| J^\pi(U^E; p, r) - J^\pi(\widehat{U}; p, r) \right|$ Zhao et al. (2024). We note that closeness under some norm implies closeness in $d^{\mathrm{all}}$:

**Proposition 4.4.** *Consider an arbitrary MDP with transition model $p$ and reward function $r$. Then, for any pair of utilities $U_1, U_2 \in \mathfrak{U}$, it holds that $d^{all}_{p,r}(U_1, U_2) \leqslant \max_{G \in \mathcal{G}^{p,r}} |U_1(G) - U_2(G)|$.*

Nonetheless, *not* all the utilities in the feasible set are close to each other in $d^{\mathrm{all}}_{p,r}$ distance:

**Proposition 4.5.** *There exists an MDP $\mathcal{M} = (\mathcal{S}, \mathcal{A}, H, s_0, p, r)$ and a policy $\pi^E$ for which there exists a pair of utilities $U_1, U_2 \in \mathcal{U}_{p,r,\pi^E}$ such that $d^{all}_{p,r}(U_1, U_2) = 1$.*

Intuitively, Propositions 4.1-4.3, 4.5, prove that demonstrations of behavior in a *single* MDP do not provide enough information on $U^E$ to obtain guarantees for applications $(i), (ii), (iii)$. Instead, the following result shows that demonstrations in *multiple* environments permit to mitigate this issue.

**Proposition 4.6** (Multiple demonstrations). *Let $\mathcal{S}, \mathcal{A}, H$ be, respectively, any state space, action space, and horizon, satisfying $S \geqslant 3, A \geqslant 2, H \geqslant 2$, and let $U^E \in \mathfrak{U}$ be any utility. If, for any possible dynamics $s_0, p$ and reward $r$, we are given the set of* all *the deterministic optimal policies of the corresponding RS-MDP $(\mathcal{S}, \mathcal{A}, H, s_0, p, r, U^E)$, then we can uniquely identify utility $U^E$.*

Simply put, through a constructive proof, Proposition 4.6 provides a sufficient condition for uniquely retrieving $U^E$, analogously to Amin & Singh (2016); Cao et al. (2021); Büning et al. (2022).

## 5 ONLINE UTILITY LEARNING WITH GENERATIVE MODEL

In the previous section, we have analyzed UL in the *exact* setting. Here, we introduce a more realistic setting for UL, and we describe two efficient algorithms with theoretical guarantees to address it.

We consider the *online UL* problem setting with demonstrations from multiple environments,[6] which we now define. Let $U^E \in \mathfrak{U}$ be the utility function of the expert. Consider $N$ MDPs $\mathcal{M}^i = (\mathcal{S}^i, \mathcal{A}^i, H, s_0^i, p^i, r^i)$, indexed by $i \in [\![N]\!]$, that share the same horizon $H$. For each $\mathcal{M}^i$, we *know* $\mathcal{S}^i, \mathcal{A}^i, H, s_0^i, r^i$, we have access to a *generative sampling oracle* (Azar et al., 2013) for the transition model $p^i$, which, given any triple $s, a, h$, returns a sample $s' \sim p_h^i(\cdot|s, a)$, and we are given a *dataset* $\mathcal{D}^{E,i} = \{(s_1^j, a_1^j, s_2^j, \ldots, s_H^j, a_H^j, s_{H+1}^j)\}_{j \in [\![\tau^{E,i}]\!]}$ of $\tau^{E,i}$ trajectories collected by executing expert policy $\pi^{E,i}$, which is optimal for the RS-MDP $\mathcal{M}^i_{U^E}$. Informally, the goal is to find $U^E$.

### 5.1 CHALLENGES AND OUR SOLUTION

To develop efficient algorithms for learning utilities *in practice*, some challenges must be addressed.

---

[6]This requirement permits to alleviate the partial identifiability issues, as shown in Proposition 4.6.

**Curse of Dimensionality.** Approximation techniques are needed for computing optimal policies in RS-MDPs (the enlarged state space is too large $|\mathcal{G}_h^{p,r}|\propto(SA)^{(h-1)}$ $\forall h$ (Wu & Xu, 2023)), for storing return distributions (whose support may grow exponentially in the horizon $|\mathcal{G}^{p,r}|\propto(SA)^H$ (Bellemare et al., 2023)), and for storing utilities in $\mathfrak{U}$ (defined over the interval $[0, H]$).

**Finite Data.** Some quantities of interest (i.e., policies and transition models) are not known exactly, but they must be *estimated* from samples, introducing an estimation error.

**Partial Identifiability.** Even in the exact setting, demonstrations of behavior are "explained" *equally well* by *infinitely* many utilities; thus, it is not clear which utility an algorithm should output.

To address these challenges, our algorithms (Sections 5.2- 5.3) adopt the following approaches.

**Curse of Dimensionality.** We combine the $(i)$ *discretization* approach in Wu & Xu (2023) for the enlarged state space, with the $(ii)$ *categorical representation* of Bellemare et al. (2023) for return distributions. Moreover, we consider $(iii)$ *discretized* utilities.

$(i)$ Fix parameter $\epsilon_0 > 0$, define sets $\mathcal{R} := \{0, \epsilon_0, 2\epsilon_0, \ldots, \lfloor 1/\epsilon_0 \rfloor \epsilon_0\}$, $\mathcal{Y}_h := \{0, \epsilon_0, 2\epsilon_0, \ldots, \lfloor (h-1)/\epsilon_0 \rfloor \epsilon_0\}$ as the $\epsilon_0$-coverings of $[0, 1]$ and $[0, h-1]$ for all $h \in \llbracket H + 1 \rrbracket$, and let $\mathcal{Y} := \mathcal{Y}_{H+1}, d := |\mathcal{Y}|$. Intuitively, note that the summation of $h$ values in $\mathcal{R}$ provides a value in $\mathcal{Y}_{h+1}$ for all $h$. Therefore, for any $i \in \llbracket N \rrbracket$, let $\overline{\mathcal{M}}_{U^E}^i := (\mathcal{S}^i, \mathcal{A}^i, H, s_0^i, p^i, \overline{r}^i, U^E)$ be the RS-MDP with reward $\overline{r}^i$, obtained from $r^i$ as $\overline{r}_h^i(s, a) := \Pi_{\mathcal{R}}[r_h^i(s,a)]$ for all $s, a, h$. In this manner, the sets of partial returns of $\overline{\mathcal{M}}^i$ satisfy $\mathcal{G}_h^{p^i, \overline{r}^i} \subseteq \mathcal{Y}_h \subseteq \mathcal{Y}$ for all $h$, thus the MDP $\mathfrak{E}[\overline{\mathcal{M}}_{U^E}^i]$ has a state space with cardinality at most $Sd \leqslant \mathcal{O}(SH/\epsilon_0)$, which is no longer exponential in the horizon. $(ii)$ Denote as $\mathcal{Q} := \{q \in \Delta^{\llbracket d \rrbracket} \mid \sum_{j \in \llbracket d \rrbracket} q_j \delta_{y_j}\}$ the set of parametric probability distributions supported on $\mathcal{Y}$, where $y_1 := 0, y_2 := \epsilon_0, \ldots, y_d := \lfloor H/\epsilon_0 \rfloor \epsilon_0$ represent the *ordered* items of set $\mathcal{Y}$. We construct the categorical representation $\text{Proj}_{\mathcal{C}}(\eta) \in \mathcal{Q}$ of an arbitrary (return) distribution $\eta \in \Delta^{[0,H]}$ through the operator $\text{Proj}_{\mathcal{C}}$, defined in Eq. (4). $(iii)$ We approximate utilities $U \in \mathfrak{U}$ with vectors $\overline{U} \in \overline{\mathfrak{U}} := \overline{\mathfrak{U}}^{\mathcal{Y}}$ so that $\overline{U}(y) = U(y)$ for all $y \in \mathcal{Y}$.

In this way, we work with tractable approximations whose complexity is controlled by parameter $\epsilon_0$.

**Finite Data.** We introduce the notion of *utility compatibility* to cope with finite data. With multiple demonstrations, the true utility $U^E$ satisfies the *hard* constraints $J^{\pi^{E,i}}(U^E; p^i, r^i) = J^*(U^E; p^i, r^i)$ for all $i \in \llbracket N \rrbracket$. However, with finite data, our *estimate* of $J^{\pi^{E,i}}(U^E; p^i, r^i) - J^*(U^E; p^i, r^i)$ might be *different from zero* for some $i$, thus, we might get wrong in recognizing $U^E$ as the true expert's utility. Crucially, collecting more (but still finite) data does *not* guarantee to obtain *exactly* zero. Drawing inspiration from Lazzati et al. (2024a), we relax these "hard" requirements by introducing a "soft" notion of constraints satisfaction, which we name *utility compatibility*:

**Definition 5.1.** *Given MDP* $\mathcal{M} = (\mathcal{S}, \mathcal{A}, H, s_0, p, r)$ *and policy* $\pi^E$, *the* (non)compatibility $\overline{\mathcal{C}}_{p,r,\pi^E} : \mathfrak{U} \to [0, H]$ *of utility* $U \in \mathfrak{U}$ *with* $\pi^E$ *in* $\mathcal{M}$ *is:* $\overline{\mathcal{C}}_{p,r,\pi^E}(U) := J^*(U; p, r) - J^{\pi^E}(U; p, r)$.

Thanks to utility compatibility, we can *quantify* the extent to which a utility $U$ is *(non)compatible* with the (multiple) demonstrations by computing $\max_{i \in \llbracket N \rrbracket} \overline{\mathcal{C}}_{p^i, r^i, \pi^{E,i}}(U)$.

**Partial Identifiability.** We *propose* to develop two *practical* algorithms to fully characterize a set of utility functions: $(i)$ A *utility classifier*,[7] that "defines" the boundaries of the set, and $(ii)$ a *utility extractor*, that extracts a utility from the set.

For a given accuracy threshold $\Delta > 0$, define the set of $\Delta$-compatible utilities as: $\mathcal{U}_\Delta := \{U \in \mathfrak{U} \mid \max_{i \in \llbracket N \rrbracket} \overline{\mathcal{C}}_{p^i, r^i, \pi^{E,i}}(U) \leqslant \Delta\}$. $(i)$ We define a *utility classifier* algorithm as a procedure that takes in input a utility $U \in \mathfrak{U}$, and outputs a boolean saying whether $U \in \mathcal{U}_\Delta$ or not. Intuitively, being the input utility arbitrary, such algorithm permits to characterize the entire set $\mathcal{U}_\Delta$. Furthermore, $(ii)$ we define a *utility extractor* algorithm as a procedure that outputs an arbitrary utility $U$ from set $\mathcal{U}_\Delta$.

### 5.2 **CATY-UL** (COMP**AT**IBILIT**Y** FOR **U**TILITY **L**EARNING)

**CATY-UL** is a *utility classifier* algorithm. It classifies utilities $U$ w.r.t. $\mathcal{U}_\Delta$ by estimating the (non)compatibility $\widehat{\mathcal{C}}^i(U) \approx \mathcal{C}_{p^i, r^i, \pi^{E,i}}(U)$ for all $i \in \llbracket N \rrbracket$, and, then, checking if $\max_{i \in \llbracket N \rrbracket} \widehat{\mathcal{C}}^i(U) \leqslant \Delta$.

---

[7]The notion of reward classifier can be found in Lazzati et al. (2024a). We extend it to utilities.

**Algorithm 1: CATY-UL**

**Input:** data $\{\mathcal{D}_i^E\}_i$, threshold $\Delta$, utility $U$, discretization $\epsilon_0$, dynamics $\{\widehat{p}^i\}_i$

1   $\overline{U}(y) \leftarrow U(y)$   for all $y \in \mathcal{Y}$
2   **for** $i \in \{1, 2, \dots, N\}$ **do**
3     // Estimate $J^{\pi^{E,i}}(U; p^i, r^i)$:
    $\widehat{\eta}^{E,i} \leftarrow \mathrm{ERD}(\mathcal{D}_i^E, r^i)$
4     $\widehat{J}^{E,i}(U) \leftarrow \sum_{y \in \mathcal{Y}} \widehat{\eta}^{E,i}(y) \overline{U}(y)$
    // Estimate $J^*(U; p^i, r^i)$:
5     $\widehat{J}^{*,i}(U), \_ \leftarrow \mathrm{PLANNING}(\overline{U}, i, \widehat{p}^i)$
    // Estimate $\overline{\mathcal{C}}_{p^i, r^i, \pi^{E,i}}(U)$:
6     $\widehat{\mathcal{C}}^i(U) \leftarrow \widehat{J}^{*,i}(U) - \widehat{J}^{E,i}(U)$
7   **end**
8   class $\leftarrow$ True **if** $\max_{i \in [\![N]\!]} \widehat{\mathcal{C}}^i(U) \leqslant \Delta$ **else** False
9   **Return** class

**Algorithm 2: TRACTOR-UL**

**Input:** data $\{\mathcal{D}_i^E\}_i$, parameters $T, K, \alpha, \overline{U}_0$, discretization $\epsilon_0$, dynamics $\{\widehat{p}^i\}_i$

1   $\widehat{\eta}^{E,i} \leftarrow \mathrm{ERD}(\mathcal{D}_i^E, r^i)$   for $i \in [\![N]\!]$
2   **for** $t = 0, \dots, T-1$ **do**
3     // Compute distributions $\{\widehat{\eta}_t^i\}_i$:
    **for** $i$ in $1, 2, \dots, N$ **do**
4      $\_, \widehat{\psi}_t^{*,i} \leftarrow \mathrm{PLANNING}(\overline{U}_t, i, \widehat{p}^i)$
5      $\mathcal{D} \leftarrow \mathrm{ROLLOUT}(\widehat{\psi}_t^{*,i}, \widehat{p}^i, \overline{r}^i, i, K)$
6      $\widehat{\eta}_t^i(y) \leftarrow \frac{1}{K} \sum_{G \in \mathcal{D}} \mathbb{1}\{G = y\}, \forall y \in \mathcal{Y}$
7     **end**
    // Update $\overline{U}_{t+1}$:
8     $g_t \leftarrow \sum_{i \in [\![N]\!]} (\widehat{\eta}_t^i - \widehat{\eta}^{E,i})$
9     $\overline{U}_{t+1} \leftarrow \Pi_{\underline{\mathfrak{U}}_L} (\overline{U}_t - \alpha g_t)$
10   **end**
11   **Return** $\frac{1}{T} \sum_{t=0}^{T-1} \overline{U}_t$

As other algorithms (Jin et al., 2020; Lazzati et al., 2024a), **CATY-UL** comprises two phases: an *exploration* phase (Algorithm 3), where we compute estimates $\{\widehat{p}^i\}_i$ by collecting $\tau^i$ samples from the generative model of each $\mathcal{M}^i$, and a *classification* phase (Algorithm 1), that takes in input a utility $U \in \mathfrak{U}$, estimates $\{\widehat{p}^i\}_i$, and datasets $\{\mathcal{D}^{E,i}\}_i$, to construct estimates $\{\widehat{\mathcal{C}}^i(U)\}_i$ for classifying $U$.

Specifically, at Line 1, we discretize the utility $U$. Next, for all $i \in [\![N]\!]$, we construct estimates $\widehat{J}^{E,i}(U) \approx J^{\pi^{E,i}}(U; p^i, r^i)$ and $\widehat{J}^{*,i}(U) \approx J^*(U; p^i, r^i)$ as follows. At Line 3, we estimate $\widehat{\eta}^{E,i} \approx \mathrm{Proj}_\mathcal{C}(\eta^{p^i, r^i, \pi^{E,i}}) \approx \eta^{p^i, r^i, \pi^{E,i}}$ through the ERD (Estimate Return Distribution) subroutine (Algorithm 5), and dataset $\mathcal{D}^{E,i}$, while at Line 4 we compute $\widehat{J}^{E,i}(U)$. At Line 5, we approximate the optimal performance $J^*(U; p^i, r^i)$ in RS-MDP $\mathcal{M}_U^i$ with the optimal performance $\widehat{J}^{*,i}(U) := J^*(\overline{U}; \widehat{p}^i, \overline{r}^i)$ in RS-MDP $\mathfrak{E}[\widehat{\mathcal{M}}_{\overline{U}}^i] := (\mathcal{S}^i, \mathcal{A}^i, H, s_0^i, \widehat{p}^i, \overline{r}^i, \overline{U})$, which is computed through *value iteration* in the enlarged state space MDP $\mathfrak{E}[\widehat{\mathcal{M}}_{\overline{U}}^i]$ using the PLANNING subroutine (Algorithm 4). Finally, at Line 6 we compute $\widehat{\mathcal{C}}^i(U)$, and at Line 8 we perform the classification. **CATY-UL** enjoys the following guarantee:

**Theorem 5.1.** *Let $\epsilon, \delta \in (0, 1)$, and let $\mathcal{U}$ be a subset of $\mathfrak{U}_L$ containing the utilities to classify. If we set $\epsilon_0 = \epsilon^2/(72HL^2)$, and if it holds that, for all $i \in [\![N]\!]$:*

$$if \ |\mathcal{U}| = 1: \qquad \tau^{E,i} \leqslant \widetilde{\mathcal{O}}\Big(\frac{H^2}{\epsilon^2} \log \frac{N}{\delta}\Big), \qquad \tau^i \leqslant \widetilde{\mathcal{O}}\Big(\frac{SAH^4}{\epsilon^2} \log \frac{SAHNL}{\delta\epsilon}\Big),$$

$$else: \qquad \tau^{E,i} \leqslant \widetilde{\mathcal{O}}\Big(\frac{H^4L^2}{\epsilon^4} \log \frac{HNL}{\delta\epsilon}\Big), \qquad \tau^i \leqslant \widetilde{\mathcal{O}}\Big(\frac{SAH^5}{\epsilon^2}\Big(S + \log \frac{SAHN}{\delta}\Big)\Big),$$

*then, w.p. at least $1 - \delta$, **CATY-UL** correctly classifies all the $U \in \mathcal{U}$ that satisfy either $\max_i \overline{\mathcal{C}}_{p^i, r^i, \pi^{E,i}}(U) < \Delta - \epsilon$ (inside $\mathcal{U}_\Delta$) or $\max_i \overline{\mathcal{C}}_{p^i, r^i, \pi^{E,i}}(U) > \Delta + \epsilon$ (outside $\mathcal{U}_\Delta$).*

Some observations are in order. First, note that $\Delta$ is arbitrary in $[0, H]$, and the sample complexity does not depend on it. If we have one utility to classify $|\mathcal{U}| = 1$, then $\propto S$ queries to the generative model suffice instead of $\propto S^2$. Note that $\epsilon_0$ represents a trade-off between approximation and estimation error. If we re-normalize utilities so that $U(H) = 1$, then some $H$ terms in the bounds disappear. Intuitively, the Lipschitzianity assumption is necessary for approximating *continuous* utilities $U \in \mathcal{U}$ with vectors in $\overline{\mathfrak{U}}$. Finally, observe that we can restrict the range of (non)compatibility $[\Delta - \epsilon, \Delta + \epsilon]$ where **CATY-UL** can make mistake with high probability (w.h.p.) by collecting more data.

## 5.3   TRACTOR-UL (ex**TRACTOR** for **U**tility **L**earning)

For simplicity, let $\overline{\mathfrak{U}}_L := \overline{\mathfrak{U}}_L^\mathcal{Y}$ for $L > 0$, and let $\underline{\mathfrak{U}}, \underline{\mathfrak{U}}_L, \underline{\overline{\mathfrak{U}}}, \underline{\overline{\mathfrak{U}}}_L, \underline{\mathcal{U}}_\Delta$ be the analogous of, respectively, $\mathfrak{U}, \mathfrak{U}_L, \overline{\mathfrak{U}}, \overline{\mathfrak{U}}_L, \mathcal{U}_\Delta$, but containing *increasing* functions instead of *strictly-increasing* functions.[8]

**TRACTOR-UL** is a *utility extractor* algorithm. For any $\Delta > 0$, it aims to extract a utility $U$ from $\underline{\mathcal{U}}_\Delta$ by performing *online gradient descent* in the space of discretized $L$-Lipschitz utilities $\underline{\mathfrak{U}}_L$. It

---

[8] Note that, for defining $\underline{\mathcal{U}}_\Delta$, we extend also the definition of (non)compatibility (Def. 5.1) to utilities in $\underline{\mathfrak{U}}$.

comprises two phases: an *exploration* phase, that coincides with that of `CATY-UL` (Algorithm 3) and aims to compute estimates $\{\widehat{p}^i\}_i$ using $\{\tau^i\}_i$ samples, and an *extraction* phase (Algorithm 2), that takes in input estimates $\{\widehat{p}^i\}_i$, and datasets $\{\mathcal{D}^{E,i}\}_i$, to construct a utility $\widehat{U} \in \underline{\mathfrak{U}}_L$ to return.

Specifically, starting from $\overline{U}_0 \in \underline{\overline{\mathfrak{U}}}_L$, we compute a sequence $\{\overline{U}_1, \ldots, \overline{U}_T\}$ of utilities in $\underline{\overline{\mathfrak{U}}}_L$ through an *online projected gradient descent* scheme, where the gradient $g_t$ is computed at Line 8, and the update is carried out at Line 9 with projection onto $\underline{\overline{\mathfrak{U}}}_L$. Intuitively, we aim to minimize function $\max_{i \in [\![N]\!]} \overline{\mathcal{C}}_{p^i, r^i, \pi^{E,i}}(U) \leqslant \sum_i \overline{\mathcal{C}}_{p^i, r^i, \pi^{E,i}}(U)$ over set $\underline{\mathfrak{U}}_L$ (we upper bound the max with the sum to work with gradients instead of subgradients), but computing the gradient $\nabla_U \sum_{i \in [\![N]\!]} \overline{\mathcal{C}}_{p^i, r^i, \pi^{E,i}}(U) = \sum_i (\nabla_U J^*(U; p^i, r^i) - \eta^{p^i, r^i, \pi^{E,i}})$ is not simple. Thus, analogously to Syed & Schapire (2007); Schlaginhaufen & Kamgarpour (2024), we replace $\nabla_U J^*(U; p^i, r^i)$ with $\nabla_U J^{\pi_t^{*,i}}(U; p^i, r^i) = \eta^{p^i, r^i, \pi_t^{*,i}}$, where $\pi_t^{*,i}$ is the (fixed) optimal policy in RS-MDP $\mathcal{M}_{U_t}^i$ ($U_t \in \underline{\mathfrak{U}}_L$ satisfies $U_t(y) = \overline{U}_t(y)$ for all $y \in \mathcal{Y}$), and we prove convergence. Therefore, Lines 1, 3-7 approximate $\sum_i (\widehat{\eta}_t^i - \widehat{\eta}^{E,i}) \approx \sum_i (\eta^{p^i, r^i, \pi_t^{*,i}} - \eta^{p^i, r^i, \pi^{E,i}})$ for all $t$. In particular, Lines 4-6 approximate $\eta^{p^i, r^i, \pi_t^{*,i}}$ by passing through $\widehat{\eta}_t^i \approx \eta^{\widehat{p}^i, \overline{r}^i, \widehat{\pi}_t^{*,i}} \approx \eta^{p^i, r^i, \pi_t^{*,i}}$, where $\widehat{\pi}_t^{*,i}$ is the optimal policy for the RS-MDP $\widehat{\mathcal{M}}_{\overline{U}_t}^i := (\mathcal{S}^i, \mathcal{A}^i, H, s_0^i, \widehat{p}^i, \overline{r}^i, \overline{U}_t)$. At Line 4 we compute through *value iteration* (`PLANNING` subroutine, Algorithm 4) the optimal policy $\widehat{\psi}_t^{*,i}$ for MDP $\mathfrak{E}[\widehat{\mathcal{M}}_{\overline{U}_t}^i]$. Then, at Line 5, we collect the return of $K$ trajectories obtained by executing $\widehat{\psi}_t^{*,i}$ in MDP $\mathfrak{E}[\widehat{\mathcal{M}}_{\overline{U}_t}^i]$ (`ROLLOUT` subroutine, Algorithm 6), which is equivalent to playing $\widehat{\pi}_t^{*,i}$ in $\widehat{\mathcal{M}}_{\overline{U}_t}^i$. Finally, at Line 6, we use this data to compute the *empirical* estimate $\widehat{\eta}_t^i$. `TRACTOR-UL` enjoys the following guarantee:

**Theorem 5.2.** *Let* $\epsilon, \delta \in (0, 1), L > 0$, *and assume that* $U^E \in \underline{\mathfrak{U}}_L$. *If we execute* `TRACTOR-UL` *with parameters* $\epsilon_0 = \epsilon^2/(80N^2L^2H), T \geqslant \mathcal{O}(N^4H^4L^2/\epsilon^4), K \geqslant \widetilde{\mathcal{O}}(N^2H^2 \log \frac{NHL}{\delta\epsilon}/\epsilon^2), \alpha = \sqrt{\lceil H/\epsilon_0 \rceil - 1}H/(2N\sqrt{T})$, *an arbitrary* $\overline{U}_0 \in \underline{\overline{\mathfrak{U}}}_L$, *and if it holds that, for all* $i \in [\![N]\!]$:

$$\tau^{E,i} \geqslant \widetilde{\mathcal{O}}\Big(\frac{H^4N^4L^2}{\epsilon^4} \log \frac{NHL}{\delta\epsilon}\Big), \qquad \tau^i \geqslant \widetilde{\mathcal{O}}\Big(\frac{N^2SAH^5}{\epsilon^2}\Big(S + \log \frac{SAHN}{\delta}\Big)\Big),$$

*then, w.p. at least* $1 - \delta$, *for any* $\Delta \geqslant \epsilon$, `TRACTOR-UL` *guarantees that all the utilities* $U \in \underline{\mathfrak{U}}_L$ *such that* $U(y) = \widehat{U}(y)$ *for all* $y \in \mathcal{Y}$ *(where* $\widehat{U} \in \underline{\overline{\mathfrak{U}}}_L$ *is the output of* `TRACTOR-UL`*) belong to* $U \in \underline{\mathcal{U}}_\Delta$.

Intuitively, any $U \in \underline{\mathfrak{U}}_L$ obtained by "interpolating" $\widehat{U}$ has a small (non)compatibility w.h.p.. We consider *increasing* utilities $\underline{\mathfrak{U}}_L$ instead of *strictly-increasing* $\mathfrak{U}_L$ to guarantee the closedness of the set onto which we project. As for `CATY-UL`, normalizing $U(H) = 1$ would remove some $H$ terms from the bounds, and the Lipschitzianity assumption cannot be dropped. Finally, projection $\Pi_{\underline{\overline{\mathfrak{U}}}_L}$ can be implemented efficiently since set $\underline{\overline{\mathfrak{U}}}_L$ is made of $\mathcal{O}(H^2/\epsilon_0^2)$ linear constraints (Appendix E.1).

## 6 NUMERICAL SIMULATIONS

In this section, we provide proof-of-concept experiments using data collected from *lab members*.

**The Data.** We asked to 15 participants to describe the actions they would play in an MDP with horizon $H = 5$ (see Appendix F), at varying of the state, the stage, and the *cumulative reward* collected. The reward has a monetary interpretation. To answer the questions, the participants have been provided with complete information about the dynamics and the reward function of the MDP.[9]

**Experiment 1 - Model validation.** We aim to answer to: Is it worthy to increase the model complexity using a learnable utility in Eq. (1) instead of the (fixed) linear utility as (Ng & Russell, 2000)? How much better do we fit the data? To measure the fitness of a utility $U$ to the data (policy $\pi$) *fairly*, we consider a *relative* notion of (non)compatibility (we omit $p, r$ for simplicity): $\overline{\mathcal{C}}_\pi^r(U) := (J^*(U) - J^\pi(U))/J^*(U)$. Intuitively, $\overline{\mathcal{C}}_\pi^r(U)$ measures the *quality* of $\pi$ as perceived by the demonstrating agent, *if* $U$ *was its true utility function*. We execute `CATY-UL` (without exploration) for the 15 participants comparing the IRL risk-neutral utility $U_{\text{linear}}$ with 3 "baselines": A risk-averse $U_{\text{sqrt}}$ (concave) and a risk-lover $U_{\text{square}}$ (convex) utilities, and the utility $U_{\text{SG}}$ fitted through the SG method (see Appendix F for details). We report the (non)compatibilities in *percentage* below:

---

[9]We have been allowed to collect these data because they are not personal.

| | 1 | 2 | 3 | 4 | 5 | 6 | 7 | 8 | 9 | 10 | 11 | 12 | 13 | 14 | 15 | mean |
|---|---|---|---|---|---|---|---|---|---|---|---|---|---|---|---|---|
| $U_{\text{linear}}$ | 39 | 58 | 18 | 1 | 9 | 33 | 25 | 62 | 1 | 56 | 1 | 16 | 16 | 25 | 60 | $28 \pm 22$ |
| $U_{\text{sqrt}}$ | 16 | 28 | 8 | 1 | 3 | 16 | 11 | 30 | 1 | 25 | 1 | 6 | 8 | 11 | 28 | $13 \pm 10$ |
| $U_{\text{square}}$ | 70 | 86 | 32 | 1 | 19 | 41 | 44 | 91 | 1 | 88 | 1 | 35 | 28 | 44 | 91 | $45 \pm 32$ |
| $U_{\text{SG}}$ | 39 | 76 | 11 | 0 | 5 | 28 | 20 | 34 | 10 | 2 | 1 | 8 | 21 | 17 | 51 | $22 \pm 21$ |

Table 1: Values of $\overline{C}_{\pi}^{\text{r}}$ of various utilities with the demonstrations of the participants in percentage.

Some observations are in order. First, *this* data shows that replacing $U_{\text{linear}}$ (i.e., IRL) in Eq. (1) with $U_{\text{sqrt}}$ reduces $\overline{C}_{\pi}^{\text{r}}(\cdot)$ from 28% to 13% on the average of the participants, answering positively to our question. Next, the (fixed) $U_{\text{sqrt}}$ outperforms the $U_{\text{SG}}$ of *each* participant. This is due to both the bounded rationality of humans, who can *not* apply the $H = 1$ utility $U_{\text{SG}}$ to $H > 1$ problems, and the fact that $U_{\text{sqrt}}$ "overfits" the simple MDP considered, but it might generalize worse than $U_{\text{SG}}$ to new environments. Finally, all the utilities are compatible with policies 4 and 11, providing empirical evidence on the *partial identifiability* of the expert's utility from single demonstrations.

**Experiment 2 - Empirical analysis of `TRACTOR-UL`.** We aim to empirically characterise `TRACTOR-UL`. First, we execute it on the MDP described earlier with different values of step size $\alpha$ and initial utility $\overline{U}_0$ to compute a compatible utility for participant 10 (chosen arbitrarily).

As shown in the figure on the side, the optimal step size $\alpha = 100$ may be very large, due to (i) the presence of compatible utilities on the boundaries of $\overline{\mathfrak{U}}_L$,[10] thus larger step sizes can converge sooner, and to (ii) the projection onto $\overline{\mathfrak{U}}_L$ that results in minimal changes of utility even with very large steps (see Appendix F.4.2). These observations do not change if we consider other participants (Appendix F.1.4). Next, we run `TRACTOR-UL` on simulated data (see Appendix F.4.3). We consider MDPs generated at random with larger state-actions spaces (increment of $S, A$), and also multiple environments (increment of $N$). To comply with the

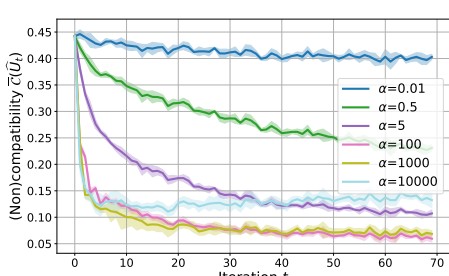

assumption that there exists a utility function for which the expert's policy is (almost) optimal, we compute, in each environment, the optimal policy for an S-shaped utility function that is convex for small returns, and concave for large returns, and then we inject some noise. The simulations show that the number of gradient iterations necessary to achieve a certain level of performance is affected by an increment of $N$, but not of $S, A$, as expected by Theorem 5.2. However, larger $S, A$ require more execution time, because of the value iteration subroutine. Moreover, we observe that the best step size when $N$ increases is smaller than $\alpha = 100$ found for the experiments with $N = 1$. Intuitively, there are less compatible utilities now, thus we need smaller gradient steps to find them.

# 7 CONCLUSION

In this paper, we proposed a novel descriptive model of behavior in MDPs, we formalized the UL problem as that of learning the risk attitude of an agent from demonstrations, and we characterised the partial identifiability of the expert's utility. In addition, we have described two provably efficient algorithms for estimating the compatibility of a utility with demonstrations, and for extracting a compatible utility. They have been empirically validated through two proof-of-concept experiments.

**Future directions.** This paper opens up many important questions. To quantify the model misspecification, to use function approximation, to conduct an empirical study on the horizon used by humans for planning (Carton et al., 2016), to combine demonstrations with other feedbacks (Jeon et al., 2020), to learn both $r$ and $U$, to extend imitation learning approaches (e.g., GAIL (Ho & Ermon, 2016)) or the maximum entropy framework with utilities, to improve the model in Eq. (1) with negative rewards and prospect theory (Kahneman & Tversky, 1979), and many others.

*We believe that most of the IRL literature shall be extended under the proposed, more expressive, framework to construct more accurate algorithms for IRL and UL.*

---

[10]$\overline{\mathfrak{U}}_L$ forces utilities to be *increasing*, i.e., with constraints $U(G_1) \leqslant U(G_2) \, \forall G_1 \leqslant G_2$. The plateau in Fig. 18 (right) indicates that $U(G_1) = U(G_2) \, \forall G_1 \leqslant G_2, G_1, G_2 \in [1, 3]$, thus, it represents a boundary.

ETHICS STATEMENT

For the numerical simulations conducted in this research, data were collected from human subjects. The data gathered are not *personal* and, as such, do not pose any privacy risks to the participants. Similar types of data, particularly regarding human risk attitudes, have been widely collected and analyzed in the field of behavioral economics throughout the last century. We are confident that both the data and the analyses conducted in this study do not raise any ethical concerns.

REPRODUCIBILITY STATEMENT

To ensure reproducibility, we have included the complete code used to conduct the experiments in the supplementary material. The folder contains a README.md file that outlines the code structure and provides step-by-step instructions on how to replicate the experiments. Further experimental details, including the specific values of the hyperparameters, are provided in Appendix F.

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

## A   ADDITIONAL RELATED WORKS

We describe here the most relevant related works. First, we describe IRL papers with risk, i.e., those works that consider MDPs, and try to learn either the reward function or the utility or both. Next, we analyze the works that aim to learning the risk attitude (i.e., a utility function) from demonstrations of behavior (potentially in problems other than MDPs). Finally, we present other connected works.

**Inverse Reinforcement Learning with risk.**   Majumdar et al. (2017) introduce the risk-sensitive IRL problem in decision problems different from MDPs. Authors analyze two settings, one in which the expert takes a single decision, and one in which there are multiple decisions in sequence. They model the expert as a risk-aware decision-making agent acting according to a *coherent risk metric* (Artzner et al., 1999), and they consider both the case in which the reward function is known, and they try to learn the risk attitude (coherent risk metric) of the expert, and the case in which the reward is unknown, and they aim to estimate both the risk attitude and the reward function. Nevertheless, the authors analyze a very simple model of environment, that they call *prepare-react model*, which is much different from an MDP, since, simply put, it is equivalent to a deterministic MDP in which the stochasticity is shared by all the state-action pairs at each stage $h \in [\![H]\!]$. Moreover, the optimal policy is markovian in this setting.

Singh et al. (2018) generalizes the work of Majumdar et al. (2017). Specifically, the biggest improvement is to consider nested optimization stages. However, the model of the environment is still much simple, and, in addition, the authors consider a maximum likelihood approach to facilitate inference.

We mention also the work of Chen et al. (2019) who extend Majumdar et al. (2017) by devising an active learning framework to improve the efficiency of their learning algorithms.

Another important work is that of Ratliff & Mazumdar (2020), who study the risk-sensitive IRL problem in MDPs, by proposing an interesting parametric model of behavior for the expert based on prospect theory Kahneman & Tversky (1979), and they devise a gradient-based inverse reinforcement learning algorithm that minimizes a loss function defined on the observed behavior. However, this work suffers from the major limitation of assuming that the expert plays actions *exactly* based on a softmax distribution, which introduces enough structure to perform maximum likelihood and to learn the parameters of the utility function. Such assumption is rather strong.

We shall mention also the recent pre-print of Cao et al. (2024) that proposes a novel stochastic control framework in continuous time that includes two utility functions and a generic discounting scheme under a time-varying rate. Assuming to know both the utilities and the discounting scheme, the authors show that, through state augmentation, the control problem is well-posed. In addition, the authors provide sufficient conditions for the identification of both the utilities and the discounting scheme given demonstrations of behavior. It should be remarked that there are many differences between this work and ours. First, they consider a continuous time environment that is rather different from an MDP. Next, when they consider MDPs to make things more concrete, they assume a utility function on the reward instead of the *return*, and they also consider the entropy-regularized setting in which the optimal policy is the Boltzmann policy, which permits to apply maximum likelihood for inferring the parameters of the utility function and the discount factor (they assume exponential discounting).

**Learning utilities from demonstrations.**   Chajewska et al. (2001) considers an approach similar to IRL Ng & Russell (2000). Their goal is not to perform *active* preference elicitation, but, similarly to us, to use demonstrations to infer preferences. Specifically, they aim to learn utilities in sequential decision-making problems from demonstrations. However, they model the problems through decision trees, which are different from MDPs, and this represents the main difference between their work and ours. Indeed, decision trees are simpler since there is no notion of reward function at intermediate states. In this manner, they are able to devise (backward induction) algorithms to learn utilities in decision trees through linear constraints similar to those devised by Ng & Russell (2000) in IRL. It is interesting to notice that they adopt a Bayesian approach to extract a single utility from the feasible set constructed, and not an heuristic like that of Ng & Russell (2000). They assume a prior $p(u)$ over the true utility function $u$, and approximate the posterior w.r.t. the feasible set of utilities $\mathcal{U}$ using Markov Chain Monte Carlo (MCMC).

Lei (2020) considers the problem of learning utilities from demonstrations similarly to Chajewska et al. (2001), but with the difference of considering *influence diagrams* instead of decision trees. Since any influence diagram can be expanded into a decision tree, authors adopt a strategy similar to Chajewska et al. (2001).

Shukla et al. (2017) faces the problem of learning human utilities from (video) demonstrations, with the aim of generating meaningful tasks based on the learned utilities. However, differently from us, they consider the stochastic context-free And-Or graph (STC-AOG) framework (Xiong et al., 2016), instead of MDPs.

**Others.** Shah et al. (2019) is similar to our work in that it aims to learn the behavioral model of the expert from demonstrations. However, they do not consider a specific model like us (i.e., Eq. (1)), but use a differentiable planner (neural network) to learn the planner. However, their approach requires a lot of demonstrations, even across multiple MDPs, and it does not consider the fact that there exist interesting models of humans in behavioral economics.

## B  ADDITIONAL NOTATION

In this appendix, we introduce additional notation that will be used in other appendices.

**Miscellaneous.** For any probability distribution $\nu \in \Delta^{\mathbb{R}}$, we denote its cumulative density function by $F_\nu$. Let $\nu \in \Delta^{\mathbb{R}}$ be a probability distribution on $\mathbb{R}$; then, for any $y \in [0,1]$, we define the *generalized inverse* $F_\nu^{-1}(y)$ as:

$$F_\nu^{-1}(y) := \inf_{x \in \mathbb{R}} \{F_\nu(x) \geqslant y\}.$$

We define the *1-Wasserstein distance* $w_1 : \Delta^{\mathbb{R}} \times \Delta^{\mathbb{R}} \to [0, \infty]$ between two probability distributions $\nu, \mu$ as:

$$w_1(\nu, \mu) := \int_0^1 \left| F_\nu^{-1}(y) - F_\mu^{-1}(y) \right| dy. \tag{2}$$

In addition, we define the *Cramér distance* $\ell_2 : \Delta^{\mathbb{R}} \times \Delta^{\mathbb{R}} \to [0, \infty]$ between two probability distributions $\nu, \mu$ as:

$$\ell_2(\nu, \mu) := \left( \int_{\mathbb{R}} (F_\nu(y) - F_\mu(y))^2 dy \right)^{1/2}. \tag{3}$$

We will use notation:

$$\mathbb{V}_{X \sim Q}[X] := \mathbb{E}_{X \sim Q}[(X - \mathbb{E}_{X \sim Q}[X])^2],$$

to denote the variance of a random variable $X \sim Q$ distributed as $Q$. Given two random variables $X \sim Q_1, Y \sim Q_2$, we denote their covariance as:

$$\text{Cov}_{X \sim Q_1, Y \sim Q_2}[X, Y] := \mathbb{E}_{X \sim Q_1, Y \sim Q_2}[(X - \mathbb{E}_{X \sim Q_1}[X])(Y - \mathbb{E}_{Y \sim Q_2}[Y])].$$

We define the *categorical projection operator* $\text{Proj}_{\mathcal{C}}$ (mentioned in Section 5), that projects onto set $\mathcal{Y} = \{y_1, y_2, \ldots, y_d\}$ (the items of $\mathcal{Y}$ are ordered: $y_1 \leqslant y_2 \leqslant \ldots \leqslant y_d$), based on Rowland et al. (2018). For single Dirac measures on an arbitrary $y \in \mathbb{R}$, we write:

$$\text{Proj}_{\mathcal{C}}(\delta_y) := \begin{cases} \delta_{y_1} & \text{if } y \leqslant y_1 \\ \frac{y_{i+1} - y}{y_{i+1} - y_i} \delta_{y_i} + \frac{y - y_i}{y_{i+1} - y_i} \delta_{y_{i+1}} & \text{if } y_i < y \leqslant y_{i+1} \ , \\ \delta_{y_d} & \text{if } y > y_d \end{cases} \tag{4}$$

and we extend it affinely to finite mixtures of $M$ Dirac distributions, so that:

$$\text{Proj}_{\mathcal{C}}\left( \sum_{j \in [\![M]\!]} q_j \delta_{z_j} \right) = \sum_{j \in [\![M]\!]} q_j \text{Proj}_{\mathcal{C}}(\delta_{z_j}), \tag{5}$$

for some set of real values $\{z_j\}_{j \in [\![M]\!]}$ and weights $\{q_j\}_{j \in [\![M]\!]}$.

**Value functions.** Given an MDP $\mathcal{M} = (\mathcal{S}, \mathcal{A}, H, s_0, p, r)$ and a policy $\pi$, we define the $V$- and $Q$-functions of policy $\pi$ in MDP $\mathcal{M}$ at every $(s, a, h) \in \mathcal{S} \times \mathcal{A} \times [\![H]\!]$ respectively as $V_h^\pi(s; p, r) := \mathbb{E}_{p,r,\pi}[\sum_{t=h}^H r_t(s_t, a_t)|s_h = s]$ and $Q_h^\pi(s, a; p, r) := \mathbb{E}_{p,r,\pi}[\sum_{t=h}^H r_t(s_t, a_t)|s_h = s, a_h = a]$. We define the optimal $V$- and $Q$-functions as $V_h^*(s; p, r) := \sup_\pi V_h^\pi(s; p, r)$ and $Q_h^*(s, a; p, r) := \sup_\pi Q_h^\pi(s, a; p, r)$.

For MDPs with an enlarged state space, e.g., $(\{\mathcal{S} \times \mathcal{Y}_h\}_h, \mathcal{A}, H, (s_0, 0), \mathfrak{p}, \mathfrak{r})$, and a policy $\psi = \{\psi_h\}_h$, for all $h \in [\![H]\!]$ and $(s, y, a) \in \mathcal{S} \times \mathcal{Y}_h \times \mathcal{A}$ we denote the $V$- and $Q$-functions respectively as $V_h^\psi(s, y; \mathfrak{p}, \mathfrak{r}) := \mathbb{E}_{\mathfrak{p},\mathfrak{r},\psi}[\sum_{t=h}^H \mathfrak{r}_t(s_t, y_t, a_t)|s_h = s, y_h = y]$ and $Q_h^\psi(s, y, a; \mathfrak{p}, \mathfrak{r}) := \mathbb{E}_{\mathfrak{p},\mathfrak{r},\psi}[\sum_{t=h}^H \mathfrak{r}_t(s_t, y_t, a_t)|s_h = s, y_h = y, a_h = a]$. We denote the optimal $V$- and $Q$-functions as $V_h^*(s, y; \mathfrak{p}, \mathfrak{r}) := \sup_\psi V_h^\psi(s, y; \mathfrak{p}, \mathfrak{r})$ and $Q_h^*(s, y, a; \mathfrak{p}, \mathfrak{r}) := \sup_\psi Q_h^\psi(s, y, a; \mathfrak{p}, \mathfrak{r})$.

Observe that the notation just introduced will be extended in a straightforward manner to MDPs (MDPs with enlarged state space) that have an estimated transition model $\widehat{p}$ ($\widehat{\mathfrak{p}}$), and/or a discretized reward function $\overline{r}$ ($\overline{\mathfrak{r}}$).

# C  ADDITIONAL RESULTS AND PROOFS FOR SECTION 3

In Appendix C.1 we present in more detail the MDP used in Example 3.1. In Appendix C.2, we present an additional motivating example explaining why including the reward into the state in Example 3.1 is not satisfactory, while in Appendix C.3 we provide the missing proofs for Section 3.

## C.1  THE MDP OF EXAMPLE 3.1

The MDP used in Example 3.1. We remark that the reward function is deterministic and is a function of the state-action space only.

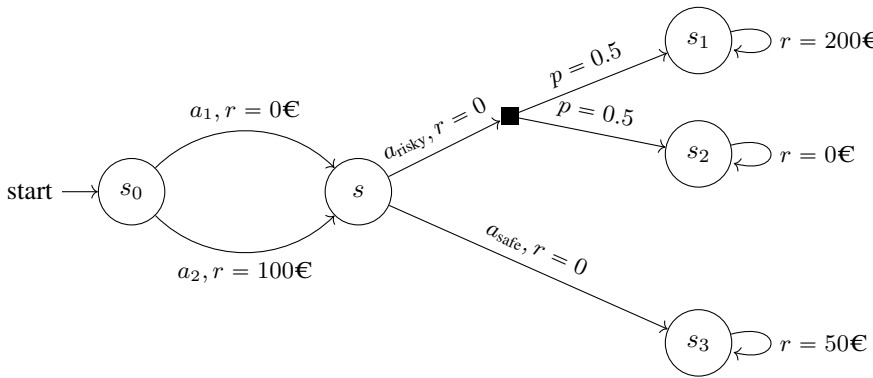

## C.2  DRAWBACKS OF RE-MODELLING THE MDP

If we re-model the MDP in Example 3.1 by including the reward into the state to make the optimal policy Markovian, then we might incur in *interpretability* and *transferability* issues. To better explain this, we make a simple example.

Consider a driving setting, where the state is the location of the car (name of the road and position inside the road), the actions permit to change the current road (only when the car is close to another road, otherwise no effect), and at every stage/timestep the position of the car advances on the current road depending on the amount of traffic in the road, which is random and modelled through the transition model of the environment.

Consider now an expert agent that *aims to reach a certain goal location $s_g$ in the minimum time/number of stages possible*, and that is *risk-averse*, in the sense that it prefers roads that always have little traffic, even though they are, on average, slower, to roads that are usually faster but sometimes have peaks of traffic that make them very very slow (since the traffic is random, there is no sequence of roads that is *always* better than others, but it is a matter of chance).

In **our model** (Eq. (1)), we can represent the expert through the reward function $r^E$ that is 0 in the goal location $s_g$, and $-1$ otherwise. In this manner, the faster a trajectory reaches $s_g$, the larger the cumulative reward. Next, we can choose the utility function $U^E$ to be some concave function in order to achieve the risk-aversion property (Bäuerle & Rieder, 2014), i.e., to make sure that the expert prefers, intuitively, roads with "smaller variance" of traffic. We remark that *our model permits to capture the preferences of the expert in a very simple yet expressive manner*. In fact, as shown in this example, $r^E$ and $U^E$ can be easily designed, and their meaning is easily interpretable. However, this does not hold if we include past rewards in the state.

In the **model with an extended state space**, the behavior of the expert is represented through a single reward $\bar{r}^E$ (defined on the expanded state space) instead of the pair reward-utility $r^E, U^E$. The intuition is that, to contain all the information present in $r^E, U^E$, the new reward $\bar{r}^E$ will be "messy". As such, designing it is also more complex. For instance, choosing $\bar{r}^E$ to be $-t$ in the (expanded) state made of the goal location $s_g$ and of $t$ timesteps, and 0 elsewhere, represents a risk-neutral agent that aims to reach $s_g$ as soon as possible, but it does not capture the risk-aversion of the expert. To make $\bar{r}^E$ model the risk-aversion, we must take it to be a concave function of $-t$, making it *more difficult to be interpreted*.

Even though the model with past rewards in the state guarantees the Markovianity of the optimal/expert policy, it suffers from major **drawbacks**:

- $\bar{r}^E$ has a size (i.e., it is defined on a number of states) that grows exponentially in the horizon in the worst case, while $r^E, U^E$ do not.
- $\bar{r}^E$ is more difficult to *interpret* (and design) than the pair $r^E, U^E$, whose meaning is immediate.
- $\bar{r}^E$ can only be transferred to problems with the same state-action (or feature) space. Instead, the utility $U^E$ can be easily *transferred* to other kinds of environments. E.g., in the considered example, $U^E$ can be used to assess how much the expert "values its time" and takes decisions based on it. Thus, we can predict the behavior of the expert in other problem settings where the time plays a role using $U^E$, even if the state-action (or feature) space is different (e.g., if the expert travels by train instead than by car, we can predict if it prefers taking a reliable train, or a faster train on average that sometimes makes huge delays).

C.3   PROOFS FOR SECTION 3

**Proposition 3.2.** *Given any RS-MDP with deterministic transition model $p$ and reward function $r$, if the utility $U$ is increasing, then, there exists a Markovian optimal policy.*

*Proof.* The objective in Eq. (1) coincides with that of a common MDP in absence of stochasticity and when $U$ is increasing. Since there always exists an optimal Markovian policy in MDPs, thus we obtain the result. □

**Proposition 3.1.** *There exists a RS-MDP with horizon $H = 4$ in which the difference between the optimal performance and the performance of the best Markovian policy is $0.5$.*

*Proof.* For reasons that will be clear later, let us define symbol $x \approx 2.6$ as the solution of $x - \frac{x^2}{3.99} - 0.1 = 1$.

Consider the RS-MDP $\mathcal{M}_U = (\mathcal{S}, \mathcal{A}, H, s_0, p, r, U)$ in Figure 2, where $\mathcal{S} = \{s_{\text{init}}, s_1, s_2, s_3, s_4, s_5, s_6\}$, $\mathcal{A} = \{a_1, a_2\}$, $H = 4$, $s_0 = s_{\text{init}}$, transition model $p$ such that:

$$p_1(s_1|s_{\text{init}}, a) = p_1(s_2|s_{\text{init}}, a) = 1/2 \quad \forall a \in \mathcal{A},$$
$$p_2(s_3|s_1, a) = p_2(s_3|s_2, a) = 1 \quad \forall a \in \mathcal{A},$$
$$p_3(s_4|s_3, a_1) = x/3.99, p_3(s_5|s_3, a_1) = 1 - x/3.99, p_3(s_6|s_3, a_2) = 1,$$

reward function $r$ defined as:

$$r_1(s_{\text{init}}, a) = 0 \quad \forall a \in \mathcal{A},$$
$$r_2(s_1, a) = 1 \quad \forall a \in \mathcal{A},$$

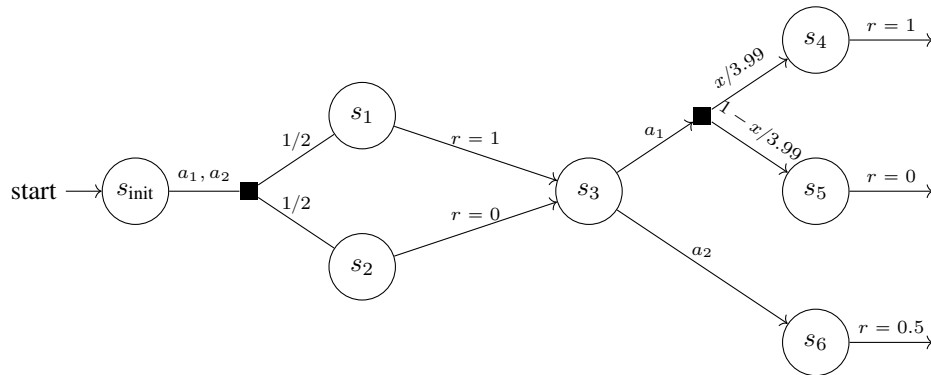

Figure 2: MDP for the proof of Proposition 3.1.

$$r_2(s_2, a) = 0 \quad \forall a \in \mathcal{A},$$
$$r_3(s_3, a) = 0 \quad \forall a \in \mathcal{A},$$
$$r_4(s_4, a) = 1 \quad \forall a \in \mathcal{A},$$
$$r_4(s_5, a) = 0 \quad \forall a \in \mathcal{A},$$
$$r_4(s_6, a) = 0.5 \quad \forall a \in \mathcal{A},$$

and utility function $U \in \mathfrak{U}$ that satisfies:

$$U(y) = \begin{cases} x - 0.1 & \text{if } y = 0.5 \\ x & \text{if } y = 1 \\ x + 0.1 & \text{if } y = 1.5 \\ 3.99 & \text{if } y = 2 \end{cases}.$$

Note that this entails that:

$$\frac{x}{3.99} U(2) + U(1) = U(0.5) + U(1.5). \tag{6}$$

Note also that the support of the return function of this (RS-)MDP is $\mathcal{G}^{p,r} = \{0, 0.5, 1, 1.5, 2\}$.

For $\alpha \in [0, 1]$, let $\pi^\alpha$ be the generic Markovian policy that plays action $a_1$ in $s_3$ w.p. $\alpha$ (the actions played in other states are not relevant). Then, its expected utility is:

$$\begin{aligned} J^{\pi^\alpha}(U; p, r) &= \frac{1}{2}\Big[\alpha\Big(\frac{x}{3.99} U(2) + (1 - \frac{x}{3.99})U(1)\Big) + (1 - \alpha)U(1.5)\Big] \\ &\quad + \frac{1}{2}\Big[\alpha\Big(\frac{x}{3.99} U(1) + (1 - \frac{x}{3.99})U(0)\Big) + (1 - \alpha)U(0.5)\Big] \\ &\stackrel{(1)}{=} \frac{1}{2}\Big[\alpha\Big(\frac{x}{3.99} U(2) + U(1)\Big) + (1 - \alpha)(U(1.5) + U(0.5))\Big] \\ &\stackrel{(2)}{=} \frac{U(1.5) + U(0.5)}{2}, \end{aligned}$$

where at (1) we have used that $U(0) = 0$, and at (2) we have used Eq. (6).

Thus, all Markovian policies $\pi^\alpha$ have the same performance. Let us consider the non-Markovian policy $\bar{\pi}$ that, in state $s_3$, plays action $a_1$ w.p. 1 if $s_3$ is reached with cumulative reward 1, and it plays action $a_2$ w.p. 1 if $s_3$ is reached with cumulative reward 0. Then, its performance is:

$$J^{\bar{\pi}}(U; p, r) = \frac{1}{2}\Big(\frac{x}{3.99} U(2) + (1 - \frac{x}{3.99})U(1)\Big) + \frac{1}{2}U(0.5).$$

The difference in performance between the optimal performance and that of $\pi^\alpha$ is:

$$\begin{aligned} J^*(U; p, r) - J^{\pi^\alpha}(U; p, r) &\geqslant J^{\bar{\pi}}(U; p, r) - J^{\pi^\alpha}(U; p, r) \\ &= \frac{1}{2}\Big(\frac{x}{3.99} U(2) + (1 - \frac{x}{3.99})U(1)\Big) + \frac{1}{2}U(0.5) - \frac{U(1.5) + U(0.5)}{2} \end{aligned}$$

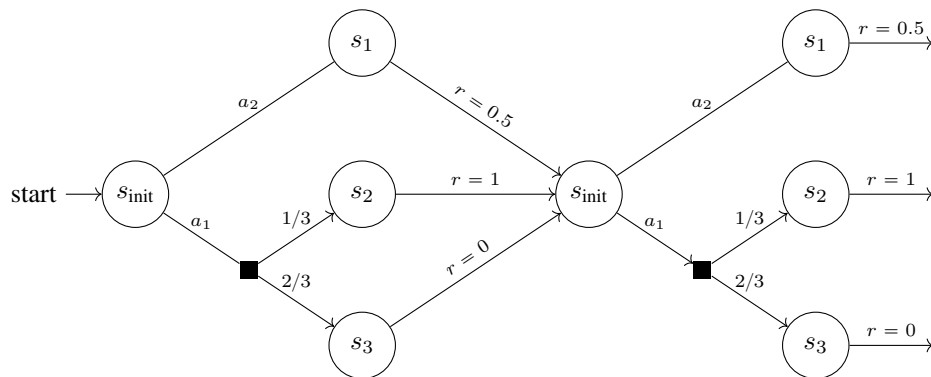

Figure 3: MDP for the proof of Proposition 3.3.

$$= \frac{1}{2}\Big(\frac{x}{3.99}U(2) + (1 - \frac{x}{3.99})U(1) - U(1.5)\Big)$$

$$\overset{(3)}{=} \frac{1}{2}\Big(x + x - \frac{x^2}{3.99} - x - 0.1\Big)$$

$$= \frac{1}{2}\Big(x - \frac{x^2}{3.99} - 0.1\Big)$$

$$\overset{(4)}{=} 0.5,$$

where at (3) we have replaced the values of utility, and at (4) we have used the definition of $x$.

$\square$

**Proposition 3.3.** *There exists a RS-MDP with stationary transition model and reward in which the* **best** *Markovian policy is non-stationary, and the best* stationary Markovian *policy is stochastic.*

*Proof.* Consider the stationary RS-MDP $\mathcal{M}_U = (\mathcal{S}, \mathcal{A}, H, s_0, p, r, U)$ depicted in Figure 3, where $\mathcal{S} = \{s_{\text{init}}, s_1, s_2, s_3\}$, $\mathcal{A} = \{a_1, a_2\}$, $H = 4$, $s_0 = s_{\text{init}}$, stationary transition model $p$ (we omit subscript because of stationarity) such that:

$$p(s_2|s_{\text{init}}, a_1) = 1 - p(s_3|s_{\text{init}}, a_1) = 1/3,$$
$$p(s_1|s_{\text{init}}, a_2) = 1,$$
$$p(s_{\text{init}}|s, a) = 1 \quad \forall s \in \{s_1, s_2, s_3\}, \forall a \in \mathcal{A},$$

reward function $r$ defined as:

$$r(s_{\text{init}}, a) = 0 \quad \forall a \in \mathcal{A},$$
$$r(s_1, a) = 0.5 \quad \forall a \in \mathcal{A},$$
$$r(s_2, a) = 1 \quad \forall a \in \mathcal{A},$$
$$r(s_3, a) = 0 \quad \forall a \in \mathcal{A},$$

and utility function $U \in \mathfrak{U}$ that satisfies:

$$U(y) = \begin{cases} 0.15 & \text{if } y = 0.5 \\ 0.2 & \text{if } y = 1 \\ 1.8 & \text{if } y = 1.5 \\ 2 & \text{if } y = 2 \end{cases}.$$

Let $\pi^{\alpha,\beta}$ denote the general non-stationary policy that plays action $a_1$ at stage 1 w.p. $\alpha \in [0, 1]$, and plays action $a_1$ at stage 2 w.p. $\beta \in [0, 1]$. The performance of policy $\pi^{\alpha,\beta}$ can be written as:

$$J^{\pi^{\alpha,\beta}}(U; p, r) = \alpha\Big\{\frac{1}{3}\Big[\beta\Big(\frac{1}{3}U(2) + \frac{2}{3}U(1)\Big) + (1 - \beta)U(1.5)\Big] + \frac{2}{3}\Big[\beta\frac{1}{3}U(1) + (1 - \beta)U(0.5)\Big]\Big\}$$

$$+ (1 - \alpha)\Big[\beta\Big(\frac{1}{3}U(1.5) + \frac{2}{3}U(0.5)\Big) + (1 - \beta)U(1)\Big]$$

$$= \alpha\beta\Big[\frac{1}{9}U(2) + \frac{13}{9}U(1) - \frac{2}{3}U(1.5) - \frac{4}{3}U(0.5)\Big]$$

$$+ (\alpha + \beta)\Big[\frac{1}{3}U(1.5) + \frac{2}{3}U(0.5) - U(1)\Big] + U(1)$$

$$= \alpha\beta\Big[\frac{2}{9} + \frac{13}{45} - \frac{18}{15} - \frac{1}{5}\Big] + (\alpha + \beta)\Big[\frac{1}{5} + \frac{1}{10} - \frac{1}{5}\Big] + \frac{1}{5}$$

$$= -\frac{8}{9}\alpha\beta + \frac{1}{10}(\alpha + \beta) + \frac{1}{5}.$$

To show that the best Markovian policy is non-stationary in this example, we show that the performance of non-stationary policy $\pi^{0,1}$ is better than the performance of all possible Markovian policies. The performance of $\pi^{0,1}$ is:

$$J^{\pi^{0,1}}(U; p, r) = \frac{1}{10} + \frac{1}{5} = 0.3.$$

Instead, the generic stationary policy is $\pi^{\alpha,\alpha}$, and has performance:

$$J^{\pi^{\alpha,\alpha}}(U; p, r) = -\frac{8}{9}\alpha^2 + \frac{1}{5}\alpha + \frac{1}{5}.$$

The value of $\alpha \in [0, 1]$ that maximizes this objective is:

$$\frac{d}{d\alpha}J^{\pi^{\alpha,\alpha}}(U; p, r) = -\frac{16}{9}\alpha + \frac{1}{5} = 0 \iff \alpha = \frac{9}{80},$$

from which we get:

$$J^{\pi^{9/80,9/80}}(U; p, r) = \frac{169}{800} \leqslant 0.22,$$

which is smaller than $0.3 = J^{\pi^{0,1}}(U; p, r)$. This concludes the proof of the first part of the proposition.

For the second part, simply observe that, in the problem instance considered, we just obtained that the best Markovian stationary policy plays action $a_1$ w.p. $9/80$, i.e., it is stochastic. $\qquad\square$

## D  ADDITIONAL RESULTS AND PROOFS FOR SECTION 4

In this appendix, we provide a more explicit formulation for the feasible utility set (Appendix D.1), and then we provide the proofs of all the results presented in Section 4 (Appendix D.2).

### D.1  A MORE EXPLICIT FORMULATION FOR THE FEASIBLE UTILITY SET

For any policy $\pi$, we denote by $\mathcal{S}^{p,r,\pi}$ the set of all $(s, h, y)$ state-stage-cumulative reward triples which are covered with non-zero probability by policy $\pi$ in the considered (RS-)MDP.

Thanks to this definition, we can rewrite the feasible set as follows:

**Proposition D.1.** *Let $\mathcal{M} = (\mathcal{S}, \mathcal{A}, H, s_0, p, r)$ be an MDP, and let $\pi^E$ be the expert policy. Then, the feasible utility set $\mathcal{U}_{p,r,\pi^E}$ contains all and only the utility functions that make the actions played by the expert policy optimal at all the $(s, h, y) \in \mathcal{S}^{p,r,\pi^E}$. Formally:*

$$\mathcal{U}_{p,r,\pi^E} = \Big\{U \in \mathfrak{U} \,\Big|\, \forall (s, h, y) \in \mathcal{S}^{p,r,\pi^E}, \forall a \in \mathcal{A} :$$

$$Q_h^*(s, y, \pi_h^E(s, y); p, r) \geqslant Q^*(s, y, a; p, r),$$

*where we used the notation introduced in Appendix B.*

*Proof.* Based on Theorem 3.1 of Bäuerle & Rieder (2014) (or Theorem 1 of Wu & Xu (2023)), we have that a utility $U \in \mathfrak{U}$ belongs to the feasible set if it makes the expert policy optimal even in the enlarged state space MDP (note that it is possible to define a policy $\psi$ for the enlarged MDP because we are considering policies $\pi$ whose non-Markovianity lies only in the cumulative reward up to now). Therefore, the result follows thanks to a proof analogous to that of Lemma E.1 in Lazzati et al. (2024b), since we are simply considering a common MDP with two variables per state. $\qquad\square$

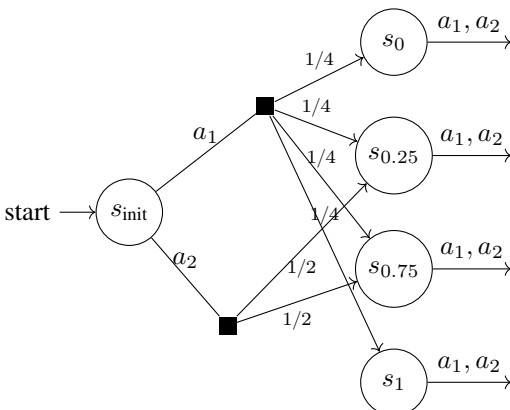

Figure 4: MDP for the proof of Proposition 4.1.

## D.2 PROOFS FOR SECTION 4

**Proposition 4.1** (Transfer to a new transition model). *There exist two MDPs $\mathcal{M} = (\mathcal{S}, \mathcal{A}, H, s_0, p, r), \mathcal{M}' = (\mathcal{S}, \mathcal{A}, H, s_0, p', r)$, with $p \neq p'$, for which there exists a policy $\pi^E$ and a pair of utilities $U_1, U_2 \in \mathfrak{U}$ such that: $U_1, U_2 \in \mathcal{U}_{p,r,\pi^E}$ and $\Pi^*_{p',r}(U_1) \cap \Pi^*_{p',r}(U_2) = \{\}$.*

*Proof.* We will prove the guarantee stated in the proposition using two different pairs of MDPs: One that that satisfies $\mathcal{G}^{p',r} = \mathcal{G}^{p,r}$, i.e., for which the support of the return function coincides, and the other that does not. Let us begin with the former.

Consider a simple MDP $\mathcal{M} = (\mathcal{S}, \mathcal{A}, H, s_{\text{init}}, p, r)$ with five states $\mathcal{S} = \{s_{\text{init}}, s_0, s_{0.25}, s_{0.75}, s_1\}$, two actions $\mathcal{A} = \{a_1, a_2\}$, horizon $H = 2$, initial state $s_{\text{init}}$, transition model $p$ such that:

$$p_1(s'|s_{\text{init}}, a_1) = \begin{cases} 1/4 & \text{if } s' = s_0 \\ 1/4 & \text{if } s' = s_{0.25} \\ 1/4 & \text{if } s' = s_{0.75} \\ 1/4 & \text{if } s' = s_1 \end{cases},$$

$$p_1(s'|s_{\text{init}}, a_2) = \begin{cases} 1/2 & \text{if } s' = s_{0.25} \\ 1/2 & \text{if } s' = s_{0.75} \end{cases},$$

and reward function $r$ that assigns $r_1(s_{\text{init}}, a_1) = r_1(s_{\text{init}}, a_2) = 0$, and:

$$r_2(s, a) = \begin{cases} 0 & \text{if } s = s_0 \wedge (a = a_1 \vee a = a_2) \\ 0.25 & \text{if } s = s_{0.25} \wedge (a = a_1 \vee a = a_2) \\ 0.75 & \text{if } s = s_{0.75} \wedge (a = a_1 \vee a = a_2) \\ 1 & \text{if } s = s_1 \wedge (a = a_1 \vee a = a_2) \end{cases}.$$

Note that the support of the return function is $\mathcal{G}^{p,r} = \{0, 0.25, 0.75, 1\}$. We are given an expert's policy $\pi^E$ that prescribes action $a_1$ at stage 1 in state $s_{\text{init}}$, and arbitrary actions in other states (the specific action is not relevant). The MDP $\mathcal{M}$ is represented in Figure 4.

Now, we show that utilities $U_1, U_2 \in \mathfrak{U}$, defined in points of the support $\mathcal{G}^{p,r}$ as (and connected in arbitrary continuous strictly-increasing manner between these points):

$$U_1(G) = \begin{cases} 0 & \text{if } G = 0 \\ 0.01 & \text{if } G = 0.25 \\ 0.02 & \text{if } G = 0.75 \\ 1.99 & \text{if } G = 1 \end{cases}, \quad U_2(G) = \begin{cases} 0 & \text{if } G = 0 \\ 0.01 & \text{if } G = 0.25 \\ 0.99 & \text{if } G = 0.75 \\ 1.99 & \text{if } G = 1 \end{cases},$$

belong to the feasible set $\mathcal{U}_{p,r,\pi^E}$, and, when transferred to the new MDP $\mathcal{M}' = (\mathcal{S}, \mathcal{A}, H, s_{\text{init}}, p', r)$, with transition model $p' \neq p$ defined as:

$$p'_1(\cdot|s_{\text{init}}, a_1) = p_1(\cdot|s_{\text{init}}, a_1),$$

$$p'_1(s'|s_{\text{init}}, a_2) = \begin{cases} 0.7 & \text{if } s' = s_0 \\ 0.3 & \text{if } s' = s_1 \end{cases},$$

impose different optimal policies, i.e., utility $U_2$ keeps making action $a_1$ optimal from state $s_{\text{init}}$ even in $\mathcal{M}'$, while $U_1$ makes action $a_2$ optimal. This proves the thesis of the proposition.

Let us begin by showing that $U_1, U_2 \in \mathcal{U}_{p,r,\pi^E}$ belong to the feasible set of $\mathcal{M}$ with policy $\pi^E$. Let $\overline{\pi}$ be the policy that plays action $a_2$ in state $s_{\text{init}}$. Then, the distribution of returns induced by policies $\pi^E$ and $\overline{\pi}$ are (we represent values only at points in $\mathcal{G}^{p,r} = \{0, 0.25, 0.75, 1\}$):

$$\eta^{p,r,\pi^E} = [1/4, 1/4, 1/4, 1/4]^\intercal$$
$$\eta^{p,r,\overline{\pi}} = [0, 1/2, 1/2, 0]^\intercal.$$

Thus, policy $\pi^E$ is optimal under some utility $U$ if and only if the values assigned by $U$ to points in $\mathcal{G}^{p,r} = \{0, 0.25, 0.75, 1\}$ (denoted, respectively, by $U^1, U^2, U^3, U^4$) satisfy:

$$U^\intercal(\eta^{p,r,\pi^E} - \eta^{p,r,\overline{\pi}}) = [1/4, -1/4, -1/4, 1/4]U = U^1 - U^2 - U^3 + U^4 \geqslant 0,$$

where we have overloaded the notation and denoted with $U := [U^1, U^2, U^3, U^4]^\intercal$ both the utility and the vector of values assigned to points in $\mathcal{G}^{p,r}$. By imposing normalization constraints ($U(0) = 0, U(2) = 2$), we get $U^1 = 0$, and by imposing also the monotonicity constraints, we get that utility $U$ is in the feasible set $\mathcal{U}_{p,r,\pi^E}$ if and only if:

$$\begin{cases} U^4 \geqslant U^2 + U^3 \\ 0 < U^2 < U^3 < U^4 < 2 \end{cases}.$$

Clearly, both utilities $U_1, U_2$ satisfy these constraints, thus they belong to the feasible set $\mathcal{U}_{p,r,\pi^E}$. Now, concerning problem $\mathcal{M}'$, the performances of $\pi^E, \overline{\pi}$ w.r.t. utilities $U_1, U_2$ are:

$$J^{\pi^E}(U_1; p', r) = \frac{1}{4}U_1(0) + \frac{1}{4}U_1(0.25) + \frac{1}{4}U_1(0.75) + \frac{1}{4}U_1(1) = 2.02/4 = 0.505,$$
$$J^{\overline{\pi}}(U_1; p', r) = 0.7U_1(0) + 0.3U_1(1) = 0.3 \times 1.99 = 0.597,$$
$$J^{\pi^E}(U_2; p', r) = \frac{1}{4}U_1(0) + \frac{1}{4}U_1(0.25) + \frac{1}{4}U_1(0.75) + \frac{1}{4}U_1(1) = 2.99/4 = 0.7475,$$
$$J^{\overline{\pi}}(U_2; p', r) = 0.7U_1(0) + 0.3U_1(1) = 0.3 \times 1.99 = 0.597.$$

Clearly, $J^{\pi^E}(U_1; p', r) < J^{\overline{\pi}}(U_1; p', r)$, but $J^{\pi^E}(U_2; p', r) > J^{\overline{\pi}}(U_2; p', r)$, thus we conclude that the set of policies induced by utilities $U_1, U_2$ in $\mathcal{M}'$ do not intersect, since they start from $s_{\text{init}}$ with different actions $\Pi^*_{p',r}(U_1) \cap \Pi^*_{p',r}(U_2) = \{\}$. This concludes the proof with an example that satisfies $\mathcal{G}^{p',r} = \mathcal{G}^{p,r}$.

If we want an example that does *not* satisfy $\mathcal{G}^{p',r} = \mathcal{G}^{p,r}$, then we can consider exactly the same example with $\mathcal{M}$ and $\mathcal{M}'$, but using $r_1(s_{\text{init}}, a_2) = 0.001$. In this manner, we see that $\mathcal{G}^{p,r} = \{0, 0.25, 0.251, 0.75, 0.751, 1\}$, and $\mathcal{G}^{p',r} = \{0, 0.001, 0.25, 0.75, 1, 1.001\}$, which are different. By choosing $U'_1, U'_2$ as:

$$U'_1(G) = \begin{cases} 0 & \text{if } G = 0 \\ 0.001 & \text{if } G = 0.001 \\ 0.01 & \text{if } G = 0.25 \\ 0.011 & \text{if } G = 0.251 \\ 0.02 & \text{if } G = 0.75 \\ 0.021 & \text{if } G = 0.751 \\ 1.99 & \text{if } G = 1 \\ 1.991 & \text{if } G = 1.001 \end{cases}, \quad U'_2(G) = \begin{cases} 0 & \text{if } G = 0 \\ 0.001 & \text{if } G = 0.001 \\ 0.01 & \text{if } G = 0.25 \\ 0.011 & \text{if } G = 0.251 \\ 0.99 & \text{if } G = 0.75 \\ 0.991 & \text{if } G = 0.751 \\ 1.99 & \text{if } G = 1 \\ 1.991 & \text{if } G = 1.001 \end{cases},$$

it can be shown that $U'_1, U'_2$ belong to the (new) feasible set of $\mathcal{M}$, and that induce different policies in $\mathcal{M}'$. This concludes the proof. $\square$

**Proposition 4.2** (Transfer to a new reward). *There exist two MDPs $\mathcal{M} = (\mathcal{S}, \mathcal{A}, H, s_0, p, r), \mathcal{M}' = (\mathcal{S}, \mathcal{A}, H, s_0, p, r')$, with $r \neq r'$, for which there exists a policy $\pi^E$ and a pair of utilities $U_1, U_2 \in \mathfrak{U}$ such that: $U_1, U_2 \in \mathcal{U}_{p,r,\pi^E}$ and $\Pi^*_{p,r'}(U_1) \cap \Pi^*_{p,r'}(U_2) = \{\}$.*

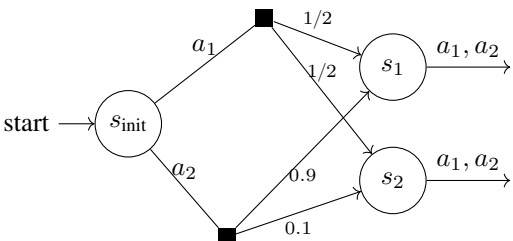

Figure 5: MDP for the proof of Proposition 4.2.

*Proof.* Similarly to the proof of Proposition 4.1, we provide two examples, one with $\mathcal{G}^{p,r'} = \mathcal{G}^{p,r}$, and the other with $\mathcal{G}^{p,r'} \neq \mathcal{G}^{p,r}$. Let us begin with the former.

Consider a simple MDP $\mathcal{M} = (\mathcal{S}, \mathcal{A}, H, s_{\text{init}}, p, r)$ with three states $\mathcal{S} = \{s_{\text{init}}, s_1, s_2\}$, two actions $\mathcal{A} = \{a_1, a_2\}$, horizon $H = 2$, initial state $s_{\text{init}}$, transition model $p$ such that:

$$p_1(s'|s_{\text{init}}, a_1) = \begin{cases} 1/2 & \text{if } s' = s_1 \\ 1/2 & \text{if } s' = s_2 \end{cases},$$

$$p_1(s'|s_{\text{init}}, a_2) = \begin{cases} 0.9 & \text{if } s' = s_1 \\ 0.1 & \text{if } s' = s_2 \end{cases},$$

and reward function $r$ that assigns $r_1(s_{\text{init}}, a_1) = 0$, $r_1(s_{\text{init}}, a_2) = 0.5$, and:

$$r_2(s, a) = \begin{cases} 0 & \text{if } s = s_1 \wedge (a = a_1 \vee a = a_2) \\ 1 & \text{if } s = s_2 \wedge (a = a_1 \vee a = a_2) \end{cases}.$$

Note that the support of the return function is $\mathcal{G}^{p,r} = \{0, 0.5, 1, 1.5\}$. We are given an expert's policy $\pi^E$ that prescribes action $a_1$ at stage 1 in state $s_{\text{init}}$, and arbitrary actions in other states (the specific action is not relevant). The MDP $\mathcal{M}$ is represented in Figure 5.

Now, we show that the utilities $U_1, U_2 \in \mathfrak{U}$, defined in points of the support $\mathcal{G}^{p,r}$ as (and connected in arbitrary continuous strictly-increasing manner between these points):

$$U_1(G) = \begin{cases} 0 & \text{if } G = 0 \\ 0.1 & \text{if } G = 0.5 \\ 0.9 & \text{if } G = 1 \\ 1.5 & \text{if } G = 1.5 \end{cases}, \qquad U_2(G) = \begin{cases} 0 & \text{if } G = 0 \\ 0.1 & \text{if } G = 0.5 \\ 0.8 & \text{if } G = 1 \\ 1.5 & \text{if } G = 1.5 \end{cases},$$

belong to the feasible set $\mathcal{U}_{p,r,\pi^E}$, and, when transferred to the new MDP $\mathcal{M}' = (\mathcal{S}, \mathcal{A}, H, s_{\text{init}}, p, r')$, with reward function $r' \neq r$ defined as:

$$r'_1(s_{\text{init}}, a_1) = 0.5, \qquad r_1(s_{\text{init}}, a_2) = 0,$$

$$r'_2(s, a) = \begin{cases} 1 & \text{if } s = s_1 \wedge (a = a_1 \vee a = a_2) \\ 0 & \text{if } s = s_2 \wedge (a = a_1 \vee a = a_2) \end{cases},$$

impose different optimal policies, i.e., utility $U_2$ keeps making action $a_1$ optimal from state $s_{\text{init}}$ even in $\mathcal{M}'$, while $U_1$ makes action $a_2$ optimal. This will demonstrate the thesis of the proposition.

Let us begin by showing that $U_1, U_2 \in \mathcal{U}_{p,r,\pi^E}$ belong to the feasible set of $\mathcal{M}$ with policy $\pi^E$. Let $\overline{\pi}$ be the policy that plays action $a_2$ in state $s_{\text{init}}$. Then, the distribution of returns induced by policies $\pi^E$ and $\overline{\pi}$ are (we represent values only at points in $\mathcal{G}^{p,r} = \{0, 0.5, 1, 1.5\}$):

$$\eta^{p,r,\pi^E} = [0.5, 0, 0.5, 0]^\intercal$$

$$\eta^{p,r,\overline{\pi}} = [0, 0.9, 0, 0.1]^\intercal.$$

Thus, policy $\pi^E$ is optimal under some utility $U$ if and only if the values assigned by $U$ to points in $\mathcal{G}^{p,r} = \{0, 0.5, 1, 1.5\}$ (denoted, respectively, by $U^1, U^2, U^3, U^4$) satisfy:

$$U^\intercal(\eta^{p,r,\pi^E} - \eta^{p,r,\overline{\pi}}) = [0.5, -0.9, 0.5, -0.1]U = 0.5U^1 - 0.9U^2 + 0.5U^3 - 0.1U^4 \geqslant 0,$$

where we have overloaded the notation and denoted with $U := [U^1, U^2, U^3, U^4]^\intercal$ both the utility and the vector of values assigned to points in $\mathcal{G}^{p,r}$. By imposing normalization constraints ($U(0) = 0, U(2) = 2$), we get $U^1 = 0$, and by imposing also the monotonicity constraints, we get that utility $U$ is in the feasible set $\mathcal{U}_{p,r,\pi^E}$ if and only if:

$$\begin{cases} U^4 \geqslant 5U^3 - 9U^2 \\ 0 < U^2 < U^3 < U^4 < 2 \end{cases}.$$

Clearly, both utilities $U_1, U_2$ satisfy these constraints, thus they belong to the feasible set $\mathcal{U}_{p,r,\pi^E}$. Now, concerning problem $\mathcal{M}'$, the performances of $\pi^E, \overline{\pi}$ w.r.t. utilities $U_1, U_2$ are:

$$J^{\pi^E}(U_1; p, r') = 0U_1(0) + 0.5U_1(0.5) + 0U_1(1) + 0.5U_1(1.5) = 1.6/2 = 0.8,$$
$$J^{\overline{\pi}}(U_1; p, r') = 0.1U_1(0) + 0U_1(0.5) + 0.9U_1(1) + 0U_1(1.5) = 0.9 \times 0.9 = 0.81,$$
$$J^{\pi^E}(U_2; p, r') = 0U_2(0) + 0.5U_2(0.5) + 0U_2(1) + 0.5U_2(1.5) = 1.6/2 = 0.8,$$
$$J^{\overline{\pi}}(U_2; p, r') = 0.1U_2(0) + 0U_2(0.5) + 0.9U_2(1) + 0U_2(1.5) = 0.9 \times 0.8 = 0.72.$$

Clearly, $J^{\pi^E}(U_1; p, r') < J^{\overline{\pi}}(U_1; p, r')$, but $J^{\pi^E}(U_2; p, r') > J^{\overline{\pi}}(U_2; p, r')$, thus we conclude that the set of policies induced by utilities $U_1, U_2$ in $\mathcal{M}'$ do not intersect, since they start from $s_{\text{init}}$ with different actions $\Pi^*_{p,r'}(U_1) \cap \Pi^*_{p,r'}(U_2) = \{\}$. This concludes the proof with an example that satisfies $\mathcal{G}^{p,r'} = \mathcal{G}^{p,r}$.

If we want an example that does *not* satisfy $\mathcal{G}^{p,r'} = \mathcal{G}^{p,r}$, then we can consider exactly the same example with $\mathcal{M}$ and $\mathcal{M}'$, but using $r'_1(s_{\text{init}}, a_2) = 0.001$. In this manner, we see that $\mathcal{G}^{p,r} = \{0, 0.5, 1, 1.5\}$, and $\mathcal{G}^{p',r} = \{0.001, 0.5, 1.001, 1.5\}$, which are different. Nevertheless, by choosing $U'_1, U'_2$ as:

$$U'_1(G) = \begin{cases} 0 & \text{if } G = 0 \\ 0.001 & \text{if } G = 0.001 \\ 0.1 & \text{if } G = 0.5 \\ 0.9 & \text{if } G = 1 \\ 0.901 & \text{if } G = 1.001 \\ 1.5 & \text{if } G = 1.5 \end{cases}, \qquad U'_2(G) = \begin{cases} 0 & \text{if } G = 0 \\ 0.001 & \text{if } G = 0.001 \\ 0.1 & \text{if } G = 0.5 \\ 0.8 & \text{if } G = 1 \\ 0.801 & \text{if } G = 1.001 \\ 1.5 & \text{if } G = 1.5 \end{cases},$$

it can be shown that $U'_1, U'_2$ still belong to the feasible set of $\mathcal{M}$ (the constraints are the same), and that induce different policies in $\mathcal{M}'$. This concludes the proof. $\qquad\square$

**Proposition 4.3.** *There exists an MDP $\mathcal{M} = (\mathcal{S}, \mathcal{A}, H, s_0, p, r)$ and a policy $\pi^E$ for which there exists a pair of utilities $U_1, U_2 \in \mathcal{U}_{p,r,\pi^E}$ such that, for any $\epsilon \geqslant 0$ smaller than some constant, there exists a policy $\pi_\epsilon$ such that $J^*(U_1; p, r) - J^{\pi_\epsilon}(U_1; p, r) = \epsilon$ and $J^*(U_2; p, r) - J^{\pi_\epsilon}(U_2; p, r) \geqslant 1$.*

*Proof.* Consider a simple MDP $\mathcal{M} = (\mathcal{S}, \mathcal{A}, H, s_{\text{init}}, p, r)$ with four states $\mathcal{S} = \{s_{\text{init}}, s_1, s_2, s_3\}$, three actions $\mathcal{A} = \{a_1, a_2, a_3\}$, horizon $H = 2$, initial state $s_{\text{init}}$, transition model $p$ such that:

$$p_1(s_2|s_{\text{init}}, a_1) = 1, \qquad p_1(s_1|s_{\text{init}}, a_3) = 1,$$
$$p_1(s'|s_{\text{init}}, a_2) = \begin{cases} 0.91 & \text{if } s' = s_1 \\ 0.09 & \text{if } s' = s_3 \end{cases},$$

and reward function $r$ that assigns $r_1(s_{\text{init}}, a_1) = r_1(s_{\text{init}}, a_2) = r_1(s_{\text{init}}, a_3) = 0$, and:

$$r_2(s, a) = \begin{cases} 0 & \text{if } s = s_1 \wedge (a = a_1 \vee a = a_2 \vee a = a_3) \\ 0.5 & \text{if } s = s_2 \wedge (a = a_1 \vee a = a_2 \vee a = a_3) \\ 1 & \text{if } s = s_3 \wedge (a = a_1 \vee a = a_2 \vee a = a_3) \end{cases}.$$

Note that the support of the return function is $\mathcal{G}^{p,r} = \{0, 0.5, 1\}$. We are given an expert's policy $\pi^E$ that prescribes action $a_1$ at stage 1 in state $s_{\text{init}}$, and arbitrary actions in other states (the specific action is not relevant). The MDP $\mathcal{M}$ is represented in Figure 6.

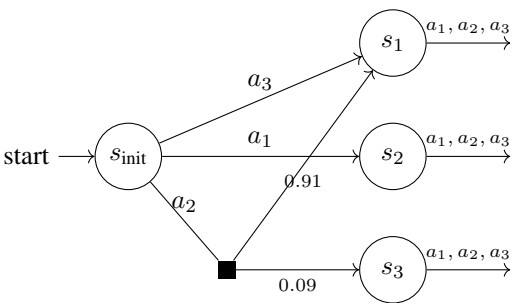

Figure 6: MDP for the proof of Proposition 4.3.

Now, we show that the utilities $U_1, U_2 \in \mathfrak{U}$, defined in points of the support $\mathcal{G}^{p,r}$ as (and connected in arbitrary continuous strictly-increasing manner between these points):

$$U_1(G) = \begin{cases} 0 & \text{if } G = 0 \\ 0.1 & \text{if } G = 0.5 \\ 0.1/0.09 & \text{if } G = 1 \end{cases}, \qquad U_2(G) = \begin{cases} 0 & \text{if } G = 0 \\ 1.099 & \text{if } G = 0.5 \\ 1.1 & \text{if } G = 1 \end{cases},$$

belong to the feasible set $\mathcal{U}_{p,r,\pi^E}$, and that, for any $\epsilon \in [0, 0.1]$, there exists a policy $\pi$ for which it holds both that $J^*(U_1; p, r) - J^\pi(U_1; p, r) = \epsilon$ and $J^*(U_2; p, r) - J^\pi(U_2; p, r) \geqslant 1$.

First, let us show that both $U_1, U_2$ belong to the feasible utility set. Let $\pi^1, \pi^2, \pi^3$ be the policies that play, respectively, action $a_1, a_2, a_3$ in state $s_{\text{init}}$ (note that $\pi^1 = \pi^E$). Then, their performances for arbitrary utility $U$ are:

$$J^{\pi^1}(U; p, r) = U(0.5),$$
$$J^{\pi^2}(U; p, r) = 0.09 U(1) + 0.91 U(0) = 0.09 U(1),$$
$$J^{\pi^3}(U; p, r) = U(0) = 0,$$

where we have used the normalization condition. Replacing $U$ with $U_1$, we get $J^*(U_1; p, r) = J^{\pi^1}(U_1; p, r) = 0.1 = J^{\pi^2}(U_1; p, r) = 0.1 > J^{\pi^3}(U_1; p, r) = 0$. Instead, replacing with $U_2$, we get $J^*(U_2; p, r) = J^{\pi^1}(U_2; p, r) = 1.099 > J^{\pi^2}(U_2; p, r) = 0.09 \times 1.1 > J^{\pi^3}(U_2; p, r) = 0$. Therefore, both $U_1, U_2 \in \mathcal{U}_{p,r,\pi^E}$.

Now, for any $\alpha \in [0, 1]$ let us denote by $\pi_\alpha$ the policy that, at state $s_{\text{init}}$, plays action $a_3$ w.p. $\alpha$, and action $a_2$ w.p. $1 - \alpha$. We show that, for any $\epsilon \in [0, 0.1]$, policy $\pi_{\epsilon/0.1}$ is $\epsilon$-optimal for utility $U_1$, and its suboptimality is at least 1 under utility $U_2$. For any $\alpha \in [0, 1]$, the expected utilities of policy $\pi_\alpha$ under $U_1$ and $U_2$ are:

$$J^{\pi_\alpha}(U_1; p, r) = (1 - \alpha) \times 0.09 \times U_1(1) = (1 - \alpha) \times 0.1,$$
$$J^{\pi_\alpha}(U_2; p, r) = (1 - \alpha) \times 0.09 \times U_2(1) = (1 - \alpha) \times 0.099,$$

from which we derive that the suboptimalities of such policy under $U_1$ and $U_2$ are:

$$J^*(U_1; p, r) - J^{\pi_\alpha}(U_1; p, r) = 0.1 - (1 - \alpha) \times 0.1 = 0.1\alpha,$$
$$J^*(U_2; p, r) - J^{\pi_\alpha}(U_2; p, r) = 1.099 - (1 - \alpha) \times 0.099 = 1 + 0.099\alpha.$$

Thus, for any $\epsilon \in [0, 0.1]$, policy $\pi_{\epsilon/0.1}$ is $\epsilon$-optimal for utility $U_1$, but it is at least 1-suboptimal for utility $U_2$.

The intuition is that utilities $U_1$ and $U_2$ assess in completely different manners the policies that play action $a_2$, although they both describe policy $\pi^E$ as optimal. This concludes the proof. $\square$

**Proposition 4.4.** *Consider an arbitrary MDP with transition model $p$ and reward function $r$. Then, for any pair of utilities $U_1, U_2 \in \mathfrak{U}$, it holds that $d^{all}_{p,r}(U_1, U_2) \leqslant \max_{G \in \mathcal{G}^{p,r}} |U_1(G) - U_2(G)|$.*

*Proof.* For the sake of simplicity, we denote the infinity norm and the 1-norm w.r.t. set $\mathcal{G}^{p,r}$ as: $\|f\|_\infty := \max_{G \in \mathcal{G}^{p,r}} |f(G)|$ and $\|f\|_1 := \sum_{G \in \mathcal{G}^{p,r}} |f(G)|$. In addition, we overload notation and use

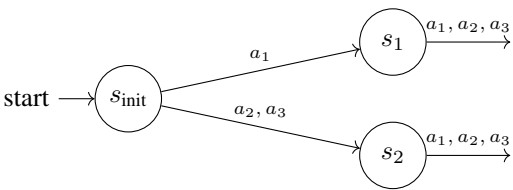

Figure 7: MDP for the proof of Proposition 4.5.

symbols $U_1, U_2$ to denote the vectors in $[0, H]^{|\mathcal{G}^{p,r}|}$ containing, respectively, the values assigned by utility functions $U_1, U_2$ to points in set $\mathcal{G}^{p,r}$. Then, we can write:

$$
\begin{aligned}
d^{\text{all}}_{p,r}(U_1, U_2) &:= \sup_{\pi \in \Pi} |J^\pi(U_1; p, r) - J^\pi(U_2; p, r)| \\
&= \sup_{\pi \in \Pi} |\mathbb{E}_{G \sim \eta^{p,r,\pi}}[U_1(G)] - \mathbb{E}_{G \sim \eta^{p,r,\pi}}[U_2(G)]| \\
&= \sup_{\pi \in \Pi} |\mathbb{E}_{G \sim \eta^{p,r,\pi}}[U_1(G) - U_2(G)]| \\
&\overset{(1)}{\leqslant} \sup_{\eta \in \Delta^{\mathcal{G}^{p,r}}} |\mathbb{E}_{G \sim \eta}[U_1(G) - U_2(G)]| \\
&\overset{(2)}{\leqslant} \sup_{\eta \in \Delta^{\mathcal{G}^{p,r}}} \mathbb{E}_{G \sim \eta}|U_1(G) - U_2(G)| \\
&\overset{(3)}{=} \|U_1 - U_2\|_\infty,
\end{aligned}
$$

where at (1) we upper bound by considering the set of all possible distributions over set $\mathcal{G}^{p,r}$ instead of just those induced by some policies in the considered MDP, at (2) we apply triangle inequality, and at (3) we have used the fact that $\|\cdot\|_1$ and $\|\cdot\|_\infty$ are dual norms. $\qquad \square$

**Proposition 4.5.** *There exists an MDP $\mathcal{M} = (\mathcal{S}, \mathcal{A}, H, s_0, p, r)$ and a policy $\pi^E$ for which there exists a pair of utilities $U_1, U_2 \in \mathcal{U}_{p,r,\pi^E}$ such that $d^{\text{all}}_{p,r}(U_1, U_2) = 1$.*

*Proof.* Consider a simple MDP $\mathcal{M} = (\mathcal{S}, \mathcal{A}, H, s_{\text{init}}, p, r)$ with three states $\mathcal{S} = \{s_{\text{init}}, s_1, s_2\}$, three actions $\mathcal{A} = \{a_1, a_2, a_3\}$, horizon $H = 2$, initial state $s_{\text{init}}$, transition model $p$ such that:

$$
p_1(s_1|s_{\text{init}}, a_1) = 1, \qquad p_1(s_2|s_{\text{init}}, a_2) = p_1(s_2|s_{\text{init}}, a_2) = 1,
$$

and reward function $r$ that assigns $r_1(s_{\text{init}}, a_1) = r_1(s_{\text{init}}, a_2) = 0$, $r_1(s_{\text{init}}, a_2) = 1$, and:

$$
r_2(s, a) = \begin{cases} 0 & \text{if } s = s_1 \wedge (a = a_1 \vee a = a_2 \vee a_3) \\ 1 & \text{if } s = s_2 \wedge (a = a_1 \vee a = a_2 \vee a_3) \end{cases}.
$$

Note that the support of the return function is $\mathcal{G}^{p,r} = \{0, 1, 2\}$. We are given an expert's policy $\pi^E$ that prescribes action $a_3$ at stage 1 in state $s_{\text{init}}$, and arbitrary actions in the other states (the specific action is not relevant). The MDP $\mathcal{M}$ is represented in Figure 7.

Consider two utilities $U_1, U_2$, that take on the following values in $\mathcal{G}^{p,r}$:

$$
U_1(G) = \begin{cases} 0 & \text{if } G = 0 \\ 0.1 & \text{if } G = 1 \\ 2 & \text{if } G = 2 \end{cases},
$$

$$
U_2(G) = \begin{cases} 0 & \text{if } G = 0 \\ 1.1 & \text{if } G = 1 \\ 2 & \text{if } G = 2 \end{cases}.
$$

It is immediate that both utilities belong to the feasible set $\mathcal{U}_{p,r,\pi^E}$. Nevertheless, if we denote by $\overline{\pi}$ the policy that plays action $a_2$ in state $s_{\text{init}}$, we see that $J^{\overline{\pi}}(U_1; p, r) = 0.1$, while $J^{\overline{\pi}}(U_2; p, r) = 1.1$, so that the difference is 1. $\qquad \square$

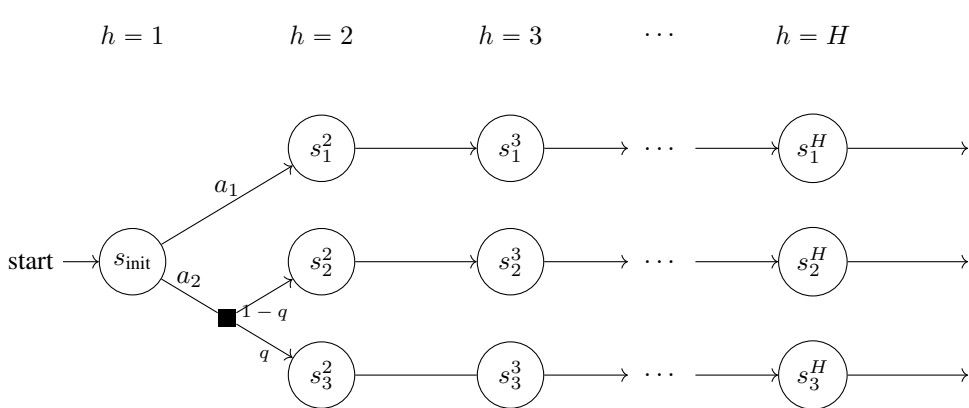

Figure 8: MDP for the proof of Proposition 4.6.

**Proposition 4.6** (Multiple demonstrations). *Let $\mathcal{S}, \mathcal{A}, H$ be, respectively, any state space, action space, and horizon, satisfying $S \geqslant 3, A \geqslant 2, H \geqslant 2$, and let $U^E \in \mathfrak{U}$ be any utility. If, for any possible dynamics $s_0, p$ and reward $r$, we are given the set of* all *the deterministic optimal policies of the corresponding RS-MDP $(\mathcal{S}, \mathcal{A}, H, s_0, p, r, U^E)$, then we can* uniquely *identify utility $U^E$.*

*Proof.* We provide a constructive proof that shows which values of $s_0, p, r$ it is sufficient to choose for recovering $U^E$ exactly. The construction is articulated into two parts. First, we aim to recover the value of $U^E(1)$, i.e., for $G = 1$; next, we recover the utility for all other possible values of return. The intuition is that we construct a Standard Gamble (SG) between two policies over the entire horizon (Wakker, 2010).

To infer $U^E(1)$, we use the $s_0, p, r$ values that provide the MDP described in Figure 8.

We consider a single initial state $s_{\text{init}}$. From here, action $a_1$ (and all actions other than $a_1$ and $a_2$) brings deterministically to state $s_1^2$, while action $a_2$ brings to state $s_3^2$ w.p. $q$ (to choose, for some $q \in [0, 1]$), and to state $s_2^2$ w.p. $1 - q$. From state $s_i^2$, for any $i \in [\![3]\!]$, all actions bring deterministically to state $s_i^3$, and so on, up to state $s_i^H$. We will call the trajectory $\{s_{\text{init}}, s_i^2, s_i^3, \ldots, s_i^H\}$ the $i$th trajectory for all $i \in [\![3]\!]$, and we will write $G(i)$ to denote the sum of rewards along such trajectory. To infer the value $U^E(1)$, we select a reward $r' : \mathcal{S} \times \mathcal{A} \times [\![H]\!] \to [0, 1]$ that provides return $G(1) = 1.5$ to the first trajectory, return $G(2) = 1$ to the second trajectory, and return $G(3) = H$ to the third trajectory (this is possible because $H \geqslant 2$). By selecting, successively, all the values of $q \in [0, 1]$, we are asking to the expert to play either action $a_1$ or action $a_2$ from the initial state $s_{\text{init}}$ (we denote policies $\pi^1, \pi^2$, respectively, the policies that play actions $a_1, a_2$ in $s_{\text{init}}$). Since we are assuming that the expert will demonstrate all the possible deterministic optimal policies, there exists a value $q' \in [0, 1]$ for which the expert demonstrates both policies $\pi^1$ and $\pi^2$. Indeed, the expected utilities of policies $\pi^1, \pi^2$ for arbitrary value of $q$ are (we write $p(q)$ as the generic transition model):

$$J^{\pi^1}(U^E; p(q), r') = U^E(1.5),$$
$$J^{\pi^2}(U^E; p(q), r') = qU^E(H) + (1 - q)U^E(1) = qH + (1 - q)U^E(1),$$

and since $U^E$ is strictly-increasing, we have $U^E(1) < U^E(1.5) < U^E(H) = H$, thus there must exist $q'$ that permits to write $U^E(1.5)$ as a convex combination of the other two. This allows us to write:

$$U^E(1.5) = q'H + (1 - q')U^E(1). \tag{7}$$

Next, we select reward $r''$ that provides returns $G(1) = 1, G(2) = 0.5, G(3) = 1.5$. Thus, there must exist a $q'' \in [0, 1]$ for which the expert demonstrates both policies $\pi^1$ and $\pi^2$, allowing us to write:

$$U^E(1) = q''U^E(1.5) + (1 - q'')U^E(0.5). \tag{8}$$

Finally, we can repeat the same step with a third reward $r'''$ that provides returns $G(1) = 0.5, G(2) = 0, G(3) = 1$, and for some $q''' \in [0, 1]$ we obtain:

$$U^E(0.5) = q''' U^E(1). \tag{9}$$

By putting together Eq. (7), Eq. (8), and Eq. (9), we can retrieve $U^E(1)$:

$$\begin{cases} U^E(1.5) = q'H + (1 - q')U^E(1) \\ U^E(1) = q''U^E(1.5) + (1 - q'')U^E(0.5) \\ U^E(0.5) = q'''U^E(1) \end{cases}.$$

Now that we know $U^E(1)$, we can infer the utility for all the returns $\overline{G} \in (1, H)$ by choosing a reward that provides returns $G(1) = \overline{G}, G(2) = 1, G(3) = H$, because for some $\overline{q} \in [0, 1]$ the expert will play both policies $\pi^1$ and $\pi^2$, which allows us to write:

$$U^E(\overline{G}) = \overline{q}H + (1 - \overline{q})U^E(1),$$

and to retrieve $U^E(\overline{G})$.

Similarly, for all $\overline{G} \in (0, 1)$, we select a reward that provides returns $G(1) = \overline{G}, G(2) = 0, G(3) = 1$, and for some $\overline{q} \in [0, 1]$ we can write:

$$U^E(\overline{G}) = \overline{q}U^E(1),$$

and retrieve $U^E(\overline{G})$.

This concludes the proof. As a final remark, we stress that the initial step for inferring $U^E(1)$ cannot be dropped because there is no reward $r : \mathcal{S} \times \mathcal{A} \times [\![H]\!] \to [0, 1]$ that provides returns $G(2) = 0$ and $G(3) = H$, because both the first and second trajectories pass through action $a_2$ in state $s_{\text{init}}$. $\qquad \square$

# E   ADDITIONAL RESULTS AND PROOFS FOR SECTION 5

This appendix is divided in 4 parts. First, we show the complexity of implementing operator $\Pi_{\overline{\mathfrak{U}}_L}$ (Appendix E.1). In Appendix E.2, we provide the pseudocode, along with a description, of algorithms EXPLORE, PLANNING, ERD, and ROLLOUT. In Appendix E.3, we provide the proof of Theorem 5.1. In Appendix E.4, we provide the proof of Theorem 5.2.

## E.1   PROJECTING ONTO THE SET OF DISCRETIZED UTILITIES

Let us use the square brackets $[]$ to denote the components of vectors. Then, note that set $\overline{\mathfrak{U}}_L$ can be represented more explicitly as:

$$\overline{\mathfrak{U}}_L = \{\overline{U} \in [0, H]^d \,|\, \overline{U}[1] = 0 \wedge \overline{U}[d] = H \wedge \overline{U}[i] \leqslant \overline{U}[i + 1] \,\forall i \in [\![d - 1]\!]$$
$$\wedge \;\; \forall i, j \in [\![d]\!] \text{ s.t. } i < j : |\overline{U}[i] - \overline{U}[j]| \leqslant L(j - i)\epsilon_0\}. \tag{10}$$

Notice that set $\overline{\mathfrak{U}}_L$ is closed and convex, since it is defined by linear constraints only. The amount of constraints scales as $\propto d^2$.

## E.2   MISSING ALGORITHMS AND SUB-ROUTINES

**EXPLORE**   In Algorithm 3, we report the pseudo-code implementing subroutine EXPLORE. Simply put, we adopt a uniform-sampling strategy, i.e., we collect $n = \lfloor \tau/(SAH) \rfloor$ samples from each $(s, a, h) \in \mathcal{S} \times \mathcal{A} \times [\![H]\!]$ triple, that we use to compute the empirical estimate of the transition model. We return such estimate.

**PLANNING**   The PLANNING sub-routine (Algorithm 4) takes in input a utility $U$, an environment index $i$, and a transition model $p$, that uses to construct the RS-MDP $\mathcal{M}_U :=$ $(\mathcal{S}^i, \mathcal{A}^i, H, s_0^i, p, \overline{r}^i, U)$. Notice that $\mathcal{M}_U \neq \mathcal{M}_{U^E}^i$, for 3 aspects. First, it uses the input transition model $p \neq p^i$; next, it consider the discretized reward $\overline{r}^i \neq r^i$; finally, it has input utility $U \neq U^E$.

---

**Algorithm 3:** EXPLORE

**Input:** samples budget $\tau$

1   $n \leftarrow \lfloor \tau/(SAH) \rfloor$

2   **for** $i \in \{1, 2, \ldots, N\}$ **do**

     // Initialize the transition model estimate:

3      $\widehat{p}_h^i(s'|s, a) = 0$ for all $(s, a, h, s') \in \mathcal{S} \times \mathcal{A} \times [\![H]\!] \times \mathcal{S}$

     // Collect samples:

4      **for** $(s, a, h) \in \mathcal{S} \times \mathcal{A} \times [\![H]\!]$ **do**

5         **for** $\_ \in \{1, 2, \ldots, n\}$ **do**

6            $s' \leftarrow$ sample from $p_h^i(\cdot|s, a)$

7            $\widehat{p}_h^i(s'|s, a) \leftarrow \widehat{p}_h^i(s'|s, a) + 1$

8         **end**

9      **end**

10     $\widehat{p}_h^i(\cdot|s, a) \leftarrow \widehat{p}_h^i(\cdot|s, a)/n$

11 **end**

12 **Return** $\{\widehat{p}^i\}_i$

---

PLANNING outputs two items. The optimal performance $J^*(U; p, r^i)$ for RS-MDP $\mathcal{M}_U$, and the optimal policy $\psi^* = \{\psi_h^*\}_h$ for the enlarged state space MDP $\mathfrak{E}[\mathcal{M}_U]$. However, it should be remarked that, instead of computing optimal policy $\psi^*$ for $\mathfrak{E}[\mathcal{M}_U]$ only at pairs $(s, y) \in \mathcal{S} \times \mathcal{G}_h^{p, \overline{r}^i}$ for all $h \in [\![H]\!]$, PLANNING computes the optimal policy $\psi^*$ at all pairs $(s, y) \in \mathcal{S} \times \mathcal{Y}_h$ for all $h \in [\![H]\!]$ (note that $\mathcal{G}_h^{p, \overline{r}^i} \subseteq \mathcal{Y}_h$).

The algorithm implemented in PLANNING for computing both $J^*(U; p, r^i)$ and $\psi^*$ is value iteration. The difference from common implementations of value iterations lies in the presence of an additional variable in the state. A similar pseudocode is provided in Algorithm 1 of Wu & Xu (2023).

---

**Algorithm 4:** PLANNING

**Input:** utility $U$, environment index $i$, transition model $p$

// Initialize the $Q$ and value function at the last stage:

1   **for** $(s, y) \in \mathcal{S}^i \times \mathcal{Y}_H$ **do**

2     **for** $a \in \mathcal{A}^i$ **do**

3        $Q_H(s, y, a) \leftarrow U(y + \overline{r}_H^i(s, a))$

4     **end**

5     $V_H(s, y) \leftarrow \max\limits_{a \in \mathcal{A}^i} Q_H(s, y, a)$

6     $\psi_H(s, y) \leftarrow \arg\max\limits_{a \in \mathcal{A}^i} Q_H(s, y, a)$                 /* Keep just one action */

7   **end**

// Backward induction:

8   **for** $h = H - 1, \ldots, 2, 1$ **do**

9     **for** $(s, y) \in \mathcal{S}^i \times \mathcal{Y}_h$ **do**

10      **for** $a \in \mathcal{A}^i$ **do**

11        $Q_h(s, y, a) \leftarrow \mathbb{E}_{s' \sim p_h(\cdot|s, a)}\Big[V_{h+1}(s', y + \overline{r}_h^i(s, a))\Big]$

12      **end**

13      $V_h(s, y) \leftarrow \max\limits_{a \in \mathcal{A}^i} Q_h(s, y, a)$

14      $\psi_h(s, y) \leftarrow \arg\max\limits_{a \in \mathcal{A}^i} Q_h(s, y, a)$          /* Keep just one action */

15     **end**

16 **end**

// Return optimal performance and policy:

17 **Return** $V_1(s_0^i, 0), \psi$

---

---

**Algorithm 5:** ERD - Estimate the Return Distribution

---

**Input:** dataset $\mathcal{D}^E$, reward $r$

       // Initialize $\widehat{\eta}$:

**1 for** $y \in \mathcal{Y}$ **do**

**2**   |   $\widehat{\eta}(y) \leftarrow 0$

**3 end**

       // Loop over all trajectories in $\mathcal{D}^E$:

**4 for** $\omega \in \mathcal{D}^E$ **do**

        |   // Compute return of $\omega = \{s_1, a_1, \ldots, s_H, a_H, s_{H+1}\}$:

**5**   |   $G \leftarrow \sum_{h=1}^{H} r_h(s_h, a_h)$

        |   // Update estimate $\widehat{\eta}$:

**6**   |   **if** $G \leqslant 0$ **then**

**7**   |  |   $\widehat{\eta}(0) \leftarrow \widehat{\eta}(0) + 1$

**8**   |   **end**

**9**   |   **else if** $G > \lfloor \frac{H}{\epsilon_0} \rfloor \epsilon_0$ **then**

**10**   |  |   $\widehat{\eta}(\lfloor \frac{H}{\epsilon_0} \rfloor \epsilon_0) \leftarrow \widehat{\eta}(\lfloor \frac{H}{\epsilon_0} \rfloor \epsilon_0) + 1$

**11**   |   **end**

**12**   |   **else**

**13**   |  |   $L \leftarrow \max_{y \in \mathcal{Y} \wedge y < G} y$

**14**   |  |   $U \leftarrow \min_{y \in \mathcal{Y} \wedge y \geqslant G} y$

**15**   |  |   $\widehat{\eta}(L) \leftarrow \widehat{\eta}(L) + \frac{U-G}{U-L}$

**16**   |  |   $\widehat{\eta}(U) \leftarrow \widehat{\eta}(U) + \frac{G-L}{U-L}$

**17**   |   **end**

**18 end**

       // Normalize:

**19** $\widehat{\eta} \leftarrow \widehat{\eta}/|\mathcal{D}^E|$

**20 Return** $\widehat{\eta}$

---

**ERD (Estimate the Return Distribution)** The ERD sub-routine (Algorithm 5) takes in input a dataset $\mathcal{D}^E = \{\omega_j\}_j$ of state-action trajectories $\omega_j \in \Omega$ and a reward function $r$, and it computes an estimate of the return distribution w.r.t. $r$.

For every trajectory $\omega_j \in \mathcal{D}^E$, ERD computes the return $G_j$ of $\omega_j$ based on the input reward $r$ (Line 5). In the next lines, ERD simply computes the categorical projection of the mixture of Dirac deltas:

$$\widehat{\eta} = \text{Proj}_{\mathcal{C}} \Big( \sum_j \frac{1}{|\mathcal{D}^E|} \delta_{G_j} \Big),$$

where the categorical projection operator $\text{Proj}_{\mathcal{C}}$ is defined in Eq. (4).

**ROLLOUT** ROLLOUT (Algorithm 6) takes in input a Markovian policy $\psi$, a transition model $p$, a reward $r$, an environment index $i$, and a number of trajectories $K$, to construct the MDP $\mathcal{M} := (\mathcal{S}^i, \mathcal{A}^i, H, s_0^i, p, r)$ obtained from MDP $\mathcal{M}^i$ by replacing the dynamics and reward $p^i, r^i$ with the input $p, r$.

ROLLOUT collects $K$ trajectories by playing policy $\psi$ in $\mathcal{M}$ for $K$ times, computes the return $G$ of each trajectory, and then returns a dataset $\mathcal{D}$ containing these $K$ returns. In other words, with abuse of notation, we say that the outputted dataset $\mathcal{D} = \{G_k\}_{k \in [\![K]\!]}$ is obtained by collecting $K$ samples $G_k$ from distribution $\eta^{p,r,\psi}$.

### E.3 ANALYSIS OF CATY-UL

**Theorem 5.1.** *Let $\epsilon, \delta \in (0, 1)$, and let $\mathcal{U}$ be a subset of $\mathfrak{U}_L$ containing the utilities to classify. If we set $\epsilon_0 = \epsilon^2/(72HL^2)$, and if it holds that, for all $i \in [\![N]\!]$:*

$$\text{if } |\mathcal{U}| = 1: \qquad \tau^{E,i} \leqslant \widetilde{\mathcal{O}}\Big( \frac{H^2}{\epsilon^2} \log \frac{N}{\delta} \Big), \qquad \tau^i \leqslant \widetilde{\mathcal{O}}\Big( \frac{SAH^4}{\epsilon^2} \log \frac{SAHNL}{\delta\epsilon} \Big),$$

$$\text{else}: \qquad \tau^{E,i} \leqslant \widetilde{\mathcal{O}}\Big( \frac{H^4L^2}{\epsilon^4} \log \frac{HNL}{\delta\epsilon} \Big), \qquad \tau^i \leqslant \widetilde{\mathcal{O}}\Big( \frac{SAH^5}{\epsilon^2} \Big( S + \log \frac{SAHN}{\delta} \Big) \Big),$$

---

**Algorithm 6:** ROLLOUT

---

**Input:** policy $\psi$, transition model $p$, reward $r$, environment index $i$, number of trajectories $K$

1   $\mathcal{D} \leftarrow \{\}$
   // Loop over the number of trajectories:
2   **for** $\_ \in \{1, 2, \ldots, K\}$ **do**
3     $s \leftarrow s_0^i$
4     $y \leftarrow 0$                         /* $y$ keeps track of the accumulated reward */
5     **for** $h = 1$ *to* $H$ **do**
6       $a \leftarrow \psi_h(s, y)$
7       $y \leftarrow y + r_h(s, a)$
8       $s \leftarrow s'$ where $s' \sim p_h(\cdot|s, a)$
9     **end**
10    $\mathcal{D} \leftarrow \mathcal{D} \cup \{y\}$
11 **end**
12 **Return** $\mathcal{D}$

---

*then, w.p. at least $1 - \delta$,* **CATY-UL** *correctly classifies all the $U \in \mathcal{U}$ that satisfy either* $\max_i \overline{\mathcal{C}}_{p^i, r^i, \pi^{E,i}}(U) < \Delta - \epsilon$ *(inside $\mathcal{U}_\Delta$) or* $\max_i \overline{\mathcal{C}}_{p^i, r^i, \pi^{E,i}}(U) > \Delta + \epsilon$ *(outside $\mathcal{U}_\Delta$).*

*Proof.* Observe that the classification carried out by **CATY-UL** complies with the statement in the theorem as long as we can demonstrate that:

$$\mathbb{P}_{\{\mathcal{M}^i\}_i, \{\pi^{E,i}\}_i} \left( \sup_{U \in \mathcal{U}} \left| \max_{i \in [\![N]\!]} \overline{\mathcal{C}}_{p^i, r^i, \pi^{E,i}}(U) - \max_{i \in [\![N]\!]} \widehat{\mathcal{C}}^i(U) \right| \leqslant \epsilon \right) \geqslant 1 - \delta,$$

where $\mathbb{P}_{\{\mathcal{M}^i\}_i, \{\pi^{E,i}\}_i}$ represents the joint probability distribution induced by the exploration phase of **CATY-UL** and the execution of each $\pi^{E,i}$ in the corresponding $\mathcal{M}^i$.

We can rewrite this expression as:

$$\sup_{U \in \mathcal{U}} \left| \max_{i \in [\![N]\!]} \overline{\mathcal{C}}_{p^i, r^i, \pi^{E,i}}(U) - \max_{i \in [\![N]\!]} \widehat{\mathcal{C}}^i(U) \right| \overset{(1)}{\leqslant} \sup_{U \in \mathcal{U}} \max_{i \in [\![N]\!]} \left| \overline{\mathcal{C}}_{p^i, r^i, \pi^{E,i}}(U) - \widehat{\mathcal{C}}^i(U) \right|$$

$$= \max_{i \in [\![N]\!]} \sup_{U \in \mathcal{U}} \left| \overline{\mathcal{C}}_{p^i, r^i, \pi^{E,i}}(U) - \widehat{\mathcal{C}}^i(U) \right|,$$

where at (1) we have upper bounded the difference of the maxima of two real-valued functions with the maximum of their difference. This shows that we can obtain the result as long as we can demonstrate that, for all $i \in [\![N]\!]$, it holds that:

$$\mathbb{P}_{p^i, r^i, \pi^{E,i}} \left( \sup_{U \in \mathcal{U}} \left| \overline{\mathcal{C}}_{p^i, r^i, \pi^{E,i}}(U) - \widehat{\mathcal{C}}^i(U) \right| \leqslant \epsilon \right) \geqslant 1 - \frac{\delta}{N}; \tag{11}$$

the statement of the theorem would then follow from a union bound. Therefore, let us omit the $i$ index for simplicity, and let us try to obtain the bound in Eq. (11). We can write:

$$\sup_{U \in \mathcal{U}} \left| \overline{\mathcal{C}}_{p, r, \pi^E}(U) - \widehat{\mathcal{C}}(U) \right| := \sup_{U \in \mathcal{U}} \left| \left( J^*(U; p, r) - J^{\pi^E}(U; p, r) \right) - \left( \widehat{J}^*(U) - \widehat{J}^E(U) \right) \right|$$

$$\overset{(2)}{\leqslant} \sup_{U \in \mathcal{U}} \left| J^{\pi^E}(U; p, r) - \widehat{J}^E(U) \right| + \sup_{U \in \mathcal{U}} \left| J^*(U; p, r) - \widehat{J}^*(U) \right|$$

$$\overset{(3)}{=} \sup_{U \in \mathcal{U}} \left| \mathbb{E}_{G \sim \eta^{p,r,\pi^E}} [U(G)] - \mathbb{E}_{G \sim \widehat{\eta}^E} [U(G)] \right.$$

$$\left. \pm \mathbb{E}_{G \sim \text{Proj}_{\mathcal{C}}(\eta^{p,r,\pi^E})} [U(G)] \right| + \sup_{U \in \mathcal{U}} \left| J^*(U; p, r) - \widehat{J}^*(U) \right|$$

$$\overset{(4)}{\leqslant} \sup_{U \in \mathcal{U}} \left| \mathbb{E}_{G \sim \eta^{p,r,\pi^E}} [U(G)] - \mathbb{E}_{G \sim \text{Proj}_{\mathcal{C}}(\eta^{p,r,\pi^E})} [U(G)] \right|$$

$$+ \sup_{U \in \mathcal{U}} \left| \mathbb{E}_{G \sim \text{Proj}_{\mathcal{C}}(\eta^{p,r,\pi^E})} [U(G)] - \mathbb{E}_{G \sim \widehat{\eta}^E} [U(G)] \right|$$

$$+ \sup_{U \in \mathcal{U}} \left| J^*(U; p, r) - \widehat{J}^*(U) \right|$$

$$\overset{(5)}{\leqslant} \sup_{f:\, f \text{ is } L\text{-Lipschitz}} \Big| \underset{G \sim \eta^{p,r,\pi^E}}{\mathbb{E}}[f(G)] - \underset{G \sim \mathrm{Proj}_{\mathcal{C}}(\eta^{p,r,\pi^E})}{\mathbb{E}}[f(G)]\Big|$$

$$+ \sup_{U \in \mathcal{U}} \Big| \underset{G \sim \mathrm{Proj}_{\mathcal{C}}(\eta^{p,r,\pi^E})}{\mathbb{E}}[U(G)] - \underset{G \sim \widehat{\eta}^E}{\mathbb{E}}[U(G)]\Big|$$

$$+ \sup_{U \in \mathcal{U}} \big|J^*(U;p,r) - \widehat{J}^*(U)\big|$$

$$\overset{(6)}{=} L \cdot w_1(\eta^{p,r,\pi^E}, \mathrm{Proj}_{\mathcal{C}}(\eta^{p,r,\pi^E}))$$

$$+ \sup_{U \in \mathcal{U}} \Big| \underset{G \sim \mathrm{Proj}_{\mathcal{C}}(\eta^{p,r,\pi^E})}{\mathbb{E}}[U(G)] - \underset{G \sim \widehat{\eta}^E}{\mathbb{E}}[U(G)]\Big|$$

$$\sup_{U \in \mathcal{U}} \big|J^*(U;p,r) - \widehat{J}^*(U)\big|,$$

where at (2) we have applied triangle inequality, at (3) we use the definition of $J^{\pi^E}(U;p,r)$, and that of $\widehat{J}^E(U)$ (Line 4 of **CATY-UL**), and we have added and subtracted a term, where operator $\mathrm{Proj}_{\mathcal{C}}$ is defined in Eq. (4). We remark that distribution $\eta^{p,r,\pi^E}$ may have a support that grows exponentially in $H$, while both $\widehat{\eta}^E$ and $\mathrm{Proj}_{\mathcal{C}}(\eta^{p,r,\pi^E})$ are supported on $\mathcal{Y}$. Note that $\widehat{\eta}^E$ and $\mathrm{Proj}_{\mathcal{C}}(\eta^{p,r,\pi^E})$ are different distributions, since the former is the projection on $\mathcal{Y}$ of an estimate of $\eta^{p,r,\pi^E}$. At (4), we apply triangle inequality, at (5) we use the hypothesis that all utilities are $L$-Lipschitz $\mathcal{U} \subseteq \mathfrak{U}_L$, and notice that $\mathfrak{U}_L$ is a subset of all $L$-Lipschitz functions $f : [0,H] \to [0,H]$, and at (6) we apply the duality formula for the 1-Wasserstein distance $w_1$ (see Eq. (6.3) in Chapter 6 of Villani (2008)).

Concerning the case $|\mathcal{U}| = 1$, we apply, for all $i \in [\![N]\!]$, Lemma E.3 with probability $\delta/(2N)$ and accuracy $\epsilon/3$, and Lemma E.5 with probability $\delta/(2N)$ and accuracy $\epsilon/3$, while we bound the 1-Wasserstein distance through Lemma E.1, to obtain, through an application of the union bound, that:

$$\underset{\{\mathcal{M}^i\}_i, \{\pi^{E,i}\}_i}{\mathbb{P}} \Big( \sup_{U \in \mathcal{U}} \Big| \max_{i \in [\![N]\!]} \overline{\mathcal{C}}_{p^i,r^i,\pi^{E,i}}(U) - \max_{i \in [\![N]\!]} \widehat{\mathcal{C}}^i(U)\Big| \leqslant$$

$$L\sqrt{2H\epsilon_0} + \epsilon/3 + HL\epsilon_0 + \epsilon/3 \Big) \geqslant 1 - \delta,$$

as long as, for all $i \in [\![N]\!]$:

$$\tau^{E,i} \geqslant \widetilde{\mathcal{O}}\Big( \frac{H^2 \log \frac{N}{\delta}}{\epsilon^2}\Big),$$

$$\tau^i \geqslant \widetilde{\mathcal{O}}\Big( \frac{SAH^4}{\epsilon^2} \log \frac{SAHN}{\delta\epsilon_0}\Big).$$

By setting $\epsilon_0 = \frac{\epsilon^2}{72HL^2}$, we obtain that:

$$L\sqrt{2H\epsilon_0} + HL\epsilon_0 = \frac{\epsilon}{6} + \frac{\epsilon^2}{72L} \leqslant \epsilon/3.$$

By putting this bound into the bound on $\tau^i$, we get the result.

When $\mathcal{U}$ is an arbitrary subset of $\mathfrak{U}_L$, we apply, for all $i \in [\![N]\!]$, Lemma E.4 with probability $\delta/(2N)$ and accuracy $\epsilon/3$, and Lemma E.13 with probability $\delta/(2N)$ and accuracy $\epsilon/3$, while we bound the 1-Wasserstein distance through Lemma E.1, to obtain, through an application of the union bound, that:

$$\underset{\{\mathcal{M}^i\}_i, \{\pi^{E,i}\}_i}{\mathbb{P}} \Big( \sup_{U \in \mathcal{U}} \Big| \max_{i \in [\![N]\!]} \overline{\mathcal{C}}_{p^i,r^i,\pi^{E,i}}(U) - \max_{i \in [\![N]\!]} \widehat{\mathcal{C}}^i(U)\Big| \leqslant$$

$$L\sqrt{2H\epsilon_0} + \epsilon/3 + HL\epsilon_0 + \epsilon/3 \Big) \geqslant 1 - \delta,$$

as long as, for all $i \in [\![N]\!]$:

$$\tau^{E,i} \geqslant \widetilde{\mathcal{O}}\Big( \frac{H^3}{\epsilon^2\epsilon_0} \log \frac{HN}{\delta\epsilon_0}\Big),$$

$$\tau^i \geqslant \tilde{\mathcal{O}}\Big(\frac{SAH^5}{\epsilon^2}\Big(S + \log \frac{SAHN}{\delta}\Big)\Big).$$

Again, by setting $\epsilon_0 = \frac{\epsilon^2}{72HL^2}$, we obtain that:

$$L\sqrt{2H\epsilon_0} + HL\epsilon_0 = \frac{\epsilon}{6} + \frac{\epsilon^2}{72L} \leqslant \epsilon/3.$$

By putting this bound into the bounds on $\tau^{E,i}$ and $\tau^i$, we get the result. $\qquad\square$

### E.3.1 Lemmas on the Expert's Return Distribution

**Lemma E.1.** *Let the projection operator $Proj_{\mathcal{C}}$ be defined as in Eq.* (4)*, over set $\mathcal{Y}$ with discretization $\epsilon_0$. Then, for all $i \in [\![N]\!]$, it holds that:*

$$w_1(\eta^{p^i,r^i,\pi^{E,i}}, Proj_{\mathcal{C}}(\eta^{p^i,r^i,\pi^{E,i}})) \leqslant \sqrt{2H\epsilon_0}.$$

*Proof.* For the sake of simplicity, we omit index $i \in [\![N]\!]$, but the following derivation can be applied to all the $N$ demonstrations.

By applying Lemma 5.2 of Rowland et al. (2024), replacing term $1/(1-\gamma)$ with horizon $H$, we get:

$$w_1(\eta^{p,r,\pi^E}, Proj_{\mathcal{C}}(\eta^{p,r,\pi^E})) \leqslant \sqrt{H}\ell_2(\eta^{p,r,\pi^E}, Proj_{\mathcal{C}}(\eta^{p,r,\pi^E})).$$

Similarly to the proof of Proposition 3 of Rowland et al. (2018), we can write:

$$
\begin{aligned}
\ell_2^2(\eta^{p,r,\pi^E}, Proj_{\mathcal{C}}(\eta^{p,r,\pi^E})) &\overset{(1)}{:=} \int_{\mathbb{R}} (F_{\eta^{p,r,\pi^E}}(y) - F_{Proj_{\mathcal{C}}(\eta^{p,r,\pi^E})}(y))^2 dy \\
&\overset{(2)}{=} \int_0^H (F_{\eta^{p,r,\pi^E}}(y) - F_{Proj_{\mathcal{C}}(\eta^{p,r,\pi^E})}(y))^2 dy \\
&\overset{(3)}{=} \sum_{j\in[\![d-1]\!]} \int_{y_j}^{y_{j+1}} (F_{\eta^{p,r,\pi^E}}(y) - F_{Proj_{\mathcal{C}}(\eta^{p,r,\pi^E})}(y))^2 dy \\
&\qquad + \int_{y_d}^H (F_{\eta^{p,r,\pi^E}}(y) - F_{Proj_{\mathcal{C}}(\eta^{p,r,\pi^E})}(y))^2 dy \\
&\overset{(4)}{\leqslant} \sum_{j\in[\![d-1]\!]} \int_{y_j}^{y_{j+1}} (F_{\eta^{p,r,\pi^E}}(y) - F_{Proj_{\mathcal{C}}(\eta^{p,r,\pi^E})}(y))^2 dy + \epsilon_0 \\
&\overset{(5)}{\leqslant} \sum_{j\in[\![d-1]\!]} \int_{y_j}^{y_{j+1}} (F_{\eta^{p,r,\pi^E}}(y_{j+1}) - F_{\eta^{p,r,\pi^E}}(y_j))^2 dy + \epsilon_0 \\
&= \sum_{j\in[\![d-1]\!]} (y_{j+1} - y_j)(F_{\eta^{p,r,\pi^E}}(y_{j+1}) - F_{\eta^{p,r,\pi^E}}(y_j))^2 + \epsilon_0 \\
&\overset{(6)}{=} \epsilon_0 \sum_{j\in[\![d-1]\!]} (F_{\eta^{p,r,\pi^E}}(y_{j+1}) - F_{\eta^{p,r,\pi^E}}(y_j))^2 + \epsilon_0 \\
&\overset{(7)}{\leqslant} \epsilon_0 \Big(\sum_{j\in[\![d-1]\!]} (F_{\eta^{p,r,\pi^E}}(y_{j+1}) - F_{\eta^{p,r,\pi^E}}(y_j))\Big)^2 + \epsilon_0 \\
&\overset{(8)}{=} \epsilon_0 (F_{\eta^{p,r,\pi^E}}(y_d) - F_{\eta^{p,r,\pi^E}}(y_1))^2 + \epsilon_0 \\
&\leqslant 2\epsilon_0,
\end{aligned}
$$

where at (1) we have applied the definition of $\ell_2$ distance (Eq. (3)), at (2) we recognize that the two distributions $\eta^{p,r,\pi^E}, Proj_{\mathcal{C}}(\eta^{p,r,\pi^E})$ are defined on $[0, H]$, at (3) we use the additivity property of the integral, using notation $\mathcal{Y} := \{0, \epsilon_0, 2\epsilon_0, \ldots, \lfloor H/\epsilon_0 \rfloor \epsilon_0\}$, $d := |\mathcal{Y}| = \lfloor H/\epsilon_0 \rfloor + 1$, $y_1 := 0, y_2 := \epsilon_0, y_3 := 2\epsilon_0, \ldots, y_d := \lfloor H/\epsilon_0 \rfloor \epsilon_0$, (notation introduced in Section 5). At (4) we upper bound $\int_{y_d}^H (F_{\eta^{p,r,\pi^E}}(y) - F_{Proj_{\mathcal{C}}(\eta^{p,r,\pi^E})}(y))^2 dy \leqslant \int_{y_d}^H dy = H - y_d = H - \lfloor H/\epsilon_0 \rfloor \epsilon_0 = \epsilon_0(H/\epsilon_0 - \lfloor H/\epsilon_0 \rfloor) \leqslant \epsilon_0$ since the difference of cumulative distribution functions is bounded by

1. At (5), thanks to the definition of the projection operator $\text{Proj}_{\mathcal{C}}$ (Eq. (4)), we notice that, for $y \in [y_j, y_{j+1}]$, it holds that $F_{\text{Proj}_{\mathcal{C}}(\eta^{p,r,\pi^E})}(y) \in [F_{\eta^{p,r,\pi^E}}(y_j), F_{\eta^{p,r,\pi^E}}(y_{j+1})]$, thus we can upper bound the integrand through the maximum, constant, difference of cumulative distribution functions. At (6) we use the definition of set $\mathcal{Y}$, i.e., an $\epsilon_0$-covering of the $[0, H]$ interval, at (7) we use the Cauchy-Schwarz's inequality $\sum_j (x_j)^2 \leqslant (\sum_j x_j)^2$ for $x_j \geqslant 0$, and noticed that the summands are always non-negative, at (8) we apply a telescoping argument.

The result follows by taking the square root of both sides. $\qquad\square$

**Lemma E.2.** *Let $i \in [\![N]\!]$, and let $f \in [0, H]^d$ be an arbitrary $d$-dimensional vector. Denote by $G_1, G_2, \ldots, G_{\tau^{E,i}} \overset{i.i.d.}{\sim} \eta^{p^i,r^i,\pi^{E,i}}$ the random variables representing the returns of the $\tau^{E,i}$ trajectories inside dataset $\mathcal{D}^{E,i}$. Let $\widehat{\eta}^{E,i}$ be the random output of Algorithm 5 that depends on the random variables $G_1, G_2, \ldots, G_{\tau^{E,i}}$. Then, it holds that:*

$$\mathbb{E}_{G_1, G_2, \ldots, G_{\tau^{E,i}} \sim \eta^{p^i,r^i,\pi^{E,i}}} \left[ \mathbb{E}_{y \sim \widehat{\eta}^{E,i}} \left[ f(y) \right] \right] = \mathbb{E}_{y \sim \text{Proj}_{\mathcal{C}}(\eta^{p^i,r^i,\pi^{E,i}})} \left[ f(y) \right].$$

*Proof.* We omit index $i$ for simplicity, but the proof can be carried out for all $i \in [\![N]\!]$ independently. To prove the statement, we use the notation described in Appendix E.2 for the Dirac delta, to provide an explicit representation of both the distribution $\text{Proj}_{\mathcal{C}}(\eta^{p,r,\pi^E})$ and the "random" distribution $\widehat{\eta}^E$.

We consider distribution $\eta^{p,r,\pi^E}$ supported on $\mathcal{Z} := \{z_1, z_2, \ldots, z_M\} \subseteq [0, H]$, while distributions $\text{Proj}_{\mathcal{C}}(\eta^{p,r,\pi^E}), \widehat{\eta}^E$ are supported on set $\mathcal{Y} = \{y_1, y_2, \ldots, y_d\} \subseteq [0, H]$.

W.r.t. distribution $\text{Proj}_{\mathcal{C}}(\eta^{p,r,\pi^E})$, we can write:

$$\text{Proj}_{\mathcal{C}}(\eta^{p,r,\pi^E}) = \text{Proj}_{\mathcal{C}} \left( \sum_{k \in [\![M]\!]} \eta^{p,r,\pi^E}(z_k) \delta_{z_k} \right)$$

$$\overset{(1)}{=} \sum_{k \in [\![M]\!]} \eta^{p,r,\pi^E}(z_k) \text{Proj}_{\mathcal{C}}(\delta_{z_k})$$

$$\overset{(2)}{=} \sum_{k \in [\![M]\!]} \eta^{p,r,\pi^E}(z_k) \left( \delta_{y_1} \mathbb{1}\{z_k \leqslant y_1\} + \delta_{y_d} \mathbb{1}\{z_k > y_d\} \right.$$

$$\left. + \sum_{j \in [\![d-1]\!]} \left( \frac{y_{j+1} - z_k}{y_{j+1} - y_j} \delta_{y_j} + \frac{z_k - y_j}{y_{j+1} - y_j} \delta_{y_{j+1}} \right) \mathbb{1}\{z_k \in (y_j, y_{j+1}]\} \right)$$

$$= \delta_{y_1} \sum_{k \in [\![M]\!]} \eta^{p,r,\pi^E}(z_k) \left( \mathbb{1}\{z_k \leqslant y_1\} + \frac{y_2 - z_k}{y_2 - y_1} \mathbb{1}\{z_k \in (y_1, y_2]\} \right)$$

$$+ \sum_{j \in \{2, \ldots, d-1\}} \delta_{y_j} \left( \sum_{k \in [\![M]\!]} \eta^{p,r,\pi^E}(z_k) \left( \frac{y_{j+1} - z_k}{y_{j+1} - y_j} \mathbb{1}\{z_k \in (y_i, y_{j+1}]\} \right. \right.$$

$$\left. \left. + \frac{z_k - y_{j-1}}{y_i - y_{j-1}} \mathbb{1}\{z_k \in (y_{j-1}, y_i]\} \right) \right)$$

$$+ \delta_{y_d} \sum_{k \in [\![M]\!]} \eta^{p,r,\pi^E}(z_k) \left( \mathbb{1}\{z_k > y_d\} + \frac{z_k - y_{d-1}}{y_d - y_{d-1}} \mathbb{1}\{z_k \in (y_{d-1}, y_d]\} \right),$$

where at (1) we have applied the extension in Eq. (5) of the projection operator $\text{Proj}_{\mathcal{C}}$ to finite mixtures of Dirac distributions, and at (2) we have applied its definition (Eq. (4)).

Concerning distribution $\widehat{\eta}^E$, based on Algorithm 5, we can write:

$$\widehat{\eta}^E = \frac{\delta_{y_1}}{\tau^E} \left( \sum_{t \in [\![\tau^E]\!]} \left( \mathbb{1}\{G_t \leqslant y_1\} + \frac{y_2 - G_t}{y_2 - y_1} \mathbb{1}\{G_t \in (y_1, y_2]\} \right) \right)$$

$$+ \sum_{j \in \{2, \ldots, d-1\}} \frac{\delta_{y_j}}{\tau^E} \left( \sum_{t \in [\![\tau^E]\!]} \left( \frac{y_{j+1} - G_t}{y_{j+1} - y_j} \mathbb{1}\{G_t \in (y_i, y_{j+1}]\} \right. \right.$$

$$\left. \left. + \frac{G_t - y_{j-1}}{y_i - y_{j-1}} \mathbb{1}\{G_t \in (y_{j-1}, y_i]\} \right) \right)$$

$$+ \frac{\delta_{y_d}}{\tau^E}\Big( \sum_{t\in[\![\tau^E]\!]} \Big( \mathbb{1}\{G_t > y_d\} + \frac{G_t - y_{d-1}}{y_d - y_{d-1}}\mathbb{1}\{G_t \in (y_{d-1}, y_d]\}\Big)\Big).$$

Now, if we take the expectation of the random vector $\widehat{\eta}^E$ w.r.t. $\eta^{p,r,\pi^E}$, we get:

$$\mathbb{E}_{G_1,G_2,\ldots,G_{\tau^E}\sim\eta^{p,r,\pi^E}}\Big[\widehat{\eta}^E\Big]$$

$$= \mathbb{E}_{G_1,G_2,\ldots,G_{\tau^E}\sim\eta^{p,r,\pi^E}}\bigg[\frac{\delta_{y_1}}{\tau^E}\Big(\sum_{t\in[\![\tau^E]\!]}\Big(\mathbb{1}\{G_t \leq y_1\} + \frac{y_2 - G_t}{y_2 - y_1}\mathbb{1}\{G_t \in (y_1, y_2]\}\Big)\Big)$$

$$+ \sum_{j\in\{2,\ldots,d-1\}}\frac{\delta_{y_j}}{\tau^E}\Big(\sum_{t\in[\![\tau^E]\!]}\Big(\frac{y_{j+1} - G_t}{y_{j+1} - y_j}\mathbb{1}\{G_t \in (y_i, y_{j+1}]\}$$

$$+ \frac{G_t - y_{j-1}}{y_i - y_{j-1}}\mathbb{1}\{G_t \in (y_{j-1}, y_i]\}\Big)\Big)$$

$$+ \frac{\delta_{y_d}}{\tau^E}\Big(\sum_{t\in[\![\tau^E]\!]}\Big(\mathbb{1}\{G_t > y_d\} + \frac{G_t - y_{d-1}}{y_d - y_{d-1}}\mathbb{1}\{G_t \in (y_{d-1}, y_d]\}\Big)\Big)\bigg]$$

$$\overset{(3)}{=} \mathbb{E}_{G\sim\eta^{p,r,\pi^E}}\bigg[\delta_{y_1}\Big(\mathbb{1}\{G \leq y_1\} + \frac{y_2 - G}{y_2 - y_1}\mathbb{1}\{G \in (y_1, y_2]\}\Big)$$

$$+ \sum_{j\in\{2,\ldots,d-1\}}\delta_{y_j}\Big(\frac{y_{j+1} - G}{y_{j+1} - y_j}\mathbb{1}\{G \in (y_i, y_{j+1}]\}$$

$$+ \frac{G - y_{j-1}}{y_i - y_{j-1}}\mathbb{1}\{G \in (y_{j-1}, y_i]\}\Big)$$

$$+ \delta_{y_d}\Big(\mathbb{1}\{G > y_d\} + \frac{G - y_{d-1}}{y_d - y_{d-1}}\mathbb{1}\{G \in (y_{d-1}, y_d]\}\Big)\bigg]$$

$$\overset{(4)}{=} \delta_{y_1}\sum_{k\in[\![M]\!]}\eta^{p,r,\pi^E}(z_k)\Big(\mathbb{1}\{z_k \leq y_1\} + \frac{y_2 - z_k}{y_2 - y_1}\mathbb{1}\{z_k \in (y_1, y_2]\}\Big)$$

$$+ \sum_{j\in\{2,\ldots,d-1\}}\delta_{y_j}\Big(\sum_{k\in[\![M]\!]}\eta^{p,r,\pi^E}(z_k)\Big(\frac{y_{j+1} - z_k}{y_{j+1} - y_j}\mathbb{1}\{z_k \in (y_i, y_{j+1}]\}$$

$$+ \frac{z_k - y_{j-1}}{y_i - y_{j-1}}\mathbb{1}\{z_k \in (y_{j-1}, y_i]\}\Big)\Big)$$

$$+ \delta_{y_d}\sum_{k\in[\![M]\!]}\eta^{p,r,\pi^E}(z_k)\Big(\mathbb{1}\{z_k > y_d\} + \frac{z_k - y_{d-1}}{y_d - y_{d-1}}\mathbb{1}\{z_k \in (y_{d-1}, y_d]\}\Big)$$

$$\overset{(5)}{=} \mathrm{Proj}_{\mathcal{C}}(\eta^{p,r,\pi^E}),$$

where at (3) we use the fact that $G_1, G_2, \ldots, G_{\tau^E}$ are independent and identically distributed, at (4) we apply the linearity of the expectation, we notice that $\delta_{y_j}$ does not depend on $G$ for all $j \in [\![d]\!]$, and we notice that, for any $y \in \mathcal{Y}$, it holds that $\mathbb{E}_{G\sim\eta^{p,r,\pi^E}}\big[\mathbb{1}\{G \leq y\}\big] = \eta^{p,r,\pi^E}(G \leq y) = \sum_{k\in[\![M]\!]}\eta^{p,r,\pi^E}(z_k)\mathbb{1}\{z_k \leq y\}$, where we have abused notation by writing $\eta^{p,r,\pi^E}(G \leq y)$ to mean the probability, under distribution $\eta^{p,r,\pi^E}$, that event $\{G \leq y\}$ happens. Moreover, similarly, we notice that, for any $y, y' \in \mathcal{Y}$, it holds that $\mathbb{E}_{G\sim\eta^{p,r,\pi^E}}\big[G \cdot \mathbb{1}\{G \in [y, y']\}\big] = \sum_{k\in[\![M]\!]}z_k\eta^{p,r,\pi^E}(z_k)\mathbb{1}\{z_k \in [y, y']\}$. At (5) we simply recognize $\mathrm{Proj}_{\mathcal{C}}(\eta^{p,r,\pi^E})$ using the previous expression.

This concludes the proof because the equality of the Dirac delta representations means that the expectations of any function w.r.t. these two distributions coincide. $\qquad\square$

**Lemma E.3.** *Let $i \in [\![N]\!]$ and let $\epsilon, \delta \in (0, 1)$. If $|\mathcal{U}| = 1$, then, with probability at least $1 - \delta$, we have:*

$$\sup_{U\in\mathcal{U}}\Big| \mathop{\mathbb{E}}_{G\sim\mathrm{Proj}_{\mathcal{C}}(\eta^{p^i,r^i,\pi^{E,i}})}[U(G)] - \mathop{\mathbb{E}}_{G\sim\widehat{\eta}^{E,i}}[U(G)]\Big| \leq \epsilon,$$

*as long as:*

$$\tau^E \geqslant c\frac{H^2 \log\frac{2}{\delta}}{\epsilon^2},$$

*where $c$ is some positive constant.*

*Proof.* Let $U$ be the only function inside $\mathcal{U}$. Let us omit index $i$ for simplicity. Then, we can write:

$$\left| \mathop{\mathbb{E}}_{G\sim\widehat{\eta}^E}[U(G)] - \mathop{\mathbb{E}}_{G\sim\mathrm{Proj}_{\mathcal{C}}(\eta^{p,r,\pi^E})}[U(G)] \right| \stackrel{(1)}{=} \left| \mathop{\mathbb{E}}_{G\sim\widehat{\eta}^E}[U(G)] - \mathop{\mathbb{E}}_{\eta^{p,r,\pi^E}}\left[ \mathop{\mathbb{E}}_{G\sim\widehat{\eta}^E}[U(G)] \right] \right|$$

$$\stackrel{(2)}{\leqslant} cH\sqrt{\frac{\log\frac{2}{\delta}}{\tau^E}},$$

where at (1) we have applied Lemma E.2, and at (2) we have applied the Hoeffding's inequality noticing that function $U$ is bounded in $[0, H]$, and denoting with $c$ some positive constant.

By imposing:

$$cH\sqrt{\frac{\log\frac{2}{\delta}}{\tau^E}} \leqslant \epsilon,$$

and solving w.r.t. $\tau^E$, we get the result. $\qquad\square$

**Lemma E.4.** *Let $i \in [\![N]\!]$ and let $\epsilon, \delta \in (0, 1)$. Then, with probability at least $1 - \delta$, we have:*

$$\sup_{U\in\mathcal{U}} \left| \mathop{\mathbb{E}}_{G\sim Proj_{\mathcal{C}}(\eta^{p^i,r^i,\pi^{E,i}})}[U(G)] - \mathop{\mathbb{E}}_{G\sim\widehat{\eta}^{E,i}}[U(G)] \right| \leqslant \epsilon,$$

*as long as:*

$$\tau^E \geqslant \widetilde{\mathcal{O}}\left(\frac{H^3}{\epsilon^2\epsilon_0}\log\frac{H}{\delta\epsilon_0}\right).$$

*Proof.* Again, let us omit index $i$ for simplicity. First, for all possible functions $U \in \mathcal{U}$, we denote by $\overline{U} \in \overline{\mathfrak{U}}_L$ the function in $\overline{\mathfrak{U}}_L$ that takes on the values that the function $U$ assigns to the points of set $\mathcal{Y}$. This permits us to write:

$$\sup_{U\in\mathcal{U}} \left| \mathop{\mathbb{E}}_{G\sim\widehat{\eta}^E}[U(G)] - \mathop{\mathbb{E}}_{G\sim\mathrm{Proj}_{\mathcal{C}}(\eta^{p,r,\pi^E})}[U(G)] \right|$$

$$= \sup_{\overline{U}\in\overline{\mathfrak{U}}_L} \left| \mathop{\mathbb{E}}_{G\sim\widehat{\eta}^E}[\overline{U}(G)] - \mathop{\mathbb{E}}_{G\sim\mathrm{Proj}_{\mathcal{C}}(\eta^{p,r,\pi^E})}[\overline{U}(G)] \right|$$

$$\stackrel{(1)}{\leqslant} \sup_{\overline{U}\in[0,H]^d} \left| \mathop{\mathbb{E}}_{G\sim\widehat{\eta}^E}[\overline{U}(G)] - \mathop{\mathbb{E}}_{G\sim\mathrm{Proj}_{\mathcal{C}}(\eta^{p,r,\pi^E})}[\overline{U}(G)] \right|$$

$$\stackrel{(2)}{=} \sup_{\overline{U}\in[0,H]^d} \left| \mathop{\mathbb{E}}_{G\sim\widehat{\eta}^E}[\overline{U}(G)] - \mathop{\mathbb{E}}_{\eta^{p,r,\pi^E}}\left[ \mathop{\mathbb{E}}_{G\sim\widehat{\eta}^E}[\overline{U}(G)] \right] \right|,$$

where at (1) we upper bound by considering all the possible vectors $\overline{U} \in [0, H]^d$, and at (2) we apply Lemma E.2.

Now, similarly to the proof of Lemma 7.2 in Agarwal et al. (2021), we construct an $\epsilon'$-covering of set $[0, H]^d$, call it $\mathcal{N}_{\epsilon'}$, with $|\mathcal{N}_{\epsilon'}| \leqslant (1 + 2H\sqrt{d}/\epsilon')^d$ such that, for all $f \in [0, H]^d$, there exists $f' \in \mathcal{N}_{\epsilon'}$ for which $\|f - f'\|_2 \leqslant \epsilon'$. By applying a union bound over all $f' \in \mathcal{N}_{\epsilon'}$ and Lemma E.3, we have that, with probability at least $1 - \delta$, for all $f' \in \mathcal{N}_{\epsilon'}$, it holds that:

$$\left| \mathop{\mathbb{E}}_{G\sim\widehat{\eta}^E}[f'(G)] - \mathop{\mathbb{E}}_{\eta^{p,r,\pi^E}}\left[ \mathop{\mathbb{E}}_{G\sim\widehat{\eta}^E}[f'(G)] \right] \right| \leqslant cH\sqrt{\frac{d\log\frac{2(1+2H\sqrt{d}/\epsilon')}{\delta}}{\tau^E}}. \tag{12}$$

Next, for any $f \in [0, H]^d$, denote its closest points (in 2-norm) from $\mathcal{N}_{\epsilon'}$ as $f'$. Then, we have:

$$
\left| \mathbb{E}_{G \sim \hat{\eta}^E}[f(G)] - \mathbb{E}_{\eta^{p,r,\pi^E}} \left[ \mathbb{E}_{G \sim \hat{\eta}^E}[f(G)] \right] \right|
$$

$$
= \left| \mathbb{E}_{G \sim \hat{\eta}^E}[f(G)] - \mathbb{E}_{\eta^{p,r,\pi^E}} \left[ \mathbb{E}_{G \sim \hat{\eta}^E}[f(G)] \right] \pm \left( \mathbb{E}_{G \sim \hat{\eta}^E}[f'(G)] - \mathbb{E}_{\eta^{p,r,\pi^E}} \left[ \mathbb{E}_{G \sim \hat{\eta}^E}[f'(G)] \right] \right) \right|
$$

$$
\overset{(3)}{\leqslant} \left| \mathbb{E}_{G \sim \hat{\eta}^E}[f'(G)] - \mathbb{E}_{\eta^{p,r,\pi^E}} \left[ \mathbb{E}_{G \sim \hat{\eta}^E}[f'(G)] \right] \right|
$$

$$
+ \left| \mathbb{E}_{G \sim \hat{\eta}^E}[f(G) - f'(G)] \right| + \left| \mathbb{E}_{\eta^{p,r,\pi^E}} \left[ \mathbb{E}_{G \sim \hat{\eta}^E}[f(G) - f'(G)] \right] \right|
$$

$$
\overset{(4)}{\leqslant} cH\sqrt{\frac{d \log \frac{2(1 + 2H\sqrt{d}/\epsilon')}{\delta}}{\tau^E}} + 2\epsilon'
$$

$$
\overset{(5)}{\leqslant} c'H\sqrt{\frac{d \log \frac{Hd\tau^E}{\delta}}{\tau^E}}
$$

where at (3) we apply triangle inequality, at (4) we apply the result in Eq. (12), and the fact that, by definition of $\epsilon'$-covering, $\|f - f'\|_2 \leqslant \epsilon'$ entails that $|f(y) - f(y')| \leqslant \epsilon'$ for all $y \in \mathcal{Y}$; at (5) we set $\epsilon' = 1/\tau^E$, and we simplify.

The result follows by upper bounding $d \leqslant H/\epsilon_0 + 1$, and then by setting:

$$
c''H\sqrt{\frac{H \log \frac{H\tau^E}{\delta \epsilon_0}}{\epsilon_0 \tau^E}} \leqslant \epsilon, \tag{13}
$$

and solving w.r.t. $\tau^E$, and noticing that for all $\tau^E$ greater than some constant, we can get rid of the logarithmic terms in $\tau^E$. $\qquad\square$

### E.3.2 LEMMAS ON THE OPTIMAL PERFORMANCE FOR SINGLE UTILITY

In this section, we will omit index $i \in [\![N]\!]$ since the following derivations can be carried out for each $i$.

We denote the arbitrary MDP in $\{\mathcal{M}^i\}_i$ as $\mathcal{M} = (\mathcal{S}, \mathcal{A}, H, s_0, p, r)$, and its analogous with discretized reward $\bar{r}$, defined at all $(s, a, h) \in \mathcal{S} \times \mathcal{A} \times [\![H]\!]$ as $\bar{r}_h(s, a) := \Pi_{\mathcal{R}}[r_h(s, a)]$, as $\overline{\mathcal{M}} := (\mathcal{S}, \mathcal{A}, H, s_0, p, \bar{r})$. We denote the analogous MDPs with empirical transition model $\hat{p}$ as $\widehat{\mathcal{M}} = (\mathcal{S}, \mathcal{A}, H, s_0, \hat{p}, r)$ and $\widehat{\overline{\mathcal{M}}} := (\mathcal{S}, \mathcal{A}, H, s_0, \hat{p}, \bar{r})$.

Given any utility $U \in \mathfrak{U}_L$, we denote the corresponding RS-MDPs, respectively, as $\mathcal{M}_U, \overline{\mathcal{M}}_U, \widehat{\mathcal{M}}_U, \widehat{\overline{\mathcal{M}}}_U$. Concerning the discretized RS-MDPs $\overline{\mathcal{M}}_U$ and $\widehat{\overline{\mathcal{M}}}_U$, we denote the corresponding enlarged state space MDPs, respectively, as $\mathfrak{E}[\overline{\mathcal{M}}_U] = (\{\mathcal{S} \times \mathcal{Y}_h\}_h, \mathcal{A}, H, (s_0, 0), \mathfrak{p}, \mathfrak{r})$ and $\mathfrak{E}[\widehat{\overline{\mathcal{M}}}_U] = (\{\mathcal{S} \times \mathcal{Y}_h\}_h, \mathcal{A}, H, (s_0, 0), \hat{\mathfrak{p}}, \mathfrak{r})$, where we decided to define such enlarged state space MDPs using the state space $\{\mathcal{S} \times \mathcal{Y}_h\}_h$ considered by Algorithm 4 (PLANNING) instead of, respectively, $\{\mathcal{S} \times \mathcal{G}_h^{p,\bar{r}}\}_h$ and $\{\mathcal{S} \times \mathcal{G}_h^{\hat{p},\bar{r}}\}_h$. Thus, the transition models $\mathfrak{p}$ and $\hat{\mathfrak{p}}$, from any $h \in [\![H]\!]$ and $(s, y, a) \in \mathcal{S} \times \mathcal{Y}_h \times \mathcal{A}$, assign to the next state $(s', y') \in \mathcal{S} \times \mathcal{Y}_{h+1}$ the probability: $\mathfrak{p}_h(s', y'|s, y, a) := p_h(s'|s, a)\mathbb{1}\{y' = y + \bar{r}_h(s, a)\}$ and $\hat{\mathfrak{p}}_h(s', y'|s, y, a) := \hat{p}_h(s'|s, a)\mathbb{1}\{y' = y + \bar{r}_h(s, a)\}$. Moreover, the reward function $\mathfrak{r}$, in any $h \in [\![H]\!]$ and $(s, y, a) \in \mathcal{S} \times \mathcal{Y}_h \times \mathcal{A}$, is $\mathfrak{r}_h(s, y, a) = 0$ if $h < H$, and $\mathfrak{r}_h(s, y, a) = U(y + \bar{r}_h(s, a))$ if $h = H$.

We will make extensive use of notation for $V$- and $Q$- functions introduced in Appendix B.

We are now ready to proceed with the analysis. In general, the analysis shares similarities to that of Theorem 3 of Wu & Xu (2023), but we use results also from Azar et al. (2013) to obtain tighter bounds.

**Lemma E.5.** *Let $\epsilon, \delta \in (0,1)$. For any fixed $L$-Lipschitz utility function $U \in \mathfrak{U}_L$, it suffices to execute* **CATY-UL** *with:*

$$\tau \leqslant \widetilde{\mathcal{O}}\Big(\frac{SAH^4}{\epsilon^2} \log \frac{SAH}{\delta \epsilon_0}\Big),$$

*to obtain $\big|J^*(U; p, r) - \widehat{J}^*(U)\big| \leqslant HL\epsilon_0 + \epsilon$ w.p. $1 - \delta$.*

*Proof.* For an arbitrary utility $U \in \mathfrak{U}_L$, we can write:

$$
\begin{aligned}
|J^*(U; p, r) - \widehat{J}^*(U)| &\overset{(1)}{=} |J^*(U; p, r) - \widehat{J}^*(U) \pm J^*(\mathfrak{p}, \mathfrak{r})| \\
&\overset{(2)}{\leqslant} |J^*(U; p, r) - J^*(\mathfrak{p}, \mathfrak{r})| + |J^*(\mathfrak{p}, \mathfrak{r}) - \widehat{J}^*(U)| \\
&\overset{(3)}{=} |J^*(U; p, r) - J^*(\mathfrak{p}, \mathfrak{r})| + |J^*(\mathfrak{p}, \mathfrak{r}) - J^*(\widehat{\mathfrak{p}}, \mathfrak{r})| \\
&\overset{(4)}{\leqslant} HL\epsilon_0 + |J^*(\mathfrak{p}, \mathfrak{r}) - J^*(\widehat{\mathfrak{p}}, \mathfrak{r})| \\
&= HL\epsilon_0 + |V_1^*(s_0, 0; \mathfrak{p}, \mathfrak{r}) - V_1^*(s_0, 0; \widehat{\mathfrak{p}}, \mathfrak{r})| \\
&\leqslant HL\epsilon_0 + \max_{h \in [\![H]\!], (s,y,a) \in \mathcal{S} \times \mathcal{Y}_h \times \mathcal{A}} |Q_h^*(s, y, a; \mathfrak{p}, \mathfrak{r}) - Q_h^*(s, y, a; \widehat{\mathfrak{p}}, \mathfrak{r})| \\
&\overset{(5)}{\leqslant} HL\epsilon_0 + \epsilon',
\end{aligned}
$$

where at (1) we add and subtract the optimal expected utility in the enlarged MDP $\mathfrak{E}[\overline{\mathcal{M}}_U]$ considered by Algorithm 4, but with the true transition model $\mathfrak{p}$. At (2) we apply triangle inequality, at (3) we recognize that the estimate $\widehat{J}^*(U)$ used in **CATY-UL** and outputted by PLANNING (Algorithm 4) is the optimal expected utility for the discretized problem with estimated dynamics $\widehat{\mathfrak{p}}$, at (4) we use Proposition 3 of Wu & Xu (2023), since $U$ is $L$-Lipschitz, and at (5) we apply Lemma E.6 to bound the distance between $Q$-functions.

By setting:

$$\underbrace{c\sqrt{\frac{H^3 \log \frac{4SAHd}{\delta}}{n}}}_{\leqslant \epsilon/3} + \underbrace{cH^2 \Big(\frac{\log \frac{16SAHd}{\delta}}{n}\Big)^{3/4}}_{\leqslant \epsilon/3} + \underbrace{cH^3 \frac{\log \frac{16SAHd}{\delta}}{n}}_{\leqslant \epsilon/3} \leqslant \epsilon,$$

and solving w.r.t. $\epsilon$:

$$\begin{cases} n \geqslant c' \dfrac{H^3 \log \frac{4SAHd}{\delta}}{\epsilon^2} \\ n \geqslant c'' \dfrac{H^{8/3} \log \frac{16SAHd}{\delta}}{\epsilon^{4/3}} \\ n \geqslant c''' \dfrac{H^3 \log \frac{16SAHd}{\delta}}{\epsilon} \end{cases}.$$

Taking the largest bound, we get:

$$n \geqslant c \frac{H^3 \log \frac{16SAHd}{\delta}}{\epsilon^2},$$

for some positive constant $c$. Since $d \leqslant H/\epsilon_0 + 1$, we can write:

$$\tau \geqslant c' \frac{SAH^4 \log \frac{c''SAH}{\delta \epsilon_0}}{\epsilon^2},$$

for some positive constants $c', c''$, where we used that $\tau = SAHn$. $\qquad\square$

The proof of the following lemma is organized in many lemmas, and is based on the proof of Theorem 1 of Azar et al. (2013).

**Lemma E.6.** *For any $\delta \in (0,1)$, we have:*

$$\max_{h \in [\![H]\!], (s,y,a) \in \mathcal{S} \times \mathcal{Y}_h \times \mathcal{A}} |Q_h^*(s, y, a; \mathfrak{p}, \mathfrak{r}) - Q_h^*(s, y, a; \widehat{\mathfrak{p}}, \mathfrak{r})| \leqslant \epsilon',$$

*w.p. at least $1 - \delta$, where $\epsilon'$ is defined as:*

$$\epsilon' := c\sqrt{\frac{H^3 \log \frac{4SAHd}{\delta}}{n}} + cH^2\left(\frac{\log \frac{16SAHd}{\delta}}{n}\right)^{3/4} + cH^3\frac{\log \frac{16SAHd}{\delta}}{n},$$

*for some positive constant c.*

*Proof.* We upper bound one side, and then the other. For all the $h \in [\![H]\!], (s, y, a) \in \mathcal{S} \times \mathcal{Y}_h \times \mathcal{A}$, it holds that:

$$Q_h^*(s, a, y; \mathfrak{p}, \mathfrak{r}) - Q_h^*(s, y, a; \widehat{\mathfrak{p}}, \mathfrak{r})$$

$$\overset{(1)}{\leqslant} \mathbb{E}_{\widehat{\mathfrak{p}}, \mathfrak{r}, \psi^*}\left[\sum_{h'=h}^{H}\sum_{s'\in\mathcal{S}}\left(p_{h'}(s'|s_{h'}, a_{h'}) - \widehat{p}_{h'}(s'|s_{h'}, a_{h'})\right)V_{h'+1}^{\psi^*}(s', y_{h'+1}; \mathfrak{p}, \mathfrak{r})\right.$$

$$\left.\left| s_h = s, y_h = y, a_h = a\right]\right.$$

$$\overset{(2)}{\leqslant} \mathbb{E}_{\widehat{\mathfrak{p}}, \mathfrak{r}, \psi^*}\left[\sum_{h'=h}^{H} c\sqrt{\frac{c_1 \mathbb{V}_{s'\sim\widehat{p}_{h'}(\cdot|s_{h'}, a_{h'})}[V_{h'+1}^{\psi^*}(s', y_{h'+1}; \widehat{\mathfrak{p}}, \mathfrak{r})]}{n}} + b_2\right.$$

$$\left.\left| s_h = s, y_h = y, a_h = a\right]\right.$$

$$= c\sqrt{\frac{c_1}{n}}\mathbb{E}_{\widehat{\mathfrak{p}}, \mathfrak{r}, \psi^*}\left[\sum_{h'=h}^{H}\sqrt{\mathbb{V}_{s'\sim\widehat{p}_{h'}(\cdot|s_{h'}, a_{h'})}[V_{h'+1}^{\psi^*}(s', y_{h'+1}; \widehat{\mathfrak{p}}, \mathfrak{r})]}\,\Big|\, s_h = s, y_h = y, a_h = a\right]$$

$$+ Hb_2$$

$$\overset{(3)}{\leqslant} c\sqrt{\frac{c_1}{n}}\sqrt{H^3} + Hb_2$$

$$= c\sqrt{\frac{H^3 \log \frac{4SAHY}{\delta}}{n}} + c'H^2\left(\frac{\log \frac{16SAHY}{\delta}}{n}\right)^{3/4} + c''H^3\frac{\log \frac{16SAHY}{\delta}}{n}$$

$$=: \epsilon',$$

where at (1) we have applied Lemma E.7, at (2) we have applied Lemma E.10 with $\delta/2$ of probability, at (3) we have applied Lemma E.12.

The proof for the other side of inequality is completely analogous, and it holds w.p. $1 - \delta/2$. The result follows through the application of a union bound. $\qquad\square$

**Lemma E.7.** *For any tuple $h \in [\![H]\!], (s, y, a) \in \mathcal{S} \times \mathcal{Y}_h \times \mathcal{A}$, it holds that:*

$$Q_h^*(s, y, a; \mathfrak{p}, \mathfrak{r}) - Q_h^*(s, y, a; \widehat{\mathfrak{p}}, \mathfrak{r}) \leqslant \mathbb{E}_{\widehat{\mathfrak{p}}, \mathfrak{r}, \psi^*}\left[\sum_{h'=h}^{H}\sum_{s'\in\mathcal{S}}\right.$$

$$\left.\left(p_{h'}(s'|s_{h'}, a_{h'}) - \widehat{p}_{h'}(s'|s_{h'}, a_{h'})\right)V_{h'+1}^{\psi^*}(s', y_{h'+1}; \mathfrak{p}, \mathfrak{r})\,\Big|\, s_h = s, y_h = y, a_h = a\right],$$

$$Q_h^*(s, y, a; \mathfrak{p}, \mathfrak{r}) - Q_h^*(s, y, a; \widehat{\mathfrak{p}}, \mathfrak{r}) \geqslant \mathbb{E}_{\widehat{\mathfrak{p}}, \mathfrak{r}, \widehat{\psi}^*}\left[\sum_{h'=h}^{H}\sum_{s'\in\mathcal{S}}\right.$$

$$\left.\left(p_{h'}(s'|s_{h'}, a_{h'}) - \widehat{p}_{h'}(s'|s_{h'}, a_{h'})\right)V_{h'+1}^{\psi^*}(s', y_{h'+1}; \mathfrak{p}, \mathfrak{r})\,\Big|\, s_h = s, y_h = y, a_h = a\right],$$

*where $\psi^*, \widehat{\psi}^*$ are the optimal policies respectively in problems $\mathfrak{p}, \mathfrak{r}$ and $\widehat{\mathfrak{p}}, \mathfrak{r}$.*

*Proof.* For any $h \in [\![H]\!], (s, y, a) \in \mathcal{S} \times \mathcal{Y}_h \times \mathcal{A}$, we can write:

$$Q_h^*(s, y, a; \mathfrak{p}, \mathfrak{r}) - Q_h^*(s, y, a; \widehat{\mathfrak{p}}, \mathfrak{r})$$

$$= Q_h^{\psi^*}(s, y, a; \mathfrak{p}, \mathfrak{r}) - Q_h^{\widehat{\psi}^*}(s, y, a; \widehat{\mathfrak{p}}, \mathfrak{r})$$

$$
\overset{(1)}{\leqslant} Q_h^{\psi^*}(s,y,a;\mathfrak{p},\mathfrak{r}) - Q_h^{\psi^*}(s,y,a;\widehat{\mathfrak{p}},\mathfrak{r})
$$

$$
\overset{(2)}{=} \mathfrak{r}_h(s,y,a) + \sum_{(s',y')\in\mathcal{S}\times\mathcal{Y}_{h+1}} \mathfrak{p}_h(s',y'|s,y,a)V_{h+1}^{\psi^*}(s',y';\mathfrak{p},\mathfrak{r})
$$

$$
- \left( \mathfrak{r}_h(s,y,a) + \sum_{(s',y')\in\mathcal{S}\times\mathcal{Y}_{h+1}} \widehat{\mathfrak{p}}_h(s',y'|s,y,a)V_{h+1}^{\psi^*}(s',y';\widehat{\mathfrak{p}},\mathfrak{r}) \right)
$$

$$
\overset{(3)}{=} \sum_{(s',y')\in\mathcal{S}\times\mathcal{Y}_{h+1}} \mathfrak{p}_h(s',y'|s,y,a)V_{h+1}^{\psi^*}(s',y';\mathfrak{p},\mathfrak{r})
$$

$$
- \sum_{(s',y')\in\mathcal{S}\times\mathcal{Y}_{h+1}} \widehat{\mathfrak{p}}_h(s',y'|s,y,a)V_{h+1}^{\psi^*}(s',y';\widehat{\mathfrak{p}},\mathfrak{r})
$$

$$
\pm \sum_{(s',y')\in\mathcal{S}\times\mathcal{Y}_{h+1}} \widehat{\mathfrak{p}}_h(s',y'|s,y,a)V_{h+1}^{\psi^*}(s',y';\mathfrak{p},\mathfrak{r})
$$

$$
= \sum_{(s',y')\in\mathcal{S}\times\mathcal{Y}_{h+1}} \left( \mathfrak{p}_h(s',y'|s,y,a) - \widehat{\mathfrak{p}}_h(s',y'|s,y,a) \right) V_{h+1}^{\psi^*}(s',y';\mathfrak{p},\mathfrak{r})
$$

$$
+ \sum_{(s',y')\in\mathcal{S}\times\mathcal{Y}_{h+1}} \widehat{\mathfrak{p}}_h(s',y'|s,y,a) \left( V_{h+1}^{\psi^*}(s',y';\mathfrak{p},\mathfrak{r}) - V_{h+1}^{\psi^*}(s',y';\widehat{\mathfrak{p}},\mathfrak{r}) \right)
$$

$$
\overset{(4)}{=} \sum_{(s',y')\in\mathcal{S}\times\mathcal{Y}_{h+1}} \Big( p_h(s'|s,a)\mathbb{1}\{y+\overline{r}_h(s,a)=y'\}
$$

$$
- \widehat{p}_h(s'|s,a)\mathbb{1}\{y+\overline{r}_h(s,a)=y'\} \Big) V_{h+1}^{\psi^*}(s',y';\mathfrak{p},\mathfrak{r})
$$

$$
+ \sum_{(s',y')\in\mathcal{S}\times\mathcal{Y}_{h+1}} \widehat{\mathfrak{p}}_h(s',y'|s,y,a) \left( V_{h+1}^{\psi^*}(s',y';\mathfrak{p},\mathfrak{r}) - V_{h+1}^{\psi^*}(s',y';\widehat{\mathfrak{p}},\mathfrak{r}) \right)
$$

$$
\overset{(5)}{=} \sum_{s'\in\mathcal{S}} \left( p_h(s'|s,a) - \widehat{p}_h(s'|s,a) \right) \sum_{y'\in\mathcal{Y}_{h+1}} \mathbb{1}\{y+\overline{r}_h(s,a)=y'\}V_{h+1}^{\psi^*}(s',y';\mathfrak{p},\mathfrak{r})
$$

$$
+ \sum_{(s',y')\in\mathcal{S}\times\mathcal{Y}_{h+1}} \widehat{\mathfrak{p}}_h(s',y'|s,y,a) \left( V_{h+1}^{\psi^*}(s',y';\mathfrak{p},\mathfrak{r}) - V_{h+1}^{\psi^*}(s';\widehat{\mathfrak{p}},\mathfrak{r}) \right)
$$

$$
\overset{(6)}{=} \sum_{s'\in\mathcal{S}} \left( p_h(s'|s,a) - \widehat{p}_h(s'|s,a) \right) V_{h+1}^{\psi^*}(s',y+\overline{r}_h(s,a);\mathfrak{p},\mathfrak{r})
$$

$$
+ \sum_{(s',y')\in\mathcal{S}\times\mathcal{Y}_{h+1}} \widehat{\mathfrak{p}}_h(s',y'|s,y,a) \left( V_{h+1}^{\psi^*}(s',y';\mathfrak{p},\mathfrak{r}) - V_{h+1}^{\psi^*}(s',y';\widehat{\mathfrak{p}},\mathfrak{r}) \right)
$$

$$
= \sum_{s'\in\mathcal{S}} \left( p_h(s'|s,a) - \widehat{p}_h(s'|s,a) \right) V_{h+1}^{\psi^*}(s',y+\overline{r}_h(s,a);\mathfrak{p},\mathfrak{r})
$$

$$
+ \sum_{(s',y')\in\mathcal{S}\times\mathcal{Y}_{h+1}} \widehat{\mathfrak{p}}_h(s',y'|s,y,a)
$$

$$
\cdot \left( Q_{h+1}^{\psi^*}(s',y',\psi_{h+1}^*(s',y');\mathfrak{p},\mathfrak{r}) - Q_{h+1}^{\psi^*}(s',y',\psi_{h+1}^*(s',y');\widehat{\mathfrak{p}},\mathfrak{r}) \right),
$$

where at (1) we have used that $\widehat{\psi}^*$ is the optimal policy in $\widehat{\mathfrak{p}},\mathfrak{r}$, and thus $Q_h^{\psi^*}(s,a;\widehat{\mathfrak{p}},\mathfrak{r}) \leqslant Q_h^{\widehat{\psi}^*}(s,a;\widehat{\mathfrak{p}},\mathfrak{r})$. At (2) we apply the Bellman equation, at (3) we add and subtract the expected under $\widehat{\mathfrak{p}}$ optimal value function under $\mathfrak{p}$, at (4) we use the definition of transition model $\mathfrak{p},\widehat{\mathfrak{p}}$, at (5) we split the summations, at (6) we recognize that the indicator function takes on value 1 only when $y+\overline{r}_h(s,a)=y'$. Finally, we unfold the recursion to obtain the result.

Concerning the second equation, for any $h\in[\![H]\!]$, $(s,y,a)\in\mathcal{S}\times\mathcal{Y}_h\times\mathcal{A}$, we can write:

$$
Q_h^*(s,y,a;\mathfrak{p},\mathfrak{r}) - Q_h^*(s,y,a;\widehat{\mathfrak{p}},\mathfrak{r})
$$

$$
= Q_h^{\psi^*}(s,y,a;\mathfrak{p},\mathfrak{r}) - Q_h^{\widehat{\psi}^*}(s,y,a;\widehat{\mathfrak{p}},\mathfrak{r})
$$

$$\overset{(7)}{=} \mathfrak{r}_h(s,y,a) + \sum_{(s',y')\in\mathcal{S}\times\mathcal{Y}_{h+1}} \mathfrak{p}_h(s',y'|s,y,a)V_{h+1}^{\psi^*}(s',y';\mathfrak{p},\mathfrak{r})$$

$$- \left( \mathfrak{r}_h(s,y,a) + \sum_{(s',y')\in\mathcal{S}\times\mathcal{Y}_{h+1}} \widehat{\mathfrak{p}}_h(s',y'|s,y,a)V_{h+1}^{\widehat{\psi}^*}(s',y';\widehat{\mathfrak{p}},\mathfrak{r}) \right)$$

$$\overset{(8)}{=} \sum_{(s',y')\in\mathcal{S}\times\mathcal{Y}_{h+1}} \mathfrak{p}_h(s',y'|s,y,a)V_{h+1}^{\psi^*}(s',y';\mathfrak{p},\mathfrak{r})$$

$$- \sum_{(s',y')\in\mathcal{S}\times\mathcal{Y}_{h+1}} \widehat{\mathfrak{p}}_h(s',y'|s,y,a)V_{h+1}^{\widehat{\psi}^*}(s',y';\widehat{\mathfrak{p}},\mathfrak{r})$$

$$\pm \sum_{(s',y')\in\mathcal{S}\times\mathcal{Y}_{h+1}} \widehat{\mathfrak{p}}_h(s',y'|s,y,a)V_{h+1}^{\psi^*}(s',y';\mathfrak{p},\mathfrak{r})$$

$$= \sum_{(s',y')\in\mathcal{S}\times\mathcal{Y}_{h+1}} \left( \mathfrak{p}_h(s',y'|s,y,a) - \widehat{\mathfrak{p}}_h(s',y'|s,y,a) \right)V_{h+1}^{\psi^*}(s',y';\mathfrak{p},\mathfrak{r})$$

$$+ \sum_{(s',y')\in\mathcal{S}\times\mathcal{Y}_{h+1}} \widehat{\mathfrak{p}}_h(s',y'|s,y,a)\left( V_{h+1}^{\psi^*}(s',y';\mathfrak{p},\mathfrak{r}) - V_{h+1}^{\widehat{\psi}^*}(s',y';\widehat{\mathfrak{p}},\mathfrak{r}) \right)$$

$$= \sum_{(s',y')\in\mathcal{S}\times\mathcal{Y}_{h+1}} \Big( p_h(s'|s,a)\mathbb{1}\{y+\overline{r}_h(s,a)=y'\}$$

$$- \widehat{p}_h(s'|s,a)\mathbb{1}\{y+\overline{r}_h(s,a)=y'\} \Big)V_{h+1}^{\psi^*}(s',y';\mathfrak{p},\mathfrak{r})$$

$$+ \sum_{(s',y')\in\mathcal{S}\times\mathcal{Y}_{h+1}} \widehat{\mathfrak{p}}_h(s',y'|s,y,a)\left( V_{h+1}^{\psi^*}(s',y';\mathfrak{p},\mathfrak{r}) - V_{h+1}^{\widehat{\psi}^*}(s',y';\widehat{\mathfrak{p}},\mathfrak{r}) \right)$$

$$= \sum_{s'\in\mathcal{S}} \left( p_h(s'|s,a) - \widehat{p}_h(s'|s,a) \right)\sum_{y'\in\mathcal{Y}_{h+1}}\mathbb{1}\{y+\overline{r}_h(s,a)=y'\}V_{h+1}^{\psi^*}(s',y';\mathfrak{p},\mathfrak{r})$$

$$+ \sum_{(s',y')\in\mathcal{S}\times\mathcal{Y}_{h+1}} \widehat{\mathfrak{p}}_h(s',y'|s,y,a)\left( V_{h+1}^{\psi^*}(s',y';\mathfrak{p},\mathfrak{r}) - V_{h+1}^{\widehat{\psi}^*}(s',y';\widehat{\mathfrak{p}},\mathfrak{r}) \right)$$

$$= \sum_{s'\in\mathcal{S}} \left( p_h(s'|s,a) - \widehat{p}_h(s'|s,a) \right)V_{h+1}^{\psi^*}(s',y+\overline{r}_h(s,a);\mathfrak{p},\mathfrak{r})$$

$$+ \sum_{(s',y')\in\mathcal{S}\times\mathcal{Y}_{h+1}} \widehat{\mathfrak{p}}_h(s',y'|s,y,a)\left( V_{h+1}^{\psi^*}(s',y';\mathfrak{p},\mathfrak{r}) - V_{h+1}^{\widehat{\psi}^*}(s',y';\widehat{\mathfrak{p}},\mathfrak{r}) \right)$$

$$\overset{(9)}{\geqslant} \sum_{s'\in\mathcal{S}} \left( p_h(s'|s,a) - \widehat{p}_h(s'|s,a) \right)V_{h+1}^{\psi^*}(s',y+\overline{r}_h(s,a);\mathfrak{p},\mathfrak{r})$$

$$+ \sum_{(s',y')\in\mathcal{S}\times\mathcal{Y}_{h+1}} \widehat{\mathfrak{p}}_h(s',y'|s,y,a)$$

$$\cdot \left( Q_{h+1}^{\psi^*}(s',y',\widehat{\psi}_{h+1}^*(s',y');\mathfrak{p},\mathfrak{r}) - Q_{h+1}^{\widehat{\psi}^*}(s',y',\widehat{\psi}_{h+1}^*(s',y');\widehat{\mathfrak{p}},\mathfrak{r}) \right),$$

where at (7) we have applied the Bellman equation, at (8) we have added and subtracted a term, and at (9) we have used that $V_{h+1}^{\psi^*}(s',y';\mathfrak{p},\mathfrak{r}) = Q_{h+1}^{\psi^*}(s',y',\psi_{h+1}^*(s',y');\mathfrak{p},\mathfrak{r}) \geqslant Q_{h+1}^{\psi^*}(s',y',\widehat{\psi}_{h+1}^*(s',y');\mathfrak{p},\mathfrak{r})$, since $\psi_{h+1}^*(s',y')$ is the optimal action under $\mathfrak{p},\mathfrak{r}$, and so, it cannot be worse than action $\widehat{\psi}_{h+1}^*(s',y')$. By unfolding the recursion, we obtain the result. $\square$

**Lemma E.8.** *For any $\delta \in (0,1)$, w.p. at least $1-\delta$, it holds that:*

$$\max_{h\in[\![H]\!],(s,y)\in\mathcal{S}\times\mathcal{Y}_h} |V_h^*(s,y;\mathfrak{p},\mathfrak{r}) - V_h^{\psi^*}(s,y;\widehat{\mathfrak{p}},\mathfrak{r})| \leqslant cH^2\sqrt{\frac{\log\frac{2SAHd}{\delta}}{n}},$$

$$\max_{h\in[\![H]\!],(s,y)\in\mathcal{S}\times\mathcal{Y}_h} |V_h^*(s,y;\mathfrak{p},\mathfrak{r}) - V_h^*(s,y;\widehat{\mathfrak{p}},\mathfrak{r})| \leqslant cH^2\sqrt{\frac{\log\frac{2SAHd}{\delta}}{n}}.$$

*where c is some positive constant.*

*Proof.* First, we observe that, for any $h \in [\![H]\!], (s, y) \in \mathcal{S} \times \mathcal{Y}_h$, by following passages similar to those in the proof of Lemma E.7:

$$|V_h^*(s, y; \mathfrak{p}, \mathfrak{r}) - V_h^{\psi^*}(s, y; \widehat{\mathfrak{p}}, \mathfrak{r})|$$

$$= |Q_h^{\psi^*}(s, y, \psi_h^*(s, y); \mathfrak{p}, \mathfrak{r}) - Q_h^{\psi^*}(s, y, \psi_h^*(s, y); \widehat{\mathfrak{p}}, \mathfrak{r})|$$

$$= \Big| \mathfrak{r}_h(s, y, \psi_h^*(s, y)) + \sum_{(s', y') \in \mathcal{S} \times \mathcal{Y}_{h+1}} \mathfrak{p}_h(s', y'|s, y, \psi_h^*(s, y)) V_{h+1}^{\psi^*}(s', y'; \mathfrak{p}, \mathfrak{r})$$

$$- \Big( \mathfrak{r}_h(s, y, \psi_h^*(s, y)) + \sum_{(s', y') \in \mathcal{S} \times \mathcal{Y}_{h+1}} \widehat{\mathfrak{p}}_h(s', y'|s, y, \psi_h^*(s, y)) V_{h+1}^{\psi^*}(s', y'; \widehat{\mathfrak{p}}, \mathfrak{r}) \Big) \Big|$$

$$= \Big| \sum_{(s', y') \in \mathcal{S} \times \mathcal{Y}_{h+1}} \mathfrak{p}_h(s', y'|s, y, \psi_h^*(s, y)) V_{h+1}^{\psi^*}(s', y'; \mathfrak{p}, \mathfrak{r})$$

$$- \sum_{(s', y') \in \mathcal{S} \times \mathcal{Y}_{h+1}} \widehat{\mathfrak{p}}_h(s', y'|s, y, \psi_h^*(s, y)) V_{h+1}^{\psi^*}(s', y'; \widehat{\mathfrak{p}}, \mathfrak{r})$$

$$\pm \sum_{(s', y') \in \mathcal{S} \times \mathcal{Y}_{h+1}} \widehat{\mathfrak{p}}_h(s', y'|s, y, \psi_h^*(s, y)) V_{h+1}^{\psi^*}(s', y'; \mathfrak{p}, \mathfrak{r}) \Big|$$

$$= \Big| \sum_{(s', y') \in \mathcal{S} \times \mathcal{Y}_{h+1}} \Big( \mathfrak{p}_h(s', y'|s, y, \psi_h^*(s, y)) - \widehat{\mathfrak{p}}_h(s', y'|s, y, \psi_h^*(s, y)) \Big) V_{h+1}^{\psi^*}(s', y'; \mathfrak{p}, \mathfrak{r})$$

$$+ \sum_{(s', y') \in \mathcal{S} \times \mathcal{Y}_{h+1}} \widehat{\mathfrak{p}}_h(s', y'|s, y, \psi_h^*(s, y)) \Big( V_{h+1}^{\psi^*}(s', y'; \mathfrak{p}, \mathfrak{r}) - V_{h+1}^{\psi^*}(s', y'; \widehat{\mathfrak{p}}, \mathfrak{r}) \Big) \Big|$$

$$= \Big| \sum_{s' \in \mathcal{S}} \Big( p_h(s'|s, \psi_h^*(s, y)) - \widehat{p}_h(s'|s, \psi_h^*(s, y)) \Big) V_{h+1}^{\psi^*}(s', y + \overline{r}_h(s, \psi_h^*(s, y)); \mathfrak{p}, \mathfrak{r})$$

$$+ \sum_{(s', y') \in \mathcal{S} \times \mathcal{Y}_{h+1}} \widehat{\mathfrak{p}}_h(s', y'|s, y, \psi_h^*(s, y)) \Big( V_{h+1}^{\psi^*}(s', y'; \mathfrak{p}, \mathfrak{r}) - V_{h+1}^{\psi^*}(s', y'; \widehat{\mathfrak{p}}, \mathfrak{r}) \Big) \Big|$$

$$= \dots$$

$$= \Big| \mathbb{E}_{\widehat{\mathfrak{p}}, \mathfrak{r}, \psi^*} \Big[ \sum_{h'=h}^{H} \sum_{s' \in \mathcal{S}} \Big( p_{h'}(s'|s_{h'}, a_{h'}) - \widehat{p}_{h'}(s'|s_{h'}, a_{h'}) \Big) V_{h'+1}^{\psi^*}(s', y_{h'+1}; \mathfrak{p}, \mathfrak{r}) $$

$$\Big| s_h = s, y_h = y \Big] \Big|$$

$$\overset{(1)}{\leq} \mathbb{E}_{\widehat{\mathfrak{p}}, \mathfrak{r}, \psi^*} \Big[ \sum_{h'=h}^{H} \Big| \sum_{s' \in \mathcal{S}} \Big( p_{h'}(s'|s_{h'}, a_{h'}) - \widehat{p}_{h'}(s'|s_{h'}, a_{h'}) \Big) V_{h'+1}^{\psi^*}(s', y_{h'+1}; \mathfrak{p}, \mathfrak{r}) \Big|$$

$$\Big| s_h = s, y_h = y \Big],$$

where at (1) we have brought the absolute value inside the expectation.

Similarly, for the other term, for any $h \in [\![H]\!], (s, y) \in \mathcal{S} \times \mathcal{Y}_h$, we can write:

$$|V_h^*(s, y; \mathfrak{p}, \mathfrak{r}) - V_h^*(s, y; \widehat{\mathfrak{p}}, \mathfrak{r})|$$

$$= |V_h^{\psi^*}(s, y; \mathfrak{p}, \mathfrak{r}) - V_h^{\widehat{\psi}^*}(s, y; \widehat{\mathfrak{p}}, \mathfrak{r})|$$

$$\overset{(2)}{=} |\max_{a \in \mathcal{A}} Q_h^{\psi^*}(s, y, a; \mathfrak{p}, \mathfrak{r}) - \max_{a \in \mathcal{A}} Q_h^{\widehat{\psi}^*}(s, y, a; \widehat{\mathfrak{p}}, \mathfrak{r})|$$

$$\overset{(3)}{\leq} \max_{a \in \mathcal{A}} |Q_h^{\psi^*}(s, y, a; \mathfrak{p}, \mathfrak{r}) - Q_h^{\widehat{\psi}^*}(s, y, a; \widehat{\mathfrak{p}}, \mathfrak{r})|$$

$$= \max_{a \in \mathcal{A}} \Big| \mathfrak{r}_h(s, y, a) + \sum_{(s', y') \in \mathcal{S} \times \mathcal{Y}_{h+1}} \mathfrak{p}_h(s', y'|s, y, a) V_{h+1}^{\psi^*}(s', y'; \mathfrak{p}, \mathfrak{r})$$

$$- \Big( \mathfrak{r}_h(s, y, a) + \sum_{(s', y') \in \mathcal{S} \times \mathcal{Y}_{h+1}} \widehat{\mathfrak{p}}_h(s', y'|s, y, a) V_{h+1}^{\widehat{\psi}^*}(s', y'; \widehat{\mathfrak{p}}, \mathfrak{r}) \Big) \Big|$$

$$
\begin{aligned}
&= \max_{a \in \mathcal{A}} \Big| \sum_{(s',y') \in \mathcal{S} \times \mathcal{Y}_{h+1}} \mathfrak{p}_h(s',y'|s,y,a) V_{h+1}^{\psi^*}(s',y';\mathfrak{p},\mathfrak{r}) \\
&\qquad - \sum_{(s',y') \in \mathcal{S} \times \mathcal{Y}_{h+1}} \widehat{\mathfrak{p}}_h(s',y'|s,y,a) V_{h+1}^{\widehat{\psi}^*}(s',y';\widehat{\mathfrak{p}},\mathfrak{r}) \\
&\qquad \pm \sum_{(s',y') \in \mathcal{S} \times \mathcal{Y}_{h+1}} \widehat{\mathfrak{p}}_h(s',y'|s,y,a) V_{h+1}^{\psi^*}(s',y';\mathfrak{p},\mathfrak{r}) \Big| \\
&= \max_{a \in \mathcal{A}} \Big| \sum_{(s',y') \in \mathcal{S} \times \mathcal{Y}_{h+1}} \Big( \mathfrak{p}_h(s',y'|s,y,a) - \widehat{\mathfrak{p}}_h(s',y'|s,y,a) \Big) V_{h+1}^{\psi^*}(s',y';\mathfrak{p},\mathfrak{r}) \\
&\qquad + \sum_{(s',y') \in \mathcal{S} \times \mathcal{Y}_{h+1}} \widehat{\mathfrak{p}}_h(s',y'|s,y,a) \Big( V_{h+1}^{\psi^*}(s',y';\mathfrak{p},\mathfrak{r}) - V_{h+1}^{\widehat{\psi}^*}(s',y';\widehat{\mathfrak{p}},\mathfrak{r}) \Big) \Big| \\
&\overset{(4)}{\leqslant} \Big| \sum_{(s',y') \in \mathcal{S} \times \mathcal{Y}_{h+1}} \Big( \mathfrak{p}_h(s',y'|s,y,\overline{a}) - \widehat{\mathfrak{p}}_h(s',y'|s,y,\overline{a}) \Big) V_{h+1}^{\psi^*}(s',y';\mathfrak{p},\mathfrak{r}) \Big| \\
&\qquad + \Big| \sum_{(s',y') \in \mathcal{S} \times \mathcal{Y}_{h+1}} \widehat{\mathfrak{p}}_h(s',y'|s,y,\overline{a}) \Big( V_{h+1}^{\psi^*}(s',y';\mathfrak{p},\mathfrak{r}) - V_{h+1}^{\widehat{\psi}^*}(s',y';\widehat{\mathfrak{p}},\mathfrak{r}) \Big) \Big| \\
&= \Big| \sum_{s' \in \mathcal{S}} \Big( p_h(s'|s,\overline{a}) - \widehat{p}_h(s'|s,\overline{a}) \Big) V_{h+1}^{\psi^*}(s',y+\Pi_{\mathcal{R}}[r_h(s,\overline{a})];\mathfrak{p},\mathfrak{r}) \Big| \\
&\qquad + \Big| \sum_{(s',y') \in \mathcal{S} \times \mathcal{Y}_{h+1}} \widehat{\mathfrak{p}}_h(s',y'|s,y,\overline{a}) \Big( V_{h+1}^{\psi^*}(s',y';\mathfrak{p},\mathfrak{r}) - V_{h+1}^{\widehat{\psi}^*}(s',y';\widehat{\mathfrak{p}},\mathfrak{r}) \Big) \Big| \\
&\leqslant \dots \\
&\overset{(5)}{\leqslant} \mathop{\mathbb{E}}_{\widehat{\mathfrak{p}},\mathfrak{r},\overline{\psi}} \Bigg[ \sum_{h'=h}^{H} \Big| \sum_{s' \in \mathcal{S}} \Big( p_{h'}(s'|s_{h'},a_{h'}) - \widehat{p}_{h'}(s'|s_{h'},a_{h'}) \Big) V_{h'+1}^{\psi^*}(s',y_{h'+1};\mathfrak{p},\mathfrak{r}) \Big| \\
&\qquad \Big| s_h = s, y_h = y \Bigg],
\end{aligned}
$$

where at (2) we have applied the Bellman optimality equation, at (3) we have upper bounded the difference of maxima with the maximum of the difference, at (4) we denote the maximal action by $\overline{a}$, and we apply triangle inequality; at (5) we have unfolded the recursion and called $\overline{\psi}$ the resulting policy.

Now, for some $\epsilon \in (0,1)$, let us denote by $\mathcal{E}$ the event defined as:

$$
\mathcal{E} := \Bigg\{ \forall h \in [\![H]\!], (s,y,a) \in \mathcal{S} \times \mathcal{Y}_h \times \mathcal{A} : \\
\Big| \sum_{s' \in \mathcal{S}} \Big( p_h(s'|s,a) - \widehat{p}_h(s'|s,a) \Big) V_{h+1}^{\psi^*}(s',y+\overline{r}_h(s,a);\mathfrak{p},\mathfrak{r}) \Big| \leqslant \epsilon \Bigg\}
$$

We can write:

$$
\begin{aligned}
\mathbb{P}(\mathcal{E}^{\complement}) &= \mathbb{P}\Bigg( \exists h \in [\![H]\!], (s,y,a) \in \mathcal{S} \times \mathcal{Y}_h \times \mathcal{A} : \\
&\qquad \Big| \sum_{s' \in \mathcal{S}} \Big( p_h(s'|s,a) - \widehat{p}_h(s'|s,a) \Big) V_{h+1}^{\psi^*}(s',y+\overline{r}_h(s,a);\mathfrak{p},\mathfrak{r}) \Big| > \epsilon \Bigg) \\
&\overset{(6)}{\leqslant} \sum_{h \in [\![H]\!], (s,y,a) \in \mathcal{S} \times \mathcal{Y}_h \times \mathcal{A}} \\
&\qquad \mathbb{P}\Bigg( \Big| \sum_{s' \in \mathcal{S}} \Big( p_h(s'|s,a) - \widehat{p}_h(s'|s,a) \Big) V_{h+1}^{\psi^*}(s',y+\overline{r}_h(s,a);\mathfrak{p},\mathfrak{r}) \Big| > \epsilon \Bigg) \\
&\overset{(7)}{\leqslant} \sum_{h \in [\![H]\!], (s,y,a) \in \mathcal{S} \times \mathcal{Y}_h \times \mathcal{A}} 2e^{\frac{-2n\epsilon^2}{H^2}}
\end{aligned}
$$

$$= 2SAHde^{\frac{-2n\epsilon^2}{H^2}},$$

where at (6) we have applied a union bound over all tuples $h \in [\![H]\!], (s, y, a) \in \mathcal{S} \times \mathcal{Y}_h \times \mathcal{A}$, and at (7) we have applied Hoeffding's inequality, by recalling that we collect $n$ samples (see Algorithm 3) for any $(s, a, h) \in \mathcal{S} \times \mathcal{A} \times [\![H]\!]$ triple, and that vector $V_{h+1}^{\psi^*}(\cdot, y + \overline{r}_h(s, a); \mathfrak{p}, \mathfrak{r})$ bounded by $[0, H]$ is independent of the randomness in $\widehat{p}_h(\cdot|s, a)$. It should be remarked that our collection of samples depends only on $\mathcal{S} \times \mathcal{A} \times [\![H]\!]$, and not on $\mathcal{Y}_h$; such term enters the expression only through the union bound, because we have to apply Hoeffding's inequality for all the value functions considered, which are as many as $|\mathcal{Y}_h|$. Note that we use $d = |\mathcal{Y}_{H+1}|$ since it is the largest $|\mathcal{Y}_h|$ among $h \in [\![H+1]\!]$.

This probability is at most $\delta$ if:

$$2SAHde^{\frac{-2n\epsilon^2}{H^2}} \leqslant \delta \iff \epsilon \geqslant H\sqrt{\frac{\log \frac{2SAHd}{\delta}}{2n}}.$$

By plugging into the previous expressions, we obtain that, w.p. $1 - \delta$:

$$|V_h^*(s, y; \mathfrak{p}, \mathfrak{r}) - V_h^{\psi^*}(s, y; \widehat{\mathfrak{p}}, \mathfrak{r})|$$

$$\leqslant \mathop{\mathbb{E}}_{\widehat{\mathfrak{p}}, \mathfrak{r}, \psi^*} \Bigg[ \sum_{h'=h}^{H} \Bigg| \sum_{s' \in \mathcal{S}} \Big( p_{h'}(s'|s_{h'}, a_{h'}) - \widehat{p}_{h'}(s'|s_{h'}, a_{h'}) \Big) V_{h'+1}^{\psi^*}(s', y_{h'+1}; \mathfrak{p}, \mathfrak{r}) \Bigg|$$

$$\Bigg| s_h = s, y_h = y \Bigg]$$

$$\leqslant \mathop{\mathbb{E}}_{\widehat{\mathfrak{p}}, \mathfrak{r}, \psi^*} \Bigg[ \sum_{h'=h}^{H} H\sqrt{\frac{\log \frac{2SAHd}{\delta}}{2n}} \Bigg| s_h = s, y_h = y \Bigg]$$

$$= H^2\sqrt{\frac{\log \frac{2SAHd}{\delta}}{2n}},$$

and also:

$$|V_h^*(s, y; \mathfrak{p}, \mathfrak{r}) - V_h^*(s, y; \widehat{\mathfrak{p}}, \mathfrak{r})|$$

$$\leqslant \mathop{\mathbb{E}}_{\widehat{\mathfrak{p}}, \mathfrak{r}, \overline{\psi}} \Bigg[ \sum_{h'=h}^{H} \Bigg| \sum_{s' \in \mathcal{S}} \Big( p_{h'}(s'|s_{h'}, a_{h'}) - \widehat{p}_{h'}(s'|s_{h'}, a_{h'}) \Big) V_{h'+1}^{\psi^*}(s', y_{h'+1}; \mathfrak{p}, \mathfrak{r}) \Bigg|$$

$$\Bigg| s_h = s, y_h = y \Bigg]$$

$$\leqslant \mathop{\mathbb{E}}_{\widehat{\mathfrak{p}}, \mathfrak{r}, \overline{\psi}} \Bigg[ \sum_{h'=h}^{H} H\sqrt{\frac{\log \frac{2SAHd}{\delta}}{2n}} \Bigg| s_h = s, y_h = y \Bigg]$$

$$= H^2\sqrt{\frac{\log \frac{2SAHd}{\delta}}{2n}}.$$

This concludes the proof.

$\square$

**Lemma E.9.** *For any $\delta \in (0, 1)$, w.p. at least $1 - \delta$, it holds that, for all $h \in [\![H]\!], (s, y, a) \in \mathcal{S} \times \mathcal{Y}_h \times \mathcal{A}$:*

$$\sqrt{\mathbb{V}_{s' \sim p_h(\cdot|s,a)}[V_{h+1}^*(s', y + \overline{r}_h(s, a); \mathfrak{p}, \mathfrak{r})]} \leqslant$$

$$\sqrt{\mathbb{V}_{s' \sim \widehat{p}_h(\cdot|s,a)}[V_{h+1}^{\psi^*}(s', y + \overline{r}_h(s, a); \widehat{\mathfrak{p}}, \mathfrak{r})]} + b_1,$$

$$\sqrt{\mathbb{V}_{s' \sim p_h(\cdot|s,a)}[V_{h+1}^*(s', y + \overline{r}_h(s, a); \mathfrak{p}, \mathfrak{r})]} \leqslant$$

$$\sqrt{\mathbb{V}_{s' \sim \widehat{p}_h(\cdot|s,a)}[V_{h+1}^*(s', y + \overline{r}_h(s, a); \widehat{\mathfrak{p}}, \mathfrak{r})]} + b_1,$$

where $b_1$ is defined as:

$$b_1 := cH\left(\frac{\log\frac{4SAHY}{\delta}}{n}\right)^{1/4} + c'H^2\sqrt{\frac{\log\frac{4SAHY}{\delta}}{n}},$$

for some positive constants $c, c'$.

*Proof.* In the following, we will use $\overline{y}$ as a label for $y + \overline{r}_h(s, a)$. We begin with the first expression. We can write, for any $h \in [\![H]\!], (s, y, a) \in \mathcal{S} \times \mathcal{Y}_h \times \mathcal{A}$:

$$\mathbb{V}_{s' \sim p_h(\cdot|s,a)}[V^*_{h+1}(s', \overline{y}; \mathfrak{p}, \mathfrak{r})]$$

$$= \mathbb{V}_{s' \sim p_h(\cdot|s,a)}[V^*_{h+1}(s', \overline{y}; \mathfrak{p}, \mathfrak{r})] \pm \mathbb{V}_{s' \sim \widehat{p}_h(\cdot|s,a)}[V^*_{h+1}(s', \overline{y}; \mathfrak{p}, \mathfrak{r})]$$

$$= \left(\mathbb{V}_{s' \sim p_h(\cdot|s,a)}[V^*_{h+1}(s', \overline{y}; \mathfrak{p}, \mathfrak{r})] - \mathbb{V}_{s' \sim \widehat{p}_h(\cdot|s,a)}[V^*_{h+1}(s', \overline{y}; \mathfrak{p}, \mathfrak{r})]\right)$$

$$+ \mathbb{V}_{s' \sim \widehat{p}_h(\cdot|s,a)}[V^*_{h+1}(s', \overline{y}; \mathfrak{p}, \mathfrak{r})]$$

$$\overset{(1)}{=} \sum_{s' \in \mathcal{S}} \left(p_h(s'|s,a) - \widehat{p}_h(s'|s,a)\right) V^{*2}_{h+1}(s', \overline{y}; \mathfrak{p}, \mathfrak{r})$$

$$- \left[\left(\sum_{s' \in \mathcal{S}} p_h(s'|s,a) V^*_{h+1}(s', \overline{y}; \mathfrak{p}, \mathfrak{r})\right)^2 \right.$$

$$\left. - \left(\sum_{s' \in \mathcal{S}} \widehat{p}_h(s'|s,a) V^*_{h+1}(s', \overline{y}; \mathfrak{p}, \mathfrak{r})\right)^2\right]$$

$$+ \mathbb{V}_{s' \sim \widehat{p}_h(\cdot|s,a)}[V^*_{h+1}(s', \overline{y}; \mathfrak{p}, \mathfrak{r}) \pm V^{\psi^*}_{h+1}(s', \overline{y}; \widehat{\mathfrak{p}}, \mathfrak{r})]$$

$$\overset{(2)}{=} \sum_{s' \in \mathcal{S}} \left(p_h(s'|s,a) - \widehat{p}_h(s'|s,a)\right) V^{*2}_{h+1}(s', \overline{y}; \mathfrak{p}, \mathfrak{r})$$

$$- \left[\left(\sum_{s' \in \mathcal{S}} p_h(s'|s,a) V^*_{h+1}(s', \overline{y}; \mathfrak{p}, \mathfrak{r})\right)^2 \right.$$

$$\left. - \left(\sum_{s' \in \mathcal{S}} \widehat{p}_h(s'|s,a) V^*_{h+1}(s', \overline{y}; \mathfrak{p}, \mathfrak{r})\right)^2\right]$$

$$+ \mathbb{V}_{s' \sim \widehat{p}_h(\cdot|s,a)}[V^*_{h+1}(s', \overline{y}; \mathfrak{p}, \mathfrak{r}) - V^{\psi^*}_{h+1}(s', \overline{y}; \widehat{\mathfrak{p}}, \mathfrak{r})]$$

$$+ \mathbb{V}_{s' \sim \widehat{p}_h(\cdot|s,a)}[V^{\psi^*}_{h+1}(s', \overline{y}; \widehat{\mathfrak{p}}, \mathfrak{r})]$$

$$+ 2\text{Cov}_{s' \sim \widehat{p}_h(\cdot|s,a)}[V^*_{h+1}(s', \overline{y}; \mathfrak{p}, \mathfrak{r}) - V^{\psi^*}_{h+1}(s', \overline{y}; \widehat{\mathfrak{p}}, \mathfrak{r}),$$

$$V^{\psi^*}_{h+1}(s', \overline{y}; \widehat{\mathfrak{p}}, \mathfrak{r})]$$

$$\overset{(3)}{\leq} \sum_{s' \in \mathcal{S}} \left(p_h(s'|s,a) - \widehat{p}_h(s'|s,a)\right) V^{*2}_{h+1}(s', \overline{y}; \mathfrak{p}, \mathfrak{r})$$

$$- \left[\left(\sum_{s' \in \mathcal{S}} p_h(s'|s,a) V^*_{h+1}(s', \overline{y}; \mathfrak{p}, \mathfrak{r})\right)^2 \right.$$

$$\left. - \left(\sum_{s' \in \mathcal{S}} \widehat{p}_h(s'|s,a) V^*_{h+1}(s', \overline{y}; \mathfrak{p}, \mathfrak{r})\right)^2\right]$$

$$+ \mathbb{V}_{s' \sim \widehat{p}_h(\cdot|s,a)}[V^*_{h+1}(s', \overline{y}; \mathfrak{p}, \mathfrak{r}) - V^{\psi^*}_{h+1}(s', \overline{y}; \widehat{\mathfrak{p}}, \mathfrak{r})]$$

$$+ \mathbb{V}_{s' \sim \widehat{p}_h(\cdot|s,a)}[V^{\psi^*}_{h+1}(s', \overline{y}; \widehat{\mathfrak{p}}, \mathfrak{r})]$$

$$+ 2\left(\mathbb{V}_{s' \sim \widehat{p}_h(\cdot|s,a)}[V^*_{h+1}(s', \overline{y}; \mathfrak{p}, \mathfrak{r}) - V^{\psi^*}_{h+1}(s', \overline{y}; \widehat{\mathfrak{p}}, \mathfrak{r})]\right.$$

$$\left. \cdot \mathbb{V}_{s' \sim \widehat{p}_h(\cdot|s,a)}[V^{\psi^*}_{h+1}(s', \overline{y}; \widehat{\mathfrak{p}}, \mathfrak{r})]\right)^{1/2}$$

$$= \sum_{s' \in \mathcal{S}} \left(p_h(s'|s,a) - \widehat{p}_h(s'|s,a)\right) V^{*2}_{h+1}(s', \overline{y}; \mathfrak{p}, \mathfrak{r})$$

$$
- \Big[ \Big( \sum_{s' \in \mathcal{S}} p_h(s'|s,a) V_{h+1}^*(s', \overline{y}; \mathfrak{p}, \mathfrak{r}) \Big)^2
$$

$$
- \Big( \sum_{s' \in \mathcal{S}} \widehat{p}_h(s'|s,a) V_{h+1}^*(s', \overline{y}; \mathfrak{p}, \mathfrak{r}) \Big)^2 \Big]
$$

$$
+ \Big[ \sqrt{\mathbb{V}_{s' \sim \widehat{p}_h(\cdot|s,a)}[V_{h+1}^*(s', \overline{y}; \mathfrak{p}, \mathfrak{r}) - V_{h+1}^{\psi^*}(s', \overline{y}; \widehat{\mathfrak{p}}, \mathfrak{r})]}
$$

$$
+ \sqrt{\mathbb{V}_{s' \sim \widehat{p}_h(\cdot|s,a)}[V_{h+1}^{\psi^*}(s', \overline{y}; \widehat{\mathfrak{p}}, \mathfrak{r})]} \Big]^2,
$$

where at (1) we have used the common formula for the variance $\mathbb{V}[X] = \mathbb{E}[X^2] - \mathbb{E}[X]^2$, at (2) we have decomposed the variance of a sum as $\mathbb{V}[X + Y] = \mathbb{V}[X] + \mathbb{V}[Y] + 2\mathrm{Cov}[X, Y]$, at (3) we have applied Cauchy-Schwarz's inequality to bound the covariance with the product of the variances $|\mathrm{Cov}[X, Y]| \leqslant \sqrt{\mathbb{V}[X]\mathbb{V}[Y]}$.

Next, observe that:

$$
\mathbb{V}_{s' \sim \widehat{p}_h(\cdot|s,a)}[V_{h+1}^*(s', \overline{y}; \mathfrak{p}, \mathfrak{r}) - V_{h+1}^{\psi^*}(s', \overline{y}; \widehat{\mathfrak{p}}, \mathfrak{r})]
$$

$$
\overset{(4)}{=} \mathbb{E}_{s' \sim \widehat{p}_h(\cdot|s,a)}[(V_{h+1}^*(s', \overline{y}; \mathfrak{p}, \mathfrak{r}) - V_{h+1}^{\psi^*}(s', \overline{y}; \widehat{\mathfrak{p}}, \mathfrak{r}))^2]
$$

$$
- \mathbb{E}_{s' \sim \widehat{p}_h(\cdot|s,a)}[V_{h+1}^*(s', \overline{y}; \mathfrak{p}, \mathfrak{r}) - V_{h+1}^{\psi^*}(s', \overline{y}; \widehat{\mathfrak{p}}, \mathfrak{r})]^2
$$

$$
\overset{(5)}{\leqslant} \mathbb{E}_{s' \sim \widehat{p}_h(\cdot|s,a)}[(V_{h+1}^*(s', \overline{y}; \mathfrak{p}, \mathfrak{r}) - V_{h+1}^{\psi^*}(s', \overline{y}; \widehat{\mathfrak{p}}, \mathfrak{r}))^2]
$$

$$
\overset{(6)}{\leqslant} \|(V_{h+1}^*(\cdot, \overline{y}; \mathfrak{p}, \mathfrak{r}) - V_{h+1}^{\psi^*}(\cdot, \overline{y}; \widehat{\mathfrak{p}}, \mathfrak{r}))^2\|_\infty
$$

$$
= \|V_{h+1}^*(\cdot, \overline{y}; \mathfrak{p}, \mathfrak{r}) - V_{h+1}^{\psi^*}(\cdot, \overline{y}; \widehat{\mathfrak{p}}, \mathfrak{r})\|_\infty^2,
$$

where at (4) we have used $\mathbb{V}[X] = \mathbb{E}[X^2] - \mathbb{E}[X]^2$, at (5) we recognize that the second term is a square, thus always positive, and we remove it, and at (6) we have upper bounded the expected value, an average, through the infinity norm.

Thanks to this expression, we can continue to upper bound the previous term as:

$$
\mathbb{V}_{s' \sim p_h(\cdot|s,a)}[V_{h+1}^*(s', \overline{y}; \mathfrak{p}, \mathfrak{r})]
$$

$$
\leqslant \sum_{s' \in \mathcal{S}} \Big( p_h(s'|s,a) - \widehat{p}_h(s'|s,a) \Big) V_{h+1}^{*^2}(s', \overline{y}; \mathfrak{p}, \mathfrak{r})
$$

$$
- \Big[ \Big( \sum_{s' \in \mathcal{S}} p_h(s'|s,a) V_{h+1}^*(s', \overline{y}; \mathfrak{p}, \mathfrak{r}) \Big)^2
$$

$$
- \Big( \sum_{s' \in \mathcal{S}} \widehat{p}_h(s'|s,a) V_{h+1}^*(s', \overline{y}; \mathfrak{p}, \mathfrak{r}) \Big)^2 \Big]
$$

$$
+ \Big[ \|V_{h+1}^*(\cdot, \overline{y}; \mathfrak{p}, \mathfrak{r}) - V_{h+1}^{\psi^*}(\cdot, \overline{y}; \widehat{\mathfrak{p}}, \mathfrak{r})\|_\infty
$$

$$
+ \sqrt{\mathbb{V}_{s' \sim \widehat{p}_h(\cdot|s,a)}[V_{h+1}^{\psi^*}(s', \overline{y}; \widehat{\mathfrak{p}}, \mathfrak{r})]} \Big]^2
$$

$$
\overset{(7)}{=} \sum_{s' \in \mathcal{S}} \Big( p_h(s'|s,a) - \widehat{p}_h(s'|s,a) \Big) V_{h+1}^{*^2}(s', \overline{y}; \mathfrak{p}, \mathfrak{r})
$$

$$
- \Big[ \Big( \sum_{s' \in \mathcal{S}} (p_h(s'|s,a) - \widehat{p}_h(s'|s,a)) V_{h+1}^*(s', \overline{y}; \mathfrak{p}, \mathfrak{r}) \Big)
$$

$$
\cdot \Big( \sum_{s' \in \mathcal{S}} (p_h(s'|s,a) + \widehat{p}_h(s'|s,a)) V_{h+1}^*(s', \overline{y}; \mathfrak{p}, \mathfrak{r}) \Big) \Big]
$$

$$
+ \Big[ \|V_{h+1}^*(\cdot, \overline{y}; \mathfrak{p}, \mathfrak{r}) - V_{h+1}^{\psi^*}(\cdot, \overline{y}; \widehat{\mathfrak{p}}, \mathfrak{r})\|_\infty
$$

$$
+ \sqrt{\mathbb{V}_{s' \sim \widehat{p}_h(\cdot|s,a)}[V_{h+1}^{\psi^*}(s', \overline{y}; \widehat{\mathfrak{p}}, \mathfrak{r})]} \Big]^2
$$

$$
\overset{(8)}{\leqslant} \sum_{s' \in \mathcal{S}} \Big( p_h(s'|s,a) - \widehat{p}_h(s'|s,a) \Big) V_{h+1}^{*^2}(s', \overline{y}; \mathfrak{p}, \mathfrak{r})
$$

$$
- \Bigg[ \bigg( \sum_{s' \in \mathcal{S}} (p_h(s'|s,a) - \widehat{p}_h(s'|s,a)) V_{h+1}^*(s', \overline{y}; \mathfrak{p}, \mathfrak{r}) \bigg)
$$

$$
\cdot \bigg( \sum_{s' \in \mathcal{S}} (p_h(s'|s,a) + \widehat{p}_h(s'|s,a)) V_{h+1}^*(s', \overline{y}; \mathfrak{p}, \mathfrak{r}) \bigg) \Bigg]
$$

$$
+ \bigg[ cH^2 \sqrt{\frac{\log \frac{4SAHd}{\delta}}{n}} + \sqrt{\mathbb{V}_{s' \sim \widehat{p}_h(\cdot|s,a)} [V_{h+1}^{\psi^*}(s', \overline{y}; \widehat{\mathfrak{p}}, \mathfrak{r})]} \bigg]^2
$$

$$
\overset{(9)}{\leqslant} \sum_{s' \in \mathcal{S}} \Big( p_h(s'|s,a) - \widehat{p}_h(s'|s,a) \Big) V_{h+1}^{*^2}(s', \overline{y}; \mathfrak{p}, \mathfrak{r})
$$

$$
+ 2H \Big| \sum_{s' \in \mathcal{S}} (p_h(s'|s,a) - \widehat{p}_h(s'|s,a)) V_{h+1}^*(s', \overline{y}; \mathfrak{p}, \mathfrak{r}) \Big|
$$

$$
+ \bigg[ cH^2 \sqrt{\frac{\log \frac{4SAHd}{\delta}}{n}} + \sqrt{\mathbb{V}_{s' \sim \widehat{p}_h(\cdot|s,a)} [V_{h+1}^{\psi^*}(s', \overline{y}; \widehat{\mathfrak{p}}, \mathfrak{r})]} \bigg]^2
$$

$$
\overset{(10)}{\leqslant} \sum_{s' \in \mathcal{S}} \Big( p_h(s'|s,a) - \widehat{p}_h(s'|s,a) \Big) V_{h+1}^{*^2}(s', \overline{y}; \mathfrak{p}, \mathfrak{r})
$$

$$
+ 2cH^2 \sqrt{\frac{\log \frac{4SAHd}{\delta}}{n}}
$$

$$
+ \bigg[ cH^2 \sqrt{\frac{\log \frac{4SAHd}{\delta}}{n}} + \sqrt{\mathbb{V}_{s' \sim \widehat{p}_h(\cdot|s,a)} [V_{h+1}^{\psi^*}(s', \overline{y}; \widehat{\mathfrak{p}}, \mathfrak{r})]} \bigg]^2
$$

$$
\overset{(11)}{\leqslant} cH^2 \sqrt{\frac{\log \frac{4SAHd}{\delta}}{n}} + 2cH^2 \sqrt{\frac{\log \frac{4SAHd}{\delta}}{n}}
$$

$$
+ \bigg[ cH^2 \sqrt{\frac{\log \frac{4SAHd}{\delta}}{n}} + \sqrt{\mathbb{V}_{s' \sim \widehat{p}_h(\cdot|s,a)} [V_{h+1}^{\psi^*}(s', \overline{y}; \widehat{\mathfrak{p}}, \mathfrak{r})]} \bigg]^2
$$

$$
= 3cH^2 \sqrt{\frac{\log \frac{4SAHd}{\delta}}{n}}
$$

$$
+ \bigg[ cH^2 \sqrt{\frac{\log \frac{4SAHd}{\delta}}{n}} + \sqrt{\mathbb{V}_{s' \sim \widehat{p}_h(\cdot|s,a)} [V_{h+1}^{\psi^*}(s', \overline{y}; \widehat{\mathfrak{p}}, \mathfrak{r})]} \bigg]^2,
$$

where at (7) we have applied the common formula $x^2 - y^2 = (x - y)(x + y)$, at (8) we have applied Lemma E.8 using probability $\delta' = \delta/2$, and noticing that, for how the discretized MDP is constructed, we have that $\overline{y} \in \mathcal{Y}$, at (9) we have upper bounded the second term with the absolute value and recognized that the value function does not exceed $H$ and the sum of probabilities is no greater than 2; at (10) we recognize that, in the proof of Lemma E.8, we had already bounded that term, thus, under the event $\mathcal{E}$ which holds w.p. $1 - \delta/2$, we have that bound; at (11) we have applied Hoeffding's inequality to all tuples $h \in [\![H]\!], (s, y, a) \in \mathcal{S} \times \mathcal{Y}_h \times \mathcal{A}$ with probability $\delta/(2SAHd)$, and noticed that the square of the value function does not exceed $H^2$.

Observe that the previous formula holds for all $h \in [\![H]\!], (s, y, a) \in \mathcal{S} \times \mathcal{Y}_h \times \mathcal{A}$ w.p. $1 - \delta$ (by summing the two $\delta/2$ through a union bound). By taking the square root of both sides, we obtain:

$$
\sqrt{\mathbb{V}_{s' \sim p_h(\cdot|s,a)} [V_{h+1}^*(s', \overline{y}; \mathfrak{p}, \mathfrak{r})]}
$$

$$
\leqslant \bigg( 3cH^2 \sqrt{\frac{\log \frac{4SAHd}{\delta}}{n}} + \bigg[ cH^2 \sqrt{\frac{\log \frac{4SAHd}{\delta}}{n}}
$$

$$
+ \sqrt{\mathbb{V}_{s' \sim \widehat{p}_h(\cdot|s,a)} [V_{h+1}^{\psi^*}(s', \overline{y}; \widehat{\mathfrak{p}}, \mathfrak{r})]} \bigg]^2 \bigg)^{1/2}
$$

$$\overset{(12)}{\leqslant} c'H\sqrt[4]{\frac{\log\frac{4SAHY}{\delta}}{n}} + cH^2\sqrt{\frac{\log\frac{4SAHY}{\delta}}{n}}$$

$$\underbrace{\phantom{c'H\sqrt[4]{\frac{\log\frac{4SAHY}{\delta}}{n}} + cH^2\sqrt{\frac{\log\frac{4SAHY}{\delta}}{n}}}}_{=:b_1}$$

$$+ \sqrt{\mathbb{V}_{s'\sim\widehat{p}_h(\cdot|s,a)}[V_{h+1}^{\psi^*}(s',\overline{y};\widehat{\mathfrak{p}},\mathfrak{r})]}$$

$$= \sqrt{\mathbb{V}_{s'\sim\widehat{p}_h(\cdot|s,a)}[V_{h+1}^{\psi^*}(s',\overline{y};\widehat{\mathfrak{p}},\mathfrak{r})]} + b_1,$$

where at (12) we have used the fact that $\sqrt{a+b} \leqslant \sqrt{a} + \sqrt{b}$.

To prove the second formula, the passages are basically the same, the only difference is that, at passage (1), we sum and subtract $V_{h+1}^{\widehat{\psi}^*}(s',\overline{y};\widehat{\mathfrak{p}},\mathfrak{r})$ instead of $V_{h+1}^{\psi^*}(s',\overline{y};\widehat{\mathfrak{p}},\mathfrak{r})$, and that at passage (8) we apply the other expression in Lemma E.8. This concludes the proof. $\qquad\square$

**Lemma E.10.** *For any $\delta \in (0,1)$, define:*

$$c_1 := \log\frac{2SAHd}{\delta},$$

$$b_2 := cH\left(\frac{\log\frac{8SAHd}{\delta}}{n}\right)^{3/4} + c'H^2\frac{\log\frac{8SAHd}{\delta}}{n},$$

*for some positive constants $c, c'$. Then, w.p. at least $1 - \delta$, we have, for all $h \in [\![H]\!], (s,y,a) \in \mathcal{S}\times\mathcal{Y}_h\times\mathcal{A}$:*

$$\sum_{s'\in\mathcal{S}}\left(p_h(s'|s,a) - \widehat{p}_h(s'|s,a)\right)V_{h+1}^*(s', y+\overline{r}_h(s,a); \mathfrak{p},\mathfrak{r})$$

$$\leqslant c''\sqrt{\frac{c_1\mathbb{V}_{s'\sim\widehat{p}_h(\cdot|s,a)}[V_{h+1}^{\psi^*}(s', y+\overline{r}_h(s,a); \widehat{\mathfrak{p}},\mathfrak{r})]}{n}} + b_2,$$

$$\sum_{s'\in\mathcal{S}}\left(p_h(s'|s,a) - \widehat{p}_h(s'|s,a)\right)V_{h+1}^*(s', y+\overline{r}_h(s,a); \mathfrak{p},\mathfrak{r})$$

$$\geqslant -c'''\sqrt{\frac{c_1\mathbb{V}_{s'\sim\widehat{p}_h(\cdot|s,a)}[V_{h+1}^*(s', y+\overline{r}_h(s,a); \widehat{\mathfrak{p}},\mathfrak{r})]}{n}} + b_2,$$

*for some positive constants $c'', c'''$.*

*Proof.* Again, we will write $\overline{y}$ instead of $y + \overline{r}_h(s,a)$ for simplicity. For all $h \in [\![H]\!], (s,y,a) \in \mathcal{S}\times\mathcal{Y}_h\times\mathcal{A}$, we can write:

$$\sum_{s'\in\mathcal{S}}\left(p_h(s'|s,a) - \widehat{p}_h(s'|s,a)\right)V_{h+1}^*(s', \overline{y}; \mathfrak{p},\mathfrak{r})$$

$$\overset{(1)}{\leqslant} \sqrt{\frac{2\mathbb{V}_{s'\sim p_h(\cdot|s,a)}[V_{h+1}^*(s',\overline{y};\mathfrak{p},\mathfrak{r})]\log\frac{2SAHd}{\delta}}{n}} + \frac{2H\log\frac{2SAHd}{\delta}}{3n}$$

$$\overset{(2)}{\leqslant} \sqrt{\frac{2\log\frac{2SAHd}{\delta}}{n}}\left(\sqrt{\mathbb{V}_{s'\sim\widehat{p}_h(\cdot|s,a)}[V_{h+1}^{\psi^*}(s',\overline{y};\widehat{\mathfrak{p}},R)]} + b_1\right) + \frac{2H\log\frac{2SAHd}{\delta}}{3n}$$

$$\overset{(3)}{=} c\sqrt{\frac{c_1\mathbb{V}_{s'\sim\widehat{p}_h(\cdot|s,a)}[V_{h+1}^{\psi^*}(s',\overline{y};\widehat{\mathfrak{p}},\mathfrak{r})]}{n}} + c'\sqrt{\frac{c_1}{n}}H\left(\frac{\log\frac{8SAHd}{\delta}}{n}\right)^{1/4}$$

$$+ c''\sqrt{\frac{c_1}{n}}H^2\sqrt{\frac{\log\frac{8SAHd}{\delta}}{n}} + c'''H\frac{c_1}{n}$$

$$\leqslant c\sqrt{\frac{c_1\mathbb{V}_{s'\sim\widehat{p}_h(\cdot|s,a)}[V_{h+1}^{\psi^*}(s',\overline{y};\widehat{\mathfrak{p}},\mathfrak{r})]}{n}} + c'H\left(\frac{\log\frac{8SAHd}{\delta}}{n}\right)^{3/4} + c''''H^2\frac{\log\frac{8SAHd}{\delta}}{n},$$

where at (1) we have applied the Bernstein's inequality using $\delta/(2SAHd)$ as probability for all $h \in [\![H]\!], (s,y,a) \in \mathcal{S} \times \mathcal{Y}_h \times \mathcal{A}$, and at (2) we have applied Lemma E.9 with $\delta/2$ of probability, and a union bound to guarantee the event to hold w.p. $1-\delta$, at (3) we use the definition of $c_1 := \log \frac{2SAHd}{\delta}$, and denoted by $c, c', c'', c'''$ some positive constants.

For the other expression, an analogous derivation can be carried out. In particular, we use the other side of the Bernstein's inequality, and the other expression in Lemma E.9. $\qquad\square$

**Lemma E.11.** *For any* $h \in [\![H]\!], (s,y,a) \in \mathcal{S} \times \mathcal{Y}_h \times \mathcal{A}$ *and deterministic policy* $\psi$, *let* $\Sigma_h^\psi(s,y,a)$ *be defined as:*

$$\Sigma_h^\psi(s,y,a) := \mathop{\mathbb{E}}_{\mathfrak{p},\mathfrak{r},\psi} \left[ \Big| \sum_{h'=h}^H \mathfrak{r}_{h'}(s_{h'}, y_{h'}, a_{h'}) - Q_h^\psi(s,y,a;\mathfrak{p},\mathfrak{r}) \Big|^2 \mid s_h = s, y_h = y, a_h = a \right].$$

*Then, function* $\Sigma$ *satisfies the Bellman equation, i.e., for any* $h \in [\![H]\!], (s,y,a) \in \mathcal{S} \times \mathcal{Y}_h \times \mathcal{A}$ *and deterministic policy* $\psi$:

$$\Sigma_h^\psi(s,y,a) = \mathbb{V}_{s' \sim p_h(\cdot|s,a)} [V_{h+1}^\psi(s', y + \overline{r}_h(s,a); \mathfrak{p}, \mathfrak{r})]$$
$$+ \mathop{\mathbb{E}}_{s' \sim p_h(\cdot|s,a)} [\Sigma_{h+1}^\psi(s', y + \overline{r}_h(s,a), \psi_{h+1}(s', y + \overline{r}_h(s,a)))].$$

*Proof.* For all $h \in [\![H]\!], (s,y,a) \in \mathcal{S} \times \mathcal{Y}_h \times \mathcal{A}$ and deterministic policy $\psi$, we can write (we denote $a' := \psi_{h+1}(s', y + \overline{r}_h(s,a))$ and $\overline{y} := y + \overline{r}_h(s,a)$ for notational simplicity, and we remark that $\overline{y}$ is *not* a random variable):

$$\Sigma_h^\psi(s,y,a) := \mathop{\mathbb{E}}_{\mathfrak{p},\mathfrak{r},\psi} \left[ \Big| \sum_{h'=h}^H \mathfrak{r}_{h'}(s_{h'}, y_{h'}, a_{h'}) - Q_h^\psi(s,y,a;\mathfrak{p},\mathfrak{r}) \Big|^2 \mid s_h = s, y_h = y, a_h = a \right]$$

$$\overset{(1)}{=} \mathop{\mathbb{E}}_{s' \sim p_h(\cdot|s,a)} \left[ \mathop{\mathbb{E}}_{\mathfrak{p},\mathfrak{r},\psi} \left[ \Big| \sum_{h'=h}^H \mathfrak{r}_{h'}(s_{h'}, y_{h'}, a_{h'}) - Q_{h+1}^\psi(s', \overline{y}, a'; \mathfrak{p}, \mathfrak{r}) \right.\right.$$
$$\left.\left. - \big( Q_h^\psi(s,y,a;\mathfrak{p},\mathfrak{r}) - Q_{h+1}^\psi(s', \overline{y}, a'; \mathfrak{p}, \mathfrak{r}) \big) \Big|^2 \right.\right.$$
$$\left.\left. \mid s_h = s, a_h = a, y_h = y, s_{h+1} = s' \right] \right]$$

$$\overset{(2)}{=} \mathop{\mathbb{E}}_{s' \sim p_h(\cdot|s,a)} \left[ \mathop{\mathbb{E}}_{\mathfrak{p},\mathfrak{r},\psi} \left[ \Big| \sum_{h'=h+1}^H \mathfrak{r}_{h'}(s_{h'}, y_{h'}, a_{h'}) - Q_{h+1}^\psi(s', \overline{y}, a'; \mathfrak{p}, \mathfrak{r}) \right.\right.$$
$$\left.\left. - \big( Q_h^\psi(s,y,a;\mathfrak{p},\mathfrak{r}) - \mathfrak{r}_h(s,y,a) - Q_{h+1}^\psi(s', \overline{y}, a'; \mathfrak{p}, \mathfrak{r}) \big) \Big|^2 \right.\right.$$
$$\left.\left. \mid s_{h+1} = s', y_{h+1} = \overline{y} \right] \right]$$

$$\overset{(3)}{=} \mathop{\mathbb{E}}_{s' \sim p_h(\cdot|s,a)} \left[ \mathop{\mathbb{E}}_{\mathfrak{p},\mathfrak{r},\psi} \left[ \Big| \sum_{h'=h+1}^H \mathfrak{r}_{h'}(s_{h'}, y_{h'}, a_{h'}) - Q_{h+1}^\psi(s', \overline{y}, a'; \mathfrak{p}, \mathfrak{r}) \Big|^2 \right.\right.$$
$$\left.\left. \mid s_{h+1} = s', y_{h+1} = \overline{y} \right] \right]$$

$$- 2 \mathop{\mathbb{E}}_{s' \sim p_h(\cdot|s,a)} \left[ \big( Q_h^\psi(s,y,a;\mathfrak{p},\mathfrak{r}) - \mathfrak{r}_h(s,y,a) - Q_{h+1}^\psi(s', \overline{y}, a'; \mathfrak{p}, \mathfrak{r}) \big) \right.$$
$$\left. \cdot \underbrace{\mathop{\mathbb{E}}_{\mathfrak{p},\mathfrak{r},\psi} \left[ \sum_{h'=h+1}^H \mathfrak{r}_{h'}(s_{h'}, y_{h'}, a_{h'}) - Q_{h+1}^\psi(s', \overline{y}, a'; \mathfrak{p}, \mathfrak{r}) \mid s_{h+1} = s', y_{h+1} = \overline{y} \right]}_{=0} \right]$$

$$+ \mathop{\mathbb{E}}_{s' \sim p_h(\cdot|s,a)} \left[ \big| Q_h^\psi(s,y,a;\mathfrak{p},\mathfrak{r}) - \mathfrak{r}_h(s,y,a) - Q_{h+1}^\psi(s', \overline{y}, a'; \mathfrak{p}, \mathfrak{r}) \big|^2 \right]$$

$$\overset{(4)}{=} \underset{s' \sim p_h(\cdot|s,a)}{\mathbb{E}} \Bigg[$$

$$\underbrace{\underset{\mathfrak{p},\mathfrak{r},\psi}{\mathbb{E}} \Big[ \Big| \sum_{h'=h+1}^{H} \mathfrak{r}_{h'}(s_{h'}, y_{h'}, a_{h'}) - Q_{h+1}^{\psi}(s', \overline{y}, a'; \mathfrak{p}, \mathfrak{r}) \Big|^2 \,\big|\, s_{h+1} = s', y_{h+1} = \overline{y} \Big]}_{= \Sigma_{h+1}^{\psi}(s', \overline{y}, a')} \Bigg]$$

$$+ \underbrace{\underset{s' \sim p_h(\cdot|s,a)}{\mathbb{E}} \Big[ \big| Q_h^{\psi}(s, y, a; \mathfrak{p}, \mathfrak{r}) - \mathfrak{r}_h(s, y, a) - Q_{h+1}^{\psi}(s', \overline{y}, a'; \mathfrak{p}, \mathfrak{r}) \big|^2 \Big]}_{=: \mathbb{V}_{s' \sim p_h(\cdot|s,a)}[Q_{h+1}^{\psi}(s', \overline{y}, a'; \mathfrak{p}, \mathfrak{r})] = \mathbb{V}_{s' \sim p_h(\cdot|s,a)}[V_{h+1}^{\psi}(s', \overline{y}; \mathfrak{p}, \mathfrak{r})]}$$

$$= \underset{s' \sim p_h(\cdot|s,a)}{\mathbb{E}} [\Sigma_{h+1}^{\psi}(s', \overline{y}, a')] + \mathbb{V}_{s' \sim p_h(\cdot|s,a)}[V_{h+1}^{\psi}(s', \overline{y}; \mathfrak{p}, \mathfrak{r})],$$

at (1) we add and subtract a term, at (2) we bring out the non-random reward received at $h$, at (3) we compute the square and use the linearity of expectation, at (4) we use the fact that $\mathbb{E}_{\mathfrak{p},\mathfrak{r},\psi} \big[ \sum_{h'=h+1}^{H} \mathfrak{r}_{h'}(s_{h'}, y_{h'}, a_{h'}) - Q_{h+1}^{\psi}(s', \overline{y}, a'; \mathfrak{p}, \mathfrak{r}) \,\big|\, s_{h+1} = s' \big] = Q_{h+1}^{\psi}(s', \overline{y}, a'; \mathfrak{p}, \mathfrak{r}) - Q_{h+1}^{\psi}(s', \overline{y}, a'; \mathfrak{p}, \mathfrak{r}) = 0$ because of linearity of expectation. $\qquad\square$

**Lemma E.12.** *Let $\psi$ be any policy, and let $\mathfrak{p}$ be any transition model associated to an arbitrary inner dynamics $p$. Then, for all $h \in [\![H]\!], (s, y, a) \in \mathcal{S} \times \mathcal{Y}_h \times \mathcal{A}$, it holds that:*

$$\Bigg| \underset{\mathfrak{p},\mathfrak{r},\psi}{\mathbb{E}} \Big[ \sum_{h'=h}^{H} \sqrt{\mathbb{V}_{s' \sim p_{h'}(\cdot|s_{h'}, a_{h'})}[V_{h'+1}^{\psi}(s', y_{h'+1}; \mathfrak{p}, \mathfrak{r})]} \,\Big|\, s_h = s, y_h = y, a_h = a \Big] \Bigg| \leqslant \sqrt{H^3}.$$

*Proof.* For all $h \in [\![H]\!], (s, y, a) \in \mathcal{S} \times \mathcal{Y}_h \times \mathcal{A}$, we can write (note that this derivation is independent of $\mathfrak{p}, p$, so we might use even $\widehat{\mathfrak{p}}, \widehat{p}$ in the proof):

$$\Bigg| \underset{\mathfrak{p},\mathfrak{r},\psi}{\mathbb{E}} \Big[ \sum_{h'=h}^{H} \sqrt{\mathbb{V}_{s' \sim p_{h'}(\cdot|s_{h'}, a_{h'})}[V_{h'+1}^{\psi}(s', y_{h'+1}; \mathfrak{p}, \mathfrak{r})]} \,\big|\, s_h = s, y_h = y, a_h = a \Big] \Bigg|$$

$$\overset{(1)}{\leqslant} \Bigg| \underset{\mathfrak{p},\mathfrak{r},\psi}{\mathbb{E}} \Big[ \sqrt{H \sum_{h'=h}^{H} \mathbb{V}_{s' \sim p_{h'}(\cdot|s_{h'}, a_{h'})}[V_{h'+1}^{\psi}(s', y_{h'+1}; \mathfrak{p}, \mathfrak{r})]} \,\big|\, s_h = s, y_h = y, a_h = a \Big] \Bigg|$$

$$\overset{(2)}{\leqslant} \sqrt{H} \sqrt{\underset{\mathfrak{p},\mathfrak{r},\psi}{\mathbb{E}} \Big[ \sum_{h'=h}^{H} \mathbb{V}_{s' \sim p_{h'}(\cdot|s_{h'}, a_{h'})}[V_{h'+1}^{\psi}(s', y_{h'+1}; \mathfrak{p}, \mathfrak{r})] \,\big|\, s_h = s, y_h = y, a_h = a \Big]}$$

$$\overset{(3)}{=} \sqrt{H} \bigg( \underset{\mathfrak{p},\mathfrak{r},\psi}{\mathbb{E}} \Big[ \sum_{h'=h}^{H} \Sigma_{h'}^{\psi}(s_{h'}, y_{h'}, a_{h'}) - \mathbb{E}_{s' \sim p_{h'}(\cdot|s_{h'}, a_{h'})} \big[ \Sigma_{h'+1}^{\psi}(s', y_{h'+1}, \psi_{h'+1}(s', y_{h'+1})) \big]$$

$$\big|\, s_h = s, y_h = y, a_h = a \Big] \bigg)^{1/2}$$

$$= \sqrt{H} \sqrt{\underset{\mathfrak{p},\mathfrak{r},\psi}{\mathbb{E}} \Big[ \sum_{h'=h}^{H} \Sigma_{h'}^{\psi}(s_{h'}, y_{h'}, a_{h'}) - \Sigma_{h'+1}^{\psi}(s_{h'+1}, y_{h'+1}, a_{h'+1}) \,\big|\, s_h = s, y_h = y, a_h = a \Big]}$$

$$\overset{(4)}{=} \sqrt{H} \sqrt{\underset{\mathfrak{p},\mathfrak{r},\psi}{\mathbb{E}} \Big[ \Sigma_h^{\psi}(s_h, y_h, a_h) - \underbrace{\Sigma_{H+1}^{\psi}(s_{H+1}, y_{H+1}, a_{H+1})}_{=0} \,\big|\, s_h = s, y_h = y, a_h = a \Big]}$$

$$= \sqrt{H} \sqrt{\Sigma_h^{\psi}(s, y, a)}$$

$$\overset{(5)}{\leqslant} \sqrt{H} \sqrt{H^2}$$

$$= \sqrt{H^3},$$

where at (1) we have applied the Cauchy-Schwarz's inequality, at (2) we have applied Jensen's inequality, at (3) we have applied Lemma E.11, at (4) we have used telescoping, and at (5) we have bounded $\Sigma_h^\psi(s, y, a) \leqslant H^2$ for all $h \in [\![H]\!], (s, y, a) \in \mathcal{S} \times \mathcal{Y}_h \times \mathcal{A}$. $\qquad\square$

### E.3.3 Lemmas on the Optimal Performance for Multiple Utilities

To prove the following results, we will make use of the notation introduced in the previous section.

**Lemma E.13.** *Let $\epsilon, \delta \in (0, 1)$. It suffices to execute* **CATY-UL** *with:*

$$\tau \leqslant \widetilde{\mathcal{O}}\Big(\frac{SAH^5}{\epsilon^2}\Big(S + \log \frac{SAH}{\delta}\Big)\Big),$$

*to obtain* $\sup_{U \in \mathfrak{U}_L} \big|J^*(U; p, r) - \widehat{J}^*(U)\big| \leqslant HL\epsilon_0 + \epsilon$ *w.p.* $1 - \delta$.

*Proof.* Similarly to the proof of Lemma E.13, we can write:

$$\sup_{U \in \mathfrak{U}_L} |J^*(U; p, r) - \widehat{J}^*(U)|$$

$$= \sup_{U \in \mathfrak{U}_L} |J^*(U; p, r) - \widehat{J}^*(U) \pm J^*(\mathfrak{p}, \mathfrak{r})|$$

$$\leqslant \sup_{U \in \mathfrak{U}_L} |J^*(U; p, r) - J^*(\mathfrak{p}, \mathfrak{r})| + \sup_{U \in \mathfrak{U}_L} |J^*(\mathfrak{p}, \mathfrak{r}) - \widehat{J}^*(U)|$$

$$= \sup_{U \in \mathfrak{U}_L} |J^*(U; p, r) - J^*(\mathfrak{p}, \mathfrak{r})| + \sup_{U \in \mathfrak{U}_L} |J^*(\mathfrak{p}, \mathfrak{r}) - J^*(\widehat{\mathfrak{p}}, \mathfrak{r})|$$

$$\leqslant HL\epsilon_0 + \sup_{U \in \mathfrak{U}_L} |J^*(\mathfrak{p}, \mathfrak{r}) - J^*(\widehat{\mathfrak{p}}, \mathfrak{r})|$$

$$\overset{(1)}{\leqslant} HL\epsilon_0 + H^2\sqrt{\frac{2}{n}\Big(\log \frac{SAH}{\delta} + (S-1)\log\big(e(1 + n/(S-1))\big)\Big)}$$

$$\leqslant HL\epsilon_0 + \epsilon,$$

where at (1) we have applied the formula in Lemma E.14.

By enforcing such quantity to be smaller than $\epsilon$, we get:

$$H^2\sqrt{\frac{2}{n}\Big(\log \frac{SAH}{\delta} + (S-1)\log\big(e(1 + n/(S-1))\big)\Big)} \leqslant$$

$$\frac{H^2\sqrt{\log\big(e(1 + n/(S-1))\big)}}{\sqrt{n}}\sqrt{2\Big(\log \frac{SAH}{\delta} + (S-1)\Big)} \leqslant \epsilon$$

$$\iff n \geqslant 2\frac{H^4}{\epsilon^2}\Big(\log \frac{SAH}{\delta} + (S-1)\Big)\log\big(e(1 + n/(S-1))\big).$$

By summing over all $(s, a, h) \in \mathcal{S} \times \mathcal{A} \times [\![H]\!]$, and by applying Lemma J.3 of Lazzati et al. (2024b), we obtain that:

$$\tau = SAHn \geqslant \widetilde{\mathcal{O}}\Big(\frac{SAH^5}{\epsilon^2}\Big(\log \frac{SAH}{\delta} + S\Big)\Big).$$

$\qquad\square$

**Lemma E.14.** *For any $\delta \in (0, 1)$, for all utility functions $U \in \mathfrak{U}_L$ at the same time, we have:*

$$|J_h^*(\mathfrak{p}, \mathfrak{r}) - J_h^*(\widehat{\mathfrak{p}}, \mathfrak{r})| \leqslant H^2\sqrt{\frac{2}{n}\Big(\log \frac{SAH}{\delta} + (S-1)\log\big(e(1 + n/(S-1))\big)\Big)},$$

*w.p. at least $1 - \delta$.*

*Proof.* Let us denote by $\mathcal{E}$ the event defined as:

$$\mathcal{E} := \Big\{\forall n \in \mathbb{N}, \forall h \in [\![H]\!], (s, y, a) \in \mathcal{S} \times \mathcal{Y}_h \times \mathcal{A} :$$

$$nKL\big(\widehat{p}_h(\cdot|s,a)\|p_h(\cdot|s,a)\big) \leqslant \log\frac{SAH}{\delta} + (S-1)\log\big(e(1+n/(S-1))\big)\bigg\}.$$

We can write:

$$\mathbb{P}(\mathcal{E}^\complement) = \mathbb{P}\bigg(\exists n \in \mathbb{N},\, \exists h \in [\![H]\!],\, (s,y,a) \in \mathcal{S} \times \mathcal{Y}_h \times \mathcal{A}:$$

$$nKL\big(\widehat{p}_h(\cdot|s,a)\|p_h(\cdot|s,a)\big) > \log\frac{SAH}{\delta} + (S-1)\log\big(e(1+n/(S-1))\big)\bigg)$$

$$\overset{(1)}{=} \mathbb{P}\bigg(\exists n \in \mathbb{N},\, \exists (s,a,h) \in \mathcal{S} \times \mathcal{A} \times [\![H]\!]:$$

$$nKL\big(\widehat{p}_h(\cdot|s,a)\|p_h(\cdot|s,a)\big) > \log\frac{SAH}{\delta} + (S-1)\log\big(e(1+n/(S-1))\big)\bigg)$$

$$\overset{(2)}{\leqslant} \sum_{(s,a,h)\in\mathcal{S}\times\mathcal{A}\times[\![H]\!]} \mathbb{P}\bigg(\exists n \in \mathbb{N},\, nKL\big(\widehat{p}_h(\cdot|s,a)\|p_h(\cdot|s,a)\big) >$$

$$\log\frac{SAH}{\delta} + (S-1)\log\big(e(1+n/(S-1))\big)\bigg)$$

$$\overset{(3)}{\leqslant} \sum_{(s,a,h)\in\mathcal{S}\times\mathcal{A}\times[\![H]\!]} \frac{\delta}{SAH}$$

$$\leqslant \delta,$$

where at (1) we realize that there is no dependence on variable $y$, thus we can drop it,[11] at (2) we have applied a union bound over all triples $(s,a,h) \in \mathcal{S} \times \mathcal{A} \times [\![H]\!]$, and at (3) we have applied Proposition 1 of Jonsson et al. (2020).

Next, for all utilities $U \in \mathfrak{U}_L$ at the same time, for all the tuples $h \in [\![H]\!]$, $(s,y) \in \mathcal{S} \times \mathcal{Y}_h$, we can write:

$$|V_h^*(s,y;\mathfrak{p},\mathfrak{r}) - V_h^*(s,y;\widehat{\mathfrak{p}},\mathfrak{r})|$$

$$\overset{(4)}{\leqslant} \mathop{\mathbb{E}}_{\widehat{\mathfrak{p}},\mathfrak{r},\overline{\psi}}\bigg[ \sum_{h'=h}^{H} \bigg| \sum_{s'\in\mathcal{S}} \big(p_{h'}(s'|s_{h'},a_{h'}) - \widehat{p}_{h'}(s'|s_{h'},a_{h'})\big) V_{h'+1}^{\psi^*}(s',y_{h'+1};\mathfrak{p},\mathfrak{r})\bigg|$$

$$\bigg| s_h = s, y_h = y, a_h = a\bigg]$$

$$\overset{(5)}{\leqslant} H \mathop{\mathbb{E}}_{\widehat{\mathfrak{p}},\mathfrak{r},\overline{\psi}}\bigg[ \sum_{h'=h}^{H} \|p_{h'}(\cdot|s_{h'},a_{h'}) - \widehat{p}_{h'}(\cdot|s_{h'},a_{h'})\|_1 \,\bigg|\, s_h = s, y_h = y, a_h = a\bigg]$$

$$\overset{(6)}{\leqslant} H \mathop{\mathbb{E}}_{\widehat{\mathfrak{p}},\mathfrak{r},\overline{\psi}}\bigg[ \sum_{h'=h}^{H} \sqrt{2KL(\widehat{p}_{h'}(\cdot|s_{h'},a_{h'})\|p_{h'}(\cdot|s_{h'},a_{h'}))} \,\bigg|\, s_h = s, y_h = y, a_h = a\bigg]$$

$$\overset{(7)}{\leqslant} H \mathop{\mathbb{E}}_{\widehat{\mathfrak{p}},\mathfrak{r},\overline{\psi}}\bigg[ \sum_{h'=h}^{H} \sqrt{\frac{2}{n}\Big(\log\frac{SAH}{\delta} + (S-1)\log\big(e(1+n/(S-1))\big)\Big)}$$

$$\bigg| s_h = s, y_h = y, a_h = a\bigg]$$

$$= H^2 \sqrt{\frac{2}{n}\Big(\log\frac{SAH}{\delta} + (S-1)\log\big(e(1+n/(S-1))\big)\Big)},$$

where at (4) we apply the formula derived in the proof of Lemma E.8 and triangle inequality, at (5) we have upper bounded with the 1-norm, defined as $\|f\|_1 := \sum_x |f(x)|$, at (6) we have applied Pinsker's inequality, at (7) we assume that concentration event $\mathcal{E}$ holds.

---

[11]Therefore, differently from the event for a single utility, now there is no dependence on $d$ in the bound. Intuitively, $d$ appeared in the case of a single utility because we had to apply Hoeffding's inequality $d$ times, because we had, potentially, $d$ different value functions (as many as the states). Since now we provide the bound for all the possible value functions (1-norm bound), then the dependence on $d$ disappears.

We remark that the guarantee provided by this theorem holds not only for $L$-Lipschitz utilities, but for all functions with the same dimensionality (since it is a bound in 1-norm). $\square$

### E.4 ANALYSIS OF `TRACTOR-UL`

**Theorem 5.2.** *Let* $\epsilon, \delta \in (0,1), L > 0$, *and assume that* $U^E \in \underline{\mathfrak{U}}_L$. *If we execute* `TRACTOR-UL` *with parameters* $\epsilon_0 = \epsilon^2/(80N^2L^2H), T \geqslant \mathcal{O}(N^4H^4L^2/\epsilon^4), K \geqslant \tilde{\mathcal{O}}(N^2H^2 \log \frac{NHL}{\delta\epsilon}/\epsilon^2), \alpha = \sqrt{\lceil H/\epsilon_0 \rceil - 1}H/(2N\sqrt{T})$, *an arbitrary* $\overline{U}_0 \in \underline{\mathfrak{U}}_L$, *and if it holds that, for all* $i \in [\![N]\!]$:

$$\tau^{E,i} \geqslant \tilde{\mathcal{O}}\Big(\frac{H^4N^4L^2}{\epsilon^4} \log \frac{NHL}{\delta\epsilon}\Big), \qquad \tau^i \geqslant \tilde{\mathcal{O}}\Big(\frac{N^2SAH^5}{\epsilon^2}\Big(S + \log \frac{SAHN}{\delta}\Big)\Big),$$

*then, w.p. at least* $1 - \delta$, *for any* $\Delta \geqslant \epsilon$, `TRACTOR-UL` *guarantees that all the utilities* $U \in \underline{\mathfrak{U}}_L$ *such that* $U(y) = \hat{U}(y)$ *for all* $y \in \mathcal{Y}$ *(where* $\hat{U} \in \underline{\overline{\mathfrak{U}}}_L$ *is the output of* `TRACTOR-UL`*) belong to* $U \in \mathcal{U}_\Delta$.

*Proof.* The proof draws inspiration from those of Syed & Schapire (2007) and Schlaginhaufen & Kamgarpour (2024).

Given any distribution $\eta$ supported on $\mathcal{Y}$, and given any two utilities $U \in \underline{\mathfrak{U}}_L, \overline{U} \in \underline{\overline{\mathfrak{U}}}_L$ (where $U$ is a function on $[0, H]$ and $\overline{U}$ is a vector on $\mathcal{Y}$), we will abuse notation and write both $U^\intercal\eta$ and $\overline{U}^\intercal\eta$, with obvious meaning.

Moreover, for $L > 0$, we define operator $\mathfrak{C}_L : \underline{\overline{\mathfrak{U}}}_L \to 2^{\underline{\mathfrak{U}}_L}$ (where $2^{\mathcal{X}}$ denotes the power set of set $\mathcal{X}$) that, given vector $\overline{U} \in \underline{\overline{\mathfrak{U}}}_L$, returns the set $\mathfrak{C}_L(\overline{U}) := \{U \in \underline{\mathfrak{U}}_L \mid \forall y \in \mathcal{Y} : U(y) = \overline{U}(y)\}$.

First of all, we observe that the guarantee provided by the theorem follows directly by the following expression:

$$\mathbb{P}_{\mathcal{M}^1, \mathcal{M}^2, \ldots, \mathcal{M}^N} \Big( \sup_{U \in \mathfrak{C}_L(\hat{U})} \max_{i \in [\![N]\!]} \overline{\mathcal{C}}_{p^i, r^i, \pi^{E,i}}(U) \leqslant \epsilon \Big) \geqslant 1 - \delta,$$

where $\mathbb{P}_{\mathcal{M}^1, \mathcal{M}^2, \ldots, \mathcal{M}^N}$ denotes the joint probability distribution obtained by the $N$ MDPs $\{\mathcal{M}^i\}_i$.

Let us denote by $\hat{U} := (\sum_{t=0}^{T-1} \overline{U}_t)/T$ the output of `TRACTOR-UL`. Note that $\hat{U} \in \underline{\overline{\mathfrak{U}}}_L$. We can write:

$$\sup_{U \in \mathfrak{C}_L(\hat{U})} \max_{i \in [\![N]\!]} \overline{\mathcal{C}}_{p^i, r^i, \pi^{E,i}}(U)$$

$$\stackrel{(1)}{\leqslant} \sup_{U \in \mathfrak{C}_L(\hat{U})} \sum_{i \in [\![N]\!]} \overline{\mathcal{C}}_{p^i, r^i, \pi^{E,i}}(U)$$

$$\stackrel{(2)}{=} \sup_{U \in \mathfrak{C}_L(\hat{U})} \sum_{i \in [\![N]\!]} \Big( J^*(U; p^i, r^i) - J^{\pi^{E,i}}(U; p^i, r^i) \pm \hat{U}^\intercal \hat{\eta}^{E,i} \Big)$$

$$\stackrel{(3)}{\leqslant} \sup_{U \in \mathfrak{C}_L(\hat{U})} \sum_{i \in [\![N]\!]} \Big( J^*(U; p^i, r^i) - \hat{U}^\intercal \hat{\eta}^{E,i} \Big) + \epsilon_1$$

$$\stackrel{(4)}{=} \sup_{U \in \mathfrak{C}_L(\hat{U})} \sum_{i \in [\![N]\!]} \Big( \max_{\eta \in \mathfrak{D}_i} U^\intercal \eta - \hat{U}^\intercal \hat{\eta}^{E,i} \Big) + \epsilon_1$$

$$\stackrel{(5)}{=} \sup_{\substack{U_0 \in \mathfrak{C}_L(\overline{U}_0), \\ \ldots, \\ U_{T-1} \in \mathfrak{C}_L(\overline{U}_{T-1})}} \frac{1}{T} \sum_{i \in [\![N]\!]} \max_{\eta \in \mathfrak{D}_i} \sum_{t=0}^{T-1} \Big( U_t^\intercal \eta - \overline{U}_t^\intercal \hat{\eta}^{E,i} \Big) + \epsilon_1$$

$$\stackrel{(6)}{\leqslant} \frac{1}{T} \sum_{t=0}^{T-1} \sup_{U_t \in \mathfrak{C}_L(\overline{U}_t)} \sum_{i \in [\![N]\!]} \Big( \max_{\eta \in \mathfrak{D}_i} U_t^\intercal \eta \pm \overline{U}_t^\intercal \hat{\eta}^i - \overline{U}_t^\intercal \hat{\eta}^{E,i} \Big) + \epsilon_1$$

$$\stackrel{(7)}{\leqslant} \frac{1}{T} \sum_{t=0}^{T-1} \sum_{i \in [\![N]\!]} \overline{U}_t^\intercal \Big( \hat{\eta}^i_t - \hat{\eta}^{E,i} \Big) \pm \frac{1}{T} \min_{\overline{U} \in \underline{\overline{\mathfrak{U}}}_L} \sum_{t=0}^{T-1} \sum_{i \in [\![N]\!]} \overline{U}^\intercal \Big( \hat{\eta}^i_t - \hat{\eta}^{E,i} \Big) + \epsilon_1 + \epsilon_2$$

$$\overset{(8)}{\leqslant} \frac{1}{T} \min_{\overline{U} \in \underline{\mathfrak{U}}_L} \sum_{t=0}^{T-1} \sum_{i \in [\![N]\!]} \overline{U}^\intercal \left( \widehat{\eta}_t^i - \widehat{\eta}^{E,i} \right) + \epsilon_1 + \epsilon_2 + \underbrace{\frac{2HN\sqrt{H/\epsilon_0}}{\sqrt{T}}}_{=:\epsilon_3}$$

$$\overset{(9)}{\leqslant} \frac{1}{T} \sum_{t=0}^{T-1} \sum_{i \in [\![N]\!]} \overline{U}^{E,\intercal} \left( \widehat{\eta}_t^i - \widehat{\eta}^{E,i} \right) \pm U^{E,\intercal} \eta^{p^i,r^i,\pi^{E,i}} + \epsilon_1 + \epsilon_2 + \epsilon_3$$

$$\overset{(10)}{\leqslant} \frac{1}{T} \sum_{t=0}^{T-1} \sum_{i \in [\![N]\!]} \overline{U}^{E,\intercal} \widehat{\eta}_t^i \pm U^{E,\intercal} \eta^{p^i,r^i,\overline{\pi}_t^i} - U^{E,\intercal} \eta^{p^i,r^i,\pi^{E,i}} + 2\epsilon_1 + \epsilon_2 + \epsilon_3$$

$$\overset{(11)}{\leqslant} \frac{1}{T} \sum_{t=0}^{T-1} \sum_{i \in [\![N]\!]} \underbrace{U^{E,\intercal} \left( \eta^{p^i,r^i,\overline{\pi}_t^i} - \eta^{p^i,r^i,\pi^{E,i}} \right)}_{\leqslant 0} + 2\epsilon_1 + \epsilon_2 + \epsilon_3 + \epsilon_4$$

$$\overset{(12)}{\leqslant} 2\epsilon_1 + \epsilon_2 + \epsilon_3 + \epsilon_4,$$

where at (1) we upper bound the maximum of positive terms with their sum, at (2) we apply the definition of (non)compatibility, at (3) we first upper bound the supremum of a sum with the sum of the supremum, and then we apply Lemma E.15 w.p. $\delta/3$, and denote $\epsilon_1 := NL\sqrt{2H\epsilon_0} + \sum_{i \in [\![N]\!]} cH\sqrt{\frac{H \log \frac{NH\tau^{E,i}}{\delta \epsilon_0}}{\epsilon_0 \tau^{E,i}}}$, at (4) we denote by $\mathfrak{D}_i$ the set of possible return distributions in environment $i$, at (5) we use the definition of $\widehat{U}$, and realize that all functions $U \in \mathfrak{C}_L(\widehat{U})$ can be constructed based on $T$ functions $U_0 \in \mathfrak{C}_L(\overline{U}_0), \ldots, U_{T-1} \in \mathfrak{C}_L(\overline{U}_{T-1})$. At (6) we upper bound the maximum of the sum with the sum of maxima, and exchange the two summations, and we add and subtract the dot product between the (discretized) utility $\overline{U}_t$ and the estimate of the return distribution computed at Line 6; moreover, we bring the sup inside the summation. At (7) we upper bound the supremum of the sum with the sum of the supremum, and we apply Lemma E.16 w.p. $\delta/3$, defining $\epsilon_2 := cNH^2\sqrt{\frac{1}{n}\left( \log \frac{SAHN}{\delta} + (S-1)\log\left(e(1+n/(S-1))\right)\right)} + NHL\epsilon_0 + c'HN\sqrt{\frac{\log \frac{NT}{\delta}}{K}}$, and we add and subtract a term, at (8) we apply Theorem H.2 from Schlaginhaufen & Kamgarpour (2024) since set $\underline{\mathfrak{U}}_L$ is closed and convex, where $D := \max_{\overline{U},\overline{U}' \in \underline{\mathfrak{U}}_L} \|\overline{U} - \overline{U}'\|_2 = \sqrt{d - 2}H = \sqrt{\lceil H/\epsilon_0 \rceil - 1}H \leqslant H\sqrt{H/\epsilon_0}$ (recall that we consider increasing and not strictly-increasing utilities),[12] and $\max_{\overline{U} \in \underline{\mathfrak{U}}_L} \|\nabla \sum_{i \in [\![N]\!]} \overline{U}^\intercal(\widehat{\eta}_t^i - \widehat{\eta}^{E,i})\|_2 = \|\sum_{i \in [\![N]\!]} \widehat{\eta}_t^i - \widehat{\eta}^{E,i}\|_2 \leqslant \sum_{i \in [\![N]\!]} \|\widehat{\eta}_t^i\|_1 + \|\widehat{\eta}^{E,i}\|_1 = 2N =: G$ (because $\widehat{\eta}_t^i$ and $\widehat{\eta}^{E,i}$ are probability distributions), with learning rate $\alpha = D/(G\sqrt{T}) = H\sqrt{d-2}/(2N\sqrt{T}) = \sqrt{\lceil H/\epsilon_0 \rceil - 1}H/(2N\sqrt{T})$, at (9) we upper bound the minimum over utilities with a specific choice of utility, $\overline{U}^E$, and we add and subtract a term; note that $\overline{U}^E \in \underline{\mathfrak{U}}_L$ corresponds to the expert's utility $U^E \in \underline{\mathfrak{U}}_L$ (by hypothesis), i.e., for all $y \in \mathcal{Y} : \overline{U}^E(y) = U^E(y)$. Note that, by hypothesis, $U^E$ makes all the expert policies optimal, i.e., $\forall i \in [\![N]\!] : U^{E,\intercal} \eta^{p^i,r^i,\pi^{E,i}} = \sup_\pi U^{E,\intercal} \eta^{p^i,r^i,\pi}$. At (10) we note that, under the good event of Lemma E.15, we can provide an upper bound using the term in Lemma E.15 (since $U^E \in \underline{\mathfrak{U}}_L$); in addition, we sum and subtract a term that depends on some policy $\overline{\pi}_t^i$, whose existence is guaranteed by Lemma E.17, which we apply at the next step. At (11) we apply Lemma E.17 w.p. $\delta/3$, and we define as $\epsilon_4$ the upper bound times $N$. Finally, at (12) we use the hypothesis that utility $U^E$ makes the expert policy optimal in all environments.

We want that $2\epsilon_1 + \epsilon_2 + \epsilon_3 + \epsilon_4 \leqslant \epsilon$. We can rewrite the sum as:

$$2\epsilon_1 + \epsilon_2 + \epsilon_3 + \epsilon_4$$

$$= \left( 2NL\sqrt{2H\epsilon_0} + \frac{3}{2}LNH\epsilon_0 \right) + c\frac{HN\sqrt{H}}{\sqrt{\epsilon_0 T}}$$

---

[12]The maximum is attained by discretized utilities $\overline{U}, \overline{U}'$ that assign, respectively, $\overline{U}(y) = 0$ and $\overline{U}'(y) = H$ to all the $y \in \mathcal{Y} \setminus \{y_1, y_d\}$.

$$+ c' \sum_{i \in [\![N]\!]} H \sqrt{\frac{H \log \frac{NH\tau^{E,i}}{\delta \epsilon_0}}{\epsilon_0 \tau^{E,i}}} + c'' NH \sqrt{\frac{\log \frac{NT}{\delta}}{K}}$$

$$+ c''' NH^2 \sqrt{\frac{1}{n} \left( \log \frac{SAHN}{\delta} + (S-1) \log \left( e(1 + n/(S-1)) \right) \right)}.$$

By imposing each term smaller than $\epsilon/5$, we find that it suffices that

$$\begin{cases} \epsilon_0 = \frac{\epsilon^2}{80 N^2 L^2 H} \\ T \geqslant \mathcal{O}\left( \frac{N^2 H^3}{\epsilon_0 \epsilon^2} \right) \geqslant \mathcal{O}\left( \frac{N^4 H^4 L^2}{\epsilon^4} \right) \\ \tau^{E,i} \geqslant \widetilde{\mathcal{O}}\left( \frac{H^3 N^2 \log \frac{NH}{\delta \epsilon_0}}{\epsilon_0 \epsilon^2} \right) \geqslant \widetilde{\mathcal{O}}\left( \frac{H^4 N^4 L^2 \log \frac{NHL}{\delta \epsilon}}{\epsilon^4} \right) & \forall i \in [\![N]\!] \\ K \geqslant \widetilde{\mathcal{O}}\left( \frac{N^2 H^2 \log \frac{NT}{\delta}}{\epsilon^2} \right) \geqslant \widetilde{\mathcal{O}}\left( \frac{N^2 H^2 \log \frac{NHL}{\delta \epsilon}}{\epsilon^2} \right) \\ \tau^i \geqslant \widetilde{\mathcal{O}}\left( \frac{N^2 SAH^5}{\epsilon^2} \left( S + \log \frac{SAHN}{\delta} \right) \right) & \forall i \in [\![N]\!] \end{cases},$$

where we have used that $\tau^i = SAHn$ for all $i \in [\![N]\!]$, and also used Lemma J.3 of Lazzati et al. (2024b).

The statement of the theorem follows through the application of a union bound. $\qquad \square$

**Lemma E.15.** *Let $\delta \in (0,1)$. Then, it holds that, w.p. at least $1 - \delta$:*

$$\sup_{U \in \underline{\mathcal{U}}_L} \sum_{i \in [\![N]\!]} \left| U^{\intercal} \widehat{\eta}^{E,i} - J^{\pi^{E,i}}(U; p^i, r^i) \right| \leqslant NL\sqrt{2H\epsilon_0} + \sum_{i \in [\![N]\!]} cH \sqrt{\frac{H \log \frac{NH\tau^{E,i}}{\delta \epsilon_0}}{\epsilon_0 \tau^{E,i}}},$$

*where $c$ is some positive constant.*

*Proof.* We can make the same derivation as in the proof of Theorem 5.1 to upper bound the objective with the sum of two terms, which can then be bounded using Lemma E.1 and the expression (Eq. (13)) obtained in the proof of Lemma E.4 w.p. $\delta/N$:

$$\sup_{U \in \underline{\mathcal{U}}_L} \sum_{i \in [\![N]\!]} \left| U^{\intercal} \widehat{\eta}^{E,i} - J^{\pi^{E,i}}(U; p^i, r^i) \right|$$

$$\leqslant L \sum_{i \in [\![N]\!]} w_1(\eta^{p^i, r^i, \pi^{E,i}}, \mathrm{Proj}_{\mathcal{C}}(\eta^{p^i, r^i, \pi^{E,i}}))$$

$$+ \sum_{i \in [\![N]\!]} \sup_{\overline{U}' \in [0,H]^d} \left| \mathbb{E}_{G \sim \mathrm{Proj}_{\mathcal{C}}(\eta^{p^i, r^i, \pi^{E,i}})} [\overline{U}'(G)] - \mathbb{E}_{G \sim \widehat{\eta}^{E,i}} [\overline{U}'(G)] \right|$$

$$\leqslant LN\sqrt{2H\epsilon_0} + \sum_{i \in [\![N]\!]} cH \sqrt{\frac{H \log \frac{NH\tau^{E,i}}{\delta \epsilon_0}}{\epsilon_0 \tau^{E,i}}}.$$

The result follows through the application of the union bound. $\qquad \square$

**Lemma E.16.** *Let $\delta \in (0,1)$. With probability at least $1 - \delta$, for all $t \in \{0, 1, \ldots, T-1\}$, for all $i \in [\![N]\!]$, it holds that:*

$$\sup_{U_t \in \mathfrak{C}_L(\overline{U}_t)} \max_{\eta \in \mathfrak{D}_i} U_t^{\intercal} \eta - \overline{U}_t^{\intercal} \widehat{\eta}_t^i \leqslant cH^2 \sqrt{\frac{1}{n} \left( \log \frac{SAHN}{\delta} + (S-1) \log \left( e(1 + n/(S-1)) \right) \right)}$$

$$+ HL\epsilon_0 + c' H \sqrt{\frac{\log \frac{NT}{\delta}}{K}},$$

*where $c, c'$ are some positive constants.*

*Proof.* We use the notation in Section 5. In particular, let policy $\widehat{\pi}_t^{*,i}$ be the optimal policy in the RS-MDP $\widehat{\mathcal{M}}_{\overline{U}_t}^i := (\mathcal{S}^i, \mathcal{A}^i, H, s_0^i, \widehat{p}^i, \overline{r}^i, \overline{U}_t)$, i.e.:

$$J^{\widehat{\pi}_t^{*,i}}(\overline{U}_t; \widehat{p}^i, \overline{r}^i) = J^*(\overline{U}_t; \widehat{p}^i, \overline{r}^i) = J^*(U_t; \widehat{p}^i, \overline{r}^i),$$

where the last passage holds trivially for all $U_t \in \mathfrak{C}_L(\overline{U}_t)$ (because there is no evaluation of utility outside $\mathcal{Y}$).

Thus, for all $t \in \{0, 1, \ldots, T-1\}$, we have:

$$\sup_{U_t \in \mathfrak{C}_L(\overline{U}_t)} \max_{\eta \in \mathfrak{D}_i} U_t^{\mathsf{T}} \eta - \overline{U}_t^{\mathsf{T}} \widehat{\eta}_t^i \pm J^*(U_t; \widehat{p}^i, \overline{r}^i)$$

$$\overset{(1)}{\leqslant} \sup_{U_t \in \mathfrak{C}_L(\overline{U}_t)} \left| J^*(U_t; p^i, r^i) - J^*(U_t; \widehat{p}^i, \overline{r}^i) \right| + \left| \overline{U}_t^{\mathsf{T}} \left( \widehat{\eta}_t^i - \eta^{\widehat{p}^i, \overline{r}^i, \widehat{\pi}_t^{*,i}} \right) \right|$$

$$\overset{(2)}{\leqslant} HL\epsilon_0 + cH^2 \sqrt{\frac{1}{n} \left( \log \frac{SAHN}{\delta} + (S-1) \log \left( e(1 + n/(S-1)) \right) \right)}$$

$$+ \left| \overline{U}_t^{\mathsf{T}} \left( \widehat{\eta}_t^i - \eta^{\widehat{p}^i, \overline{r}^i, \widehat{\pi}_t^{*,i}} \right) \right|$$

$$\overset{(3)}{\leqslant} HL\epsilon_0 + cH^2 \sqrt{\frac{1}{n} \left( \log \frac{SAHN}{\delta} + (S-1) \log \left( e(1 + n/(S-1)) \right) \right)}$$

$$+ c'H \sqrt{\frac{\log \frac{NT}{\delta}}{K}},$$

where at (1) we have applied the triangle inequality, and realized that in the second term there is no dependence on the value of utility outside of $\mathcal{Y}$; moreover, we have used that $J^*(U_t; \widehat{p}^i, \overline{r}^i) = \overline{U}_t^{\mathsf{T}} \eta^{\widehat{p}^i, \overline{r}^i, \widehat{\pi}_t^{*,i}}$ by definition of policy $\widehat{\pi}_t^{*,i}$. At (2) we apply Lemma E.13 (our $J^*(U_t; \widehat{p}^i, \overline{r}^i)$ has the same meaning of $\widehat{J}^*(U)$ in the lemma, and we upper bound $\sup_{U_t \in \mathfrak{C}_L(\overline{U}_t)}$ with $\sup_{U \in \underline{\mathfrak{U}}_L}$) w.p. $\delta/(2N)$,[13] and we keep the confidence bound explicit, and we upper bound $d \leqslant H/\epsilon_0 + 1$, and at (3) we observe that $\widehat{\eta}_t^i$ is the empirical estimate of distribution $\eta^{\widehat{p}^i, \overline{r}^i, \widehat{\pi}_t^{*,i}}$ (see Line 6) obtained through the sampling of $K$ sample returns $G_1, G_2, \ldots, G_K \overset{\text{i.i.d.}}{\sim} \eta^{\widehat{p}^i, \overline{r}^i, \widehat{\pi}_t^{*,i}}$. Indeed, note that the policy $\widehat{\psi}_t^{*,i}$, computed at Line 4 and optimal for $\mathfrak{E}[\widehat{\mathcal{M}}_{\overline{U}_t}^i] = (\{\mathcal{S}^i \times \mathcal{Y}_h\}_h, \mathcal{A}^i, H, s_0^i, \widehat{\mathfrak{p}}^i, \mathfrak{r}_t^i)$,[14] provides policy $\widehat{\pi}_t^{*,i}$ through the formula in Section 2, thus Line 5 is actually simulating $\widehat{\pi}_t^{*,i}$ in MDP $\widehat{\mathcal{M}}^i$. Therefore, we can apply Hoeffding's inequality (e.g., see Lemma E.3) w.p. $\delta/(2TN)$.

The result follows through the application of the union bound.

We remark that in one case we use probability $\delta/(2N)$ (without $T$) while in the other we use $\delta/(2NT)$ (with $T$), because in the former we provide a guarantee for all possible utilities w.r.t. the optimal performance, thus all the $T$ steps are already included; instead, in the latter, we provide a guarantee for a single utility and for a single policy at a specific $t \in \{0, \ldots, T-1\}$, thus we have to compute a union bound with $T$. $\qquad\square$

**Lemma E.17.** *Let $\delta \in (0,1)$. With probability at least $1-\delta$, for all $i \in [\![N]\!]$ and $t \in \{0, \ldots, T-1\}$, under the good event in Lemma E.16, there exists a policy $\overline{\pi}_t^i$ such that:*

$$\overline{U}^{E,\mathsf{T}} \widehat{\eta}_t^i - U^{E,\mathsf{T}} \eta^{p^i, r^i, \overline{\pi}^i} \leqslant LH\epsilon_0/2 + cH \sqrt{\frac{\log \frac{NT}{\delta}}{K}}$$

$$+ c'H^2 \sqrt{\frac{1}{n} \left( \log \frac{SAHN}{\delta} + (S-1) \log \left( e(1 + n/(S-1)) \right) \right)},$$

*where $c, c'$ are positive constants.*

---

[13]We remark that, in doing so, we can still apply Proposition 3 of Wu & Xu (2023) inside the proof of Lemma E.13 even though we consider *increasing* utilities instead of *strictly-increasing* utilities; indeed, it is trivial to observe that the proof of Proposition 3 of Wu & Xu (2023) does not depend on such property.

[14]See Section 2 for the meaning of $\widehat{\mathfrak{p}}^i$ and $\mathfrak{r}_t^i$; we use $\mathcal{Y}_h$ for all $h$ in the state space instead of the sets of partial returns $\{\mathcal{G}_h^{\widehat{p}^i, \overline{r}^i}\}_h$ in order to obtain policy $\widehat{\psi}_t^{*,i}$ supported on the entire $\mathcal{S} \times \mathcal{Y}_h$ space, and to make it compliant with Algorithm 4

*Proof.* First, simply observe that $\widehat{\eta}_t^i$ is the empirical estimate (see Line 6) of $\eta^{\widehat{p}^i, \overline{r}^i, \widehat{\pi}_t^{*,i}}$, thus, similarly to the proof of Lemma E.16, for all $i \in [\![N]\!]$ and $t \in \{0, 1, \dots, T-1\}$, we can apply Hoeffding's inequality w.p. $\delta/(2TN)$:

$$\left| \overline{U}^{E,\intercal} \left( \widehat{\eta}_t^i - \eta^{\widehat{p}^i, \overline{r}^i, \widehat{\pi}_t^{*,i}} \right) \right| \leqslant cH \sqrt{\frac{\log \frac{NT}{\delta}}{K}}.$$

Now, we compare distributions $\eta^{\widehat{p}^i, \overline{r}^i, \widehat{\pi}_t^{*,i}}$ and $\eta^{p^i, \overline{r}^i, \widehat{\pi}_t^{*,i}}$. Through straightforward passages, we can write:

$$\begin{aligned}
|U^{E,\intercal} &\left( \eta^{\widehat{p}^i, \overline{r}^i, \widehat{\pi}_t^{*,i}} - \eta^{p^i, \overline{r}^i, \widehat{\pi}_t^{*,i}} \right)| \\
&= |J^{\widehat{\pi}_t^{*,i}}(\overline{U}^E; \widehat{p}^i, \overline{r}^i) - J^{\widehat{\pi}_t^{*,i}}(\overline{U}^E; p^i, \overline{r}^i)| \\
&= \Big| \sum_{s' \in \mathcal{S}} p_1^i(s'|s_0^i, \widehat{\pi}_{t,1}^{*,i}(s_0^i)) V_2^{\widehat{\pi}_t^{*,i}}(s'; p^i, \overline{r}^i) \\
&\qquad - \sum_{s' \in \mathcal{S}} \widehat{p}_1^i(s'|s_0^i, \widehat{\pi}_{t,1}^{*,i}(s_0^i)) V_2^{\widehat{\pi}_t^{*,i}}(s'; \widehat{p}^i, \overline{r}^i) \Big| \\
&\leqslant \Big| \sum_{s' \in \mathcal{S}} \left( p_1^i(s'|s_0^i, \widehat{\pi}_{t,1}^{*,i}(s_0^i)) - \widehat{p}_1^i(s'|s_0^i, \widehat{\pi}_{t,1}^{*,i}(s_0^i)) \right) V_2^{\widehat{\pi}_t^{*,i}}(s'; p^i, \overline{r}^i) \Big| \\
&\qquad + \sum_{s' \in \mathcal{S}} \widehat{p}_1^i(s'|s_0^i, \widehat{\pi}_{t,1}^{*,i}(s_0^i, 0)) \Big| V_2^{\widehat{\pi}_t^{*,i}}(s'; p^i, \overline{r}^i) - V_2^{\widehat{\pi}_t^{*,i}}(s'; \widehat{p}^i, \overline{r}^i) \Big| \\
&\leqslant \dots \\
&\leqslant \mathbb{E}_{\widehat{p}^i, \overline{r}^i, \widehat{\pi}_t^{*,i}} \Big[ \sum_{h'=1}^{H} \Big| \sum_{s' \in \mathcal{S}} \left( p_{h'}^i(s'|s_{h'}, a_{h'}) - \widehat{p}_{h'}^i(s'|s_{h'}, a_{h'}) \right) V_{h'+1}^{\widehat{\pi}_t^{*,i}}(s'; p^i, \overline{r}^i) \Big| \\
&\qquad \Big| s_1 = s_0^i \Big] \\
&\leqslant H \mathbb{E}_{\widehat{p}^i, \overline{r}^i, \widehat{\pi}_t^{*,i}} \Big[ \sum_{h'=1}^{H} \left\| p_{h'}^i(\cdot|s_{h'}, a_{h'}) - \widehat{p}_{h'}^i(\cdot|s_{h'}, a_{h'}) \right\|_1 \Big| s_1 = s_0^i \Big] \\
&\leqslant H \mathbb{E}_{\widehat{p}^i, \overline{r}^i, \widehat{\pi}_t^{*,i}} \Big[ \sum_{h'=1}^{H} \sqrt{2\mathrm{KL}(p_{h'}^i(\cdot|s_{h'}, a_{h'}) \| \widehat{p}_{h'}^i(\cdot|s_{h'}, a_{h'}))} \Big| s_1 = s_0^i \Big],
\end{aligned}$$

where at the last passage we applied the Pinsker's inequality. Note that the previous derivation was possible as long as as policy $\widehat{\pi}_t^{*,i}$ is defined over all the possible pairs state-cumulative reward $(s, y) \in \mathcal{S} \times \mathcal{Y}_h$ for all $h \in [\![H]\!]$. Since we construct it through policy $\widehat{\psi}_t^{*,i}$, obtained at Line 4, i.e., over the entire enlarged state space $\{\mathcal{S} \times \mathcal{Y}_h\}_h$, then policy $\widehat{\pi}_t^{*,i}$ satsifies such property. Now, in the proof of Lemma E.16 we used Lemma E.14, in which event $\mathcal{E}$ bounds the KL-divergence between transition models. Therefore, under the application of Lemma E.16, it holds that:

$$|U^{E,\intercal} \left( \eta^{\widehat{p}^i, \overline{r}^i, \widehat{\pi}_t^{*,i}} - \eta^{p^i, \overline{r}^i, \widehat{\pi}_t^{*,i}} \right)| \leqslant H^2 \sqrt{\frac{2}{n} \left( \log \frac{SAHN}{\delta} + (S-1)\log\left(e(1 + n/(S-1))\right) \right)},$$

where $n$ is the number of samples takes at each $(s, a, h) \in \mathcal{S} \times \mathcal{A} \times [\![H]\!]$ in the $i \in [\![N]\!]$ MDP.

Therefore, we can finally write:

$$\begin{aligned}
\overline{U}^{E,\intercal} \widehat{\eta}_t^i &- U^{E,\intercal} \eta^{p^i, r^i, \overline{\pi}^i} \pm \overline{U}^{E,\intercal} \eta^{\widehat{p}^i, \overline{r}^i, \widehat{\pi}_t^{*,i}} \pm \overline{U}^{E,\intercal} \eta^{p^i, \overline{r}^i, \widehat{\pi}_t^{*,i}} \\
&= U^{E,\intercal} \left( \eta^{p^i, \overline{r}^i, \widehat{\pi}_t^{*,i}} - \eta^{p^i, r^i, \overline{\pi}^i} \right) + \overline{U}^{E,\intercal} \left( \eta^{\widehat{p}^i, \overline{r}^i, \widehat{\pi}_t^{*,i}} - \eta^{p^i, \overline{r}^i, \widehat{\pi}_t^{*,i}} \right) \\
&\qquad + \overline{U}^{E,\intercal} \left( \widehat{\eta}_t^i - \eta^{\widehat{p}^i, \overline{r}^i, \widehat{\pi}_t^{*,i}} \right) \\
&\overset{(1)}{\leqslant} U^{E,\intercal} \left( \eta^{p^i, \overline{r}^i, \widehat{\pi}_t^{*,i}} - \eta^{p^i, r^i, \overline{\pi}^i} \right) + cH \sqrt{\frac{\log \frac{NT}{\delta}}{K}}
\end{aligned}$$

$$+ c'H^2\sqrt{\frac{2}{n}\Big(\log\frac{SAHN}{\delta} + (S-1)\log\big(e(1+n/(S-1))\big)\Big)}$$

$$\overset{(2)}{\leqslant} LH\epsilon_0/2 + cH\sqrt{\frac{\log\frac{NT}{\delta}}{K}}$$

$$+ c'H^2\sqrt{\frac{2}{n}\Big(\log\frac{SAHN}{\delta} + (S-1)\log\big(e(1+n/(S-1))\big)\Big)},$$

where at (1) we have used the bounds derived earlier, and at (2) we have applied Lemma E.18, noticing that we can choose policy $\overline{\pi}^i$ as we wish, and using that $k \leqslant \epsilon_0/2$.

$\square$

**Lemma E.18.** *Let $\mathcal{M}_1 = (\mathcal{S}, \mathcal{A}, H, s_0, p, r^1)$ and $\mathcal{M}_2 = (\mathcal{S}, \mathcal{A}, H, s_0, p, r^2)$ be two MDPs with deterministic rewards that differ only in the reward function $r^1 \neq r^2$, and assume that, for all $(s, a, h) \in \mathcal{S} \times \mathcal{A} \times [\![H]\!]$, it holds that $|r_h^1(s,a) - r_h^2(s,a)| \leqslant k$, for some $k \geqslant 0$. Let $\pi^1$ be an arbitrary (potentially non-Markovian) policy that induces, in $\mathcal{M}_1$, the distribution over returns $\eta^{p,r^1,\pi^1}$. Then, there exists a policy $\pi^2$ that induces in $\mathcal{M}_2$ the distribution $\eta^{p,r^2,\pi^2}$ such that:*

$$\sup_{U \in \underline{\mathfrak{U}}_L} \Big| \mathbb{E}_{G\sim\eta^{p,r^1,\pi^1}}[U(G)] - \mathbb{E}_{G\sim\eta^{p,r^2,\pi^2}}[U(G)] \Big| \leqslant LHk.$$

*Proof.* A non-Markovian policy like $\pi^1$, in its most general form, prescribes actions at stages $h \in [\![H]\!]$ depending on the sequence of state-action-reward $(s_1, a_1, r_1, s_2, a_2, r_2, \ldots, s_{h-1}, a_{h-1}, r_{h-1}, s_h)$ received so far. Since, by hypothesis, the reward functions are deterministic (see also Section 2), then it is clear that the information contained in the rewards received so far ($\{r_1, r_2, \ldots, r_{h-1}\}$) is already contained in the state-action pairs received $(s_1, a_1, s_2, a_2, \ldots, s_{h-1}, a_{h-1}, s_h)$ (indeed, for deterministic reward $r^1$, we have that $r_1 = r_1^1(s_1, a_1), r_2 = r_2^1(s_2, a_2)$, and so on). This means that, for any non-Markovian policy in the MDP $\mathcal{M}_1$, since it coincides with $\mathcal{M}_2$ except for the deterministic reward function, it is possible to construct a policy $\pi^2$ that induces the same distribution over *state-action* trajectories, i.e., for any state-action trajectory $\omega = (s_1, a_1, s_2, a_2, \ldots, s_{H-1}, a_{H-1}, s_H, a_H, s_{H+1}) \in \Omega$, it holds $\mathbb{P}_{p,r^1,\pi^1}(\omega) = \mathbb{P}_{p,r^2,\pi^2}(\omega)$.

Therefore, we can write:

$$\sup_{U \in \underline{\mathfrak{U}}_L} \Big| \mathbb{E}_{G\sim\eta^{p,r^1,\pi^1}}[U(G)] - \mathbb{E}_{G\sim\eta^{p,r^2,\pi^2}}[U(G)] \Big|$$

$$\overset{(1)}{=} \sup_{U \in \underline{\mathfrak{U}}_L} \Big| \sum_{\omega\in\Omega} \mathbb{P}_{p,r^1,\pi^1}(\omega) U\Big( \sum_{(s,a,h)\in\omega} r_h^1(s,a) \Big)$$

$$- \sum_{\omega\in\Omega} \mathbb{P}_{p,r^2,\pi^2}(\omega) U\Big( \sum_{(s,a,h)\in\omega} r_h^2(s,a) \Big) \Big|$$

$$\overset{(2)}{=} \sup_{U \in \underline{\mathfrak{U}}_L} \Big| \sum_{\omega\in\Omega} \mathbb{P}_{p,r^1,\pi^1}(\omega) U\Big( \sum_{(s,a,h)\in\omega} r_h^1(s,a) \Big)$$

$$- \sum_{\omega\in\Omega} \mathbb{P}_{p,r^1,\pi^1}(\omega) U\Big( \sum_{(s,a,h)\in\omega} r_h^2(s,a) \Big) \Big|$$

$$= \sup_{U \in \underline{\mathfrak{U}}_L} \Big| \sum_{\omega\in\Omega} \mathbb{P}_{p,r^1,\pi^1}(\omega) \Big( U\Big( \sum_{(s,a,h)\in\omega} r_h^1(s,a) \Big) - U\Big( \sum_{(s,a,h)\in\omega} r_h^2(s,a) \Big) \Big) \Big|$$

$$\overset{(3)}{\leqslant} \sup_{U \in \underline{\mathfrak{U}}_L} \sum_{\omega\in\Omega} \mathbb{P}_{p,r^1,\pi^1}(\omega) \Big| U\Big( \sum_{(s,a,h)\in\omega} r_h^1(s,a) \Big) - U\Big( \sum_{(s,a,h)\in\omega} r_h^2(s,a) \Big) \Big|$$

$$\overset{(4)}{\leqslant} \sum_{\omega\in\Omega} \mathbb{P}_{p,r^1,\pi^1}(\omega) L \Big| \sum_{(s,a,h)\in\omega} (r_h^1(s,a) - r_h^2(s,a)) \Big|$$

$$\overset{(5)}{\leqslant} \sum_{\omega\in\Omega} \mathbb{P}_{p,r^1,\pi^1}(\omega) L \sum_{(s,a,h)\in\omega} \Big| r_h^1(s,a) - r_h^2(s,a) \Big|$$

$$\overset{(6)}{\leqslant} \sum_{\omega \in \Omega} \mathbb{P}_{p,r^1,\pi^1}(\omega) L \sum_{(s,a,h) \in \omega} k$$

$$= LHk,$$

where at (1) we use the fact that the expected utility w.r.t. the distribution over returns can be computed using the probability distribution over state-action trajectories (since the rewards are deterministic), at (2) we use that policy $\pi^2$ is constructed exactly to match the distribution over state-action trajectories, at (3) we apply triangle inequality, at (4) we use the fact that all utilities $U \in \underline{\mathfrak{U}}_L$ are $L$-Lipschitz, i.e., for all $x, y \in [0, H]$: $|U(x) - U(y)| \leqslant L|x - y|$, at (5) we apply again the triangle inequality, and at (6) we use the hypothesis that $r^1, r^2$ are close to each other by parameter $k$. $\qquad\square$

## F    EXPERIMENTAL DETAILS

In this appendix, we collect additional information about the experiments described in Section 6. Appendix F.1 presents formally the MDP used for the collection of the data along with the questions posed to the participants. Appendix F.2 describes what is a Standard Gamble (Wakker, 2010) and how it has been used to construct the utility $U_{\text{SG}}$ of the participants. Finally, Appendices F.3 and F.4 contain, respectively, additional details on Experiment 1 and 2.

### F.1    DATA DESCRIPTION

Below, we describe the data collected.

#### F.1.1    CONSIDERED MDP.

The 15 participants analyzed in the study have been provided with complete access to the MDP in Figure 9, which we will denote by $\mathcal{M}$. In other words, the participants *know the transition model and the reward function of $\mathcal{M}$ everywhere*.

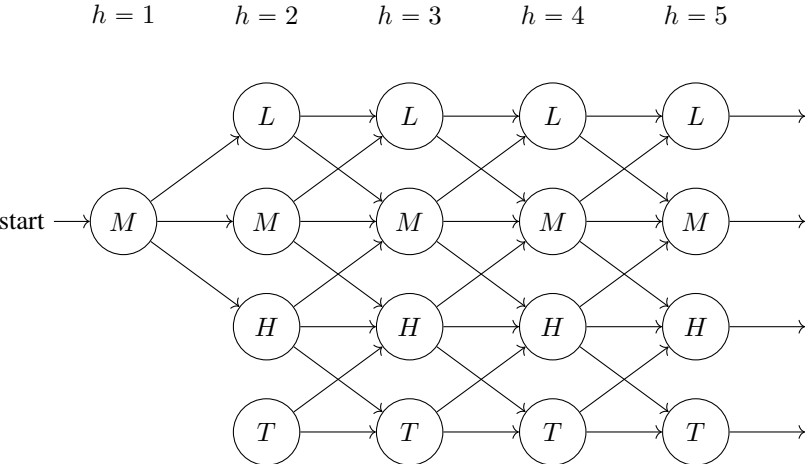

Figure 9: The MDP used for data collection.

Intuitively, states L (Low), M (Medium), H (High), and T (Top), represent 4 "levels" so that the received reward increases when playing actions in "higher" states instead of "lower" states. Formally, MDP $\mathcal{M} = (\mathcal{S}, \mathcal{A}, H, s_0, p, r)$ has four states $\mathcal{S} = \{L, M, H, T\}$, and three actions for each state $\mathcal{A} = \{a_0, a_+, a_-\}$. The horizon is $H = 5$, i.e., the agent has to take 5 actions. The initial state is $s_0 = M$. The transition model $p$ is stationary, i.e., it does not depend on the stage $h \in [\![H]\!]$. Specifically, $p$ is depicted in Table 2. The intuition is that action $a_0$ keeps the agent in the same state deterministically, while action $a_+$ tries to bring the agent to the higher state with probability $1/3$, and action $a_-$ sometimes make the agent "fall down" to the lower state with probability $1/5$.

| $p$ | $L$ | $M$ | $H$ | $T$ |
|---|---|---|---|---|
| $(L, a_0)$ | 1 | 0 | 0 | 0 |
| $(L, a_+)$ | 2/3 | 1/3 | 0 | 0 |
| $(L, a_-)$ | 1 | 0 | 0 | 0 |
| $(M, a_0)$ | 0 | 1 | 0 | 0 |
| $(M, a_+)$ | 0 | 2/3 | 1/3 | 0 |
| $(M, a_-)$ | 1/5 | 4/5 | 0 | 0 |
| $(H, a_0)$ | 0 | 0 | 1 | 0 |
| $(H, a_+)$ | 0 | 0 | 2/3 | 1/3 |
| $(H, a_-)$ | 0 | 1/5 | 4/5 | 0 |
| $(T, a_0)$ | 0 | 0 | 0 | 1 |
| $(T, a_+)$ | 0 | 0 | 0 | 1 |
| $(T, a_-)$ | 0 | 0 | 1/5 | 4/5 |

Table 2: The transition model $p$ of MDP $\mathcal{M}$.

The reward function $r : \mathcal{S} \times \mathcal{A} \times [\![H]\!] \to \mathbb{R}$ is deterministic, stationary, and depends only the state-action pair played. The specific values are depicted in Table 3. Note that we have written the reward values as numbers in $[0€, 1000€]$, to provide a monetary interpretation. Nevertheless, we will rescale the interval to $[0, 1]$ during the analysis for normalization. Observe that the same actions played in "higher" states (e.g., $H$ or $T$) provide higher rewards than when played in "lower" states (e.g., $L$ or $M$). Moreover, notice that action $a_+$, which is the only action that tries to increase the state, does not provide reward at all, while the risky action $a_-$, which sometimes decreases the state, always provides double the reward than "default" action $a_0$.

| | $L$ | $M$ | $H$ | $T$ |
|---|---|---|---|---|
| $a_0$ | 0€ | 30€ | 100€ | 500€ |
| $a_+$ | 0€ | 0€ | 0€ | 0€ |
| $a_-$ | 0€ | 60€ | 200€ | 1000€ |

Table 3: The reward function $r$ of MDP $\mathcal{M}$.

### F.1.2 INTUITION BEHIND AGENTS BEHAVIOR.

The reward is interpreted as money. Playing MDP $\mathcal{M}$ involves a trade-off between playing action $a_+$, which gives no money but potentially allows to collect more money in the future (by reaching "higher" states), and action $a_-$, which provides the greatest amount of money immediately, but potentially reduces the amount of money which can be earned in the future. Action $a_0$, being deterministic, provides a reference point, so that deterministically playing action $a_0$ for all the $H = 5$ stages gives to the agent $30 \times 5 = 150€$. Thus, playing actions $a_+, a_-$ other than $a_0$ means that the agent accepts some risk to try to increase its earnings.

### F.1.3 QUESTIONS ASKED TO THE PARTICIPANTS

We remark that the participants have enough background knowledge to understand the MDP described. To each participant, we ask which action in $\{a_0, a_+, a_-\}$ it would play if it was in a certain state $s$, stage $h$, with cumulative reward up to now $y$, for many different values of triples $(s, h, y) \in \mathcal{S} \times [\![H]\!] \times [0€, 5000€]$. Specifically, the values of triples $s, h, y$ considered are:

$(M, 1, 0€)$ $\quad$ $(M, 2, 0€)$ $\quad$ $(M, 2, 30€)$ $\quad$ $(M, 2, 60€)$ $\quad$ $(H, 2, 0€)$

$(M, 3, 0€)$ $\quad$ $(M, 3, 30€)$ $\quad$ $(M, 3, 60€)$ $\quad$ $(M, 3, 200€)$ $\quad$ $(H, 3, 0€)$

$(H, 3, 30€)$ $\quad$ $(H, 3, 60€)$ $\quad$ $(H, 3, 200€)$ $\quad$ $(T, 3, 0€)$ $\quad$ $(M, 4, 0€)$

$(M, 4, 30€)$ $\quad$ $(M, 4, 60€)$ $\quad$ $(M, 4, 90€)$ $\quad$ $(M, 4, 120€)$ $\quad$ $(M, 4, 150€)$

$(M, 4, 180€)$ $\quad$ $(M, 4, 300€)$ $\quad$ $(M, 4, 400€)$ $\quad$ $(H, 4, 0€)$ $\quad$ $(H, 4, 30€)$

$(H, 4, 60€)$ $\quad$ $(H, 4, 100€)$ $\quad$ $(H, 4, 130€)$ $\quad$ $(H, 4, 200€)$ $\quad$ $(H, 4, 300€)$

$(H, 4, 1000€)$ $\quad$ $(T, 4, 0€)$ $\quad$ $(T, 4, 60€)$.

From state $L$, we assume all participants always play action $a_+$ since it is the only rational strategy. Moreover, from stage $h = 5$, we assume that all participants always play action $a_-$ since, again, it is the only rational strategy.

In all other possible combinations of values of $s, h, y$, we "interpolate" by considering the action recommended by the participant in the closest $y'$ to $y$, in the same $s, h$.

### F.1.4 THE RETURN DISTRIBUTION OF THE PARTICIPANTS' POLICIES

We now present the return distribution of the policies prescribed by the participants. Specifically, we have simulated 10000 times the policies of the participants, and we have computed the empirical estimate of their return distributions. Such values are reported in Figures 10, 11, 12, 13, and 14, where we use notation $\eta_i^E$ to denote the return distribution of participant $i$, with $i \in [\![15]\!]$.

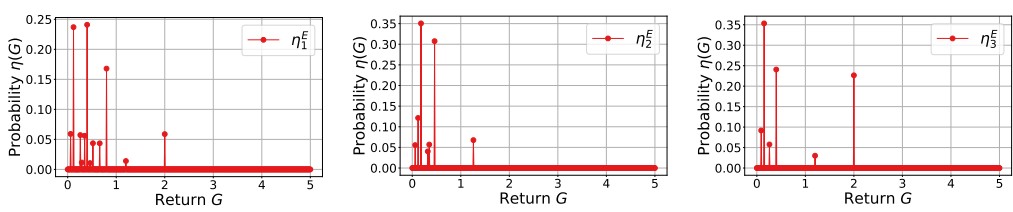

Figure 10: Plot of $\eta_1^E, \eta_2^E$, and $\eta_3^E$.

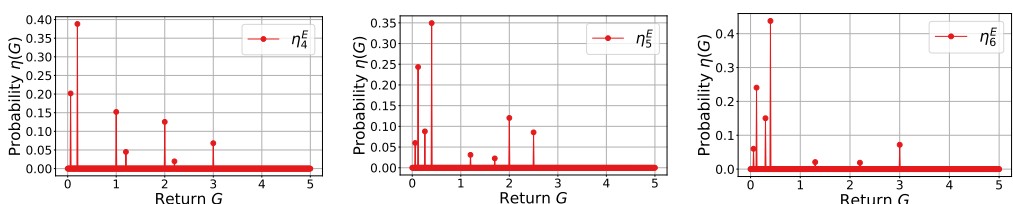

Figure 11: Plot of $\eta_4^E, \eta_5^E$, and $\eta_6^E$.

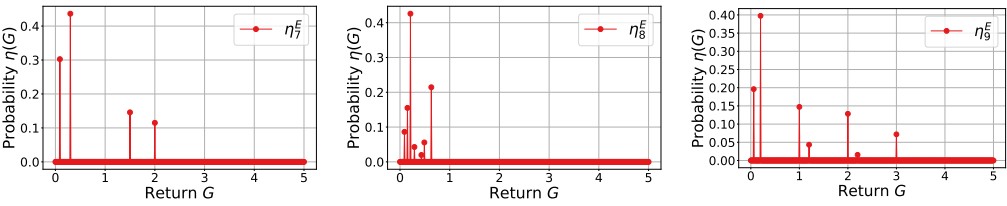

Figure 12: Plot of $\eta_7^E, \eta_8^E$, and $\eta_9^E$.

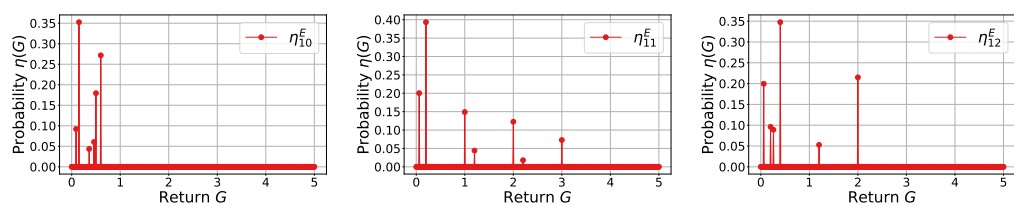

Figure 13: Plot of $\eta_{10}^E, \eta_{11}^E$, and $\eta_{12}^E$.

### F.2 STANDARD GAMBLE DATA

**Standard Gamble (SG).** The Standard Gamble (SG) method (e.g., see Section 2.5 of Wakker (2010)) is a common method for inferring the von Neumann-Morgenstern (vNM) utility function of

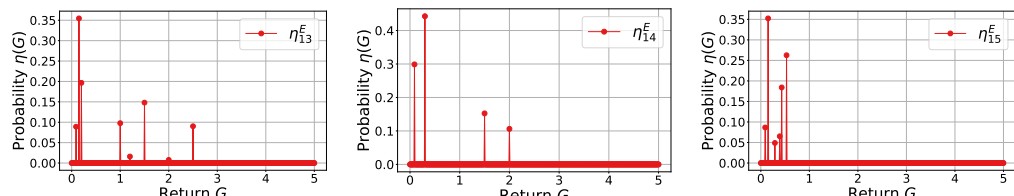

Figure 14: Plot of $\eta_{13}^E, \eta_{14}^E$, and $\eta_{15}^E$.

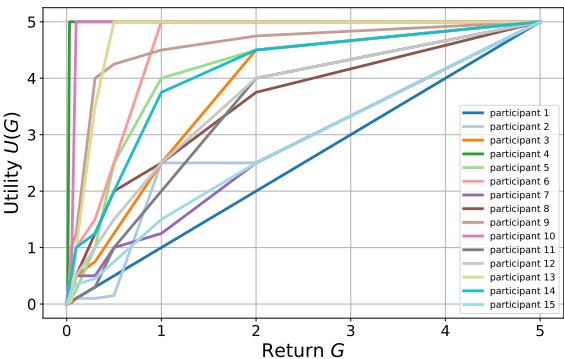

Figure 15: The SG utilities of the participants.

an agent. Observe Figure 16. In a SG, the agent has to decide between two options: A sure option (e.g., $x = 30€$), in which the prize is obtained with probability 1, and a lottery between two prizes (e.g., 5000€ and 0€), in which the best prize (5000€) is received with probability $p$. For any value of $x$, the agent has to answer what is the probability $p$ that, from his perspective, makes the two options (i.e., $x$ for sure, or the lottery) *indifferent*.

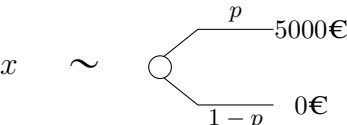

Figure 16: The SG used for data collection.

Given the probability $p$, we have that the utility $U$ of the agent for $x$ is:
$$U(x) = p \cdot U(5000) + (1-p) \cdot U(0) = p,$$
since, by normalization conditions, we have $U(0) = 0$ and $U(5000) = 1$.

**Our SG.** We have asked the 15 participants to the study to answer some SG questions, which allows us to fit a vNM utility function for each of them (which we call $U_{\text{SG}}$). Specifically, we have asked to answer 8 different SG questions, in which the $x$ value in Figure 16 has been replaced by:
$$10€, 30€, 50€, 100€, 300€, 500€, 1000€, 2000€.$$
Next, we linearly interpolate the computed utilities, obtaining the functions in Figure 15.

It should be remarked that this model considers single decisions (i.e., $H = 1$), while in MDPs there is a sequence of decisions to be taken over time, specifically over a certain time horizon $H$.

### F.3 DETAILS EXPERIMENT 1

The utilities $U_{\text{sqrt}}, U_{\text{square}}$, and $U_{\text{linear}}$ can be formally defined as: $U_{\text{sqrt}}(G) := \sqrt{5G}, U_{\text{square}}(G) := G^2/5, U_{\text{linear}}(G) := G$. They are depicted in Figure 17.

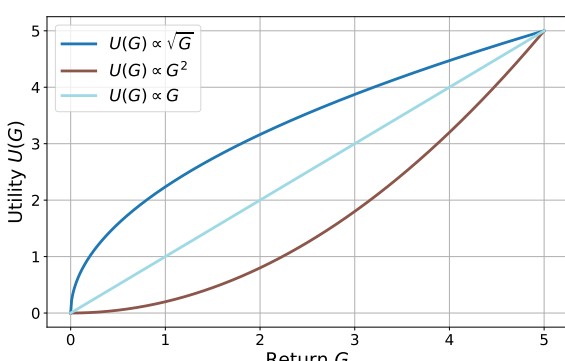

Figure 17: A plot of utilities $U_{\text{sqrt}}, U_{\text{square}}, U_{\text{linear}}$.

The experiment has been conducted collecting 10000 trajectories to estimate the return distribution of each participant's policy, and 10000 trajectories for estimating the return distribution of the optimal policy, which has been computed exactly through value iteration. We have executed 5 simulations with different seeds, and the relative (non)compatibility values written in the table in Section 6 are the average over the 5 simulations.

For the experiment, we use the true transition model, and we remark that the reward function considered, when discretized, coincides with itself, i.e., we do not incur in estimation error of the transition model nor in approximation error for the discretization.

The experiment has been conducted in less than 1 hour on a personal computer with processor AMD Ryzen 5 5500U with Radeon Graphics (2.10 GHz), with 8,00 GB of RAM.

## F.4   DETAILS EXPERIMENT 2

Experiment 2 is made of two parts, the first in which we execute it on the MDP adopted also in Experiment 1, and the other where we use simulated data. We describe here the former, while we present the latter more in detail in Appendix F.4.3.

We consider the policy of the 10th participant to the survey, and we execute **TRACTOR-UL** multiple times with varying values of the input parameters, specifically: we always use $K = 10000$ trajectories for estimating the return distribution of the 10th participant's policy, and the return distribution of the optimal policies computed along the way; we make 5 runs with each combination of parameters with different seeds. We execute for $T = 70$ iterations using Lipschitz constant $L = 10$, which means that we consider only utilities $U \in \overline{\mathfrak{U}}_L$ satisfying $|U(G) - U(G')| \leqslant 10|G - G'|$ for all $G, G' \in [0,5]$ (the horizon is 5). As initial utility $\overline{U}_0$, we try $U_{\text{sqrt}}, U_{\text{square}}$, and $U_{\text{linear}}$, and as learning rates we try $0.01, 0.5, 5, 100, 1000, 10000$.

The experiment has been conducted on the same personal computer as experiment 1, in some hours.

We note that the choice of $\overline{U}_0$ is rather irrelevant for the shape of the extracted $\widehat{U}$, but it matters for its "location", as shown in Fig. 18.

To view the sequence of utilities extracted by **TRACTOR-UL** during the run, see Appendix F.4.1, while in Appendix F.4.2 we explain better why the best learning rate is large.

### F.4.1   THE SEQUENCE OF UTILITIES EXTRACTED BY **TRACTOR-UL**

We now present some plots representing the sequence of utilities extracted by **TRACTOR-UL** during its execution. Specifically, we consider initial utility $\overline{U}_0 = U_{\text{square}}$, and we use learning rates $\alpha \in [0.01, 0.5, 5, 100, 1000, 10000]$. We plot the sequence of utilities considered by **TRACTOR-UL** during its execution in Figures 19, 20, and 21, where we adopt notation that $U_t$ denotes the utility extracted at iteration $t$, and the number in the legend represents the (non)compatibility of that utility. We consider again participant 10.

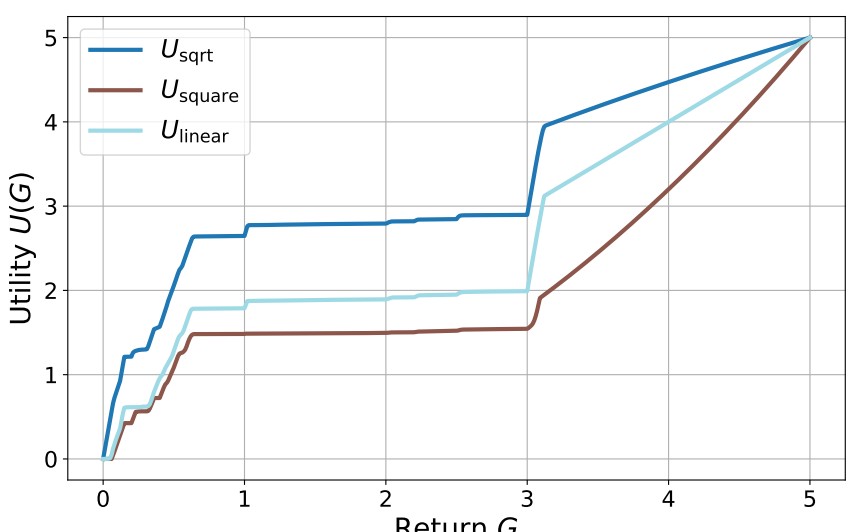

Figure 18: Utilities computed by **TRACTOR-UL** starting with the $\overline{U}_0$ in the legend ($\alpha = 100$).

we observe that for smaller learning rates (e.g., $\alpha \in [0.01, 0.5, 5]$), the utilities as well as the (non)compatibilities) do not change much (Figure 19 and Figure 20 left), while for larger learning rates, we obtain more consistent changes (Figure 20 left and Figure 21).

Clearly, larger learning rates require less iterations to achieve small values of (non)compatibilities. Nevertheless, too large values (e.g., $\alpha = 10000$) are outperformed by intermediate values (e.g., $\alpha = 100$).

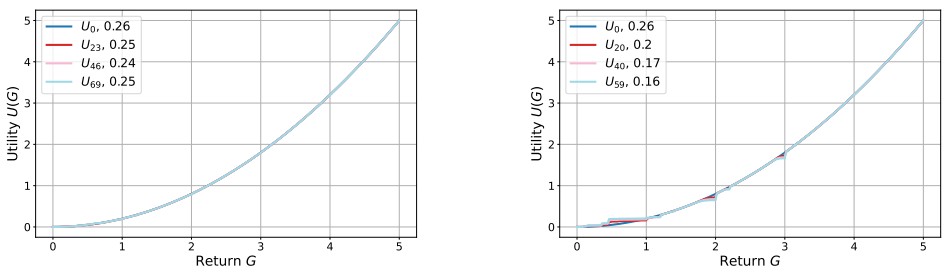

Figure 19: (Left) $\alpha = 0.01$. (Right) $\alpha = 0.5$.

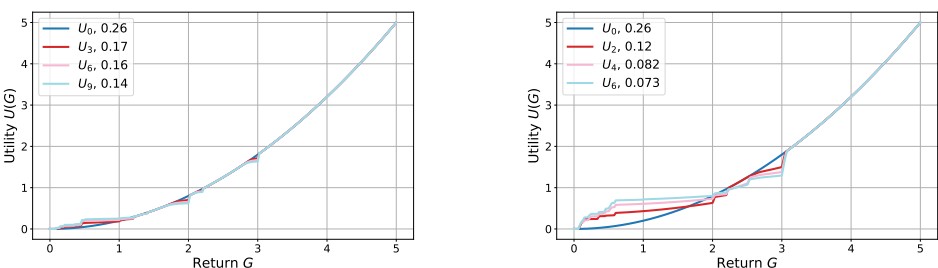

Figure 20: (Left) $\alpha = 5$. (Right) $\alpha = 100$.

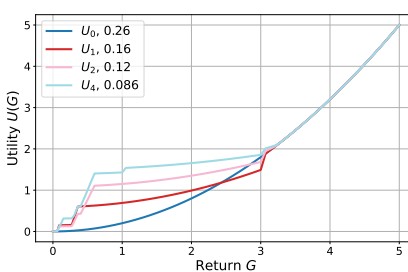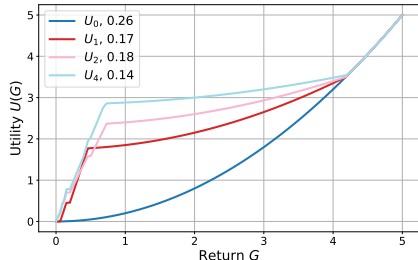

Figure 21: (Left) $\alpha = 1000$. (Right) $\alpha = 10000$.

### F.4.2 A VISUAL EXPLANATION FOR A LARGE LEARNING RATE

Now, we show that the projection update represented by operator $\Pi_{\overline{\underline{\mathfrak{U}}}_L}$ crucially neglects small variations in the (non-projected) utilities, requiring us to increase the step size.

Thus, the intuition is that we need a large learning rate because the projection step neglects small variations. To show this, we take as initial utility $\overline{U}_0 = U_{\text{sqrt}}$, two return distributions $\eta_0^*, \eta^E$, where $\eta^*$ coincides with the distribution of an optimal policy for $U_{\text{sqrt}}$, and $\eta^E$ is the return distribution of the policy played by participant 10. These distributions are plotted in Figure 22 left, and their difference is plotted in Figure 22 right. In particular, we note that the two distributions are rather different, with the expert's distribution $\eta^E$ that is more risk-averse, in that it provides higher probability to returns around $G = 0.5$, while the optimal distribution $\eta_0^*$ is more risk-lover, in that it assigns some probability to higher returns $G \geqslant 1$, but suffering from also high probability to small returns $G \leqslant 0.3$.

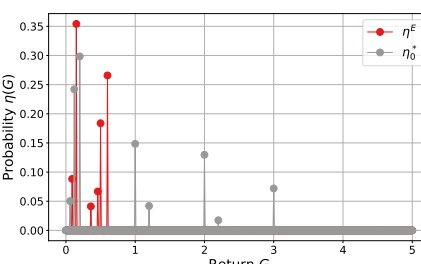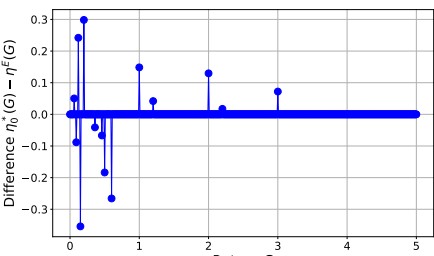

Figure 22: (Left) Plot of $\eta_0^*$ and $\eta^E$. (Right) Plot of $\eta_0^* - \eta^E$.

We aim to perform the **TRACTOR-UL** update rule:

$$\overline{U}_1' \leftarrow \overline{U}_0 - \alpha(\eta^* - \eta^E),$$

with some learning rate $\alpha$, and then to perform the projection:

$$\overline{U}_1 \leftarrow \Pi_{\overline{\underline{\mathfrak{U}}}_L}[\overline{U}_1'].$$

We execute the update with the following values of steps size: $\alpha \in \{0.01, 0.5, 5, 100, 1000, 10000\}$, and we plot the corresponding updated utilities $\overline{U}_1'$ and $\overline{U}_1$ in Figures 23, 24, and 25.

As we can see from Figures 23, 24, and 25, the update $\overline{U}_0 \to \overline{U}_1$ obtained with step sizes $< 5$ are rather neglectable, so that the return distribution of the new optimal policy $\eta_1^*$ for $\overline{U}_1$ still coincides with the previous one $\eta_0^*$, and the gradient at the next step is the same. For $\alpha = 5$, we begin to notice some changes. See Figure 26.

Instead, with larger gradients, we observe a non-neglectable change in utility, which provides a consistent change in the return distribution for $\alpha = 100$, and a huge change for $\alpha \in [1000, 10000]$ (see Figure 27).

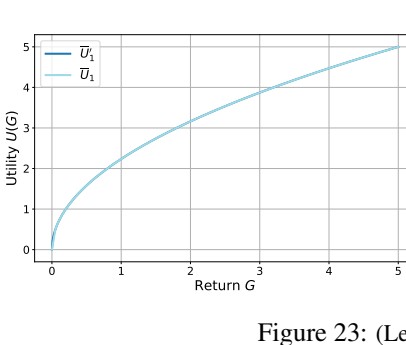
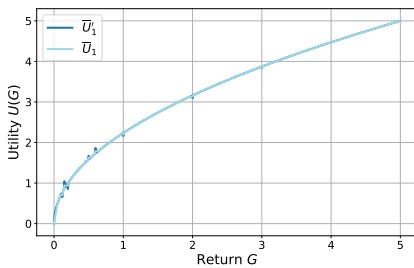

Figure 23: (Left) $\alpha = 0.01$. (Right) $\alpha = 0.5$.

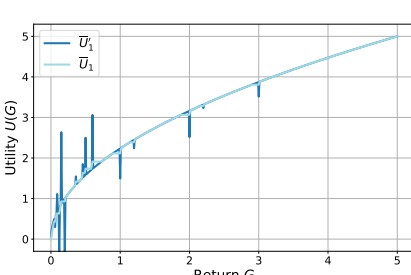
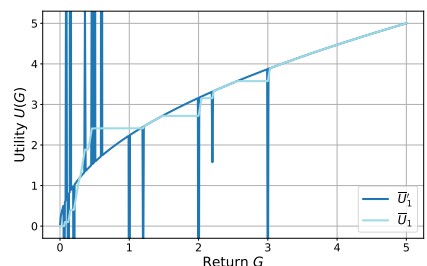

Figure 24: (Left) $\alpha = 5$. (Right) $\alpha = 100$.

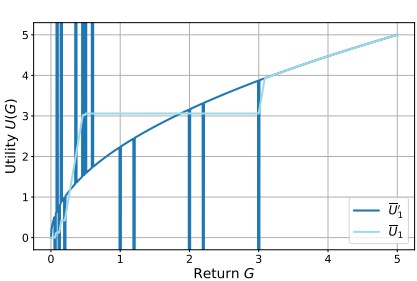
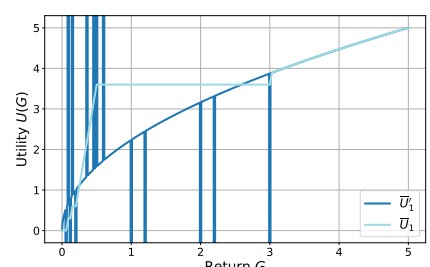

Figure 25: (Left) $\alpha = 1000$. (Right) $\alpha = 10000$.

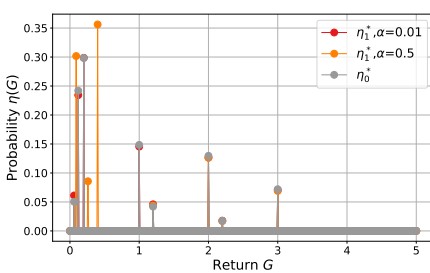
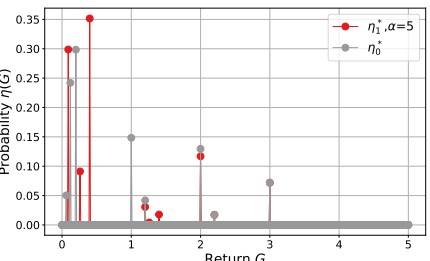

Figure 26: (Left) Comparison of the return distributions $\eta_1^*$ obtained with $\alpha = 0.01$ and $\alpha = 0.5$, with $\eta_0^*$. (Right) Comparison of the return distribution $\eta_1^*$ obtained with $\alpha = 5$, with $\eta_0^*$.

Since neglectable changes in both the utility and the optimal return distribution (obtained with small learning rates) mean that we have to update the utility many times along the same direction, then the update is equivalent to performing a single update in that direction with a huge step size. This justifies the use of large learning rates.

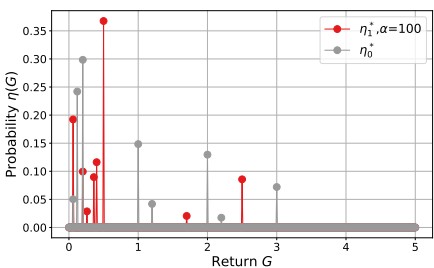 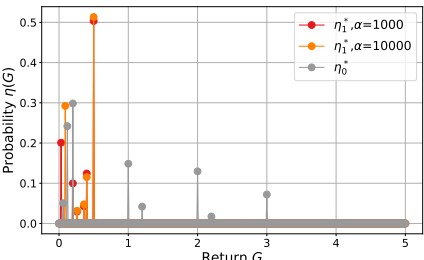

Figure 27: (Left) Comparison of the return distribution $\eta_1^*$ obtained with $\alpha = 100$, with $\eta_0^*$. (Right) Comparison of the return distributions $\eta_1^*$ obtained with $\alpha = 1000$ and $\alpha = 10000$, with $\eta_0^*$.

### F.4.3 ANALYSIS ON SIMULATED DATA

We have executed **TRACTOR-UL** on MDPs generated at random. Below we report the truncated (non)compatibility values of the utilities extracted by the algorithm as a function of the number of iterations, in the five different experiments conducted. For the experiments, we executed for $T = 70$ gradient iterations, with parameters $K = 10000$ and $L = 10$, as in the first part of the experiment. We found that the best learning rate is $\alpha = 1$.

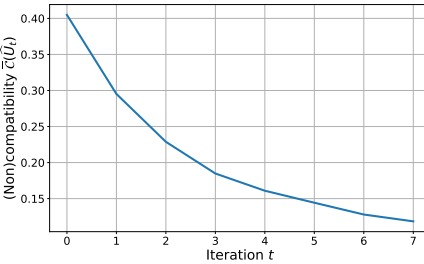 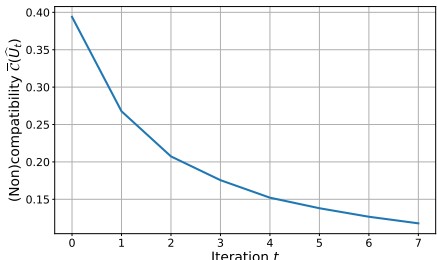

Figure 28: (Left) Simulation with $S = 20$ and $A = 5$. (Right) Simulation with $S = 100$ and $A = 10$.

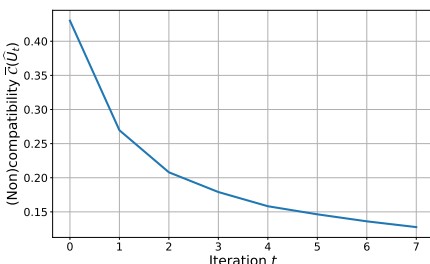 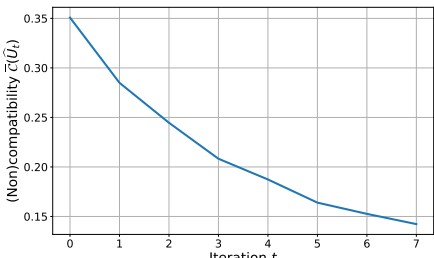

Figure 29: (Left) Simulation with $S = 1000$ and $A = 20$. (Right) Simulation with $N = 5$.

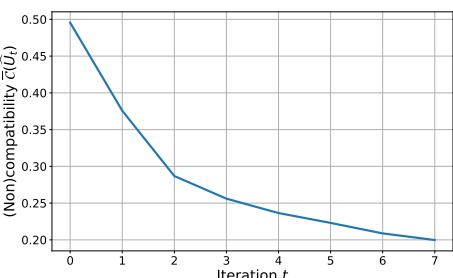

Figure 30: Simulation with $N = 20$.

