# OpenReview forum: "Learning Utilities from Demonstrations in Markov Decision Processes"
_ICLR.cc/2025/Conference — Submitted to ICLR 2025_

### Official Review · Reviewer_Koed · 2024-10-31

**Soundness:** 3
**Presentation:** 3
**Contribution:** 3
**Rating:** 6
**Confidence:** 4

**Summary:**

This work proposes utility learning as a more expressive framework than IRL to infer reward functions and risk attitude from demonstrations. The authors formally define the problem setup and prove basic properties of the utility learning problem. They focus on the case where the reward function is known and only the utility function (which is a function of the cumulative reward) is unknown and must be learned from demonstrations. Due to the issue of partial identifiability, the authors consider the case where the learner gets to observe demonstrations in multiple environments. The proposed algorithms are theoretically analyzed. The paper also includes a small empirical evaluation with human participants.

**Strengths:**

- The paper introduces a novel model of human behavior in MDPs with a focus on modeling the risk appetite of humans using a general utility function (that depends on the cumulative reward).
- The paper provides a clear overview over the proposed framework of learning utility functions from demonstrations. In particular, the paper proves basic, intuitive properties of the utility learning problem that are analogous to what is well-known in IRL. While technically straightforward, these theoretical results are interesting and useful (to understand the differences and similarities to IRL).
- Theoretical guarantees for the proposed algorithms are provided.
- The algorithms are evaluated in a small-scale experiment with human participants. Even though the experiments show that the proposed approach underperforms compared to fixing a utility function by hand instead of learning it, the results are interesting and well-explained.
- The paper is well-written and overall clear.

**Weaknesses:**

- Most importantly, I'm not yet convinced that explicitly learning risk-sensitive utility functions is necessary. Playing the devil's advocate here, I'd argue that a reward function is able to capture risk appetite by extending the state space (as noted by the authors themselves in footnote 3). In said footnote, you argue that we shouldn't alter the state representation, because we couldn't transfer the reward function to new environments. This is quite unconvincing in my opinion, because, after all, we get to design the state space (usually), and any similar environments that we would like to transfer the reward function to could also have, say, a feature that refers to the money we currently have in our pocket. I think that it is quite standard to alter a state representation this way (e.g., think about maze environments where the agent has to pick up a key first in order to go through a door).
Thus, Example 3.1 may be too simple an example to highlight why we should model some problems as utility learning instead of IRL. I do hope that this prompts the authors to make a stronger case for the necessity of utility learning. It also seems important that the utility is a function of the total return and this could be explained in more detail.

- There are some additional minor weaknesses:
	- Even though this work wants to provide a more expressive / realistic model of human behavior in MDPs, it assumes that the expert acts *optimally* w.r.t. its utility (equation 1).
	- It would be great to complement the existing experiments with some evaluations on simulated data just to see how well the proposed algorithms work. It would also allow you discuss more thoroughly any scaling issues the current approaches have. I do acknowledge that this is primiarly theoretical work and the experiments are not a priority.
	- In the experiments, you should include how the participants were recruited (e.g., lab members or others). This may somewhat limit the validity and integrity of the experiments, but this is a fairly minor point of concern to me.

**Questions:**

- Please comment on the weaknesses that I mentioned as you find appropriate.
- Are there any alternatives to enlarging the state space to compute the optimal policy in RS-MDPs?
- What follows are some minor suggestions and pointers to literature that you might have overlooked.
	- Right before and in Section 5, you refer to the situation where you get demonstrations in multiple environments as the "setting with multiple demonstrations". This could be interpreted as observing the several trajectories in the same environment (instead of trajectories in different environments). I think it woul be better to consistently refer to this as learning from demonstrations in multiple environments. There is also related work in IRL that has referred to this as such and studied learning from multiple environments: [Identifiability and Generalizability from Multiple Experts in IRL](https://proceedings.neurips.cc/paper_files/paper/2022/file/03bdba50e3741ac5e3eaa0e55423587e-Paper-Conference.pdf), [Environment Design for IRL](https://openreview.net/forum?id=Ar0dsOMStE).
	- Note that similar results to your Proposition 4.6 have been derived for IRL: [Identifiability in IRL](https://proceedings.neurips.cc/paper/2021/hash/671f0311e2754fcdd37f70a8550379bc-Abstract.html) shows that the reward function can become identifiable when you have demonstrations under different discount factors or transitions (Theorem 2); [Interactive IRL](https://proceedings.mlr.press/v162/buning22a/buning22a.pdf) shows that any reward function is identifiable under some transition model (Theorem 2, Remark 1)
	- In line 243 you refer to Skalse et al. 2023 in the context of reward identifiability in IRL. I'm a bit surprised because Skalse et al. 2023 studies human model misspecification. It would be more appropriate to refer to [Identifiability in IRL](https://proceedings.neurips.cc/paper/2021/hash/671f0311e2754fcdd37f70a8550379bc-Abstract.html) and [Reward Identification in IRL](https://proceedings.mlr.press/v139/kim21c.html) who both study the identifiability issue (up to an appropriate equivalence class) in IRL.

---

> ### Author Response · Authors · 2024-11-18
>
> We are glad that the Reviewer appreciated the novelty of the model presented, the significance of the theoretical and empirical results provided, and also acknowledged that the paper is well-written and overall clear. Below, we report answers to the Reviewer's comments.
>
> ## Weaknesses
>
> > Most importantly, I'm not yet convinced that explicitly learning risk-sensitive utility functions is necessary. Playing the devil's advocate here, I'd argue that a reward function is able to capture risk appetite by extending the state space (as noted by the authors themselves in footnote 3). In said footnote, you argue that we shouldn't alter the state representation, because we couldn't transfer the reward function to new environments. This is quite unconvincing in my opinion, because, after all, we get to design the state space (usually), and any similar environments that we would like to transfer the reward function to could also have, say, a feature that refers to the money we currently have in our pocket. I think that it is quite standard to alter a state representation this way (e.g., think about maze environments where the agent has to pick up a key first in order to go through a door). Thus, Example 3.1 may be too simple an example to highlight why we should model some problems as utility learning instead of IRL. I do hope that this prompts the authors to make a stronger case for the necessity of utility learning.
>
> We thank the Reviewer for the important question, that allows us to clarify a crucial point of the paper. Specifically, **if we re-model the MDP by including the reward into the state to make the optimal policy Markovian, then we might incur in *interpretability* and *transferability* issues**. To better explain this, let us make a **simple example**.
>
> Consider a driving setting, where the state is the location of the car (name of the road and position inside the road), the actions permit to change the current road (only when the car is close to another road, otherwise no effect), and at every stage/timestep the position of the car advances on the current road depending on the amount of traffic in the road, which is random and modelled through the transition model of the environment.
>
> Consider now an expert agent that *aims to reach a certain goal location $s_g$ in the minimum time/number of stages possible*, and that *is risk-averse*, in the sense that it prefers roads that always have little traffic, even though they are, on average, slower, to roads that are usually faster but sometimes have peaks of traffic that make them very very slow (since the traffic is random, there is no sequence of roads that is *always* better than others, but it is a matter of chance).
>
> In **our model** (Eq. (1)), we can represent the expert through the reward function $r^E$ that is $0$ in the goal location $s_g$, and $-1$ otherwise. In this manner, the faster a trajectory reaches $s_g$, the larger the cumulative reward. Next, we can choose the utility function $U^E$ to be some *concave* function in order to achieve the risk-aversion property [1], i.e., to make sure that the expert prefers, intuitively, roads with "smaller variance" of traffic. We remark that *our model permits to  capture the preferences of the expert in a very simple yet expressive manner*. In fact, as shown in this example, $r^E$ and $U^E$ can be easily designed, and their meaning is easily interpretable. However, this does not hold if we include past rewards in the state.
>
> In the **model with an extended state space**, the behavior of the expert is represented through a single reward $\overline{r}^E$ (defined on the expanded state space) instead of the pair reward-utility $r^E,U^E$. The intuition is that, to contain all the information present in $r^E,U^E$, the new reward $\overline{r}^E$ will be "messy". As such, designing it is also more complex. For instance, choosing $\overline{r}^E$ to be $-t$ in the (expanded) state made of the goal location $s_g$ and of $t$ timesteps, and $0$ elsewhere, represents a *risk-neutral* agent that aims to reach $s_g$ as soon as possible, but it does *not* capture the risk-aversion of the expert. To make $\overline{r}^E$ model the risk-aversion, we must take it to be a *concave* function of $-t$, making it *more difficult to be interpreted*.

---

> > ### Author Response · Authors · 2024-11-18
> >
> > Even though the model with past rewards in the state guarantees the Markovianity of the optimal/expert policy, it suffers from major **drawbacks**:
> > - $\overline{r}^E$ has a size (i.e., it is defined on a number of states) that grows *exponentially in the horizon* in the worst case, while $r^E,U^E$ do not.
> > - $\overline{r}^E$ is more difficult to *interpret* (and design) than the pair $r^E,U^E$, whose meaning is immediate.
> > - $\overline{r}^E$ can only be transferred to problems with the same state-action (or feature) space. Instead, the utility $U^E$ can be easily *transferred* to other kinds of environments. E.g., in the considered example, $U^E$ can be used to assess how much the expert "values its time" and takes decisions based on it. Thus, we can predict the behavior of the expert in other problem settings where the time plays a role using $U^E$, even if the state-action (or feature) space is different (e.g., if the expert travels by train instead than by car, we can predict if it prefers taking a reliable train, or a faster train on average that sometimes makes huge delays).
> >
> > For these reasons, we believe that expanding the state space by including the past rewards is not satisfactory in terms of interpretability and transferability. We will make this point clearer in the paper. If the Reviewer has additional questions or doubts, please feel free to ask.
> >
> > > It also seems important that the utility is a function of the total return and this could be explained in more detail.
> >
> > Again, we thank the Reviewer for the interesting question, that allows us to better explain the importance of this modelling choice.
> >
> > Specifically, **applying the utility to the total return** is equivalent to assuming that **the expert has a *planning horizon* $H^E$ that coincides with the *horizon* $H$ of the MDP**. When humans take decisions, i.e., perform planning, since they are *boundedly rational* [2], they are not able to "look too many steps into the future" [3] (see also Section 6 and Section 7). We call *planning horizon* $H^E$ the number of steps the expert uses for planning. In this paper, because of Eq. (1), we implicitly considered the case where $H^E=H$, i.e., where the expert is *intelligent enough* for planning over the entire horizon of the MDP.
> >
> > In other words, as explained at line 189 and in footnote 4 (line 215), applying the utility to the *total* return **can also be interpreted through the lens of Expected Utility Theory** [4,5], as having an expert that "inspects" all the policies based on the distribution/lottery over returns (prizes) they induce [6], and then it chooses the one it prefers.
> >
> > > There are some additional minor weaknesses:
> > Even though this work wants to provide a more expressive / realistic model of human behavior in MDPs, it assumes that the expert acts optimally w.r.t. its utility (equation 1).
> >
> > We note that: $(i)$ a model is a simplified representation of the reality, thus assuming the expert to act optimally is always reasonable at an abstraction level. $(ii)$ How much it is realistic depends on the extent to which it is corroborated by empirical data, and we provide some "corroborating" data in Experiment 1. $(iii)$ The most popular IRL models also assume the expert to act optimally, either in an MDP [7], or in an entropy-regularized MDP [8,9]. $(iv)$ Our model can be easily extended to consider experts that act *suboptimally* by replacing Eq. (1) with:
> > $$
> > \pi^E \text{ s.t.: }\quad\mathbb{E}\_{\pi^E}\Big[U\Big(\sum_h r_h\Big)\Big]
> > \ge \max_\pi\mathbb{E}_{\pi}\Big[U\Big(\sum_h r_h\Big)\Big]-\eta,
> > $$
> > i.e., by considering the expert to act at most $\eta$-suboptimally, for some (potentially unknown) $\eta\ge 0$. Note that the algorithms can be adjusted correspondingly. We leave the analysis of this model to future works.

---

> > > ### Author Response · Authors · 2024-11-18
> > >
> > > > It would be great to complement the existing experiments with some evaluations on simulated data just to see how well the proposed algorithms work. It would also allow you discuss more thoroughly any scaling issues the current approaches have. I do acknowledge that this is primiarly theoretical work and the experiments are not a priority.
> > >
> > > We agree with the Reviewer that a more extensive empirical evaluation of the proposed algorithms may provide interesting additional insights. *Nevertheless*, as also observed by the Reviewer, we believe that additional simulations are out of the scope of this work. In fact, the paper aims to introduce a *novel* problem setting (*Utility Learning*), to understand its theoretical difficulties in the simplest possible setting (tabular MDPs), and then to prove the existence of algorithms with theoretical guarantees of sample and computational complexity (CATY-UL and TRACTOR-UL). For these reasons, we believe that an ***extensive* empirical validation is not necessary** in this work. The proof-of-concept numerical simulations provided in the paper aim to show that our algorithms work reasonably in a simple setting.
> > >
> > > Anyway, we have conducted some *additional experiments* on simulated data in larger problems, as requested by the Reviewer. We have considered MDPs with larger state and actions spaces (increment of $S$ and $A$), and we have considered also demonstrations in multiple environments (increment of $N$). Specifically, we have generated larger MDPs at random. To guarantee the assumption that there exists a utility function for which the expert's policy is (almost) optimal, we have not generated $\pi^E$ at random, but we have computed, in each environment, the optimal policy for an S-shaped utility function that is convex for small returns, and concave for large returns. Such utility is one of the most popular (and realistic) utility functions considered in behavioral economics [2]. We have additionally injected some noise on the computed policies to make them more realistic.
> > >
> > > Below we report the truncated (non)compatibility values of the utilities extracted by the algorithm as a function of the number of iterations, in the five different experiments conducted. For the experiments, we executed for $T=70$ gradient iterations, with parameters $K=10000$ and $L=10$, as in the paper.
> > >
> > > **Experiment 1**: $N=1,S=20,A=5$:
> > >
> > > | iteration $i$ | $\overline{\mathcal{C}}(U_i)$ |
> > > | --------- | ------------------------ |
> > > | $1$      | $0.40$     |
> > > | $10$      | $0.30$     |
> > > | $20$      | $0.23$     |
> > > | $30$      | $0.18$     |
> > > | $40$      | $0.16$     |
> > > | $50$      | $0.14$     |
> > > | $60$      | $0.13$     |
> > > | $70$      | $0.12$     |
> > >
> > > **Experiment 2**: $N=1,S=100,A=10$:
> > >
> > > | iteration $i$ | $\overline{\mathcal{C}}(U_i)$ |
> > > | --------- | ------------------------ |
> > > | $1$      | $0.39$     |
> > > | $10$      | $0.27$     |
> > > | $20$      | $0.21$     |
> > > | $30$      | $0.18$     |
> > > | $40$      | $0.15$     |
> > > | $50$      | $0.14$     |
> > > | $60$      | $0.13$     |
> > > | $70$      | $0.12$     |
> > >
> > > **Experiment 3**: $N=1,S=1000,A=20$:
> > >
> > > | iteration $i$ | $\overline{\mathcal{C}}(U_i)$ |
> > > | --------- | ------------------------ |
> > > | $1$      | $0.43$     |
> > > | $10$      | $0.27$     |
> > > | $20$      | $0.21$     |
> > > | $30$      | $0.18$     |
> > > | $40$      | $0.15$     |
> > > | $50$      | $0.14$     |
> > > | $60$      | $0.13$     |
> > > | $70$      | $0.12$     |
> > >
> > > **Experiment 4**: $N=5,S=4,A=3$:
> > >
> > > | iteration $i$ | $\overline{\mathcal{C}}(U_i)$ |
> > > | --------- | ------------------------ |
> > > | $1$      | $0.35$     |
> > > | $10$      | $0.28$     |
> > > | $20$      | $0.24$     |
> > > | $30$      | $0.21$     |
> > > | $40$      | $0.19$     |
> > > | $50$      | $0.16$     |
> > > | $60$      | $0.15$     |
> > > | $70$      | $0.14$     |
> > >
> > > **Experiment 5**: $N=20,S=4,A=3$:
> > >
> > > | iteration $i$ | $\overline{\mathcal{C}}(U_i)$ |
> > > | --------- | ------------------------ |
> > > | $1$      | $0.50$     |
> > > | $10$      | $0.38$     |
> > > | $20$      | $0.29$     |
> > > | $30$      | $0.26$     |
> > > | $40$      | $0.24$     |
> > > | $50$      | $0.22$     |
> > > | $60$      | $0.21$     |
> > > | $70$      | $0.20$     |

---

> ### Author Response · Authors · 2024-11-18
>
> Some comments:
> - The **first three experiments** analyze the performance when we increase the size of the state-action space. Note that the number of gradient iterations necessary to achieve a certain level of performance is not affected by an increment of $S,A$, *as already shown in our Theorem 5.2*. Nevertheless, larger $S,A$ require more execution time, because of the value iteration subroutine, that has a computational complexity that grows with $S,A$.
> - The **last two experiments** analyze what happens when we consider demonstrations in multiple ($N=5$ and $N=20$) environments. We note that, as expected, the number of iterations required to find a utility with small (non)compatibility increases with $N$: after $T=70$ iterations, we have $\overline{\mathcal{C}}(U_T)=0.07$ for $N=1$ (see Section 6), $\overline{\mathcal{C}}(U_T)=0.14$ for $N=5$, $\overline{\mathcal{C}}(U_T)=0.20$ for $N=20$. This result is *expected* from the theoretical guarantees of Theorem 5.2, and also from the intuition. In addition, we observe that the best learning rate when $N$ increases (we found $\alpha=1$) is much smaller than $\alpha=100$ used for the experiments with $N=1$. Again, this result is *expected* (we mentioned it in the paper), since now we aim to find a utility that has to satisfy more *constraints* (more environments), and, thus, there are less utilities with small (non)compatibility. Intuitively, "finding" them is more difficult, and we cannot make large gradient steps as when $N=1$.
>
> We hope that the Reviewer finds these experiments satisfactory; we will include them in the final version of the paper. Please do not hesitate to contact us for any further explanations.
>
> > In the experiments, you should include how the participants were recruited (e.g., lab members or others). This may somewhat limit the validity and integrity of the experiments, but this is a fairly minor point of concern to me.
>
> Thank you for pointing out. The participants to the experiments are **lab members**. As such, they have the competences necessary to understand effectively the questions asked to them. We will add this point to the paper.
>
> ## Questions
>
> > Are there any alternatives to enlarging the state space to compute the optimal policy in RS-MDPs?
>
> As far as we know, **there are *not* other approaches in literature**. Moreover, we believe that no alternative can even be devised, for the following, intuitive, reason. [1] have proved that the non-Markovianity of the optimal policy is limited to the cumulative reward up to now. Thus, in the most general case, the optimal policy $\pi^*$ is a function of the current state and stage, and of the amount of reward accumulated so far: $\pi^*: \mathcal{S}\times[H]\times\mathcal{Y}\to\mathcal{A}$, where $\mathcal{Y}$ is the set of possible cumulative reward values (see definition in the paper). Thus, intuitively, since $\pi^*$ has domain $\mathcal{S}\times[H]\times\mathcal{Y}$, then we need to assign a value (action) to all the triples $(s,h,y)$. Since $\mathcal{S}\times\mathcal{Y}$ coincides with the enlarged state space, then in the worst case this approach is necessary.
>
> > Right before and in Section 5, you refer to the situation where you get demonstrations in multiple environments as the "setting with multiple demonstrations". This could be interpreted as observing the several trajectories in the same environment (instead of trajectories in different environments). I think it woul be better to consistently refer to this as learning from demonstrations in multiple environments. There is also related work in IRL that has referred to this as such and studied learning from multiple environments: Identifiability and Generalizability from Multiple Experts in IRL, Environment Design for IRL.
>
> We thank the Reviewer for the suggestion. We agree that the expression "*learning from demonstrations in multiple environments*" is more suitable to the considered problem setting, given the analogies with the mentioned papers. Our simplified expression "*setting with multiple demonstrations*" can be misleading. We will fix this in the final version.

---

> > ### Author Response · Authors · 2024-11-18
> >
> > > Note that similar results to your Proposition 4.6 have been derived for IRL: Identifiability in IRL shows that the reward function can become identifiable when you have demonstrations under different discount factors or transitions (Theorem 2); Interactive IRL shows that any reward function is identifiable under some transition model (Theorem 2, Remark 1)
> >
> > Thank you for the observation. We are aware of the relation between those papers and Proposition 4.6. We note that similar findings have been proven in IRL also in [10]. We will mention all these papers in the related works section.
> >
> > > In line 243 you refer to Skalse et al. 2023 in the context of reward identifiability in IRL. I'm a bit surprised because Skalse et al. 2023 studies human model misspecification. It would be more appropriate to refer to Identifiability in IRL and Reward Identification in IRL who both study the identifiability issue (up to an appropriate equivalence class) in IRL.
> >
> > Actually, line 243 points to [11], and not to [12]. We decided to mention [11] instead of [13,14] because $(i)$ it is more recent, and $(ii)$ it studies the partial identifiability of the expert's reward from the more general setting of *reward learning*, that comprises IRL as a special case. We will add references to [13,14] to make the presentation more complete.
> >
> > [1] Nicole Bauerle and Ulrich Rieder. More risk-sensitive markov decision processes.
> >
> > [2] Stuart Russell and Peter Norvig. Artificial Intelligence: A Modern Approach.
> >
> > [3] Daniel Carton, Verena Nitsch, Dominik Meinzer, and DirkWollherr. Towards Assessing the Human Trajectory Planning Horizon.
> >
> > [4] John von Neumann and Oskar Morgenstern. Theory of Games and Economic Behavior.
> >
> > [5] Daniel Kahneman and Amos Tversky. Prospect theory: An analysis of decision under risk.
> >
> > [6] Marc G. Bellemare, Will Dabney, and Mark Rowland. Distributional Reinforcement Learning.
> >
> > [7] Andrew Y. Ng and Stuart J. Russell. Algorithms for inverse reinforcement learning.
> >
> > [8] Brian D. Ziebart. Modeling Purposeful Adaptive Behavior with the Principle of Maximum Causal Entropy.
> >
> > [9] Tuomas Haarnoja, Haoran Tang, Pieter Abbeel, and Sergey Levine. Reinforcement Learning with Deep Energy-Based Policies.
> >
> > [10] Kareem Amin, and Satinder Singh. Towards Resolving Unidentifiability in Inverse Reinforcement Learning.
> >
> > [11] Joar Skalse, Matthew Farrugia-Roberts, Stuart Russell, Alessandro Abate, and Adam Gleave. Invariance in policy optimisation and partial identifiability in reward learning.
> >
> > [12] Joar Skalse and Alessandro Abate. Misspecification in inverse reinforcement learning.
> >
> > [13] Haoyang Cao, Samuel N. Cohen, and Lukasz Szpruch. Identifiability in inverse reinforcement learning.
> >
> > [14] Kuno Kim, Kirankumar Shiragur, Shivam Garg, and Stefano Ermon. Reward Identification in Inverse Reinforcement Learning.

---

> > > ### Comment · Reviewer_Koed · 2024-11-22
> > >
> > > Thank you for your response and thank you for conducting additional simulations. I suggest to add some of them to the main text and the remaining ones to the appendix. Please also note that you can revise your paper during the discussion phase already, which would make the promised changes more concrete to us.
> > >
> > > After your response I'm now more convinced of the merit of utility learning, though in the example you described we could still, without explicitly tracking past rewards, extend the state space minimally to include features that correlate with risk (e.g., average traffic, congestion indicators, or other metrics). For instance, if certain roads exhibit high variance in travel time, this information can be encoded into the state representation. Also, I don't find Example 3.1 as it is stated in the paper currently as convincing as the example you outlined in your response. However, if the other reviewers are convinced by the example in the text, this is fine by me.
> > >
> > > Your rebuttal did improve my opinion of the contributions and the significance of the work, and I'll consider increasing my score before the end of the discussion phase, but prefer for all the discussions to come to end before doing so.

---

> > > > ### Author Response · Authors · 2024-11-22
> > > >
> > > > We thank again the Reviewer for the insightful comments and questions, and we are glad that they will consider to increase the score.
> > > >
> > > > We have uploaded a revised version of the paper containing the promised changes (highlighted in purple), including the suggestions of the Reviewer. Specifically:
> > > > - We have added the driving example to Appendix C.1, and we have referenced it inside footnote 3, to better explain the importance of our model.
> > > > - We included references to partial identifiability papers in IRL in Section 4.
> > > > - In Section 5, we use the expression "learning from demonstrations in multiple environments" instead of "setting with multiple demonstrations".
> > > > - We mention in Section 6 that the participants to the study are lab members.
> > > > - We describe the additional simulations in Section 7, with additional comments in Appendix F.

---

### Official Review · Reviewer_Tkgg · 2024-11-03

**Soundness:** 3
**Presentation:** 2
**Contribution:** 3
**Rating:** 5
**Confidence:** 3

**Summary:**

This paper presents a model to learn utility functions from expert demonstrations within Markov Decision Processes (MDPs) which represents the agent’s risk attitude through a utility function. For this model, the paper proposes two algorithms to infer an agent’s utility based on observed behaviors and analyzes their sample complexity results.

**Strengths:**

1.The paper characterizes the partial identifiability of the expert’s utility by offering counterexamples in different aspects.

2.The paper proposes algorithms with sample complexity results, and conducts experiment to test the algorithms.

**Weaknesses:**

1. The theorems on identifiability and compatibility of utility function seem to be more like a mathematical game. In particular, I cannot think of any practical example for which Proposition 4.6 will hold. Proposition 4.6 basically says that demonstrations in multiple MDP environments have exactly the same utility function. However, I don't think that can be true in general. Utility functions, which represent risk attitudes, will change in different environments, even for the same agent. For example, the same person might have different risk attitudes when gambling with different amount of money.

2. The numerical experiment only shows the results of the proposed algorithms for different classes of utility functions.There is no comparison with other benchmarks.

3. The paper is heavy in math notations and can be hard to read.

**Questions:**

1.The papers claims that it is a new framework to consider risk-sensitive IRL. I find some related papers. Could you compare with them to state the novelty in your framework, such as:
Majumdar A, Singh S, Mandlekar A, et al. Risk-sensitive Inverse Rein-
forcement Learning via Coherent Risk Models[C]//Robotics: science and
systems. 2017, 16: 117.
Grze´skiewicz M. Uncovering Utility Functions from Observational Data
with Revealed Preference Inverse Reinforcement Learning[D]. UCL (Uni-
versity College London), 2024.

In particular, could you  clarify how your framework advances the state-of-the-art in risk-sensitive IRL beyond these existing approaches?

2. Multiple demonstrations are used to address the partial identifiability issues. Of course, an oracle to demonstrations under different
environment helps to recover the utility. But there are two issues. First, as mentioned in the Weaknesses above, why would it be reasonable to assume the same utility function in multiple demonstrations? Could you provide specific examples of real-world scenarios where the same utility function would reasonably apply across multiple environments? Second, we may not have access to multiple demonstrations in reality. Could you analyze how their method’s performance degrades as the number of available demonstrations decreases, and provide guidance on the minimum number needed for reasonable results?

3. The paper uses the gradient of the upper bound $\sum_i \overline{\mathcal{C}}$ instead of the exact subgradient of $\max_i {\overline{\mathcal{C}}}$. If you are able to know which one is  maximal,  you can  estimate this subgradient in a way similar to your method. Can you explain your rationale for using the upper bound gradient instead of the exact subgradient? If there are technical difficulties in finding the maximal, could you elaborate on what those challenges are?

4. As stated in Weaknesses, I suggest you add some other benchmarks (if they exist) to compare with in the numerical experiment. Could you identify and compare against 2-3 specific baseline methods from prior work on risk-sensitive IRL or utility learning. If relevant benchmarks do not exist, could you justify why and discuss how they could otherwise demonstrate the empirical advantages of their approach. Also, could you conduct additional experiments varying the number of demonstrations from 1 to N, and report how this affects the method’s performance?

---

> ### Author Response · Authors · 2024-11-18
>
> We thank the Reviewer for recognizing the importance of our results on the partial identifiability of the expert's utility, and for appreciating our theoretical and empirical analysis of the proposed algorithms. Below, we answer to the Reviewer's comments and questions.
>
> ## Weaknesses
>
> > The theorems on identifiability and compatibility of utility function seem to be more like a mathematical game.
>
> We remark that, although we derived them in illustrative problems, **our results on the partial identifiability of the expert's utility establish the barriers of the Utility Learning (UL) problem**. Specifically, Propositions 4.1-4.3, 4.5, demonstrate that UL can*not* be used for predicting, imitating, or "assessing" the behavior of the expert when we are given demonstrations in a *single* environment. Instead, Proposition 4.6 proves that the availability of demonstrations in *multiple* environments permits successfully apply UL in applications. We also remark that the criticality of the identifiability issue in the connected IRL problem setting has been highlighted even in the seminal papers [1,2]. Since then, many crucial results on the identifiability properties of the setting have appeared [3,4,5,6,7,8,9,10].
>
> > In particular, I cannot think of any practical example for which Proposition 4.6 will hold.
>
> We agree with the Reviewer that, since Proposition 4.6 requires the knowledge of the expert's policy in an infinite amount of different MDPs, then it cannot hold in practice. *However*, **its importance is to show that observing expert demonstrations in more than a single environment permits to reduce the unidentifiability of the expert's utility**, i.e., it permits to compute more accurate estimates. Observe that, under additional regularity assumptions, it would be possible to obtain accurate estimates of the expert's utility even with a finite number of environments. Note also that there exist analogous results in IRL, like [3,4,8,9].
>
> >  Proposition 4.6 basically says that demonstrations in multiple MDP environments have exactly the same utility function. However, I don't think that can be true in general. Utility functions, which represent risk attitudes, will change in different environments, even for the same agent.
>
> We remark that the **risk attitude of a person does *not* change across different environments when the underlying reward function is the same**. Consider the following two examples where reward is represented by "life" and "money", respectively.
> 1. A person who does not drive fast because she is worried for her life (she is very risk-averse with her life), then, reasonably, will not play dangerous sports, like skydiving, for the same reason. Namely, she exhibits the same risk-averse behavior with life in both the environments.
> 2. A person who loves gambling (risk-lover for money), is less likely to buy optional insurances than a person who hates gambling.
>
> Since the *utility function* represents the risk attitude of a person, and the risk attitude does *not* change across different environments, **then also the utility function does not change**.
>
> We also mention that the problem setting in which demonstrations in multiple MDP environments have exactly the same reward function (a different kind of "utility function") is very common in the IRL literature [3,4,6,7,8,9,10].
>
>
> >  For example, the same person might have different risk attitudes when gambling with different amount of money.
>
> We remark that  different "risk behaviors" associated with the same person gambling with different amount of money can be indeed represented **using the *same* utility function**. This is well-known in choice theory [11,12].
>
> To make a simple example, consider a person who loves making small bets (e.g., <10€) on football matches (i.e., who is risk-lover for small amounts of money), but who would never gamble more than 10€ (i.e., she is risk-averse for larger amounts of money). Clearly, this person has a *risk attitude* that exhibits love for risking small amounts of money, and aversion for risking larger amounts of money. Nevertheless, she has *one* risk attitude for money, that is easily captured by a single utility function, that is convex before 10€ and concave afterwards.
>
> If the Reviewer has additional questions or doubts, please feel free to ask.

---

> > ### Author Response · Authors · 2024-11-18
> >
> > > The numerical experiment only shows the results of the proposed algorithms for different classes of utility functions. There is no comparison with other benchmarks.
> >
> > We observe that **Experiment 1 does not test the performance of the algorithms**, but it aims to test how much the proposed *model* of behavior (Eq. (1)) is realistic. As such, it does not make sense to consider the algorithms of other papers here.
> >
> > Concerning **Experiment 2**, that provides a proof-of-concept analysis of the empirical performance of algorithm TRACTOR-UL, we note that **there is no benchmark in literature** because the algorithms proposed in the related works have different learning targets as we are introducing this learning problem (UL) for the first time in this paper. Specifically, the algorithm in [13] aims to compute a *coherent risk measure* instead of a utility function, while the algorithms in [14] and [15] output the parameters of certain notions of utilities under different *models of behavior* than ours. Intutively, this is the same issue that arises when directly comparing different IRL algorithms (e.g., when comparing [1] with [16]), because they aim to extract *different rewards* under *different assumptions*.
> >
> > > The paper is heavy in math notations and can be hard to read.
> >
> > We agree with the Reviewer that the mathematical notation adopted might appear rather cumbersome and complicated at first sight. *However*, we remark that **it is necessary and it cannot be dropped without losing formality or even clarity**. Every symbol introduced in the main paper is necessary to convey some important concept. An intuitive justification for this "strong" notation is that we have to integrate three different formalisms: IRL, risk-sensitive RL [17], and distributional RL [18].
> >
> > ## Questions
> >
> > > The papers claims that it is a new framework to consider risk-sensitive IRL. I find some related papers. Could you compare with them to state the novelty in your framework, such as: Majumdar A, Singh S, Mandlekar A, et al. Risk-sensitive Inverse Rein- forcement Learning via Coherent Risk Models[C]//Robotics: science and systems. 2017, 16: 117. Grze´skiewicz M. Uncovering Utility Functions from Observational Data with Revealed Preference Inverse Reinforcement Learning[D]. UCL (Uni- versity College London), 2024. In particular, could you clarify how your framework advances the state-of-the-art in risk-sensitive IRL beyond these existing approaches?
> >
> > Differently from all other works present in literature, we present the first paper that **recognizes the importance of non-Markovian policies in modelling the behavior of (human) agents** in presence of stochasticity/risk in MDPs for the inverse problem. We provide an intuitive justification in Example 3.1, and a proof-of-concept empirical demonstration in Experiment 1 (see Section 6), where we show that, in a toy MDP, the majority of the participants to the study exhibit a non-Markovian behavior.
> >
> > Moreover, we present the **first model of (human) behavior in MDPs that complies with non-Markovian policies** (see Eq. (1)), and we show its superior empirical validity over "Markovian" models in a small-scale study (Experiment 1, Section 6). No author in the IRL literature has analyzed neither this nor another model compliant with non-Markovian policies.
> >
> > Concerning the two papers mentioned by the Reviewer, observe that we have already analyzed and compared paper [19] to our work in both Section 1 and Appendix A. W.r.t. the Ph.D. thesis [20], which seems to be published 2 months before the ICLR deadline, it is not of public access, thus we cannot analyze it in-depth. Anyway, we recognize from its abstract that it seems to consider a specific economic setting, which makes it rather different from our work.

---

> > > ### Author Response · Authors · 2024-11-18
> > >
> > > > Multiple demonstrations are used to address the partial identifiability issues. Of course, an oracle to demonstrations under different environment helps to recover the utility. But there are two issues. First, as mentioned in the Weaknesses above, why would it be reasonable to assume the same utility function in multiple demonstrations? Could you provide specific examples of real-world scenarios where the same utility function would reasonably apply across multiple environments?
> > >
> > > Please, see the comments to the Weaknesses above. Another real-world example beyond the ones provided above considers a utility function that represents the risk attitude of a person w.r.t. *arriving late*. E.g., let the person be risk-averse (and so her utility is concave). Then, the person prefers taking a train that arrives at destination in $1$h for sure, over a train that sometimes arrives in $30$ minutes and sometimes in $1$h $30$m. Clearly, this utility function can be transferred to all the environments where the timing plays a role, like choosing the road when driving (time depends on traffic) or taking a plane for instance.
> > >
> > > If the Reviewer desires more scenarios, we can provide them.
> > >
> > > > Second, we may not have access to multiple demonstrations in reality. Could you analyze how their method’s performance degrades as the number of available demonstrations decreases, and provide guidance on the minimum number needed for reasonable results?
> > >
> > > When we have access to expert's demonstrations in a single environment ($N=1$), then we suffer from the *partial identifiability* issues presented in Propositions 4.1, 4.2, 4.3, and 4.5. When we can use demonstrations in all the possible environments ($N \rightarrow +\infty$), then we can identify the expert's utility $U^E$ exactly, as shown in Proposition 4.6. When the number of environments is in-between, i.e., $N\in(1,+\infty)$, then the *identifiability* of the expert's utility $U^E$ depends on the specific environments observed. In other words, **there does *not* exist a minimum number of environments $N\ge\overline{N}$** that guarantees a certain accuracy in the estimation of $U^E$, because it depends on how much *informative* are the observed environments. We remark that analyzing how the identifiability of $U^E$ degrades as $N$ decreases is challenging from a technical perspective, and we leave it for future works.
> > >
> > > > The paper uses the gradient of the upper bound $\sum_i \overline{\mathcal{C}}$ instead of the exact subgradient of $\max_i \overline{\mathcal{C}}$. If you are able to know which one is maximal, you can estimate this subgradient in a way similar to your method. Can you explain your rationale for using the upper bound gradient instead of the exact subgradient? If there are technical difficulties in finding the maximal, could you elaborate on what those challenges are?
> > >
> > > The choice made in the paper of upper bounding $\max_i \overline{\mathcal{C}}_i(U)\le\sum_i \overline{\mathcal{C}}_i(U)$ for all utilities $U$ is made **for the sake of simplicity**. Indeed, analyzing from a theoretical perspective the algorithm that uses the gradient instead of the subgradient requires *simpler notation*. In any case, as observed by the Reviewer, extending the algorithm and Theorem 5.2 to the usage of the subgradient is straightforward. We will add a comment in the final version.
> > >
> > > > As stated in Weaknesses, I suggest you add some other benchmarks (if they exist) to compare with in the numerical experiment. Could you identify and compare against 2-3 specific baseline methods from prior work on risk-sensitive IRL or utility learning. If relevant benchmarks do not exist, could you justify why and discuss how they could otherwise demonstrate the empirical advantages of their approach.
> > >
> > > As explained when providing feedback on one of the previous comments made by the Reviewer, **there are no benchmarks** in literature for algorithms CATY-UL and TRACTOR-UL. Indeed, CATY-UL is the first of its kind in **this setting, which is novel and has been introduced in this paper for the first time**, while TRACTOR-UL, even if it shares some similarities with the risk-sensitive IRL algorithms in [13,14,15], has a different learning target, and as such cannot be directly compared with them.
> > >
> > > We would also like to remark that **the goal of the paper is not to *show* that our algorithms "work well in practice"**. The paper aims to introduce a *novel* problem setting (*Utility Learning*), to understand its theoretical difficulties in the simplest possible setting (tabular MDPs), and then to prove the existence of algorithms with theoretical guarantees of sample and computational complexity (CATY-UL and TRACTOR-UL). For these reasons, we believe that an *extensive* empirical validation **is not necessary** in this work. The proof-of-concept numerical simulations provided in the paper aim to show that our algorithms work reasonably in a simple problem.

---

> > > > ### Comment · Reviewer_Tkgg · 2024-11-22
> > > >
> > > > I would like to thank the authors' detailed answers to my questions. Most questions got clarified, but I still think it is too strong to assume demonstrations in multiple MDP environments have exactly the same utility function. Taking the example given by the authors, a person's risk attitute w.r.t. arriving late can be different in multiple environments. If it's an important appointment, the person will be very risk averse to arriving late; but if it's a casual date, the person may be very okay with arrving late. Of course one may argue that the utility function can be defined as a function not only on the lateness but also on the importance of the task, but I think this is shifting concept. In summary, I think it is not realistic to assume that demonstrations in multiple MDP environments have exactly the same utility function.

---

> > > > > ### Author Response · Authors · 2024-11-22
> > > > >
> > > > > We thank the Reviewer for their feedback. We would like to clarify that the scenario described by the Reviewer can indeed be represented using **a unique reward function and utility function**. Specifically, the importance of the appointment, and consequently the impact of arriving late, is captured by the magnitude of the reward function. For less important appointments, the negative rewards for being late are smaller in magnitude, whereas for more important ones, the negative rewards are larger. The agent's utility function is then linear (risk-neutral) for small rewards and concave (risk-averse) for larger rewards. In this manner, the utility can be transferred and it considers the risk-attitude of being late only. The transferability of the underlying reward is a widely-accepted assumption in literature [3,4,6,7,8,9,10].

---

> > > > > > ### Comment · Reviewer_Tkgg · 2024-11-25
> > > > > >
> > > > > > I would like to thank the authors for the clarification. However, I decide to keep my original rating.

---

> ### Author Response · Authors · 2024-11-18
>
> > Also, could you conduct additional experiments varying the number of demonstrations from 1 to N, and report how this affects the method’s performance?
>
> As requested by the Reviewer, we have conducted two additional experiments on *simulated* data, one with $N=5$ environments, and the other with $N=20$ environments. For each experiment, we have first *randomly* generated $N$ MDPs. Then, in order to consider *reasonable* expert's policies that (almost) satisfy the assumption that they are generated by the same utility function, we have computed, in each environment, the optimal policy for the *same* utility function, that we took to be convex for small returns, and concave for large returns, i.e., it is S-shaped as the most popular (and realistic) utility functions considered in behavioral economics [12]. We have additionally injected some noise on these policies to make them more realistic.
>
> We have executed TRACTOR-UL on these two problem instances for $T=70$ iterations to compare with the performance reported in Section 6 for $N=1$. We have set $\overline{U}_0$ (the initial utility) to be the linear utility, and $K=10000$ and $L=10$ as in the experiments in the paper. In both the simulations, we have used a learning rate of $\alpha=1$, smaller than that used with $N=1$ (this choice is explained in the paper, and is motivated later on). Below we report the truncated (non)compatibility values of the utilities extracted by the algorithm as a function of the number of iterations.
>
> For $N=5$:
>
> | iteration $i$ | $\overline{\mathcal{C}}(U_i)$ |
> | --------- | ------------------------ |
> | $1$      | $0.35$     |
> | $10$      | $0.28$     |
> | $20$      | $0.24$     |
> | $30$      | $0.21$     |
> | $40$      | $0.19$     |
> | $50$      | $0.16$     |
> | $60$      | $0.15$     |
> | $70$      | $0.14$     |
>
> For $N=20$:
> | iteration $i$ | $\overline{\mathcal{C}}(U_i)$ |
> | --------- | ------------------------ |
> | $1$      | $0.50$     |
> | $10$      | $0.38$     |
> | $20$      | $0.29$     |
> | $30$      | $0.26$     |
> | $40$      | $0.24$     |
> | $50$      | $0.22$     |
> | $60$      | $0.21$     |
> | $70$      | $0.20$     |
>
> Some comments:
> - The best learning rate found is $\alpha=1$, much smaller than $\alpha=100$ used for the experiments with $N=1$. The reason is that, since now we aim to find a utility that has to satisfy more *constraints* (i.e., the feasible set is smaller), then there are less utilities with small (non)compatibility. Therefore, intuitively, "finding" them is more difficult,  and we cannot make large gradient steps as when $N=1$. Note that we already mention this fact in the paper.
> - The number of iterations required to find a utility with small (non)compatibility increases with $N$, as *expected*. After $T=70$ iterations, we have $\overline{\mathcal{C}}(U_T)=0.07$ for $N=1$ (see Experiment 2), $\overline{\mathcal{C}}(U_T)=0.14$ for $N=5$, $\overline{\mathcal{C}}(U_T)=0.20$ for $N=20$. This result is *expected* from the theoretical guarantees of Theorem 5.2, and also from the intuition.
>
> We will include these results in the final version of the paper. If the Reviewer desires additional details or insights on these experiments, please feel free to ask.

---

> > ### Author Response · Authors · 2024-11-18
> >
> > [1] Andrew Y. Ng and Stuart J. Russell. Algorithms for inverse reinforcement learning.
> >
> > [2] Stuart Russell. Learning agents for uncertain environments (extended abstract).
> >
> > [3] Haoyang Cao, Samuel N. Cohen, and Lukasz Szpruch. Identifiability in inverse reinforcement learning.
> >
> > [4] Kareem Amin, and Satinder Singh. Towards Resolving Unidentifiability in Inverse Reinforcement Learning.
> >
> > [5] Alberto Maria Metelli, Filippo Lazzati, and Marcello Restelli. Towards theoretical understanding of inverse reinforcement learning.
> >
> > [6] Andreas Schlaginhaufen, and Maryam Kamgarpour. Identifiability and Generalizability in Constrained Inverse Reinforcement Learning.
> >
> > [7] Paul Rolland, Luca Viano, Norman Schuerhoff, Boris Nikolov, and Volkan Cevher. Identifiability and generalizability from multiple experts in Inverse Reinforcement Learning.
> >
> > [8] Joar Max Viktor Skalse, Matthew Farrugia-Roberts, Stuart Russell, Alessandro Abate, and Adam Gleave. Invariance in policy optimisation and partial identifiability in reward learning.
> >
> > [9] Kuno Kim, Kirankumar Shiragur, Shivam Garg, and Stefano Ermon. Reward Identification in Inverse Reinforcement Learning.
> >
> > [10] Thomas Kleine Buening, Victor Villin, Christos Dimitrakakis. Environment Design for Inverse Reinforcement Learning.
> >
> > [11] David M. Kreps. Notes On The Theory Of Choice.
> >
> > [12] Daniel Kahneman and Amos Tversky. Prospect theory: An analysis of decision under risk.
> >
> > [13] Sumeet Singh, Jonathan Lacotte, Anirudha Majumdar, and Marco Pavone. Risk-sensitive inverse reinforcement learning via semi- and non-parametric methods.
> >
> > [14] Lillian J. Ratliff and Eric Mazumdar. Inverse risk-sensitive reinforcement learning.
> >
> > [15] Haoyang Cao, Zhengqi Wu, and Renyuan Xu. Inference of utilities and time preference in sequential decision-making.
> >
> > [16] Brian D. Ziebart, Andrew Maas, J. Andrew Bagnell, and Anind K. Dey. Maximum entropy inverse reinforcement learning.
> >
> > [17] Nicole Bauerle and Ulrich Rieder. More risk-sensitive markov decision processes.
> >
> > [18] Marc G. Bellemare, Will Dabney, and Mark Rowland. Distributional Reinforcement Learning.
> >
> > [19] Anirudha Majumdar, Sumeet Singh, Ajay Mandlekar, and Marco Pavone. Risk-sensitive inverse reinforcement learning via coherent risk models.
> >
> > [20] Marta Grześkiewicz. Uncovering Utility Functions from Observational Data with Revealed Preference Inverse Reinforcement Learning.

---

### Official Review · Reviewer_hwPu · 2024-11-04

**Soundness:** 2
**Presentation:** 2
**Contribution:** 2
**Rating:** 5
**Confidence:** 3

**Summary:**

The authors work on the inverse RL problem for non-risk-neutral agents. They address this problem of learning the risk attitude of the agent by learning the agent's utility function. They develop two algorithms for learning the utility function from data, and also test their approach in a toy setting.

**Strengths:**

This work identifies an important and interesting problem setting, as existing inverse RL does not consider that experts/agents to learn from can have non-neutral risk attitudes. It presents this motivation well and it explores natural questions regarding identifiability of utility. The proposed algorithms also appear to be novel.

**Weaknesses:**

* I felt the notation used in this paper might be unnecessarily complicated, which makes it more difficult to read. The paper starting from section 4 could also be more organized.
* I did not understand why the authors require a "deterministic reward function" (line 97), does this exclude any settings where the reward has noise? This seems to directly contradict the motivating example the authors present in Example 3.1 where the reward is noisy.
* I was not convinced of the practical application of this problem framing and approach.
    * When would one have access to agent demonstrations in multiple MDP environments, for each agent, in a practical application?
    * The authors criticize previous works for not being useful in practice, and also say that their algorithm works well in practice, but the empirical evaluations are weak. For example, it is hard to tell if the calculated utility is reasonable.
    * The Algorithms both require knowing the Lipschitz constant of the utility function U to use (Algorithm 1 needs it to set the discretization, and Algorithm 2 needs it to update the utility function). This seems like it would be difficult in practice.

**Questions:**

* In Example 3.1: why not just expand your state to include past rewards?
* Line 312 ("We consider the online UL problem setting"): In what way are your algorithms focused on the online setting? They are given historical trajectories.
* I had difficulty finding the definition of $\hat{p}^i$, which is necessary for Algorithm 1. How do these estimates need to be formed for your Theorem 5.1 to hold?

---

> ### Author Response · Authors · 2024-11-18
>
> We are glad that the Reviewer found the problem setting both interesting and important for the IRL community, and also that the Reviewer appreciated the analysis of the partial identifiability of the expert's utility. We provide detailed replies to their questions/comments below.
>
> ## Weaknesses:
>
> > I felt the notation used in this paper might be unnecessarily complicated, which makes it more difficult to read. The paper starting from section 4 could also be more organized.
>
> We agree with the Reviewer that the mathematical notation adopted might appear rather cumbersome and complicated at first sight. *However*, we remark that **it is necessary to avoid losing formality or even clarity**. Every symbol introduced in the main paper is necessary to convey some important concept. An intuitive justification for the "strong" notation is that we have to integrate three different formalisms: IRL, risk-sensitive RL, and distributional RL [1].
>
> If the Reviewer has any suggestions for improving the organization of the paper from Section 4 on, we would greatly appreciate your feedback.
>
> > I did not understand why the authors require a "deterministic reward function" (line 97), does this exclude any settings where the reward has noise? This seems to directly contradict the motivating example the authors present in Example 3.1 where the reward is noisy.
>
> We remark that **Example 3.1 does not require a noisy/stochastic reward function**. Indeed, the stochasticity of the transition model suffices to obtain the lotteries over returns described in the example, even with a deterministic reward function. Simply, add two next states, one with a *deterministic* reward of $0$€ and the other with $200$€, after playing $a_{\text{risky}}$, and make them reachable with probability $1/2$ from it (i.e., make the transition model *stochastic*).
>
> We have decided to **avoid considering stochastic rewards for the sake of simplicity**, since it would complicate the notation further without adding much value to the paper. Indeed, observe that **extending the results of the paper and also the algorithms to the setting with a stochastic reward function is trivial**. In particular, the proofs of Propositions 4.1-4.3, 4.5, 4.6, would still hold, and algorithms CATY-UL and TRACTOR-UL can be *easily* extended to such setting by adding a line of code for computing the *empirical* estimate (i.e., sample mean) of the reward function at each state-action pair of the space, and then by replacing the current usage of the deterministic reward value with such estimate. Next, the proofs of Theorems 5.1 and 5.2 could be adjusted by simply adding a term that quantifies the estimation error of the stochastic reward, and then by bounding it using the Hoeffding's inequality.
>
> > I was not convinced of the practical application of this problem framing and approach.
> When would one have access to agent demonstrations in multiple MDP environments, for each agent, in a practical application?
>
> In literature, there is **plenty of papers that analyze the setting** in which an agent provides demonstrations of behavior in multiple environments. Just to mention a few: [2,3,4,5,6,7,8]. To give some toy examples, you can think of an expert agent who demonstrates the task of driving safely on *multiple cars* (that have different dynamics), or on *multiple roads* (each road with its own traffic distribution), or a table tennis player that demonstrates how to play against *multiple adversaries*.

---

> > ### Author Response · Authors · 2024-11-18
> >
> > > The authors criticize previous works for not being useful in practice, and also say that their algorithm works well in practice, but the empirical evaluations are weak. For example, it is hard to tell if the calculated utility is reasonable.
> >
> > We would like to clarify this point. In the paper, **we do not claim that previous works are useless in practice**, but we claim that "*they either make demanding assumptions [...], or consider rather limited settings*" (lines 60-63). Simply put, if an algorithm makes strong assumptions, then it can only be applied to the (few) scenarios where they hold. Similarly, if a paper considers a limited setting, then its analysis cannot be applied to more general scenarios. Our algorithms, instead, lie on *milder assumptions* and can be applied to *more general settings*. We will rewrite this sentence to clarify the meaning.
> >
> > Concerning our algorithms, we remark that **the goal of the paper is not to *show* that they "work well in practice"**. The paper aims to introduce a novel problem setting (*Utility Learning*), to understand its theoretical difficulties in the simplest possible setting (tabular MDPs), and then to prove the existence of algorithms with theoretical guarantees of sample and computational complexity (CATY-UL and TRACTOR-UL). For these reasons, we believe that an *extensive* empirical validation **is not necessary** in this work. The proof-of-concept numerical simulations provided in the paper aim to show that our algorithms work reasonably in a simple problem.
> >
> > Finally, note that understanding if the calculated utility is reasonable is trivial. Indeed, **utility functions are highly interpretable**. As explained in any textbook on behavioral economics (e.g., see [9]), a convex utility represents a "risk-seeking" agent, while a concave utility represents a "risk-averse" agent. The more the convexity/concavity, the more the risk-sensitivity of the agent. If the utility is linear, then the agent is "risk-neutral". If a utility function is convex for certain amounts of money (or, more in general, reward) and concave for others, then this means that the represented agent has different risk preferences dependent on the amount of money at stake (this is very common in humans [10]).
> >
> > > The Algorithms both require knowing the Lipschitz constant of the utility function U to use (Algorithm 1 needs it to set the discretization, and Algorithm 2 needs it to update the utility function). This seems like it would be difficult in practice.
> >
> > We agree with the Reviewer that the knowledge of the Lipschitz constant is unreasonable in practice. *Nonetheless*: $(i)$ it is **common practice** for a variety of papers in RL and IRL when one aims to provide *theoretical* guarantees (e.g., see [11]), $(ii)$ it is well-known that, to perform **online learning with continuous actions**, it is **necessary** to know the regularity of the function to optimize [12], $(iii)$ **any upper bound to $L$ is fine** for our algorithms. In practical implementations, we set $\epsilon_0$ irrespective of $L$, and we choose $L$ based on the maximum of smoothness desired for the utility function to learn.

---

> > > ### Author Response · Authors · 2024-11-18
> > >
> > > ## Questions:
> > >
> > > > In Example 3.1: why not just expand your state to include past rewards?
> > >
> > > As mentioned in footnote 3 (lines 214-215), if we re-model the MDP by including the reward into the state to make the optimal policy Markovian, then we might incur in *interpretability* and *transferability* issues. To better explain this, let us make a **simple example**.
> > >
> > > Consider a driving setting, where the state is the location of the car (name of the road and position inside the road), the actions permit to change the current road (only when the car is close to another road, otherwise no effect), and at every stage/timestep the position of the car advances on the current road depending on the amount of traffic in the road, which is random and modelled through the transition model of the environment.
> > >
> > > Consider now an expert agent that *aims to reach a certain goal location $s_g$ in the minimum time/number of stages possible*, and that *is risk-averse*, in the sense that it prefers roads that always have little traffic, even though they are, on average, slower, to roads that are usually faster but sometimes have peaks of traffic that make them very very slow (since the traffic is random, there is no sequence of roads that is *always* better than others, but it is a matter of chance).
> > >
> > > In **our model** (Eq. (1)), we can represent the expert through the reward function $r^E$ that is $0$ in the goal location $s_g$, and $-1$ otherwise. In this manner, the faster a trajectory reaches $s_g$, the larger the cumulative reward. Next, we can choose the utility function $U^E$ to be some *concave* function in order to achieve the risk-aversion property [13], i.e., to make sure that the expert prefers, intuitively, roads with "smaller variance" of traffic. We remark that *our model permits to  capture the preferences of the expert in a very simple yet expressive manner*. In fact, as shown in this example, $r^E$ and $U^E$ can be easily designed, and their meaning is easily interpretable. However, this does not hold if we include past rewards in the state.
> > >
> > > In the **model with past rewards in the state**, the behavior of the expert is represented through a single reward $\overline{r}^E$ (defined on the expanded state space) instead of the pair reward-utility $r^E,U^E$. The intuition is that, to contain all the information present in $r^E,U^E$, the new reward $\overline{r}^E$ will be "messy". As such, designing it is also more complex. For instance, choosing $\overline{r}^E$ to be $-t$ in the (expanded) state made of the goal location $s_g$ and of $t$ timesteps, and $0$ elsewhere, represents a *risk-neutral* agent that aims to reach $s_g$ as soon as possible, but it does *not* capture the risk-aversion of the expert. To make $\overline{r}^E$ model the risk-aversion, we must take it to be a *concave* function of $-t$, making it *more difficult to be interpreted*.
> > >
> > > Even though the model with past rewards in the state guarantees the Markovianity of the optimal/expert policy, it suffers from major **drawbacks**:
> > > - $\overline{r}^E$ has a size (i.e., it is defined on a number of states) that grows *exponentially in the horizon* in the worst case, while $r^E,U^E$ do not.
> > > - $\overline{r}^E$ is more difficult to *interpret* (and design) than the pair $r^E,U^E$, whose meaning is immediate.
> > > - $\overline{r}^E$ can only be transferred to problems with the same state-action (or feature) space. Instead, the utility $U^E$ can be easily *transferred* to other kinds of environments. E.g., in the considered example, $U^E$ can be used to assess how much the expert "values its time" and takes decisions based on it. Thus, we can predict the behavior of the expert in other problem settings where the time plays a role using $U^E$, even if the state-action (or feature) space is different (e.g., if the expert travels by train instead than by car, we can predict if it prefers taking a reliable train, or a faster train on average that sometimes makes huge delays).
> > >
> > > For these reasons, expanding the state space by including the past rewards is not satisfactory in terms of interpretability and transferability. We will make this point clearer in the paper.

---

> > > > ### Author Response · Authors · 2024-11-18
> > > >
> > > > > Line 312 ("We consider the online UL problem setting"): In what way are your algorithms focused on the online setting? They are given historical trajectories.
> > > >
> > > > As explained at the beginning of Section 5, we consider a setting in which we are given batch (i.e., offline) datasets $\mathcal{D}^{E,i}$, that give information on the expert's policies $\pi^{E,i}$, *and* **we can actively (i.e., online) explore the environment** through generative sampling models [14], that provide information on the transition models $p^i$. Thus, the setting is online because we can actively explore the environment to estimate its dynamics. Note that this "mixed" setting with a batch dataset for the expert's policy and an online setting for the transition model is widespread in IRL and imitation learning (IL). For instance, GAIL [15] adopts it.
> > > >
> > > > > I had difficulty finding the definition of $\hat{p}^i$ which is necessary for Algorithm 1. How do these estimates need to be formed for your Theorem 5.1 to hold?
> > > >
> > > > As mentioned at lines 395 and 432, these estimates are constructed through Algorithm 3. Simply put, these estimates are the *empirical* estimates (i.e., sample means) of the transition model $\hat{p}^i_h(\cdot|s,a)\approx p^i_h(\cdot|s,a)$ at all $(s,a,h)\in\mathcal{S}\times\mathcal{A}\times[H]$, for all $i\in[N]$. This kind of estimates are widespread in the RL and IRL literature (e.g., see [14]).
> > > >
> > > > [1] Marc G. Bellemare, Will Dabney, and Mark Rowland. Distributional Reinforcement Learning.
> > > >
> > > > [2] Haoyang Cao, Samuel N. Cohen, and Lukasz Szpruch. Identifiability in inverse reinforcement learning.
> > > >
> > > > [3] Kareem Amin, and Satinder Singh. Towards Resolving Unidentifiability in Inverse Reinforcement Learning.
> > > >
> > > > [4] Kareem Amin, Nan Jiang, and Satinder Singh. Repeated Inverse Reinforcement Learning.
> > > >
> > > > [5] Andreas Schlaginhaufen and Maryam Kamgarpour. Towards the transferability of rewards recovered via regularized inverse reinforcement learning.
> > > >
> > > > [6] Joar Max Viktor Skalse, Matthew Farrugia-Roberts, Stuart Russell, Alessandro Abate, and Adam Gleave. Invariance in policy optimisation and partial identifiability in reward learning.
> > > >
> > > > [7] Kuno Kim, Kirankumar Shiragur, Shivam Garg, and Stefano Ermon. Reward Identification in Inverse Reinforcement Learning.
> > > >
> > > > [8] Thomas Kleine Buening, Victor Villin, Christos Dimitrakakis. Environment Design for Inverse Reinforcement Learning.
> > > >
> > > > [9] David M. Kreps. Notes On The Theory Of Choice.
> > > >
> > > > [10] Daniel Kahneman and Amos Tversky. Prospect theory: An analysis of decision under risk.
> > > >
> > > > [11] Dongsheng Ding, Kaiqing Zhang, Tamer Basar, and Mihailo Jovanovic. Natural Policy Gradient Primal-Dual Method for Constrained Markov Decision Processes.
> > > >
> > > > [12] Robert Kleinberg, Aleksandrs Slivkins, and Eli Upfal. Bandits and Experts in Metric Spaces.
> > > >
> > > > [13] Nicole Bauerle and Ulrich Rieder. More risk-sensitive markov decision processes.
> > > >
> > > > [14] Mohammad Gheshlaghi Azar, Remi Munos, and Hilbert Kappen. Minimax PAC bounds on the sample complexity of reinforcement learning with a generative model.
> > > >
> > > > [15] Jonathan Ho, and Stefano Ermon. Generative Adversarial Imitation Learning.

---

> > > > > ### Author Response · Authors · 2024-11-25
> > > > >
> > > > > Dear Reviewer,
> > > > >
> > > > > thank you again for the effort in reviewing our paper. We take the liberty to ask if our rebuttal has adequately addressed your concerns. We are happy to provide further clarification and engage in any discussion.
> > > > >
> > > > > Thank you!
> > > > >
> > > > > The Authors.

---

> > > > > > ### Comment · Reviewer_hwPu · 2024-11-26
> > > > > >
> > > > > > Thank you for your detailed response and clarifications! Here are my follow-up comments:
> > > > > >
> > > > > > - Regarding your response about the notation being difficult to read, even if notation is necessary, it is not presented in a way that is easy to follow for a general ML audience. We would recommend making it easier to find the definition of various quantities and, if possible, moving a more detailed discussion of certain points that are more secondary to the contribution of the paper to the appendix (especially points that involve introducing a lot of new notation).
> > > > > > - Regarding your suggested fix to Example 3.1, you are suggesting to define the reward as a function of the current state, the action, and the next state. However, you do not define this to be the case in display (1) of your paper. If this is a critical motivating example for your work, we think it is critical that your example fits in your formal problem setup. It's not clear to us that allowing rewards to be stochastic or depend on the next state follows trivially from your work.

---

> > > > > > > ### Author Response · Authors · 2024-11-26
> > > > > > >
> > > > > > > Concerning the follow-up comments, we address them below:
> > > > > > > - Thank you for the suggestion, we will implement it for the final version by moving to the appendix some parts with lots of notation that do not represent the core of our contributions.
> > > > > > > - Actually, we are not suggesting to redefine the reward as a function of the state-action-next state. What we mean is that we can simply construct Example 3.1 with a deterministic reward $r:\mathcal{S}\times\mathcal{A}\to\mathbb{R}$, as shown in the picture that we added to Appendix C.1 in the revised version of the paper.

---

> > > > > > > > ### Author Response · Authors · 2024-11-29
> > > > > > > >
> > > > > > > > Dear Reviewer,
> > > > > > > >
> > > > > > > > has our explanation equipped with the picture in Appendix C.1 clarified your doubt? Please, let us know.
> > > > > > > >
> > > > > > > > Thank you!
> > > > > > > >
> > > > > > > > The Authors.

---

### Official Review · Reviewer_7YNb · 2024-11-04

**Soundness:** 3
**Presentation:** 3
**Contribution:** 3
**Rating:** 6
**Confidence:** 3

**Summary:**

This paper introduces a model for learning the utility functions of agents in Markov Decision Processes (MDPs) based on observed demonstrations, with a focus on capturing agents' risk attitudes through utility functions. The authors formalize the Utility Learning (UL) problem and propose two provably efficient algorithms, CATY-UL and TRACTOR-UL, to infer utilities with finite data and theoretical guarantees on sample complexity. Proof-of-concept experiments are also conducted to validate the approach.

**Strengths:**

The problem of inferring agents' risk attitudes is important in sequential decision-making, where traditional Inverse Reinforcement Learning (IRL) models fall short by assuming risk-neutral agents.

**Weaknesses:**

1. While the paper extends utility-based approaches to MDPs, the core contributions, such as incorporating risk attitudes in utility functions and the algorithms for UL, rely heavily on prior work in IRL and risk-sensitive models. This might limit the novelty of the proposed approach. Maybe highlight the technical novelty in main text can help.
2.  The assumptions made to guarantee compatibility under finite data might not hold in more complex settings. I hope the author can provide comprehensive discussion on this issue and some intuition on possible relaxation.

**Questions:**

1. Could the authors clarify under which specific conditions the CATY-UL and TRACTOR-UL algorithms remain effective in highly stochastic environments, especially where reward distributions are complex?
2. How do the partial identifiability limitations impact the interpretability of learned utilities? Are there ways to mitigate this in practical applications?
3. Can the authors expand on how the proposed approach compares with recent risk-sensitive IRL methods in terms of computational efficiency and empirical performance?

---

> ### Author Response · Authors · 2024-11-18
>
> We are glad that the Reviewer appreciated the significance of the considered problem setting, as a powerful extension of IRL when interacting with risk-sensitive agents. Below, we report answers to the Reviewer's comments.
>
> ## Weaknesses
>
> > While the paper extends utility-based approaches to MDPs, the core contributions, such as incorporating risk attitudes in utility functions and the algorithms for UL, rely heavily on prior work in IRL and risk-sensitive models. This might limit the novelty of the proposed approach. Maybe highlight the technical novelty in main text can help.
>
> We remark that the paper brings several novel contributions that do not simply rely on prior work in IRL. We summarize them in the following:
> 1) We are the first researchers that **recognize the importance of non-Markovian policies** in modelling the behavior of (human) agents in presence of stochasticity/risk in MDPs for inverse problems. We provide an intuitive justification in Example 3.1 and a proof-of-concept empirical demonstration in Experiment 1 (see Section 6), where we show that, in a toy MDP, the majority of the participants to the study exhibit a non-Markovian behavior.
> 2) We present the **first model** of (human) behavior in MDPs that **complies with non-Markovian policies** (see Eq. (1)), and we show its superior empirical validity (in a small-scale study) over "Markovian" models in Section 6, Experiment 1. No author in the IRL literature has analyzed either this or another model compliant with non-Markovian policies. Indeed, most of the authors focus on a Boltzmann rationality model [1,2], that considers Markovian policies. In addition, we remark that even Eq. (1) in itself represents a novel contribution **as a formulation of the inverse problem**. In fact, we use it for modelling the behavior of (boundedly rational) agents (*inverse problem*), while works like [3,4] adopt it for constructing rational agents (*control problem*), i.e., for prescribing how an "intelligent" agent should behave in face of stochasticity/risk. The difference is subtle but clear, because it leads to two *very different* problem settings and algorithmic solutions.
> 3) We prove the first results that **characterize the properties** of this original setting when it comes to applications like *predicting*, *imitating*, or "*assessing*" behavior (see Propositions 4.1-4.3, 4.5 and 4.6). Note that these results, as well as their proofs, are novel, and they demonstrate an interesting non-trivial *connection with IRL*.
> 4) We propose to **extend two IRL algorithms** because of the connection with IRL proved in Section 4. However, the extension demands *technical effort to integrate algorithmic solutions* from distributional RL [5], risk-sensitive control [4], and IRL [6,7], which are needed to face the challenges highlighted in Section 5.1. Inter alia, we **refine the analysis** in [4] using the Bernstein's inequality to improve the sample complexity rate.
> 5) We provide a **proof-of-concept empirical analysis** of the proposed algorithms and model in a toy environment with real-world data.
>
> We will make the contributions clearer in the paper.
>
> > The assumptions made to guarantee compatibility under finite data might not hold in more complex settings. I hope the author can provide comprehensive discussion on this issue and some intuition on possible relaxation.
>
> We are unsure about which assumptions the Reviewer is referring to. Anyway, as mentioned also in Section 6, when the horizon $H$ becomes very large, assuming that the observed human expert plays an *optimal* risk-sensitive policy $\pi^E$, i.e., a policy that satisfies Eq. (1) *exactly*, might be too demanding. Indeed, due to bounded rationality, people are often not able to "look too many steps into the future" [8], and, thus, they may demonstrate a suboptimal policy. For this reason, we have left to future works (see lines 531-532) to analyze what are the maximum values of the horizon $H$ that make our model realistic. Nevertheless, we remark that **the optimality assumption is standard in IRL** for all the values of the horizon. E.g., [9] assumes the expert to be optimal in an *MDP*, while [10] assumes the expert to be optimal in an *entropy-regularized MDP*. Analogously, in this paper, we assume the expert to be optimal in a *risk-sensitive MDP*.

---

> > ### Author Response · Authors · 2024-11-18
> >
> > ## Questions
> >
> > > Could the authors clarify under which specific conditions the CATY-UL and TRACTOR-UL algorithms remain effective in highly stochastic environments, especially where reward distributions are complex?
> >
> > First of all, we remark that  CATY-UL and TRACTOR-UL  have been designed and analyzed (proofs of Theorems 5.1 and 5.2) for **(arbitrarily) stochastic transition models** and **deterministic reward functions**. As common in RL, assuming a deterministic reward function is just for simplicity and does not represent a loss of generality. Nevertheless, **CATY-UL and TRACTOR-UL can be easily extended to environments where the reward functions are (arbitrarily) stochastic**, as follows. First, we add a line of code for computing the *empirical* estimate (i.e., sample mean) of the reward function at each state-action pair of the space, and then we replace the current usage of the deterministic reward value with such estimate. Next, we modify the proofs of Theorems 5.1 and 5.2 by simply adding a term that quantifies the estimation error of the stochastic reward, and then we bound it using standard concentration inequalities (e.g., Hoeffding's inequality). We remark that the extension is *trivial* from the technical viewpoint, but it can be *tedious* from a notational perspective.
> >
> > > How do the partial identifiability limitations impact the interpretability of learned utilities? Are there ways to mitigate this in practical applications?
> >
> > The **major disadvantage** of the partial identifiability issue is that, even if we find a "*risk-averse*" utility function $U_{\text{risk-averse}}$ compatible with the observed expert's behavior $\pi^E$, i.e., a utility inside the feasible utility set $\mathcal{U}\_{\pi^E}$ (Definition 4.1) that tells us that the expert is risk-averse, then we cannot conclude that the expert's behavior is risk-averse, because there might be another utility function $U_{\text{risk-lover}}$ inside the feasible set that is "*risk-lover*". However, in such case, we could not even conclude that the expert likes risk because of the presence of the "*risk-averse*" utility in the set. This is the simplest example that illustrates how the interpretability of any learned utility is affected by partial identifiability.
> >
> > We propose **two main methods for mitigating this problem**: $(i)$ *increasing the amount of data* (i.e., information) that we have, by asking for expert demonstrations in additional environments. Indeed, as we prove in Proposition 4.6, demonstrations in multiple environments permit to consistently reduce the amount of utilities inside the feasible set. When collecting more data is not possible, $(ii)$ we can *impose more structure* to the problem by making stronger assumptions. For instance, we might assume that the expert's policy, beyond satisfying Eq. (1), also maximizes some notion of margin, so that we can identify it (almost) uniquely. This second method is analogous to popular strategies for mitigating partial identifiability in IRL [9,11].
> >
> > We will remark this in the final version of the paper.

---

> > > ### Author Response · Authors · 2024-11-18
> > >
> > > > Can the authors expand on how the proposed approach compares with recent risk-sensitive IRL methods in terms of computational efficiency and empirical performance?
> > >
> > > Concerning the **computational complexity**, CATY-UL always converges after $\mathcal{O}\Big(\frac{NSAH^3L^2}{\epsilon^2}\Big)$ basic operations. Instead, TRACTOR-UL requires $\mathcal{O}\Big(\frac{N^4H^4L^2}{\epsilon^4}\Big)$ gradient iterations for guaranteeing accuracy $\epsilon$ with high probability (see Theorem 5.2). In the experiments conducted in Section 6, on a toy MDP with $N=1,H=5, L=1, \epsilon\approx 1e-2$, with an accurate choice of learning rate, we have observed that TRACTOR-UL achieves an error smaller than $\epsilon$ in less than 100 iterations *on average* over multiple runs with different seeds.
> > >
> > > Before mentioning the computational efficiency of the main risk-sensitive IRL algorithms in literature, we remark that these algorithms have *different learning targets* w.r.t. our algorithms and among each other, thus *making a direct comparison of the number of iterations/running time would not be fair*. Intutively, this is the same issue that arises when directly comparing different IRL algorithms (e.g., when comparing [9] with [12]), because they aim to extract *different rewards* under *different assumptions*.
> > >
> > > The algorithms proposed in [13] require solving a linear program for every state-action pair demonstrated by the expert (in an environment simpler than an MDP) to learn a coherent risk metric (and not a utility function). The algorithm in [1] is a gradient-based method, but the authors do not mention in the paper the number of iterations used to learn the parameters of a certain, parametric, utility function with a desired accuracy. Finally, the experiments conducted in [2] show that a number of gradient iterations in-between 100 and 1500 suffice to obtain a good estimate of their targets in the experimental setting considered.
> > >
> > > Concerning the **empirical performance**, again we notice that we cannot compare these algorithms directly because they extract different quantities. We can make a simple *intuitive* comparison using the data considered in Experiment 1 (Section 6). Since most of the participants to the study demonstrate *non-Markovian* strategies in this setting and since our algorithms are the only ones that try to "fit" a non-Markovian policy to the data, then it appears natural that, *if all the considered algorithms were solving the same task (i.e., extracting the same quantity), then our algorithms should perform better.* Of course, to make this statement formal, one should try to modify the algorithms in [1,2,13] to our, novel, setting, which is not immediate, and out of the scope of this paper.
> > >
> > > Finally, we mention that we cannot compare the **sample complexity** of the various algorithms, because our algorithms are the *only* ones that come with *theoretical guarantees*.
> > >
> > > [1] Lillian J. Ratliff and Eric Mazumdar. Inverse risk-sensitive reinforcement learning.
> > >
> > > [2] Haoyang Cao, Zhengqi Wu, and Renyuan Xu. Inference of utilities and time preference in sequential decision-making.
> > >
> > > [3] Nicole Bauerle and Ulrich Rieder. More risk-sensitive markov decision processes.
> > >
> > > [4] Zhengqi Wu and Renyuan Xu. Risk-sensitive markov decision process and learning under general utility functions.
> > >
> > > [5] Marc G. Bellemare, Will Dabney, and Mark Rowland. Distributional Reinforcement Learning.
> > >
> > > [6] Filippo Lazzati, Mirco Mutti, and Alberto Maria Metelli. How to scale inverse rl to large state spaces? a provably efficient approach.
> > >
> > > [7] Andreas Schlaginhaufen and Maryam Kamgarpour. Towards the transferability of rewards recovered via regularized inverse reinforcement learning.
> > >
> > > [8] Daniel Carton, Verena Nitsch, Dominik Meinzer, and DirkWollherr. Towards Assessing the Human Trajectory Planning Horizon.
> > >
> > > [9] Andrew Y. Ng and Stuart J. Russell. Algorithms for inverse reinforcement learning.
> > >
> > > [10] Brian D. Ziebart. Modeling purposeful adaptive behavior with the principle of maximum causal entropy
> > >
> > > [11] Nathan D. Ratliff, J. Andrew Bagnell, and Martin A. Zinkevich. Maximum margin planning.
> > >
> > > [12] Brian D. Ziebart, Andrew Maas, J. Andrew Bagnell, and Anind K. Dey. Maximum entropy inverse reinforcement learning.
> > >
> > > [13] Sumeet Singh, Jonathan Lacotte, Anirudha Majumdar, and Marco Pavone. Risk-sensitive inverse reinforcement learning via semi- and non-parametric methods.

---

> > > > ### Author Response · Authors · 2024-11-25
> > > >
> > > > Dear Reviewer,
> > > >
> > > > thank you again for the effort in reviewing our paper. We take the liberty to ask if our rebuttal has adequately addressed your concerns. We are happy to provide further clarification and engage in any discussion.
> > > >
> > > > Thank you!
> > > >
> > > > The Authors.

---

> > > > ### Comment · Reviewer_7YNb · 2024-11-27
> > > >
> > > > Thank you for your detailed response. Most of my concerns have been well addressed. Therefore, I will increase my score to 6. I hope the author could incorparate discussion to further improve the paper in future versions.

---

> > > > > ### Author Response · Authors · 2024-11-27
> > > > >
> > > > > Thank you. We are glad that our answers and comments have well addressed the concerns of the Reviewer. We will add the important points of our discussion in the final version of the paper.

---

### Author Response · Authors · 2024-12-02
**Discussion Ending**

Dear Reviewers,

As the discussion phase approaches its conclusion, we would like to take this opportunity to thank you for your efforts and valuable feedback so far.

We also wish to ask if there are any remaining concerns that we can address. Additionally, in light of the rebuttal and the discussions, we kindly ask if you would consider revisiting and potentially adjusting your scores if deemed appropriate.

Thank you once again for your time and thoughtful contributions.

Best regards,

The Authors

---

### Meta-Review · Area_Chair_SdzW · 2024-12-24

**Metareview:**

In this paper, the authors propose a model for learning the utility function of an agents in MDPs based on observed demonstrations. The goal is to capture the risk attitude of the agent in learning the utility function. They formalize the problem of utility learning, propose two algorithms, CATY-UL and TRACTOR-UL, to infer utilities, and prove sample complexity for them. They also provide experimental results to validate their approach.

The reviewers agree that the problem of inferring the agent's risk attitude in sequential decision-making is interesting and novel. However, they are not convinced about the usefulness/practicality/necessity of this setting (explicitly learning risk-sensitive utility functions). Moreover, they found the main assumption behind this work, i.e., having access to the agent's demonstrations in multiple MDP environments and more importantly these demonstrations having exactly the same utility function, very strong and perhaps even unrealistic. There are also concerns about the readability of the work due to complicated notations, and the weakness of empirical evaluations.

**Additional Comments On Reviewer Discussion:**

The authors addressed some of the reviewers' concerns, especially those about the related work. They convinced the reviewers that the setting studied in the paper is novel and the related work mentioned by them does not study the exact same setting. However, they were not fully successful in convincing the reviewers about the usefulness of the setting and the assumptions used being realistic (as described in the meta-review).

---

### Decision · Program_Chairs · 2025-01-22

Reject